palaeontology/evolution/taxonomy and systematics

Scincidae, Egerniinae, Oligocene, Miocene, palaeontology

**Author for correspondence:**
K. M. Thorn
e-mail: kailah.thorn@uwa.edu.au

†Present address: Edward de Courcy Clarke Earth Science Museum, School of Earth Sciences M004, University of Western Australia, 35 Stirling Highway, Crawley 6009, Australia.

# A new species of *Proegernia* from the Namba Formation in South Australia and the early evolution and environment of Australian egerniine skinks

K. M. Thorn[1,2,†], M. N. Hutchinson[1,2], M. S. Y. Lee[1,2],
N. J. Brown[1], A. B. Camens[1] and T. H. Worthy[1]

[1]College of Science and Engineering, Flinders University, Bedford Park 5042, South Australia
[2]South Australian Museum, North Terrace, Adelaide 5000, South Australia

KMT, 0000-0002-0645-835X; THW, 0000-0001-7047-4680

The diverse living Australian lizard fauna contrasts greatly with their limited Oligo-Miocene fossil record. New Oligo-Miocene fossil vertebrates from the Namba Formation (south of Lake Frome, South Australia) were uncovered from multiple expeditions from 2007 to 2018. Abundant disarticulated material of small vertebrates was concentrated in shallow lenses along the palaeolake edges, now exposed on the western of Lake Pinpa also known from Billeroo Creek 2 km northeast. The fossiliferous lens within the Namba Formation hosting the abundant aquatic (such as fish, platypus *Obdurodon* and waterfowl) and diverse terrestrial (such as possums, dasyuromorphs and scincids) vertebrates and is hereafter recognized as the Fish Lens. The stratigraphic provenance of these deposits in relation to prior finds in the area is also established. A new egerniine scincid taxon *Proegernia mikebulli* sp. nov. described herein, is based on a near-complete reconstructed mandible, maxilla, premaxilla and pterygoid. Postcranial scincid elements were also recovered with this material, but could not yet be confidently associated with *P. mikebulli*. This new taxon is recovered as the sister species to *P. palankarinnensis*, in a tip-dated total-evidence phylogenetic analysis, where both are recovered as stem Australian egerniines. These taxa also help pinpoint the timing of the arrival of scincids to Australia, with egerniines the first radiation to reach the continent.

# 1. Introduction

Major biogeographical events have shaped the Australian flora and fauna, since the separation from Antarctica approximately 45 Ma, through to Pleistocene glacial cycles [1,2]. Australia's herpetofauna can be broadly traced to two origins: relics from the break-up of Gondwana (the southern supercontinent), or recent arrivals from Asia to Australasia [3,4]. Gondwanan origins are inferred for diplodactyloid geckos, chelid turtles and some frogs [5,6]. The Asian route appears to have been taken by agamid lizards, and elapid and typhlopid snakes sometime during the Oligo-Miocene [3,7,8]. The ages and origins of Australian scincid lizard radiations are still relatively poorly constrained, and molecular data are unclear beyond a few island-dwelling extant relatives in Southeast Asia [9].

The Australian scincids comprise three subfamilies: the Egerniinae Welch, 1982, Sphenomorphinae Welch, 1982 and Eugongylinae Welch, 1982 [10]. While these three clades have been included in large all-squamate phylogenies, detailed molecular phylogenetic analyses have also been conducted for the sphenomorphine [11,12] and egerniine radiations [13], but no recent molecular phylogeny of the Australian eugongyline skinks has been published (see [14], which includes some Australian taxa). A molecular phylogenetic analysis of Australian skinks [3] produced a node-calibrated molecular clock estimate of the divergence of all three subfamilies. Those results suggested that the Australian Egerniinae arrived before 18.2 Ma, Eugongylinae before 22.9 Ma and the Sphenomorphinae were potentially the first scincid clade to reach Australia, before 25.3 Ma. However, analyses using mid-Miocene fossil calibrations indicated a much earlier origin for the Australian egerniines, minimally 34.61 Ma [15]. This new date prompted the current investigation to seek fossil evidence that might further demarcate the timing of the arrival of (potentially) Australia's first scincids.

The oldest Australian fossil with distinctive scincid characters, the egerniine *Proegernia palankarinnensis* Martin *et al.*, 2004 [16] is from the Etadunna Formation, Lake Eyre Basin, at the Oligo-Miocene boundary 25–26 Ma [16]. An Eocene femur loosely assigned to Scincomorpha by Hocknull is not convincingly scincid [17], so is not robust evidence for inferring the temporal origin of the Australian Scincidae. Other Oligo-Miocene Australian material from the Etadunna Formation was referred to Egerniinae but remains undescribed, including a dentary (UCR 20814), broken parietal (UCR 20815), partial maxilla (UCR 20816), an assortment of vertebrae and a broken scapulocoracoid [18]. Both the *P. palankarinnensis* holotype and the material referred by Estes [18] cannot currently be located. New collections of Oligo-Miocene material are required to resolve the composition and relationships of Australia's oldest scincid faunas. Recent expeditions into central South Australia have unearthed new Oligo-Miocene fossil squamates from the Namba Formation at Lake Pinpa and Billeroo Creek, southeast of Lake Frome. The Namba Formation is of a similar age to the Etadunna Formation [19], and the vertebrate fossil material collected from the lowest layers of this formation are termed the Pinpa Local Fauna [20]. This investigation revises the stratigraphic and palaeoecological setting of this deposit, and places the new Oligo-Miocene egerniine fossils into phylogenetic context, alongside *P. palankarinnensis* from the Etadunna Formation, and the Miocene species of *Egernia* and *Tiliqua* analysed in Thorn *et al.* [15], to infer a more robust estimate for the age of the Australian radiation of egerniines, which represents the latest possible date for their arrival in Australia.

# 2. Material and methods

## 2.1. Stratigraphy, excavation and fossil identification

Sediment samples from excavated trenches were taken to examine the stratigraphy of both Billeroo Creek (BC2) and Site 12, Lake Pinpa (LP12). Mineral composition was determined by a combination of three approaches: (i), the elements K, Na, Mg, Ca, Mn, Fe and Al were quantified in an analysis of soluble minerals by inductively coupled plasma–mass spectrometry (ICP-MS). A 0.25 g sample of sediment was microwave digested with 5 ml of aqua regia then made up to 50 ml. Insolubles were removed by centrifuging and their percentage mass calculated. Each sample was analysed in triplicate by ICP-MS; (ii) X-ray diffraction (XRD) was used for semi-quantitative mineralogy; and (iii) X-ray fluorescence (XRF) analyses measured major and minor trace elements. These analyses, conducted by Flinders Analytical, aimed to identify the minerals present, particularly if dolomite was present, and in what proportion, as this is indicative of changing environments. Nested sieves with mesh apertures of 6, 3 and 1 mm were used to create sediment fractions from which the fossil material was sorted from the dolomitic clays. The micro-vertebrate material was sorted into fish, mammal, bird, turtle, crocodile,

frog and squamate. Of the 48 squamate cranial specimens recovered, 43 were scincids, and five identified as geckos (see electronic supplementary material, Information for the squamate specimen list). Most fossils discussed in this investigation were recovered by sieving excavated sediment and sorting the concentrates, but some were collected from the ground surface having been exposed by erosion.

All descriptive terminology of the cranial elements follows Evans [21], Richter [22] and Kosma [23] for tooth crown features. Appendicular skeleton terminology follows Russell & Bauer [24], and Hoffstetter & Gasc [25] for vertebrae.

To justify referrals of specimens to the new species, we took a pragmatic approach and prioritized apomorphies when available, but also used distinct combinations of characters (some of uncertain polarity or plesiomorphic) in order to eliminate some competing possibilities. We assumed that the taxon most plausibly belongs to one of the extant clades of Australian squamates, and that the broad patterns of morphological variation seen in these living Australian squamate clades are a reliable guide to the limits of variation in the late Oligocene.

## 2.2. Scanning electron microscopy

Cranial material identified to Scincidae was further cleaned in water, dried, and then imaged using scanning election microscopy (SEM). The minute size of some specimens is beyond the capabilities of Micro-CT for resolution of morphological features required for identification and descriptions. No specimens required sputter coating. All SEM work was conducted at Flinders University Microscopy facilities using an Inspect FEI F50 SEM. Maximum voltage used for image taking was limited to 2 kV and a spot size of 3–4. Measurements of tooth crowns and features of the dentary mentioned in the text were taken using the SEM software.

## 2.3. Phylogenetic analyses

In order to better understand the timing of the Australian colonization by the Egerniinae, both molecular and morphological data (including fossils) are required to generate tip-dated phylogenies. Undated parsimony and tip-dated Bayesian analyses infer, respectively, the phylogeny with the least homoplasy, and the most probable dated phylogeny.

### 2.3.1. Morphological characters

Morphological characters used in the following analyses consisted of 102 discrete and 48 continuous traits, forming an expanded matrix from Thorn *et al*. [15]. The expanded character list is available as electronic supplementary information accompanying this article, titled 'Morphological character list accompanying Thorn *et al*. pdf'. Continuous characters, derived from the measurements of the individual bones or teeth from the dentaries and maxillae, were taken from either Micro-CT scan data in Avizo Lite (v. 9.0) or SEM at Flinders Microscopy, to the nearest micrometre, or with digital callipers to the nearest 10 micrometres. All measurements were converted to ratios of either dentary or maxilla length to standardize for size. Continuous character states were linearly scaled to values spanning 0–2 to replicate the mean number of discrete character states (three), for analyses in both TNT [26] and BEAST 1.8.3 [27], so that they do not have a disproportionate weight.

### 2.3.2. Molecular partitions

Molecular data sourced from Tonini *et al*. [28] and Gardner *et al*. [13] were analysed using PartitionFinder 2 [29] to find optimal partitions and substitution models [15]. The same six molecular (gene) partitions, 12s (412 base pairs [bps]), 16s (681 bps), ND4 (693 bps), BDNF (699 bps), CMOS (835 bps) and B-fibrinogen (1051 alignable bps) and substitution models chosen in that study [15] are used again here.

### 2.3.3. Maximum parsimony

The parsimony analyses for the combined discrete morphological, continuous morphological, and molecular data were performed using TNT v. 1.5 [26]. *Eutropis multifasciata* was set as the most distant outgroup following the phylogenetic interpretations of Gardner *et al*. [13] and Thorn *et al*. [15]. The

**Table 1.** Fossil calibrations used, their minimum and maximum ages (Ma) and references for the age dates or species descriptions.

| fossil calibration | min age | max age | references for age |
| --- | --- | --- | --- |
| Zone A, Etadunna Formation | 25.5 | 25.7 | Woodburne MO et al. [19] |
| Pinpa Local Fauna, Namba Formation | 25.5 | 25.7 | Woodburne MO et al. [19] |
| AL 90 Locality, Carl Creek Limestone | 14.17 | 15.11 | Woodhead J et al. [37] |
| Gag Locality, Carl Creek Limestone | 14.47 | 16.86 | Woodhead J et al. [37] |

most parsimonious tree (MPT) for the combined data was found using 1000 replicates of tree-bisection-reconnection (TBR) with up to 1 000 000 trees held.

To assess clade support, 200 partitioned bootstrap replicates (with discrete characters, continuous characters, and each gene locus treated as a separate resampling partition), were performed using TNT, using new search methods (XMULT) with 1000 replicates and 1 000 000 trees held. The MPT and bootstrap trees from TNT were exported in nexus format, and continuous and discrete characters were traced (in Mesquite; [30]). The executable files for finding the MPT, and for performing 200 reps of Partitioned Bootstrap resamples can be found in the Dryad Digital Repository 'Character set and phylogenetic analyses of the living and fossil egerniine scincids of Australia'.

### 2.3.4. Bayesian analysis

The discrete and continuous morphological data, and molecular data were simultaneously analysed in BEAST v. 1.8.4 using tip-dated Bayesian approaches [27]. *Eutropis multifasciata* was again set as the furthest outgroup. Polymorphic discrete morphological data were treated exactly as coded rather than as unknown, i.e. if coded as states (0,1) it was treated as 0 or 1, but not 2. The discrete character set was analysed using the Mkv-model with correction for non-sampling of constant characters [31,32]. Despite recent disputes over the effectiveness of this model [33], it is well tested [34,35] and is still widely accepted and applied to morphological data [36]. Continuous characters, transformed to span values between 0 and 2, were analysed with the Brownian motion model. Bayes factors were used to test the need to accommodate among-character rate variability for both discrete and continuous morphological characters (i.e. gamma parameter).

The stratigraphic data used for tip-dating analyses were derived from fossil taxa and their associated stratigraphy noted in table 1. No node age constraints were imposed in this analysis, all dates are retrieved from the morphological and stratigraphic age ranges from the noted fossil taxa (tips). The most appropriate available model in BEAST v. 1.8.4, birth-death serial sampling [38], was applied. An uncorrelated relaxed clock [39] was separately applied to the molecular and morphological data.

Each Bayesian analysis was run for 100 000 000 generations with a burn-in of 20%. The analysis was conducted four times to confirm stationarity. The post-burnin samples of all four runs were examined in Tracer 1.7.1 [40] to ensure convergence was achieved. All four runs were combined in LogCombiner, and the consensus tree produced by TreeAnnotator [27]. The executable .xml file for BEAST, all output log files, and the final consensus tree file (.tree) are available in the Dryad Digital Repository.

### 2.3.5. Systematic background

The higher taxonomic framework used below follows the consensus taxonomic arrangement used in the Reptile Data Base [10], which subdivides the previously widely used subfamily Lygosominae [41] into five clades [42–44].

# 3. Geological setting

## 3.1. The Namba Formation

The Namba Formation from northeastern South Australia (figures 1 and 2) shares an unconformable lower boundary with the Eyre Formation (Palaeocene-Eocene) within our study area in the Frome Basin; and is unconformably overlain by the Pleistocene Eurinilla Formation (figure 1). The Namba

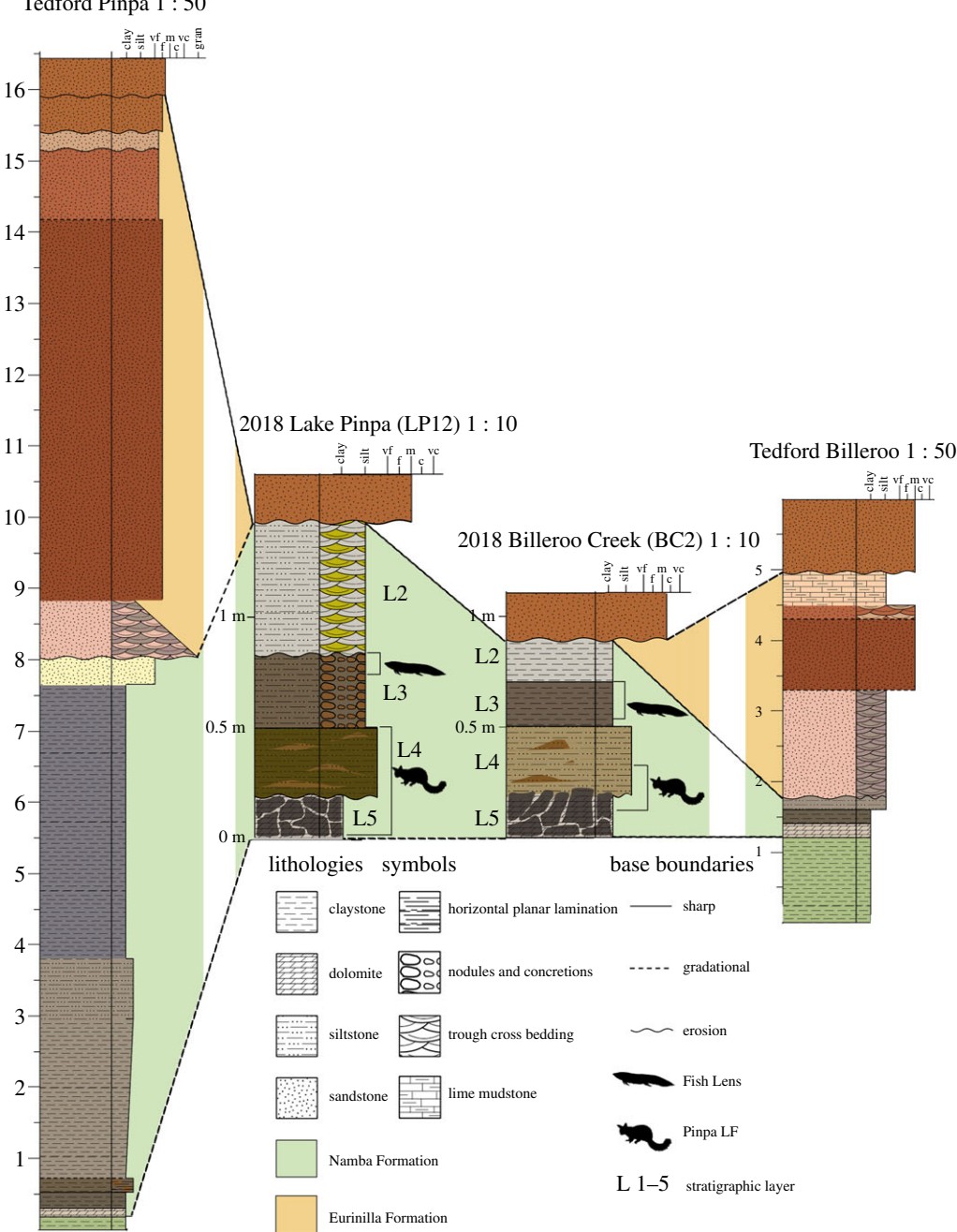

**Figure 1.** Stratigraphic sections through the Namba Formation at Lake Pinpa at Site 12 (figure 2) and Billeroo Creek, Site 2, constructed in SedLog 3.1 from authors' field observations in comparison with R. H. Tedford's sections described in notes and figures in [20,45], height in metres. Grain sizes were determined in the field. Colours are converted from Munsell figures to corresponding RGB values using [46].

sequence is a lateral equivalent to the Etadunna Formation from the northwestern Lake Eyre Basin [19,47]. The Namba Formation is divided into two members [20]; green claystones and dolomitic claystones at the top of the lower member host a locally abundant vertebrate fauna, termed the Pinpa Local Fauna [20]. This is biostratigraphically correlated with Zone A of the Etadunna Formation to be 25.5–25.7 Ma [19]. Fieldwork in the late 1970s–early 1980s, led by R. H. Tedford, T. H. Rich and others, mapped multiple fossil localities exposed in Lakes Pinpa, Namba, Tarkarooloo, Yanda and Tinko and Billeroo Creek in the Lake Frome Basin [48]; added to these are new numbered sites from expeditions carried out in 2007 and between 2015 and 2018 led by T. H. Worthy and A. B. Camens.

All of the sites from Lake Pinpa and Billeroo Creek yielding scincid material for this investigation expose and sample fluvio-lacustrine sites from the Namba Formation. Five expeditions collected new

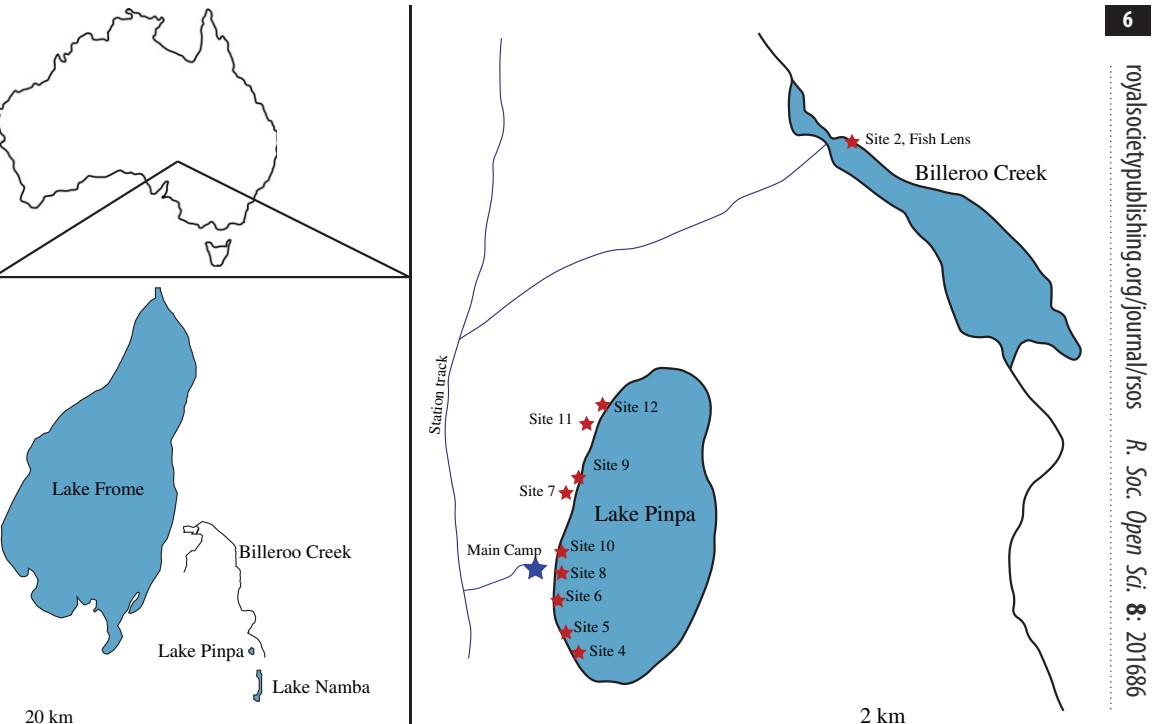

**Figure 2.** Location maps of the Lake Pinpa and Billeroo Creek sites, Frome Downs Station, South Australia. Stratigraphic columns in figure 1 were compiled from locations Site 12 (= LP12) and Site 2 Fish Lens (=BC2) marked above.

material from the Namba Formation from 2007 to 2018. Over this period, numerous squamate fragments were collected from multiple sites at Lake Pinpa and Billeroo Creek. Two sites have yielded the majority of the material described herein (see §3.2 below); Site 12 at Lake Pinpa, and the 'Fish Lens' at Billeroo Creek (within Wells' Bog Site of Tedford's, and later T. H. Rich's, expeditions).

Deposits and the fossils therein contributing to the Pinpa Local Fauna (LF) may be classed into two clear taphonomic groups: those containing isolated bones or partial skeletons of terrestrial vertebrates (marsupials, predatory birds, wading birds and meiolaniid turtles), and those containing localized concentrated bone accumulations which were first thought to be derived from crocodile coprolites, encompassing mostly aquatic vertebrates (mostly fish, but including also turtle, crocodile, dolphin and rare terrestrial vertebrates). Collection of both categories of this material was predominantly from surface exposures along the western edge of Lake Pinpa and northeastern Billeroo Creek [48], with the occasional excavation of articulated or associated marsupial skeletons. In slightly younger overlying/ incised fluviatile units exposed as channel fill deposits, the Ericmas Local Fauna has been derived from large-scale excavations at Ericmas and South Prospect quarries (Lake Namba, 5 km south of Lake Pinpa) in the 1970s and 1980s [48]. Tom O's Quarries excavated in fluviatile units at Lake Tarkarooloo unearthed the Tarkarooloo Local Fauna which is biochronologically similar to the Ericmas LF, by bulk processing and screening sediments on several expeditions led by T. H. Rich of Museums Victoria, once with the help of the Australian Army [49]. The quarried Ericmas LF sites contained predominantly terrestrial vertebrate remains, but until the 2007 Worthy and Camens expedition, no squamate material was recorded from the Namba Formation.

Squamate remains have predominantly been recovered alongside the smallest mammal taxa and were found in lenses of densely concentrated bones of aquatic vertebrates dominated by fish (Actinopterygii and *Neoceratodus* spp.). These fish lenses lie a few centimetres above dolomite of unknown depth (details below). The dolomite beds have revealed numerous fossils, including associated or partly articulated skeletons of birds and mammals, but fish bones are rare. This dolomitic bed was better exposed towards the middle of the lake bed in the 1970s when it was extensively sampled, but has since at least 2007 been buried by in-washed Quaternary sands. Both layers contain the same mammal and bird species, and so the faunas from each are collectively referred to the Pinpa Local Fauna [20,50]. Both the fish layer and underlying dolomites are exposed at Wells' Bog Site in Billeroo Creek shown in figure 1.

## 3.2. Fossil sites

Fossil sites in the Lake Pinpa–Billeroo Creek area were numbered chronologically on the 2007 trip in order of discovery, not based on geographical location (figure 2).

### 3.2.1. Lake Pinpa

Site 6 (LP6) SAMA P43058 a posterior fragment of a left scincid maxilla, was recovered from Site 6 on the 2007 expedition. The Fish Lens was observed eroding out in patches over a broad area (200 by 50 m) here around the margins of the overlying massive grey clay (Layer 2 in figure 1).

*Site 9* (LP9) SAMA P43057, a posterior fragment of a left scincid dentary, was recovered from Site 9 on the 2007 expedition. Here, the Fish Lens was exposed in a narrow zone at the base and edge of eroding massive clay Layer 2.

*Site 12* (LP12) Specimens SAM P57544 (partial humerus) and SAM P57545 (maxilla fragment) were found at Site 12 on the edge of Lake Pinpa. Bone was found eroding out on the surface, along the lake edge, from the lowest silty clay layer. Associated and articulated skeletons occur in both this layer and the dolomitic clay (Layer 5) beneath. An undulating clean erosional boundary was observed between these two units.

### 3.2.2. Billeroo Creek Site 2 Fish Lens

Billeroo Creek Site 2 (BC2) is located on the northern side of the creek (figure 2) and is a part of the more expansive Wells' Bog Site. Fossils derive from a concentrated lens of predominantly fish bone with limonite inclusions, within the top of Layer 3 of the Namba Formation (figure 1). The base of this fossiliferous 'Fish Lens' is not flat, but undulated relating to depth of semi-discrete lenses exposed over an area of roughly 10 by 3 m, excavated over three trips, these lenses with a maximum thickness of approximately 150 mm. The lens sits stratigraphically more than 200 mm above the basal dolomite (Layer 5). The dolomite layer is laterally more extensive than the Fish Lens at Billeroo Creek, due to erosion of the overlying layers, and is the layer from which the majority of skeletal fossils at the site have been collected.

## 3.3. Stratigraphy

*Layer 1*—The top layer of the stratigraphic section at LP12 (figure 1), designated Layer 1, is not part of the Namba Formation. The red sands are reworked sediment from the nearby Quaternary dunes that unconformably overlie the Namba exposures around many of the lakes in the area. At Billeroo Creek (BC2), the fluvial Eurinilla Formation lies unconformably on the Namba Formation and eroded sediments from both it and overlying dunes mantled the Namba Formation where the section was excavated.

*Layer 2*—Erosion of an unknown amount of the upper part of Layer 2 means its original depth at both BC2 and LP12 cannot be assessed, but near LP07, exposures as documented in the section by Tedford *et al*. [20] show it was minimally approximately 6 m thick. It sits unconformably on Layer 3. At LP12, Layer 2 is composed of interbedded light green-grey (7/1/10Y) and yellow (5/1/10Y) medium silts that display cross-bedding with very fine laminations. From a distance, they have an overall uniform, pale-grey appearance. The layer at BC2 is similar but with white (8/1/10YR) medium silt rather than yellow. No inclusions or fossils have been found in this unit, but locally vertically aligned gypsum crystal plates occur. Concentrations of most soluble minerals are lowest in this stratigraphic horizon.

*Layer 3*—The upper boundary of Layer 3 at LP12 is erosional, the troughs filled with sediments from Layer 2. Easily distinguishable from Layer 2, it is composed of an olive-grey (4/2/5Y) clay with strong brown (4/6/7.5YR) limonite inclusions throughout. Maximum thickness at Lake Pinpa was 150 mm. The same layer at BC2 was a dark greyish-brown (4/2/2.5Y) clay that had a maximum thickness of approximately 100 mm and limonite presence was less consistent. The results of the XRD supported field observations of the clay content of the sediment, with clay minerals smectite and palygorskite combining to make up 79% of Layer 3 at Lake Pinpa. Limonite inclusions were noted in the stratigraphic section in Layer 3. The XRD analysis of Layer 3 sediment from Lake Pinpa identified the iron ore mineral goethite (9% of total mineral content) a common component of limonite; this was supported by the XRF analyses which found 19.82% iron (II) and soluble Fe content greater than 16%

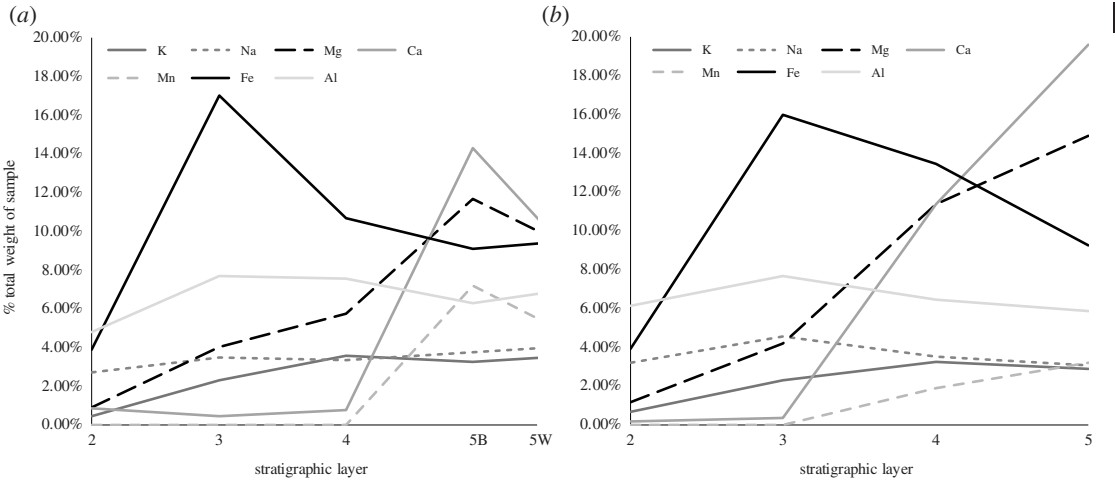

**Figure 3.** Results of the soluble mineral analyses of Layers 2–4 and the white (5W) and black (5B) sediments from Layer 5 at Lake Pinpa Site 12 (*a*); and Billeroo Creek Site 2 Layers 2–5 (*b*).

**Table 2.** Major and trace elements detected by X-ray fluorescence analysis of sediments sampled Layer 3 and 5 from stratigraphic sections of Lake Pinpa Site 12 and Billeroo Creek Site 2. Values are %, minor trace elements making the total of each sample 100 are not shown.

|  | MgO | Al$_2$O$_3$ | SiO$_2$ | CaO | MnO | Fe$_2$O$_3$ |
|---|---|---|---|---|---|---|
| Pinpa 12 L3 | 4.043 | 8.815 | 46.53 | 0.238 | 0.2303 | 19.82 |
| Pinpa 12 L5 | 10.45 | 7.444 | 43.64 | 7.901 | 4.227 | 8.105 |
| Billeroo Creek L5 | 13.65 | 5.875 | 36.08 | 13.02 | 1.634 | 6.982 |

at both sites. This spike in iron is not reached by any of the preceding layers. The Fish Lens at BC2 had variable thickness within the top 50 mm of this layer, but in some cases, Layer 2 sits directly on Layer 3 sediments, with no Fish Lens separating the sediments of each layer. At Lake Pinpa, the Fish Lens is usually much thinner (1–10 mm thick) than at Billeroo Creek and occurs near the top of Layer 3. Exceptions occur at LP6, where the Fish Lens is locally much thicker, sometimes up to 50–120 mm thick in areas of a couple of square metres scattered over several hundred square metres on the lake bed. Layer 3 sits conformably on Layer 4 in stratigraphic sections at BC2, but the boundary is unconformable at LP12.

*Layer 4*—150 mm of an olive (5Y 5/3) and orange (10YR 4/6) mottled clay that sits unconformably on Layer 5 at LP12 and BC2. Fossils were found in the lower portion of this layer, on the boundary with Layer 5 at both sites. At Billeroo Creek, Layer 4 is marked by a sharp increase in calcium, not present until Layer 5 at Lake Pinpa.

*Layer 5*—This layer is composed of a dolomitic white (10YR 8/1) mudstone with extensive black manganese staining giving a mottled appearance overall. Sixty-six per cent of sediment in Layer 5 at Pinpa and 54% of Layer 5 at Billeroo Creek (table 3) is derived of clay minerals smectite and palygorskite. Layer 5 had less Fe, but much more Mg, Ca and Mn than L3 and L4 (figure 3 and table 2), reflecting that it included the minerals dolomite/ankerite (table 3), not present in the overlying layers at either site. The mottled appearance of the dolomitic Layer 5 in both sections is explained by the transition from dolomite (CaMg(CO$_3$)$_2$) to ankerite (Ca(Fe,Mg,Mn)(CO$_3$)$_2$) with the partial replacement of magnesium with iron (II) and manganese. Carbonate presence was confirmed with dilute hydrochloric acid in the field, and reaffirmed by the soluble mineral result from both sites (figure 3). The primary source of fossils contributing to the Pinpa Local Fauna at Lake Pinpa and Billeroo Creek (excluding Fish Lens) is the assemblage occurring at the bottom of Layer 4 and in the top 100 mm of Layer 5.

# 4. Results

Our excavations have expanded the taxonomic diversity of the Pinpa Local Fauna (first recorded by Tedford *et al.* [20]). Notably, in 2015–2018 we recovered both associated skeletons and isolated bones

**Table 3.** Mineral components of sediment samples taken from Layers 3 and 5 at Lake Pinpa and Layer 5 at Billeroo Creek Site 2. The clay minerals are predominantly palygorskite and smectite.

| | clay minerals | dolomite/ankerite | goethite | quartz | other | total |
|---|---|---|---|---|---|---|
| Pinpa 12 L3 | 79 | — | 9 | 11 | 1 | 100 |
| Pinpa 12 L5 | 66 | 23 | 3 | 6 | 2 | 100 |
| Billeroo Creek L5 | 54 | 38 | 2 | 5 | 1 | 100 |

from the base of Layer 4 and the top of the dolomitic clay layer (Layer 5) at BC2 and LP12. Excavations at LP12 revealed predominantly terrestrial taxa with numerous remains of marsupials (vombatiforms, phalangeriforms, macropodiforms) and birds. In comparison, the fauna from the Fish Lens over various sites is more aquatic, a possible current-concentrated, lake-edge accumulation with bony fish abundant and lungfish, dolphins, flamingos, rails, turtles, crocodiles and the platypus *Obdurodon* represented, in addition to terrestrial marsupials (mainly dasyuromorphs, phalangeriforms) and small scincids.

# 5. Systematic palaeontology

Order: Squamata Oppel, 1811 [51]

 Family: Scincidae Gray, 1825 [52]

 Subfamily: Egerniinae Welch, 1982 [53]

 Genus: *Proegernia* Martin *et al.*, 2004 [16]

 *Proegernia mikebulli* sp. nov.

 Zoobank ID: urn:lsid:zoobank.org:act:068F625A-B537-43C7-A1B1-4F20DD5F6BF9

 **Holotype**—SAM P57502, a near complete right dentary; 27 tooth loci, 14 of which bear teeth.

 **Diagnosis**—The species is referred to the subfamily Egerniinae because of two features of the dentary: the Meckel's groove is closed except for a small anterior section, and the presence of a large inferior alveolar foramen. It is referred to the genus *Proegernia* because the tooth crowns widen anteroposteriorly from the shaft, the crista lingualis and crista mesialis are near horizontal and converge on an apex slightly posterior to the centre of the tooth, and there are medially prominent cristae on the anterior and posterior culmen laterales. *Proegernia* is further distinguished from other members of the Egerniinae (species in *Egernia*, *Bellatorias*, *Liopholis*, *Lissolepis*, *Cyclodomorphus*, *Tiliqua*, *Tribolonotus* and *Corucia*) by the combination of the following traits: a more anteriorly positioned apex of the splenial notch [16,54] at greater than 50% the anteroposterior length of the dentary; more than 22 tooth loci on the dentary and 20 on the maxilla; minimal anteroposterior flaring of the tooth crown with lateral compression of the medial face; a small sliver of open Meckel's groove immediately posterior to the symphysis and a concave ventral face on the pterygoid body. *Proegernia mikebulli* differs from *P. palankarinnensis* in lacking lateral tooth striae, in having up to five more tooth loci for a total of 27 on the dentary, and a much more medially inflected anterior tip to the dentary, making it more curved in dorsal view.

 **Type locality**—Fish Lens, subsite Billeroo Creek 2, Wells' Bog Site, northern side of Billeroo Creek, GPS coordinates 31°6′11.76″ S, 140°13′53.70″ E (WGS 84), figure 2.

 **Stratigraphy/Age**—Namba Formation, in Layers 3–5 (figure 1) as exposed in Billeroo Creek and Lake Pinpa; Pinpa Local Fauna. This Local Fauna has been biostratigraphically correlated with the 'Wynyardiid' or Minkina Fauna, (Zone A) of the late Oligocene Etadunna Formation, 25.5–25.7 Ma [19,55].

 **Paratype**—SAM P57499 a partial right dentary from BC2, with a complete coronoid process, 23 tooth loci and 10 teeth.

 **Referred specimens**—The following specimens are referred to this taxon based on the combination of their appropriate size, stratigraphic co-occurrence, and having a general similarity to the equivalent elements in other egerniines. Details of tooth crown shape also enable robust referral of tooth-bearing bones to the same taxon. Two right post-dentary compound bones; SAM P57543 from Lake Pinpa (LP6) preserving the dorsal surface of the surangular and glenoid, broken part-way through the retroarticular process; and SAM P57542 from Billeroo Creek (BC2) which preserves the ventral and medial face of the articular, the glenoid entirely and most of the retroarticular process. Two right partial maxillae both from BC2, SAM P57541 representing an anterior fragment with premaxillary

process intact and the first nine tooth loci holding three teeth; and SAM P57503, which preserves the posterior majority of the maxilla with 16 loci and nine teeth, the facial process is broken above the row of maxillary foramina. An intact left premaxilla, SAM P57542 from BC2, with osteoderm fragments on the internasal process, preserving two teeth from four loci. A left pterygoid SAM P57512 from BC2, the quadrate process broken beneath the epipterygoid notch.

*Etymology*—The species is named after Professor Michael Bull (1947–2016) of Flinders University, South Australia, who devoted decades to documenting the ecology of Australia's egerniine skinks. Mike supervised a generation of Australian ecologists and his lectures inspired countless students to become biologists. His studies of Australian egerniine skinks and their parasites are model long-term ecological studies, and led to major discoveries such as the existence of monogamous pairs in *Tiliqua rugosa*, and parental care and family living in *Egernia* spp.; and the establishment of a successful breeding and reintroduction programme for the endangered pygmy bluetongue, *Tiliqua adelaidensis*.

## 5.1. Description

**Dentary**—The most diagnostic and commonly recovered element is the dentary bone of the lower jaw. Seventeen incomplete dentaries were recovered from the 2007–2018 trips to Lake Pinpa and Billeroo Creek. A reconstruction of a near-complete lower jaw of *Proegernia mikebulli* was made, with the dentary portion based on the holotype SAM P57502 and paratype SAM P57499 (figure 4). From the anterior tip of the symphysis to the posterior tip of the coronoid process, missing only the angular process, the reconstructed dentary is 12.7 mm long. In medial aspect, the dentary is convex ventrally. It is 1.6 mm wide and 2.2 mm tall at mid-length of the tooth row, and the dental sulcus is shallowly concave. From occlusal view, the symphysis is directed medially to articulate with the opposing dentary at an angle of 32°. The dental sulcus is clearly differentiated, SAM P57502 preserving 27 loci and 14 pleurodont teeth. The tooth row is 10.8 mm long.

The symphysis, preserved in its entirety on SAM P57502, has a flattened, reverse '7' shape in medial view. The symphysis has two caudally directed branches; a narrow, ventrally directed sliver along the anterior edge of the dentary; and, dorsally, a wider process that terminates in a sharp point. Maximally, the symphysis is 1.8 mm long and 1.3 mm deep. In the notch between the caudally directed branches of the symphysis a symphysial foramen extends ventrally into a 1 mm long and less than 0.2 mm wide anterior opening of Meckel's groove. The groove is open and aligned parallel with the dorsal side of the ventral section of the symphysis; further posteriorly it is closed and the bone fused.

The inferior alveolar foramen and the dorsal and ventral margin of the splenial notch are preserved to the posterior end of the tooth row in SAM P57502 (figure 4). The inferior alveolar foramen lies on the ventral margin of the splenial notch, below the mid-point of the tooth row. The shape of the splenial notch is narrow and roughly parallel-sided for the anterior 40% of the preserved length, expanding dorsoventrally with a convex upper edge and straight ventral margin for the remaining posterior section of length. Where the notch expands, the dorsal edge preserves a concave face, allowing the splenial to medially overlap the dentary. Posterior to this face, beneath the third last tooth, the dental sulcus is broken.

The coronoid process of the dentary is preserved on the paratype SAM P57499 and ascends posterodorsally from the position of the last tooth to a tip projecting dorsally above the posterior tooth. The ventral margin of this process preserves the anteromedial articular facet for the coronoid, which therefore can be seen to overlap the splenial and dentary anterior of the third last tooth position. The angular process, although not preserved entirely on either dentary specimen, can be reconstructed with some confidence using the angle of the intact edge immediately beneath the coronoid process. The total length of this process is limited by the absence of a facet for its articulation on either of the recovered post-dentary compound bones.

The lateral face of the dentary is slightly convex and preserves a single row of eight mental foramina extending posteriorly from the anterior top of the dentary to below the 17th tooth position. The largest foramen in the row is at the posterior end.

**Post-dentary (compound) bone**—The post-dentary complex or compound bone is the fused surangular and articular (including prearticular) bones making up the posterior half of the scincid mandible. This fusion develops ontogenetically; complete fusion of these elements with no traces of a suture internally is evidence of adulthood in extant Australian scincids [56]. Two right post-dentary complexes were recovered from the Namba Formation, SAM P57543 from a right mandible at Lake Pinpa (LP6; figure 4 3a–c) and SAM P57542 from Billeroo Creek (BC2; figure 4 4a–c); these are used in the reconstruction of the complete lower jaw (figures 4 and 5). These elements are referred to this

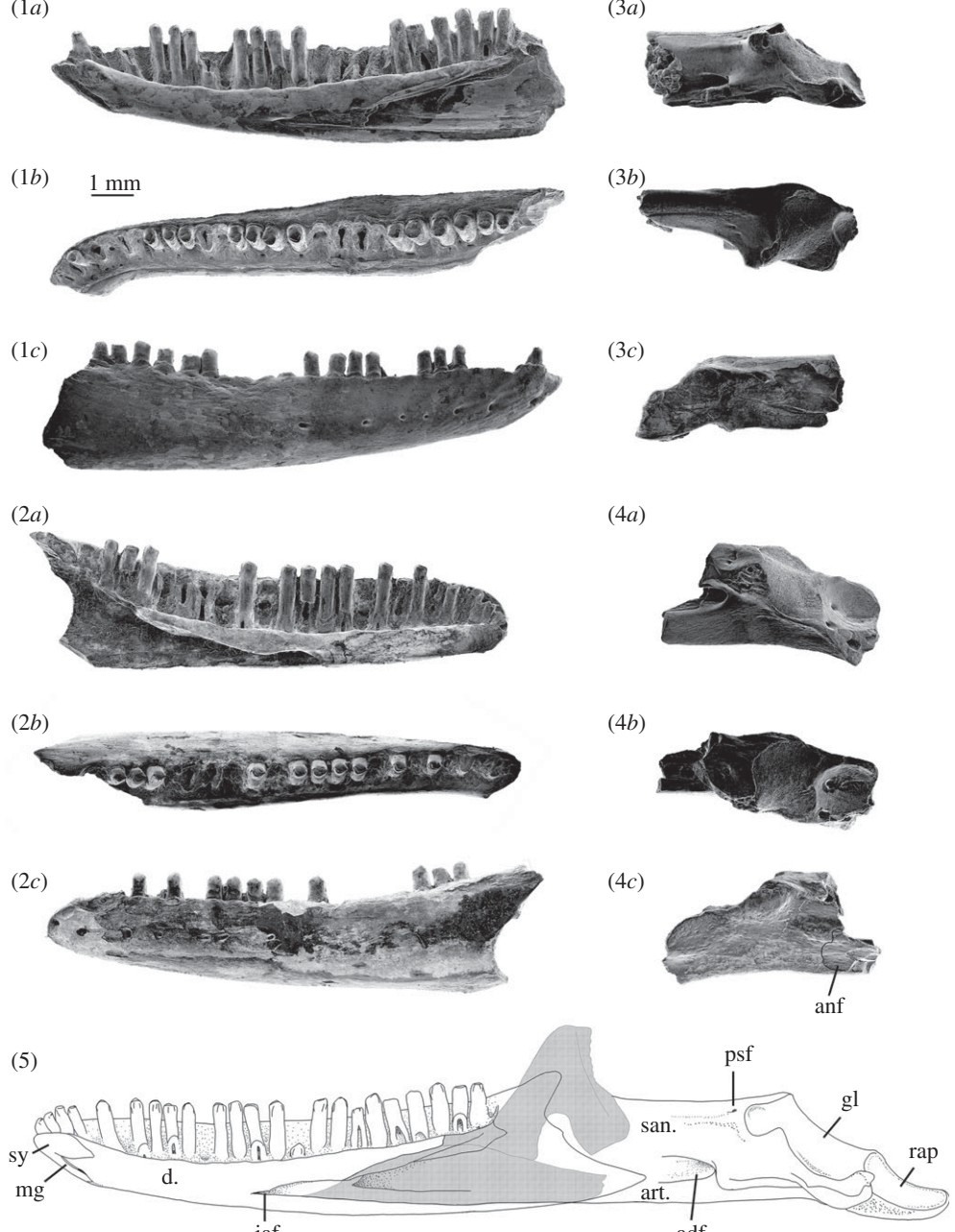

**Figure 4.** (1*a–c*) Holotype, a right dentary, SAM P57502, (2*a–c*) paratype; a left dentary SAM P57499; two right post-dentary compound bones (3*a–c*) SAM P57543 and (4*a–c*) SAM P57542 of *Proegernia mikebulli* sp. nov. from the Fish Lens at BC2; and (5) complete reconstruction in medial view of the right mandible of *Proegernia mikebulli*, the coronoid and splenial (hatched area) are reconstructed based on modern egerniine specimens, no fossil representatives of these elements are known. adf, adductor fossa; anf, angular facet; art., articular; d., dentary; gl, glenoid fossa; iaf, inferior alveolar foramen; mg, Meckel's groove; psf, posterior surangular foramen; rap, retroarticular process; san., surangular; and sy, symphysis.

taxon as they are of appropriate size, confirmed as scincid material by the presence of a facet for the angular bone, which is not present in gekkotans; and the angle of the retroarticular process. The shape of the retroarticular process is similar to that of extant egerniines. When aligned horizontally in the *in vivo* position, the angular in egerniines is wide and extends further laterally than the narrow ventrally positioned angular of eugongylines (e.g. *Emoia longicauda* SAMA R2352, *Eugongylus rufescens* R36735). Sphenomorphines generally have a simple, straight post-dentary complex without medial torsion of the articular, and the retroarticular is not inflected medially posterior of the glenoid. All scincid bones from Billeroo Creek and Lake Pinpa are consistent with the presence of a single egerniine taxon; there is no variation among elements that might indicate more than one species being represented, so both compound bones are referred to the taxon named from the dentaries.

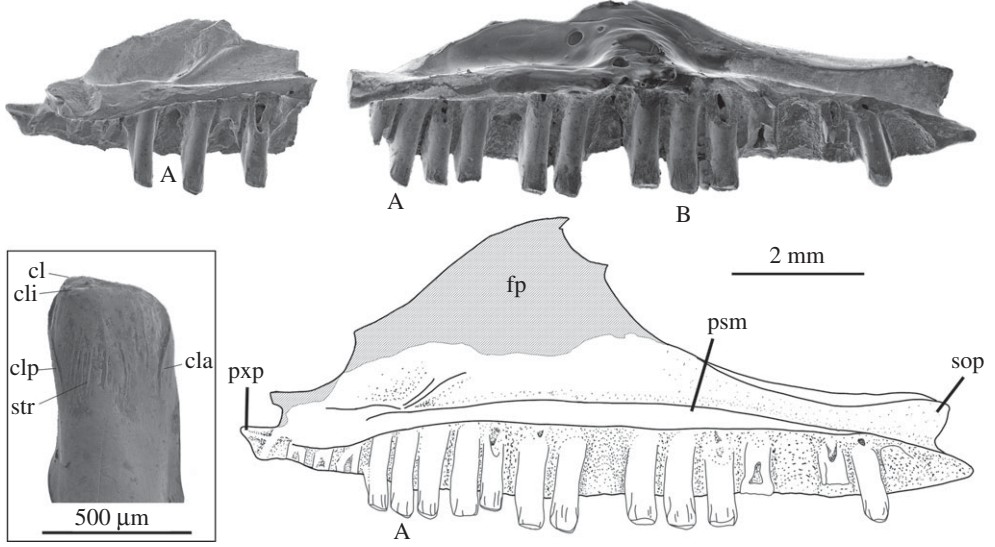

**Figure 5.** Right maxilla of *Proegernia mikebulli* sp. nov. reconstructed from SAM P57541 (anterior fragment; top left) and SAM P57503 (near complete; top right). 'A' marks the same tooth locus on each specimen. Enclosed by the box is the tooth 'B'. cl, *cuspis labialis*; cla, *culmen lateralis anterior*; cli, *cuspis lingualis*; clp, *culmen lateralis posterior*; fp, facial process; psm, palatal shelf of the maxilla; pxp, premaxillary process; sop, suborbital process; and str, striae.

SAM P57543 is a near complete right adult post-dentary preserving completely fused surangular and articular bones. Anteriorly the areas for articulation with the dentary and coronoid are not preserved, neither are the angular or the anterior edge of the adductor fossa. A facet for articulation of the angular is present on the ventral half of the lateral face of the articular, not extending to reach the posterior surangular process. The posterior majority of the retroarticular process is broken off SAM P57543, so a description of its complete shape is based on SAM P57542. The dorsal surface of the surangular is relatively straight and rises slightly to meet the dorsal tip of the mandibular condyle or glenoid fossa where the quadrate articulates with the mandible. The maximum preserved length of this dorsal surface is 2.9 mm. The glenoid fossa faces posterodorsally, the surface is convex mediolaterally and concave anteroposteriorly, reaching a maximum length of 1.8 mm. The lower third of the glenoid fossa becomes slightly concave ventrally, curving up to meet the medial articular process of the surangular. The surface of the glenoid fossa has a pitted texture suggestive of articular cartilage. Posteroventrally the glenoid fossa ends in a clearly defined ridge between it and the retroarticular process. The preserved section of the retroarticular process is concave (SAM P57542). Medially, the foramen for the *chorda tympani* nerve is preserved in SAM P57542.

Two foramina are preserved dorsally on a flattened surface, and a posterior surangular foramen is present on the lateral face immediately anterior of the glenoid fossa. No sign of an anterior surangular foramen is present on either SAM P57543 or SAM P57542. From the flattened dorsal surface of the surangular, the bone curves sharply ventrally, leaving a flat medial surface above the dorsal margin of the adductor fossa. The adductor fossa is in the lower 50% of the medial face of the complex. The anterior margin of the adductor fossa is not preserved on either post-dentary compound bone. Viewed medially, the posterior edge of the narrow, oval, fossa terminates anterior to the glenoid. The lateral face of the post-dentary complex of the mandible is convex. A discernible ridge runs from just anterior of the posterior surangular foramen, anteriorly to the broken anterior edge of SAM P57543, marking the posteroventral edge of the *M. adductor mandibulae externus*.

**Maxilla**—Two incomplete right maxillae (SAM P57503, SAM P57541) recovered from BC2 are referred to *Proegernia mikebulli*. SAM P57503 (figure 5, right) preserves a near complete tooth row, a bifid posterior termination of the suborbital process (an apomorphy of Scincoidea, see [57]), and complete tooth crowns from both the anterior tooth form and posterior tooth form on the same row. This specimen was recovered from sieved material in two halves that fitted together when articulated, and so was joined with Paraloid B72 before SEM images were taken. The smoothed hemispherical shape on the dorsal edge of the palatine process of the maxilla (figure 5) is Paraloid and not a feature of the bone.

Anteriorly, the maxilla SAM P57503 lacks the tip of the premaxillary process. The anteriormost tooth positions are missing on SAM P57541; tooth and loci counts are based on the reconstruction also using

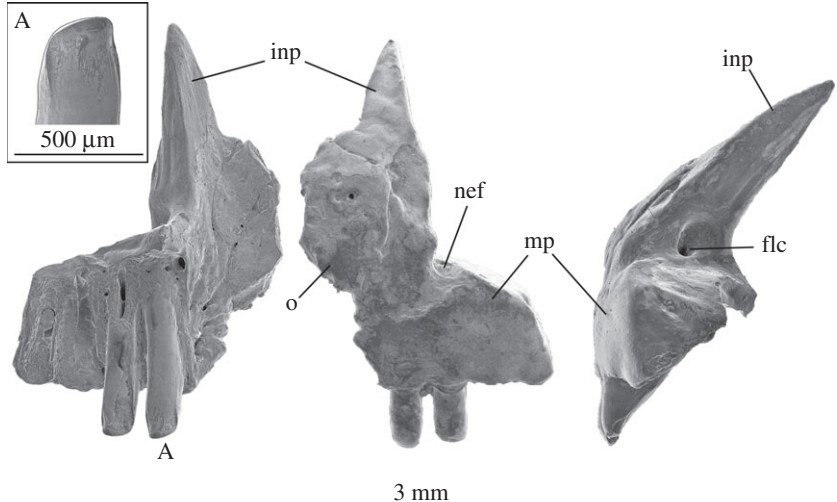

**Figure 6.** Left premaxilla (SAM P57542) of *Proegernia mikebulli* sp. nov. from posterior (left), anterior (centre), and lateral (right) view. A, tooth one, enlarged in box. flc, foramen of longitudinal canal; inp, internasal process; mp, maxillary process; nef, notch of the ethmoidal foramen; and o, osteoderm.

SAM P57541. The facial process of the maxilla is incomplete preserving only 2.3 mm above the tooth row. The unbroken edge of the orbit can be traced from above the level of the 14th tooth position, to the broken dorsal tip of the suborbital process. The ventral tip of the bifid suborbital process is intact as are the last tooth positions. The tooth row preserves 16 of 20 loci, and nine pleurodont teeth are still in position.

The lateral face of the maxilla is slightly convex in the anterior-to-posterior plane when viewed dorsally. The maxilla preserves nine primary foramina on the lateral face, the largest is the most anterior. Above the primary row, 11 more foramina, the largest one-third the size of the primaries, are present in two rows. Medially, the medial edge of the palatal shelf is broken. The palatal shelf thins posteriorly and ends in a fine point at the ventral tip of the bifid suborbital process. The tooth row occlusal surface is slightly convex posteriorly, with larger teeth beginning at the ninth position. The last two teeth decrease in size posteriorly, creating the trailing edge of the convex occlusal line.

The reconstructed maxilla is 11.1 mm long from the premaxillary process to the suborbital process, with 20 tooth positions (figure 5). The original height of the facial process was not preserved on any specimen.

**Premaxilla**—A single complete left premaxilla, SAM P57542, was recovered from Fish Lens at BC2 (figure 6). An osteoderm is fused to the anterior face of the internasal process, and would have overlapped the right premaxilla when the pair were articulated. Paired, unfused premaxillae in adulthood eliminate the possibility that this element belongs to a crown gekkotan. The ascending internasal process has a flattened medial side for articulation with its paired element. Viewed laterally, a foramen for the longitudinal canal is situated immediately above the maxillary process in the lateral face of the internasal process. Sharply pointed dorsally, the internasal process widens ventrally and terminates at the notch for the exit of the ethmoidal foramen laterally, and tooth row medially. The internasal process of the premaxilla rises towards the nasals at a 40° angle. The total length of the internasal process is 3.1 mm. The maxillary process is curved posterolaterally towards the maxilla and is approximately one-third of the height of the internasal process, reaching 1.6 mm in mediolateral width. The lateral edge of the maxilla process curves ventrally to finish beside the fourth tooth position. Four tooth loci are present and the first two teeth are preserved. These teeth are 0.53 mm tall measured from the lateral face of the maxillary process, and 0.28 mm wide. What remains of the worn crowns (see inset A, figure 6) match those of the anterior-most teeth of the maxilla specimen SAM P57541. There are weak striae on the medial face of the crown, a mediolateral compression of the crown below the *crista lingualis*, and a sharply curved, prominent, *culmen lateralis*.

**Dentition**—The dentition of *Proegernia mikebulli* is described using the two dentaries SAM P57502 and SAM P57499, the maxillae SAM P57541 and SAM P57503, and premaxilla SAM P57542, together comprising a near-complete upper and lower tooth row. The upper tooth row contains a total of 24–25 teeth, four or five on the left and right premaxilla, and 20 on the maxilla. The dentary tooth row has 27 tooth positions, beginning above the symphysis anteriorly and terminating just anterior of

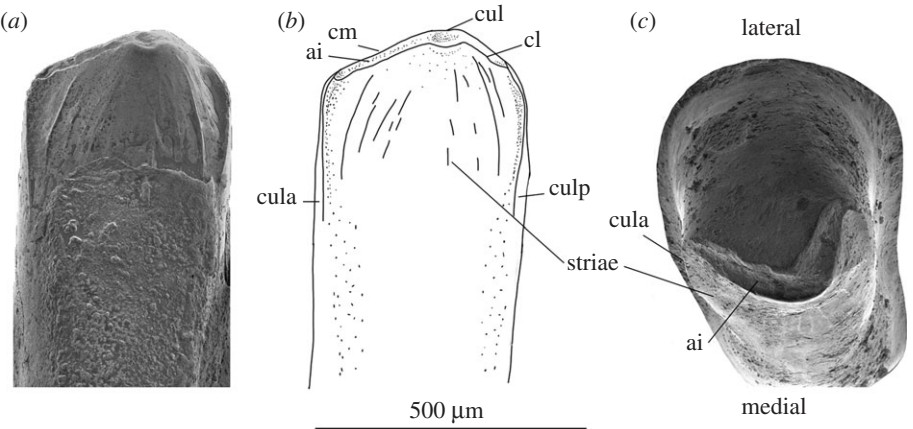

**Figure 7.** (*a,b*) 24th tooth on the dentary SAM P57502, medial view, and (*c*) the 16th tooth in occlusal view with lateral and medial facies marked. ai, *antrum intercristatum*; cl, *crista lingualis*; cm, *crista mesialis*; cul, *cuspis labialis*; cula, *culmen lateralis* anterior; and culp, *culmen lateralis* posterior.

the coronoid process. The occlusal profile of the maxilla and dentary are both convex, with slightly larger teeth in the posterior half of the tooth row.

On the dentary teeth, the tooth crown is similar in width to the shaft, expanding anteroposteriorly with slight mediolateral compression for the last 30% of the tooth height. Prominent anterior and posterior *culmina laterales* extend from the tips of the *crista mesialis*, turning medially and ventrally on the dentary, with a sharp angle producing 'shoulders' notable in medial view of the tooth (figure 7). From the central cusp, the anterior cristae dip ventrally 20°, and posterior cristae 45°. The cusp (*cuspis labialis*; figure 7) is slightly posterior of the centre of the tooth, and in occlusal view is positioned just posteromedial to the centre. This medial shift of the cusp creates a convex dorsal surface to the tooth, directing both cristae medially. Striae are only located on the medial face of the tooth crown, angled dorsoventrally from the off-centre cusp. The two most prominent striae run nearly parallel to the *culmina laterales* (both anterior and posterior), all others are weaker in profile with staggered lengths. Tooth crown morphology is similar on the maxilla and premaxilla.

Tooth wear occurs first on the cusp, forming a shallow rounded depression, gradually deepening medially. The *antrum intercristatum* expands in width from the central cusp. Wear is thus more noticeable in the centre of the tooth crown between the cusps, than anteriorly or posteriorly along the cristae, creating a thickening 'v' shape.

**Pterygoid**—A single left pterygoid attributed to *P. mikebulli* was recovered from Fish Lens at BC2 (SAM P57512; figure 8). It is near complete, missing only the anterior-most tip of the palatine process and distal tip of the quadrate process. The pterygoid articulates with the palatine anteriorly with a pointed process extending anteriorly from a fanned, v-shaped pterygoid body to meet the ectopterygoid and palatine. Laterally to the palatine process, the ectopterygoid process extends towards the ectopterygoid with a concave, curved facet for articulation with this element on the dorsal surface of the pterygoid head. Between these two processes, the fan-shaped pterygoid body has a concavity on the ventral surface that is 0.7 mm wide and 1.7 mm long. Within this concavity are three foramina, one in the deepest area of the concavity, and two are paired anteriorly, near the palatine process. Posteriorly, the pterygoid body narrows to become a parallel-sided rod, bending medially at an angle of 35° towards the epipterygoid notch. The epipterygoid notch is an oval concavity, with raised margins, on the dorsal surface of the pterygoid at the anterior end of the quadrate process. A sharp ridge extends from the anterior edge of the epipterygoid notch to the corner of the bend towards the pterygoid head. Posterior to the epipterygoid notch, the quadrate process becomes L-shaped in cross section and extends 1.5 mm posteriorly before terminating at a break. The pterygoid body is 2.7 mm wide between the palatine and ectopterygoid processes. Total length of this element is unknown.

## 5.2. Postcranial material potentially attributable to *Proegernia mikebulli*

Several procoelous vertebrae and assorted vertebral fragments were recovered from the Fish Lens at BC2. The best preserved of these is SAM P57514, a presacral vertebra (figure 9). This specimen is nearly

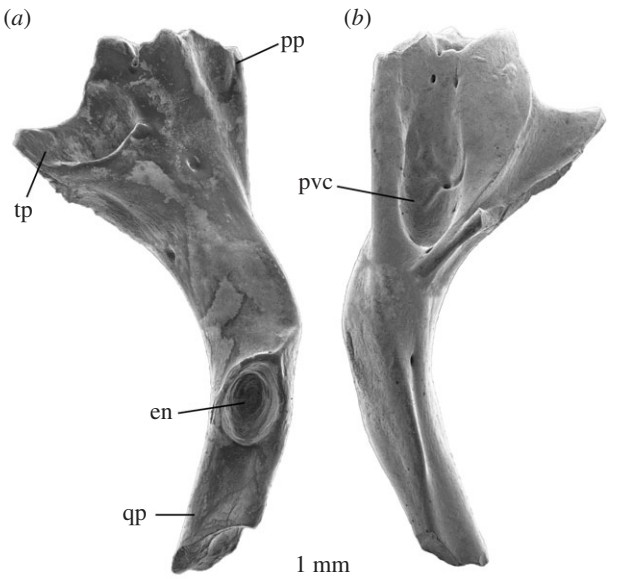

**Figure 8.** Left pterygoid of *Proegernia mikebulli* sp. nov., SAM P57512 in dorsal (*a*) and ventral (*b*) views. en, epipterygoid notch; pp, palatine process; pvc pterygoid ventral concavity; qp, quadrate process; and tp, ectopterygoid process.

complete, missing only the posterodorsal tip of the neural spine and the lateral tip of the right postzygapophysis. A thin crack runs posterolaterally on the ventral face of the centrum from just posterior of the ventral foramina, to the broken zygapophysis. The vertebra has an elongate centrum; the ventral surface is concave in lateral view. The neural canal is similar in diameter to the cotyle/condyle with a flat ventral surface broadening laterally before coming to a tear drop point when viewed anteriorly or posteriorly. Synapophyses are present immediately posterior to the prezygapophyses on the lateral facies. The neural spine rises posteriorly, steadily at an angle of 20°, from the dorsal margin of the neural canal until reaching the broken extremity. The prezygapophyses are angled at 32° and the postzygapophyses at 28°. Overall shape of the vertebra is relatively elongate, approximately one third longer than it is wide. Overall centrum length is 4.2 mm, the cotyle maximum width is 1.6 mm; the vertebra height anteriorly is 2.4 mm, posteriorly including the broken neural spine the height is a maximum of 3.7 mm. The widest point of the vertebra is between the synapophyses, totalling 3.3 mm.

This element is identified as a lizard based on procoely, absence of a zygosphene/zygantrum, minimal dorsoventral compression of the rounded cotyle and condyle, elongated centrum, and the minimal size of the ventral foramina. Snake vertebrae have shorter centra, a round cotyle/condyle with no compression, a set of zygosphenes and zygantra, and a ventral hypapophysis. None of the features described are apomorphic within Scincidae and so referral to species is not possible. However, referral to *Varanus* is excluded by the taller, less-compressed centrum. A procoelous centrum devoid of a hollow space for the notochord eliminates the possibility of the vertebra belonging to a non-pygopodid gekkotan. Gekkotans inclusive of pygopodids also have enlarged ventral foramina (personal observation). Procoelous Gekkota vertebrae (mostly restricted to pygopodids) tend to have smaller condyles and cotyles, the cotylar hollow often not visible in ventral view because the dorsoventral inclination is minimal [25].

A proximal right femur, SAM P57540 (figure 9*f*–*h*), was recovered from the Fish Lens at BC2. Broken immediately beneath the intertrochanteric fossa, shaft width and length are unknown. All other proximal features are intact. The specimen is most likely from an adult as the epiphyses are fully ossified and fused to the diaphysis. The articular surface of the femoral condyle is semi-circular viewed anteriorly, and oval in proximal profile. The femoral head extends medially away from the shaft axis. The internal trochanter is much shorter, only rising slightly above the edge of the intertrochanteric fossa. The intertrochanteric fossa is a concave surface between the condyle and trochanter on the ventral face of the proximal femur. The dorsal side also preserves a concave surface, but this is steeper-sided with a prominent 'v' marking where the femoral shaft begins. The femoral condyle is widest in anterior view, measuring 2.2 mm at the suture with the epiphysis. The internal trochanter is 1.3 mm tall from the base of the intertrochanteric fossa. Total length preserved is 3.8 mm.

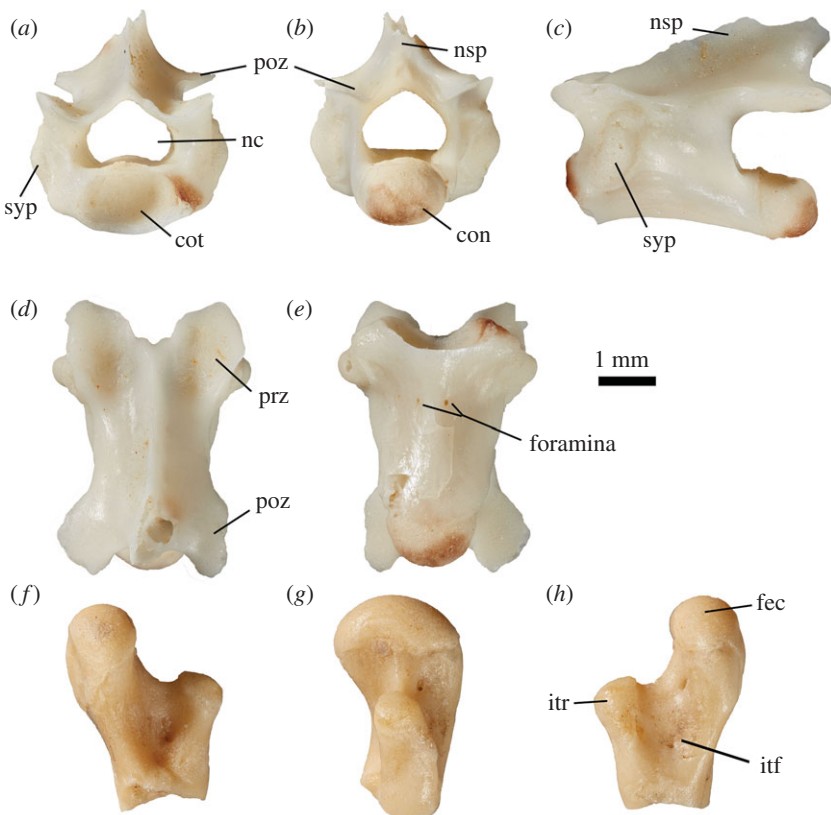

**Figure 9.** A single vertebra (SAM P57514) potentially referable to *Proegernia mikebulli* sp. nov., recovered from Fish Lens, BC2. (*a*) anterior view; (*b*) posterior view; (*c*) lateral view, anterior to the left; (*d*) dorsal view; and (*e*) ventral view. A proximal fragment of a right femur potentially referable to *Proegernia mikebulli* sp. nov. SAM P57540. (*f*) Dorsal; (*g*) anterior; and (*h*) ventral views.

The scincid apomorphy of the trochanter major gradually tapering in height distally (see Lee *et al.* [58]) is not visible on this specimen, as the shaft is broken immediately proximal to it.

## 5.3. Comparisons

**Comparison of new taxon with *Proegernia palankarinnensis*—**The type species of *Proegernia* (by monotypy) is *P. palankarinnensis*. While the type specimen is currently missing, the existing description [16] suggested *Proegernia* as being representative of a transitional form between a plesiomorphic scincid dentary morphology (e.g. as exemplified by the mabuyine *Eutropis*) and the derived egerniine condition. The plesiomorphic traits shared by both *P. palankarinnensis* and *P. mikebulli* are the small (about 1 mm long) opening of Meckel's groove immediately posterior of the symphysial foramen and the more anterior extent of the splenial notch into the anterior half of the tooth row. Both species of *Proegernia* also have egerniine features emerging in the prominent 'shoulders' of the *culmen lateralis* on their unicuspid tooth crowns, and a relatively deep splenial notch, extending anteriorly further than other Australasian scincids. *Proegernia palankarinnensis* was described as having 22 tooth loci, with a possible maximum of 23 teeth, uncertain due to a broken symphysial region. The specimens of *Proegernia mikebulli* preserve up to 27 dentary tooth positions (see SAM P57502, figure 4). Both taxa share an anterior section of crowded, smaller teeth in their lower dentition. The spacing of teeth in *P. palankarinnensis* is slightly wider than *P. mikebulli* at the posterior end of the tooth row. Tooth shape varies between the two taxa: *P. palankarinnensis* has tooth shafts narrowing at the base, whereas tooth shafts in *P. mikebulli* narrow beneath the crown and expand again at the base. Martin *et al.* [16] noted that the tooth crowns of *Proegernia* SAMA P39204 preserve distinct striations on both the lateral and medial faces. Tooth crowns for *Proegernia mikebulli* have weak striae on the medial face of the crown beneath the *crista lingualis*, but the lateral tooth face is smooth.

Although occurring almost concurrently, comparisons of the shape of the type dentaries of *P. palankarinnensis* and *P. mikebulli* (figure 10) show that the two species differ notably in the

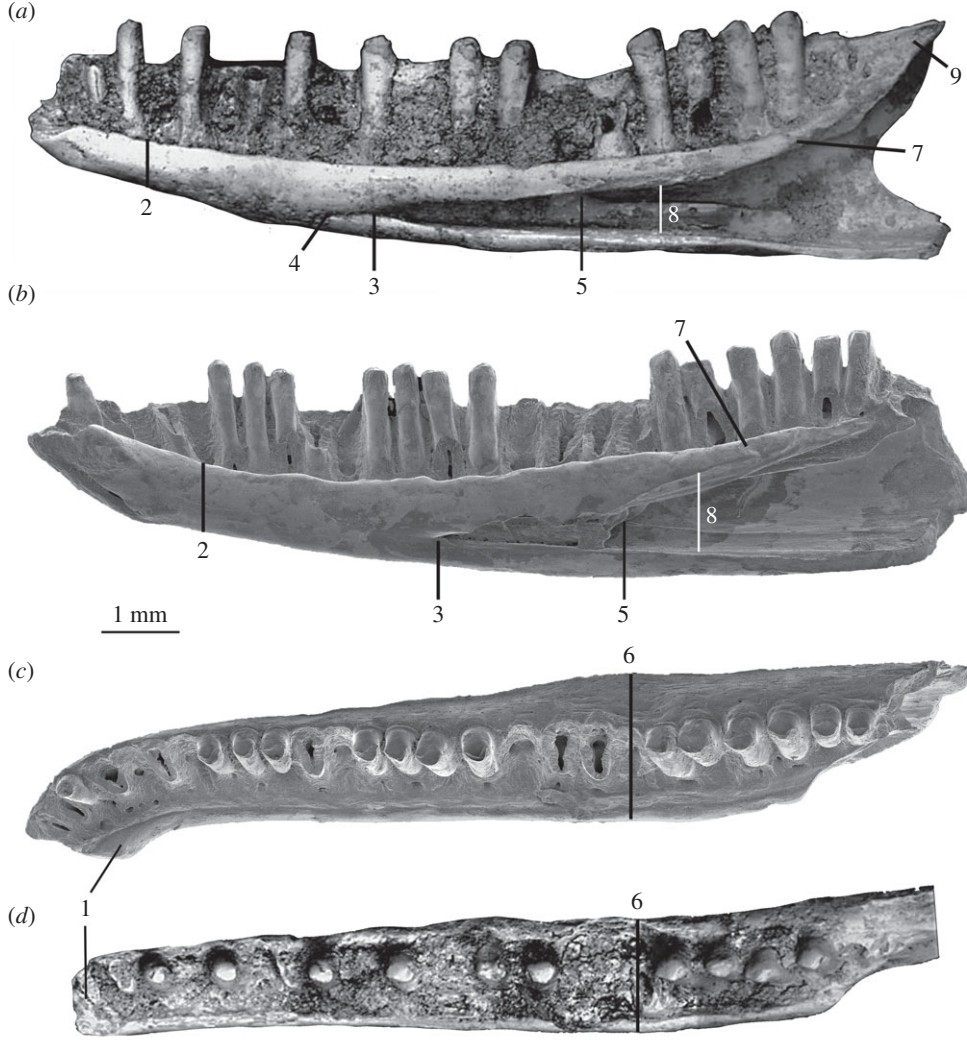

**Figure 10.** Comparative SEM of *Proegernia mikebulli* sp. nov. (*b,c*) and photomicrographs of *P. palankarinnensis* (*a,d*) from Martin *et al.* [16], in medial (above) and occlusal (below) views. Note the anterior extension of the splenial notch (3) in *P. palankarinnensis* and the increased the number of teeth/loci in *P. mikebulli*. Features described in text are labelled: 1 medial inflection of the symphysis; 2 dentary depth; 3 position of the inferior alveolar foramen; 4 Meckel's groove extension anterior of the inferior alveolar foramen, present in *P. palankarinnensis* and absent in *P. mikebulli*; 5 anterior extent of the internal septum similar in both taxa; 6 overall robustness as dentary width; 7 facet for articulation with anterior ramus of coronoid; 8 splenial notch height and 9 height of the coronoid process relative to the last tooth.

following characteristics: (1) the medial inflection of the symphysis is considerably less in *P. palankarinnensis* than in *P. mikebulli*, even after taking into account the effects of erosion. (2) The inferior alveolar foramen is positioned more anteriorly in *P. palankarinnensis*, below the 12th tooth position; that in *P. mikebulli* is level with the posterior edge of the 14th tooth. (3) *Proegernia mikebulli* lacks a short extension of Meckel's groove anterior of the inferior alveolar foramen. An anterior extension was noted [16] on the holotype of *P. palankarinnensis* (figure 10*a*) and interpreted as a plesiomorphic character. (4) The overall shape of the dentary of *P. palankarinnensis* is less robust, it being narrower between the medial edge of the dental sulcus and lateral face at the height of the mental foramina. (5) The facet for articulation of the coronoid terminates beneath the second-last tooth in *P. palankarinnensis* and the third to last tooth locus in *P. mikebulli*. (6) The dorsal edge of the splenial notch is widened more abruptly in *P. mikebulli* with a curve; in *P. palankarinnensis* the same space is sharply v-shaped. (7) The coronoid process preserved on *P. palankarinnensis* does not extend above the height of the final tooth crown, unlike the process on SAM P57499 of *P. mikebulli*.

## 5.4. Other egerniines and other scincids

We compared *P. mikebulli* with extant egerniines including *Lissolepis coventryi* (SAMA R57317) and *Liopholis multiscutata* (FUR168), as well as a Southeast Asian mabuyine outgroup representative *Eutropis multifasciata* (SAMA R35693), and three plesiomorphic 'scincine' scincids *Eumeces schneideri* (SAMA R6695), *Brachymeles schadenbergi* (SAMA R8853) and *Plestiodon fasciatus* (SAMA R66784), to determine how similar or derived the fossil taxon is from the generalized scincine (=inferred plesiomorphic) condition. Four characters notably vary among the chosen representative egerniines, the outgroup mabuyine and the extinct *P. mikebulli*: the anterior extent of the splenial notch and the inferior alveolar foramen; the shape and robustness of the symphysis, dentary depth and overall width; and the number of teeth in the tooth row. The anterior extent of the splenial notch varies within the egerniine radiation; both species of *Lissolepis* share a further anterior-reaching inferior alveolar foramen than any other egerniine genus. The outgroup scincid condition, represented by the examined taxa *Eumeces schneideri* (SAMA R6695), *Brachymeles schadenbergi* (SAMA R8853), *Plestiodon fasciatus* (SAMA R66784) and *Eutropis multifasciata* (SAMA R35693) have a splenial notch that extends further than 60% anteriorly along the length of the tooth row, or one that stretches anteriorly into an elongate open Meckel's groove. The anterior extension of the splenial notch is a plesiomorphic state for this character, also shared by both *Proegernia* taxa.

The mandibular symphysial joint in scincids varies between subfamilies in its overall size and robustness. Within egerniines, the variation is in the anteroposterior length of its posteroventral branch, and the depth of the upper branch. In egerniine species that have a reinforced symphysial joint and manipulate harder food, the lower branch of the symphysis is extended posteriorly and sometimes ventromedially (especially in *Tiliqua*, *Cyclodomorphus* and larger species of *Egernia*), to form a 'chin'. The dorsal branch of the symphysis deepens in *Liopholis* and in larger, omnivorous and herbivorous species of *Egernia*. This may be related to functional morphology to handle stresses on the chin during feeding, rather than a phylogenetic signal as these taxa do not form a clade. *Proegernia mikebulli* has a short and narrow posteroventral extension of the symphysis, possibly due to the presence of a foramen in the caudal notch between the limbs of the symphysis. This foramen exposes a short section of Meckel's cartilage in sphenomorphines and scincines. In egerniines, this foramen is usually absent. Both *P. palankarinnensis* and *P. mikebulli* have dentaries where a slightly elongate foramen would expose Meckel's cartilage, but in neither does this opening extend beyond the posterior edge of the symphysis.

Increased dentary depth and width, in relation to overall length, increases the robustness of the mandible. Variation in these measurements occur between genera of egerniines; species within *Liopholis* often have shorter, deeper skulls and corresponding dentaries, than similar-sized lizards within the genus *Egernia* (see the electronic supplementary material from [15]). *Lissolepis* has a more gracile dentary (figure 11). Outside of the egerniines, *Eutropis* and other Southeast Asian scincids with an unmodified insectivorous diet are even more gracile. *Eutropis multifasciata* has a longer dentary tooth row relative to snout–vent length (SVL) than all egerniines. Derived morphologies related to dietary adaptations as seen in the herbivorous and durophagous members of the Egerniinae (i.e. *Corucia zebrata*, *Tiliqua* and *Cyclodomorphus*) result in more robust dentary dimensions and/or modified dental morphology. *Proegernia palankarinnensis* and *P. mikebulli* present dentary depths most similar to those of *Lissolepis* spp., between the ancestral shallow insectivorous *Eutropis*, and the deeper dentary typical of *Liopholis* spp. Deepening of the dentary bone and shortening of the snout noted in numerous species within the genus *Liopholis* is possibly related to their affinity for digging burrows [59].

## 5.5. Dentition

The teeth of *Eutropis multifasciata* (SAMA R35693) and *Proegernia mikebulli* show increasing derivation from the plesiomorphic insectivorous skink tooth type as described by Richter [22] and Kosma [23] and represented by *Plestiodon fasciatus* (SAMA R66784; figure 12). Although many species within the extant radiations of eugongyline, sphenomorphine and egerniine scincids have modified tooth shapes for dietary specializations, taxa exhibiting the most plesiomorphic tooth crown shapes were chosen for comparisons with fossil taxa.

*Proegernia mikebulli* has departed from the crown shape of basal skinks (see [22,23]) by having the cristae lingualis et mesialis directing medially, creating a sharp angle with the *culmina laterales*, and less-prominent striae mark the medial face of the tooth. *Eutropis* retain the prominent striae running dorsoventrally on the medial face of the tooth, these striae are almost all equally prominent, while *P. mikebulli* appears to have slightly stronger striae immediately adjacent and parallel to the *culmina laterales*. The tooth crowns of *Eugongylus rufescens* also demonstrate prominent *culmina laterales* and weakened striae, although the tooth

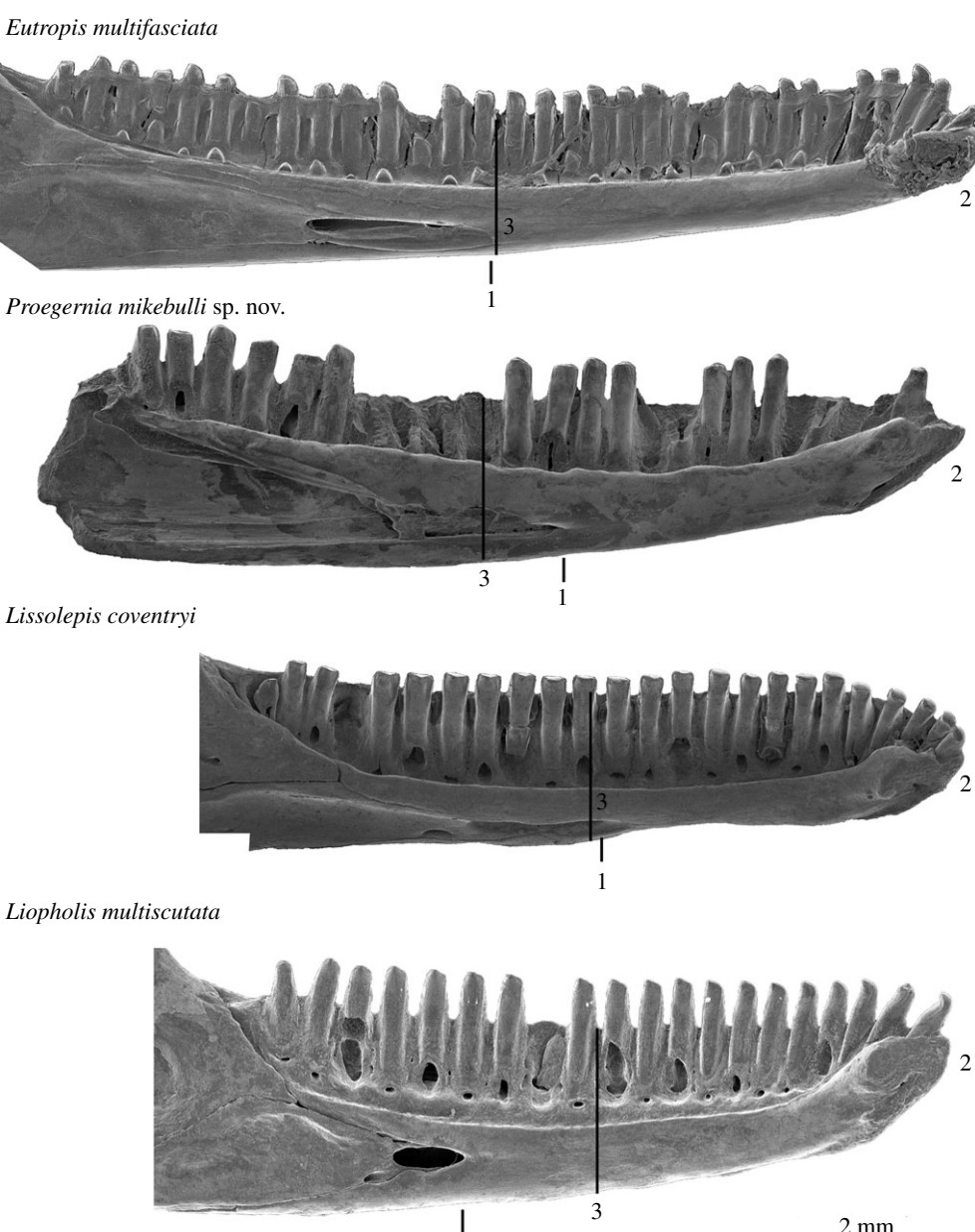

**Figure 11.** Comparative medial views of the dentaries of *Eutropis multifasciata* (SAMA R35693), *Proegernia mikebulli* sp. nov., *Lissolepis coventryi* (SAMA R57317) and *Liopholis multiscutata* (FUR168). 1 Anterior extent of splenial; 2 symphysis shape and robustness; and 3 dentary depth at mid-tooth row length.

shape differs; the shaft widening beneath the crown slightly and wear patterns make obvious the less medially directed cristae. The basal sphenomorphine condition is that of a narrow tooth shaft and crown, with sharply angled cristae creating a more pointed tooth profile. The medial face of the sphenomorphine tooth is anteroposteriorly convex and marked with prominent striae that do not approach the height of the lingual cristae but sit a third of the depth of the crown lower. The striae on *P. mikebulli* extend from almost directly in contact with the lingual cristae to the ventral tips of the *culmina laterales*.

The tooth crown in *P. mikebulli* is similar to that of the extant *Lissolepis coventryi* sharing medially directed cristae, sharp 'shoulders' to the *culmen laterales*, weak medial striae and absent lateral striae, and an expanded crown width on a narrower tooth shaft.

## 6. Phylogenetic relationships

Parsimony and Bayesian analyses retrieved broadly similar trees, placing both species of *Proegernia* outside of, but close to, living (crown) Australian egerniines.

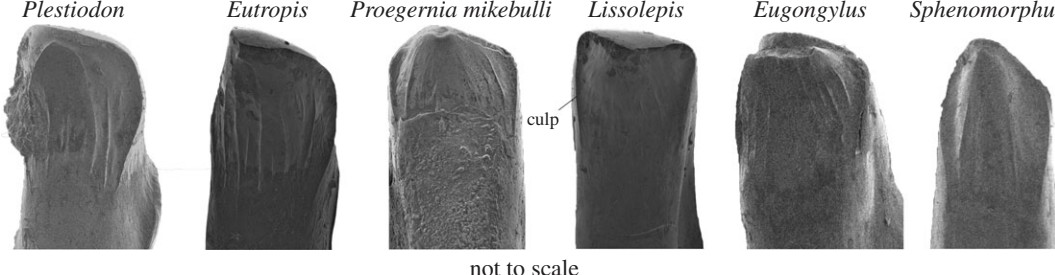

*Plestiodon*　　*Eutropis*　　*Proegernia mikebulli*　　*Lissolepis*　　*Eugongylus*　　*Sphenomorphus*

culp

not to scale

**Figure 12.** Medial view of left dentary, mid-tooth row, tooth crowns of *Plestiodon fasciatus* (SAMA R66784), *Eutropis multifasciata* (SAMA R35693), *Proegernia mikebulli* sp. nov. (SAM P57502), *Lissolepis coventryi* (SAMA R57317), *Eugongylus rufescens* (SAMA R36735) and *Sphenomorphus jobiensis* (SAMA R6736). *Plestiodon* represents the basal skink tooth condition as described by Richter [22]. *Culmen lateralis* posterior is labelled culp.

royalsocietypublishing.org/journal/rsos　　*R. Soc. Open Sci.* **8**: 201686

Parsimony analysis of all morphological and molecular data recovered one MPT (figure 13) with a best score of 4726.25, found 60 times out of 1000 replicate searches. Confidence on most of the nodes indicated by the bootstrap analyses is low; only support for the clades *Bellatorias*, *Liopholis* and *Tiliqua* + *Cyclodomorphus* are above 50. This is almost certainly due to an unstable position of the fossils, which are missing all DNA and most morphological data (111 missing characters for *P. palankarinnensis* and 73 for *P. mikebulli*).

*Proegernia mikebulli* is recovered basal to the extant Australian Egerniinae (spanning *Lissolepis* to *Tiliqua*), and *P. palankarinnensis* is retrieved as sister to the Solomon Islands' *Corucia zebrata*, but both with bootstrap support less than 50%. These relationships are conservatively interpreted as an effective polytomy between crown egerniines, *Corucia*, *Proegernia* and *Lissolepis*. Thus, rather than erect a new genus, we provisionally assign the new species to *Proegernia* (due to geographical and stratigraphic links with the type of that genus, and the Bayesian results below).

The position of both taxa outside the crown Australian egerniines is due to plesiomorphic scincid character states such as a partially open Meckel's groove (char. 11) in both fossil taxa. This feature also separates them from the outgroup taxon *Eutropis multifasciata* which has an entirely open Meckel's groove. The anterior extent of the splenial (char. 21), and the extension of Meckel's groove anterior of the splenial notch are features supporting the separation between *Lissolepis* and *Proegernia*; *Lissolepis* with a splenial reaching approximately 50% of the tooth row length anteriorly, and *Proegernia* reaching beyond to two-thirds the tooth row length.

The crown Australian Egerniinae spanning *Lissolepis* to *Tiliqua* is diagnosed by a completely closed Meckel's groove (char. 11), an absence of pterygoid teeth (char. 131), and a pterygoid quadrate ramus that is arcuate in cross section (char. 134). *Proegernia* (and *Corucia*) have plesiomorphic or alternative states for these characters, resulting in their position outside this clade. Potential morphological synapomorphies of the basal living Australian genus *Lissolepis* that are also present in *Proegernia* are the termination of the dentary coronoid process (char. 17), the orientation of the retroarticular process in dorsal view (char. 46), the plesiomorphic tooth crown shape (char. 55), minimal dental cementum (char. 60), the divot in the ventral surface of the pterygoid body (char 135). These shared characters conflict with the position of *Proegernia* outside the Australian crown group, resulted in the low support for the relationships between *Corucia*, *Proegernia* and *Lissolepis*.

The Bayesian analysis of all morphological and molecular data produced the topology shown in figure 14: the consensus of four separate runs of 100 000 000 generations, when combined in LogCombiner, then summarized with TreeAnnotator with a burnin of 20%.

The extant egerniine radiation forms a well-supported clade. Within this, *Tribolonotus* is strongly supported as the sister to remaining taxa. Of those, *Corucia* is strongly supported as the sister of the remaining living (Australian) egerniines. Both late Oligocene fossil species of *Proegernia* are weakly supported as the sister to *Corucia* with a posterior probability (PP) of 0.25. All Australian extant egerniines again form a clade that excludes both *Proegernia*. The extant Australian clade is a weakly supported (PP = 0.51) clade and retrieved as originating in the early Oligocene (figure 14).

*Proegernia palankarinnensis* and *P. mikebulli* are retrieved as sister taxa, but with little support (PP = 0.44), not unexpected given extensive missing data. Character changes affirming the sister relationship include the position of the mental foramina 50% along the length of the tooth row (char. 13), and the presence of fewer than 20 tooth crown striae (char. 61). Both of these features are also shared with

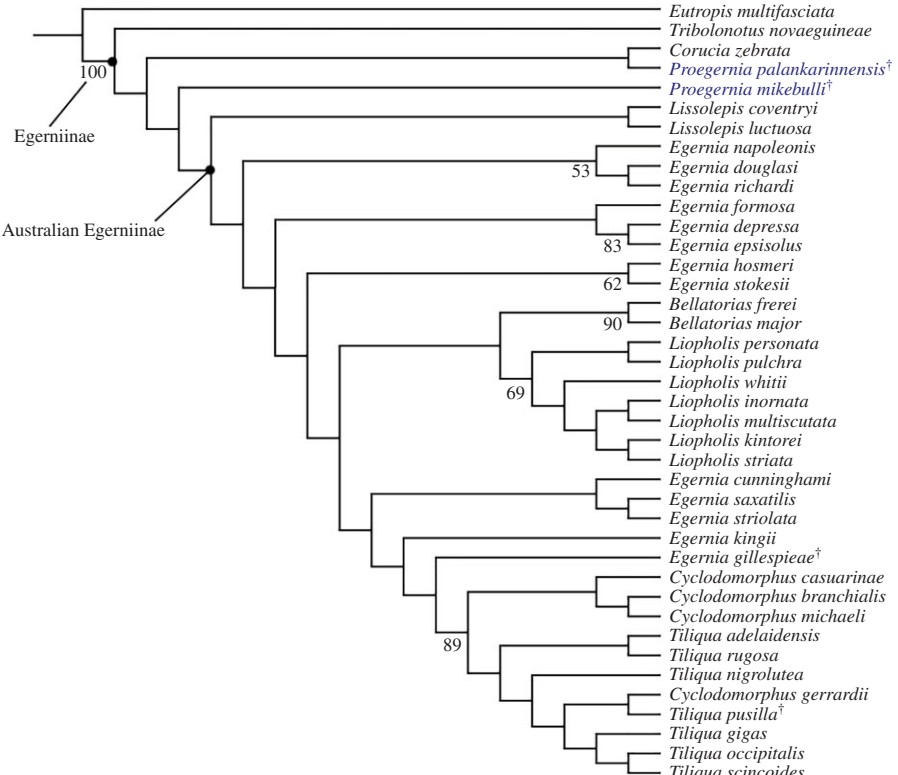

**Figure 13.** Egerniine phylogeny based on the single most parsimonious tree produced by TNT [26]. Bootstrap values greater than 50 shown. † denotes extinct taxon.

*Lissolepis*, but not with their putative sister clade *Corucia zebrata*. The similar positioning of *Corucia* and *Proegernia* (i.e. basal to crown Australian egerniines) is supported by the same characters as those noted after the parsimony analysis.

The node age for all Egerniinae is retrieved as 50.12 Ma (95% HPD 40.67–58.14), with *Tribolonotus* as the extant early-branching taxon representing the arrival of the subfamily to northern Australasia (Australia, New Guinea and surrounding islands). The last common ancestor of *Corucia* and crown Australian egerniines lived 40.47 Ma (95% HPD 31.92–49.64).

# 7. Discussion

Recent exploration of the Namba Formation outcropping at Lake Pinpa and Billeroo Creek has unearthed a new egerniine scincid lizard, *Proegernia mikebulli* sp. nov. Previous excavations and collection of surface material at these localities had not recovered any squamate elements. This identification has increased the taxonomic diversity of the Pinpa Local Fauna in the Namba Formation to include at least one skink, with further material yet to be described attributed to the Gekkota. These discoveries are attributed to the use of fine-mesh aperture sieves and bulk screening of concentrated lenses of fossiliferous sediments.

## 7.1. Namba environment

The excavation and subsequent analyses of the sediment profile at both Billeroo Creek (BC2) and Lake Pinpa (LP12) has allowed an interpretation of palaeoenvironmental conditions in the Frome Basin during the Oligo-Miocene. Dolomite in freshwater conditions requires a flooded alkaline environment to precipitate, as does calcite [60], so the presence of these minerals in Layers 4–5 (absent in Layers 2 and 3) are most likely the result of the seasonal fluctuations of the water level in the palaeolakes in which the sediments comprising the Namba Formation were deposited during the late Oligocene– early Miocene (see [44]). Fluctuations in the rainfall and evaporation contributed to the formation of palygorskite in the lower Layers 3–5. Smectite is the result of the weathering of minerals sourced from older rock outcrop (most likely from the nearby Flinders, Barrier and Olary Ranges) deposited into the Namba area as swampy soils on the lake edge. Layers where dolomite is absent, i.e. Layer 3

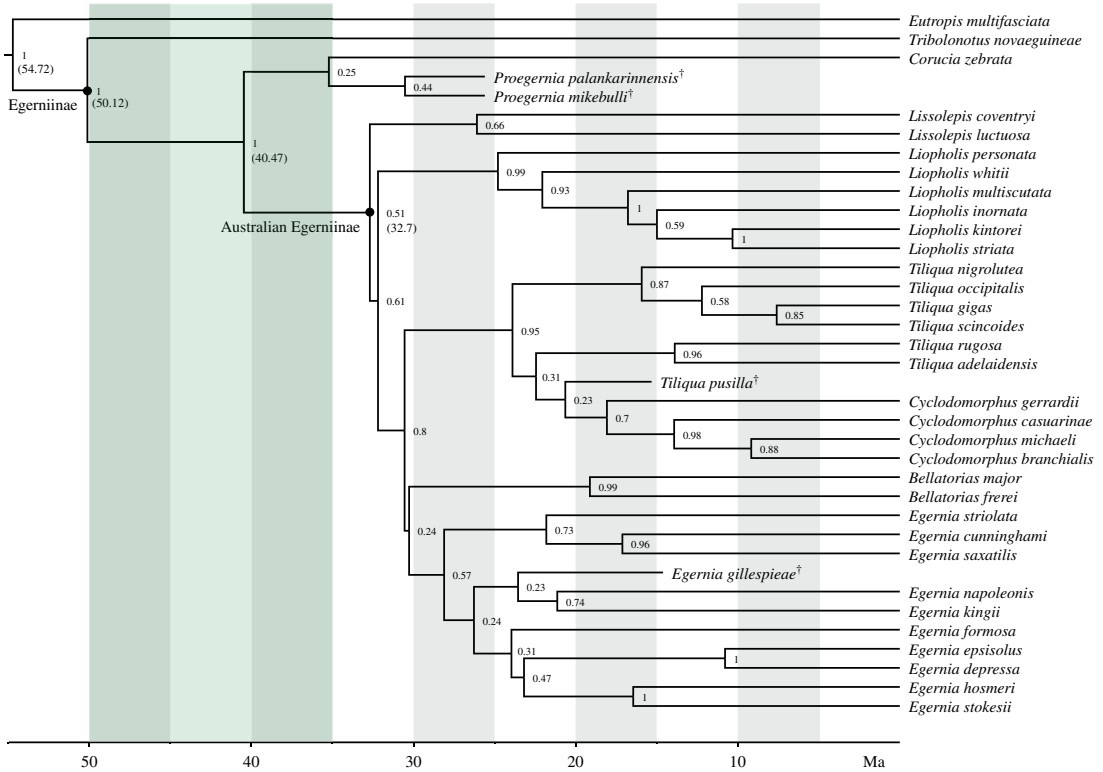

**Figure 14.** Egerniine phylogeny based on consensus total evidence Bayesian inference tree from BEAST. †denotes extinct taxon. Small numbers at nodes denote posterior probability. Reconstructed node ages for key clades are in brackets, the green shading denotes the Eocene, after which Australia became isolated from Gondwana [4].

according to the XRD result, reflect a drier phase in the palaeoenvironment without an alkaline waterbody at the surface.

Callen [47] noted that the Namba basin was once flooded, forming a palaeolake larger than the present Lake Frome. The palaeolake Namba supported a number of aquatic vertebrates including lungfish, *Obdurodon* platypus, crocodilians, cetaceans and various waterbirds [50,61–63]; their combined presence is indicative of a deeper aquatic environment. The lake was subject to varying rainfall and evaporation rates, resulting in receding lake edges, creating cycles of dolomite precipitation (recorded in Layer 5). Callen [47] concluded that sufficient evaporation of surface water (conditions required for dolomite precipitation) could not occur in a consistently high-rainfall climate. Instead, a modern analogue might be where cool moist winters and hot dry summers result in a moderate seasonal rainfall and high evaporation rates that result in dolomite and calcite deposits such as in the Coorong lacustrine-estuarine system in southern South Australia [64]. There, lake levels rise after dry summers due to groundwater recharge, resulting in the dolomite precipitation.

The sedimentary evidence from palaeolake Namba reveals periods of evaporation and alkaline water bodies. This suggests the occurrence of periodic droughts during which fish die-offs might be expected in the evaporating water bodies and terrestrial fauna might die and accumulate along the lake-edge. The articulated skeletons found at LP12 and BC2 alongside concentrations of fish bone may be the result of these lake deaths. The remains of *Proegernia mikebulli* and other small terrestrial vertebrates are preserved in lenses of concentrated bones of aquatic vertebrates (fish, turtle and crocodiles) within deposits reflecting these ephemeral lacustrine conditions. These lenses, as exemplified by the Fish Lens at BC2, and others exposed discontinuously at Lake Pinpa, were probably formed by littoral longshore currents disarticulating, mixing and concentrating bones. The material would not have to have been transported very far in order to mix animals from the two starkly different environments as the deposition site was in the littoral zone. Accumulation of small vertebrate material alongside associated larger vertebrate skeletons (at LP12) indicate that the transportation of material was not from a flash flood typical of tropical monsoon seasonality but rather accumulated along a shallow or ephemeral lake edge. Some small terrestrial vertebrate material is within a size range attributable to raptor predation [65,66] and some specimens show corrosion thinning typical of that formed after

raptor ingestion. Such material may be derived from raptor pellet accumulations washing into the lake where they disaggregated and mixed with the bones of the more common freshwater taxa. Undescribed accipitrids are known from the Pinpa Local Fauna and a skeleton of one was excavated from LP12 by Worthy/Camens-led expeditions, and is the subject of current research.

## 7.2. *Proegernia mikebulli* and crown group egerniines

*Proegernia mikebulli* was small and gracile compared to most living egerniines. Based on dentary length, an adult individual would be of a similar size to *Lissolepis coventryi*, around 100 mm SVL [67]. Examinations of tooth crown features across the type specimens reveal they share features with both *Lissolepis* and species of *Liopholis* (cf. *L. multiscutata* and *L. whitii* both of which have a cusp slightly posterior to the centre of the tooth crown). Tooth crown features of scincids potentially preserve phylogenetic and dietary signals. Detailed studies within and between the extant subfamilies may help elucidate any dietary preferences suggested by fossil dentition. Tooth striae are present on teeth in most egerniines with a specialized diet and no pattern in the number of striae between insectivorous, durophagous, or herbivorous species was obvious (see [23]). Within genera, the number and patterning of striae was fairly similar. *Cyclodomorphus* and *Tiliqua* increase the number of striae from other egerniines, partly due to the change in tooth shape leading to striae radiating on all sides from a central cusp. *Lissolepis* and *Liopholis* have fewer striae and all are restricted to the lingual face of the tooth crown, below the lingual cristae. The two fossil species of *Proegernia* have few striae, similar to *Lissolepis* and *Liopholis*. The buccal striae on the crowns of *P. palankarinnensis* are unusual and were not observed on extant *Eutropis multiscutata*, *Sphenomorphus jobiensis* or *Eugongylus rufescens*. Examination of fine-scale tooth crown features may prove useful characters for morphological phylogenetic analyses of fragmentary or incomplete tooth-bearing scincid material in future.

From what can be deduced of the palaeoenvironment surrounding Lake Pinpa and Billeroo Creek in the late Oligocene–early Miocene, *Proegernia mikebulli* lived in a cool, moist climate region with seasonal hot, dry summers with high evaporation rates. Other squamates known from this region and time period include a small gekkotan and possibly a constricting snake [50]. Although no evidence of *P. palankarinnensis* was found in the Namba Formation sites, the presence of at least two Australian scincids, from neighbouring basins, covering similar time periods and environments may be an indication of the early diversity of Australian squamates. Extant Australian scincids are difficult to separate based on isolated cranial material. The same cryptic diversity may have been present in the Oligo-Miocene, so more than two species are likely to be present in both the Etadunna and Namba Formations.

Although neither analysis produced firm support for the phylogenetic position of *Proegernia mikebulli*, both parsimony and Bayesian analyses placed the new taxon outside of living (crown) Australian Egerniinae. *Proegernia* species have a larger tooth count than crown Australian egerniines, a splenial notch placed more anteriorly, and approximately 1 mm long opening of Meckel's groove anterior to the splenial notch. Each of these character states are representative of the predicted transitional form between the plesiomorphic outgroup morphology and the majority of crown Australian egerniines. These characters have pulled *Proegernia* outside of the Australian crown group, but not beyond the more basal New Guinea genera *Corucia* and *Tribolonotus*. However, their precise position outside this crown group is uncertain. If relationships of one or both *Proegernia* with *Corucia* are correct (figures 13 and 14), then there were two migrations of Egerniinae into Australasia 40.47–32.7 Ma, or a single invasion followed by emigration of *Corucia* to the Solomon Islands. However, a large amount of missing data for these fossil taxa and low support for precise relationships means that it is possible that these trees are wrong, and that homoplasy with *Corucia* is what pulls these taxa away from the crown Australian group. It is then possible that *Proegernia* lies on the immediate stem to the Australian crown group, in which case only a single dispersal event to Australia need be assumed.

The tip-dated phylogeny inclusive of the two Oligo-Miocene *Proegernia* taxa retrieves the age of the Egerniinae as 50.12 Ma. The last common ancestor of Egerniinae and *Eutropis* (*Mabuyinae*) was most likely living in the far southeastern edge of Asia during the early Eocene. The first egerniine evolved elsewhere when Australia was far south of its current latitude [68]; there is no fossil evidence of their presence in Australia at this time. Crown Egerniinae most likely originated either in southeastern Asia or on an island arc, before connecting with the Australian landmass in the late Eocene (32.7 Ma). Similar origins have been hypothesized for agamids (22 Ma; [69]), pythons (*ca* 35 Ma; [8]), sphenomorphine skinks (*ca* 25 Ma; [12]) and elapids (*ca* 10 Ma; [8]).

# 8. Conclusion

Dated phylogenies with new fossils suggest that egerniines are the oldest radiation of skinks in Australia. The diversification of the crown Australian egerniines (*ca* 33 Ma) occurred nearly 10 million years before the estimated diversification of other Australian skinks, i.e. the Sphenomorphinae (*ca* 25 Ma) and Eugongylinae (*ca* 20 Ma; [3]). However, their diversity lags behind these other, potentially more recent arrivals. The first egerniines would have shared the continent with pygopodid, diplodactylid and carphodactylid geckos, and madtsoiid snakes [5]. The search for more Palaeogene lizard material to document the early evolution of Australia's diverse herpetofauna continues, with investigations of Eocene collections from Murgon in northern Queensland [70], and early Oligocene vertebrates from Pwerte Marnte Marnte [71].

Data accessibility.
LSIDurn:lsid:zoobank.org:pub:9C892991-E95A-4ABA-93B9-9480308A7E73.

All SAMA P specimens are registered in the Palaeontology Collection of the South Australian Museum, FUR specimen numbers refer to the Flinders University (Palaeontology) Reference collection, Micro-CT files are housed in the SAM Herpetology digital collection. The morphological character list has been uploaded as electronic supplementary information to this article. All executable files for phylogenetic analyses are available in Dryad Data Repository: doi:10.5061/dryad.3n5tb2rg7 [72].

Authors' contributions. K.M.T. conceived the project, conducted fieldwork, collected the data, performed analyses, produced all figures and wrote the manuscript. T.H.W. and A.B.C. funded the project, conducted fieldwork, contributed to discussions and commented on multiple drafts of the manuscript; M.N.H. and M.S.Y.L. commented on multiple drafts of the manuscript and contributed to discussions. N.B. collected the stratigraphic samples and information as part of a separate honours project.

Competing interests. We have no competing interests.

Funding. K.M.T. was supported by an Australian Postgraduate Research Training Stipend. The Mark Mitchell Foundation funded part of the fieldwork component of this project, and SEM time.

Acknowledgements. Sue Double and Jenny Worthy for screening and sorting the fossil material. Rod Wells, Colin Doudy, Bob and Sue Tulloch, Amy Tschirn, Warren Handley, Jacob Blokland, Carey Burke and Ellen Mather were all part of the collaborative field team. Dr Jason Gascooke for training on, and use of, the SEM for imaging and EDAX sedimentary analysis at Flinders Microscopy, an Australian Microscopy and Microanalysis Research Facility. We thank Sarah Harmer-Bassel and Jason Young of Flinders Analytical, for their ICPMS and X-ray diffraction (XRD) analyses of the sediment from the studied fossil sites. We thank Mary-Anne Binnie and Carolyn Kovach of the South Australian Museum, and Tim Ziegler from Museums Victoria, for access to collections, and loans of fossil and extant comparative material. The Willi Hennig Society for making TNT freely available. We especially thank Andrew Black (Blackie), manager, and Alec Wilson (prior owner) for access to Frome Downs Station that allowed this work. The authors thank Paul Oliver and two anonymous reviewers for their feedback on this manuscript.

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
