## [Peer Review File · Royal Society Open Science]

Review History

RSOS-201686.R0 (Original submission)

Review form: Reviewer 1 (Kristen T. Smith)

Is the manuscript scientifically sound in its present form?

Yes

Are the interpretations and conclusions justified by the results?

Yes

Is the language acceptable?

Yes

Do you have any ethical concerns with this paper?

No

Have you any concerns about statistical analyses in this paper?

No

Recommendation?

Accept with minor revision (please list in comments)

Comments to the Author(s)

The present paper is a careful work that combines paleontology and modern systematic approaches. It provides a great new perspective on the evolution of a charismatic clade of skinks native to Australia and the Solomon Islands. In some respects the results are surprising; for instance, with the new fossils Sahul clades have greater inferred ages, but the divergence of Sahul clades from *Tribolonotus* is younger in the present analysis (compared to Skinner, A., Hugall, A. F., & Hutchinson, M. N. (2011). Lygosomine phylogeny and the origins of Australian scincid lizards. *Journal of Biogeography*, 38(6), 1044-1058.).

My comments below are minor, or easily addressed.

One issue is crucial for reproducibility: the character list (missing, unless I'm much mistaken). The morphological data set is expanded from 95/40 to 102/48 discrete/continuous characters. The new discrete characters are not explained, and the new continuous characters can only vaguely be gleaned from the BEAST xlm input file. Please provide a character list similar to that in Thorn et al. (2019) on *Egernia* (unless I've overlooked something).

I think the title sells the paper short. To be sure, nothing in the title is not also found in the paper, but there's also a lot of apparently novel information about the environment that should be reflected in the title too. Maybe: "... and the early evolution and environment of Australian *egerniine* skinks" ? It might broaden the impact of the paper.

Some clarification of the XRF analyses might be useful. I had a geological training but don't remember encountering the term 'soluble minerals' as used here. A literature search suggests they are a principal concern of soil science. As I understand, the authors used XRF to measure heavier elemental concentrations (Na, Mg, Ca, K, Cl), which are somehow equated with "soluble salts". To the uninitiated (me), it seems like these are probably ions and occur in lots of minerals, some readily soluble, some not. But at the end it becomes clear that the authors are concerned especially with dolomite, the formation of which is related to paleoenvironment. Correct? Some additional clarification of goals in the methods (3a) would be helpful.

There are a number of cases where characters are stated or implied to be apomorphic at some level where this is inaccurate. For instance:

A fused Meckelian groove is more broadly distributed in Scincidae (*Lygosominae* sensu Pyron et al. 2013, including *Egerniinae* and *Eugongylyinae* of the present authors). Thus, the first sentence of the Diagnosis (!) is difficult to follow.

The "bifid posterior termination of the suborbital process (an apomorphy of Scincidae)" - Gauthier et al. (2012, char. 123/1) found this to be a synapomorphy of *Scincoidea*. See also below.

To justify referrals of other specimens to the new species, the authors list characters that are frequently not apomorphic. For examples, see below. It seems that geography and process of elimination has played a role in these identifications. In other words, we assume it's a member of an extant Australian clade; we rule out *Gekkota* (and *Varanus*), leaving only scincid, and the (usually plesiomorphic) characters are consistent with that. I think the authors' conclusions are likely valid, but this line of argumentation should be made more explicit.

The term "*Lygosominae*" has been used in the literature for a very different (and more inclusive) set of taxa. The authors appear to follow a taxonomy promulgated in an obscure publication (in

the journal “Herptile”) to which I do not have access. Perhaps a brief summary of that taxonomy and what the authors mean by various terms would be helpful for the reader.

I find myself confused by the term “splenial notch” (which is not defined or labelled). I think of a notch as a nick or short incision, like at the back end of an arrow. So when I read “splenial notch” I think of the V-shaped remnant of the Meckelian groove into which the splenial inserts. But really I think the authors mean the dorsal articulation facet of the splenial on the dentary (along the upper margin of the Meckelian groove). Please clarify.

In Section 8 the word “stem” is sometimes used for extant taxa that branched basal to a clade in question (like Australian Egerniinae). I think “stem” can only refer to extinct forms. Please re-write: for instance, “earlier-branching” or something similar (I know some people have strong opinions).

Part 3c(i). The scaling of continuous character weight (Thorn et al. 2019) was, in my opinion, a clever way to deal with the issue (moving beyond the equal-character / equal-state weighting dichotomy). So the mean or median number of discrete character states per character is 3 (i.e., 0-1-2)? The value should be stated.

Part 3c(ii). The molecular data come from other sources. The authors state that Partition Finder 2 was used, but that the same partitions were “used again” in this work. Did the authors re-run Partition Finder 2 on the data and come up with the same results? If so, maybe write “... were discovered again here.” Otherwise, maybe refer to the previous works for details of data partitioning.

Part 3c(iii). Please compare file names to supplementary files, which do not match.

Part 3c(iv). I don’t see the log files mentioned at the end. These would help follow synapomorphies.

First citations of Figure 1 and 2 reversed. However, the map first makes sense, so I’d consider switching the figures.

“GPS coordinates” - since coordinates are not presented on the map (figure 2), I think it ought to be stated which datum is being referred to (WGS 84 ?).

An angular bone (under description of Post-dentary / compound bone) is plesiomorphic, so its presence can only rule out Gekkota if we assume that there are no other groups present.

I would suggest “compound bone” (also used by the author) over “post-dentary”, as the latter could also be understood to include the angular.

The maxilla seems to show an unusual feature that is not mentioned (or shown in the reconstruction): the convexity of the palatal shelf just in front of the tooth position labeled “B” (fig. 5). It does not seem like this is artifactual. The ATOL scan of Egernia striolata suggest that there might be a similar convexity there; less clearly similar is the convexity in this position in certain Xantusiidae. Also, where is the superior alveolar foramen with respect to this structure? I do not see it.

See suggestion on dental terminology to clarify expression of tooth dimensions.

The pterygoid is a left one. Also, I'm not familiar with the term "pterygoid head" - Oelrich's (1956) (triangular) "anterior part" would be preferable.

Identification of vertebra. While I concur that an attribution to the scincid species is likely, it should be stated that none of the features listed are apomorphic at this level. The authors proceed by eliminating other extant Australian taxa to which the vertebra could belong. That's fine, but it should be stated explicitly.

Section 6c describes the "more anterior extent of the splenial notch" as being a plesiomorphic trait (shared with the type species) and an "egerniine" (presumably apomorphic) feature. Please clarify.

To play Devil's advocate: the striae claimed to be present labially in Martin et al.'s *Proegernia palankarinnensis* aren't clearly visible in their micrographs....

The labeling of figure sub-parts is inconsistent. Figure 4 uses -1, -2, etc., whereas Figure 10 uses A, B, etc. and Figure 11 uses nothing.

In Figure 12, it would be helpful to state where (approximately!) in the tooth row these teeth are from (e.g., middle of the tooth row).

Section 7 - abbreviation PP is for Posterior Probability

In addition to the trivial comments on language below, I have made some remarks in the annotated PDF (Appendix A).

Check abbreviations of names (e.g., T H Rich vs. T. H. Rich) for consistency throughout. Also, if the author's wish to use the abbreviation LF for Local Fauna, please introduce it early, explain it, and use it consistently.

The filler statement at the beginning of part 4c "Need an introductory sentence or two" should be deleted / replaced

Why are most mineral names and sometimes element names (Calcium) capitalized?

Generally: delete hyphen between adverb (-ly) and the adjective it modifies, e.g., "caudally-directed".

The word "ovular" (egg-shaped) is used where I think the authors mean "oval".

Check "ed.^eds." in references - EndNote doesn't seem to have done these properly. They might have to be manually corrected.

Review form: Reviewer 2

Is the manuscript scientifically sound in its present form?

Yes

Are the interpretations and conclusions justified by the results?

Yes

Is the language acceptable?

Yes

Do you have any ethical concerns with this paper?

No

Have you any concerns about statistical analyses in this paper?

No

Recommendation?

Accept with minor revision (please list in comments)

Comments to the Author(s)

I have reviewed the manuscript entitled "A new species of Proegernia from the Namba Formation in South Australia and the early evolution of Australian egeriine skinks" by Thorn and co-authors. I want to begin expressing that the paper is very well written and illustrated, and is a good example of a multidisciplinary approach that can result in a highly interesting results, mainly taking into account the fragmentary nature of the studied specimens. The provided geological context is important for potential future paleontological expeditions to the area, and sets a framework for interpreting the paleoecological context. Descriptions and comparisons are extensive, but not excessive. The phylogenetic analyses (using parsimony and Bayesian methods) are sound and provide interesting results. I agree with the identification of the fossils described, and with the erection of the new species. The discussion and conclusions are justified by the results, and represent an interesting starting point for (re-)interpreting the timing of events that led to the current composition of scincid lizard faunas in Australia. As I said above, images are good, although I provide in the annotated pdf a few comments regarding the resolution of a few of them (mainly figure 3, but also fig. 4.5 and fig. 12. I recommend publication of this manuscript after the authors have considered the minor revisions suggested in the annotated pdf (Appendix B). I just want to congratulate the authors for a well done work, and I look forward to see this published.

Review form: Reviewer 3 (Paul Oliver)

Is the manuscript scientifically sound in its present form?

Yes

Are the interpretations and conclusions justified by the results?

Yes

Is the language acceptable?

Yes

Do you have any ethical concerns with this paper?

No

Have you any concerns about statistical analyses in this paper?

Yes

Recommendation?

Accept with minor revision (please list in comments)

Comments to the Author(s)

First off I want to apologise to the editor and authors for my lengthy delay in reviewing this MS - last year was a special kind of hell!

Overall I think this is a really important contribution to the Australian herpetological literature. I have not made many comments on the morphology or paleontology as these are not areas that I know well. I have made a few comments and suggestions in on the pdf mainly around the biogeographic interpretations and also the framing.

In the abstract (and perhaps intro as well) I would suggest making the nature of Australia's Oligocene squamate fauna the clear central opening theme, and then finishing on this theme as well. This allows you to emphasise the broadly congruent emergent picture across molecules and paleo that most components of Australia's extent squamate biota were likely not present much before the Miocene, but with added growing evidence that the Egernia group may be an older component. This is also perhaps the inference that is most broadly interesting.

With respect to dates, I think the comparison of ages across papers should be a bit more guarded - there are lots of methodological reasons why these dates could vary - so certainly comparing the dates estimated here for Egernia group with dates for Sphenomorphines is an ok thing to do, but it needs to be caveated. In that context suggest emphasising the congruence - Skinner et al suggested that unlike other skinks and especially Sphenomorph group - the Egernia group has been in the Australia region for 50 million years, your analyses are complementing that by showing that they are also in the Australia for ~30 million years.

In comparisons with Adams 2011 paper (para II) - he dated the ages of crown radiations, this is different to arrival dates (although they may of course be linked)

I have had a little bit of a speil about Sahul in place - especially going back beyond the Miocene I think this term which describes a contemporary conglomerate, runs the risk of confounding rather than elucidating patterns - one alternative might be Australian Craton vs Melanesia island arcs and Terranes??

Re island arcs etc - latest analyses we did (Tallowin et al 2018), suggest agamids actually colonised arcs/proto-Papua from Australia, just to complicate matters further. But of course with all these ancestral state analyses, it only takes one or two extinction events to reorient the story.

Re ages of skinks see if you can track down a copy of Charles Linkem's PhD thesis (has been available online). This provides clear evidence that the Australian sphenos are recently derived from SEA groups, while the Melanesian sphenos (multiple lineages) can have much longer and deeper histories, comparable timing to Tribolonotus.

Have made a bunch of further comments in the pdf (Appendix C) of manuscript and picked up a few small typos etc. Overall good work and great to see it, will be good to see published.

Decision letter (RSOS-201686.R0)

The editorial office reopened on 4 January 2021. We are working hard to catch up after the festive break. If you need advice or an extension to a deadline, please do not hesitate to let us know -- we

will continue to be as flexible as possible to accommodate the changing COVID situation. We wish you a happy New Year, and hope 2021 proves to be a better year for everyone.

Dear Dr Thorn

On behalf of the Editors, we are pleased to inform you that your Manuscript RSOS-201686 "A new species of *Proegernia* from the Namba Formation in South Australia and the early evolution of Australian egeriine skinks" has been accepted for publication in Royal Society Open Science subject to minor revision in accordance with the referees' reports. Please find the referees' comments along with any feedback from the Editors below my signature.

Please submit your revised manuscript and required files (see below) no later than 7 days from today's (ie 07-Jan-2021) date. Note: the ScholarOne system will 'lock' if submission of the revision is attempted 7 or more days after the deadline. If you do not think you will be able to meet this deadline please contact the editorial office immediately.

on behalf of Prof Kevin Padian (Subject Editor)
openscience@royalsociety.org

Reviewer comments to Author:
Reviewer: 1

Comments to the Author(s)

The present paper is a careful work that combines paleontology and modern systematic approaches. It provides a great new perspective on the evolution of a charismatic clade of skinks native to Australia and the Solomon Islands. In some respects the results are surprising; for instance, with the new fossils Sahul clades have greater inferred ages, but the divergence of Sahul clades from *Tribolonotus* is younger in the present analysis (compared to Skinner, A., Hugall, A. F., & Hutchinson, M. N. (2011). Lygosomine phylogeny and the origins of Australian scincid lizards. *Journal of Biogeography*, 38(6), 1044-1058.).

My comments below are minor, or easily addressed.

One issue is crucial for reproducibility: the character list (missing, unless I'm much mistaken). The morphological data set is expanded from 95/40 to 102/48 discrete/continuous characters. The new discrete characters are not explained, and the new continuous characters can only vaguely be gleaned from the BEAST xlm input file. Please provide a character list similar to that in Thorn et al. (2019) on *Egernia* (unless I've overlooked something).

I think the title sells the paper short. To be sure, nothing in the title is not also found in the paper, but there's also a lot of apparently novel information about the environment that should be reflected in the title too. Maybe: "... and the early evolution and environment of Australian *egerniine* skinks" ? It might broaden the impact of the paper.

Some clarification of the XRF analyses might be useful. I had a geological training but don't remember encountering the term 'soluble minerals' as used here. A literature search suggests they are a principal concern of soil science. As I understand, the authors used XRF to measure heavier elemental concentrations (Na, Mg, Ca, K, Cl), which are somehow equated with "soluble salts". To the uninitiated (me), it seems like these are probably ions and occur in lots of minerals, some readily soluble, some not. But at the end it becomes clear that the authors are concerned especially with dolomite, the formation of which is related to paleoenvironment. Correct? Some additional clarification of goals in the methods (3a) would be helpful.

There are a number of cases where characters are stated or implied to be apomorphic at some level where this is inaccurate. For instance:

A fused Meckelian groove is more broadly distributed in Scincidae (*Lygosominae* sensu Pyron et al. 2013, including *Egerniinae* and *Eugongylinae* of the present authors). Thus, the first sentence of the Diagnosis (!) is difficult to follow.

The "bifid posterior termination of the suborbital process (an apomorphy of Scincidae)" - Gauthier et al. (2012, char. 123/1) found this to be a synapomorphy of *Scincoidea*. See also below.

To justify referrals of other specimens to the new species, the authors list characters that are frequently not apomorphic. For examples, see below. It seems that geography and process of elimination has played a role in these identifications. In other words, we assume it's a member of an extant Australian clade; we rule out *Gekkota* (and *Varanus*), leaving only scincid, and the (usually plesiomorphic) characters are consistent with that. I think the authors' conclusions are likely valid, but this line of argumentation should be made more explicit.

The term "*Lygosominae*" has been used in the literature for a very different (and more inclusive) set of taxa. The authors appear to follow a taxonomy promulgated in an obscure publication (in the journal "*Herpetile*") to which I do not have access. Perhaps a brief summary of that taxonomy and what the authors mean by various terms would be helpful for the reader.

I find myself confused by the term "splenial notch" (which is not defined or labelled). I think of a notch as a nick or short incision, like at the back end of an arrow. So when I read "splenial notch" I think of the V-shaped remnant of the Meckelian groove into which the splenial inserts. But really I think the authors mean the dorsal articulation facet of the splenial on the dentary (along the upper margin of the Meckelian groove). Please clarify.

In Section 8 the word "stem" is sometimes used for extant taxa that branched basal to a clade in question (like Australian *Egerniinae*). I think "stem" can only refer to extinct forms. Please re-write: for instance, "earlier-branching" or something similar (I know some people have strong opinions).

Part 3c(i). The scaling of continuous character weight (Thorn et al. 2019) was, in my opinion, a clever way to deal with the issue (moving beyond the equal-character / equal-state weighting dichotomy). So the mean or median number of discrete character states per character is 3 (i.e., 0-1-2)? The value should be stated.

Part 3c(ii). The molecular data come from other sources. The authors state that Partition Finder 2 was used, but that the same partitions were “used again” in this work. Did the authors re-run Partition Finder 2 on the data and come up with the same results? If so, maybe write “... were discovered again here.” Otherwise, maybe refer to the previous works for details of data partitioning.

Part 3c(iii). Please compare file names to supplementary files, which do not match.

Part 3c(iv). I don't see the log files mentioned at the end. These would help follow synapomorphies.

First citations of Figure 1 and 2 reversed. However, the map first makes sense, so I'd consider switching the figures.

“GPS coordinates” - since coordinates are not presented on the map (figure 2), I think it ought to be stated which datum is being referred to (WGS 84 ?).

An angular bone (under description of Post-dentary / compound bone) is plesiomorphic, so its presence can only rule out Gekkota if we assume that there are no other groups present.

I would suggest “compound bone” (also used by the author) over “post-dentary”, as the latter could also be understood to include the angular.

The maxilla seems to show an unusual feature that is not mentioned (or shown in the reconstruction): the convexity of the palatal shelf just in front of the tooth position labeled “B” (fig. 5). It does not seem like this is artifactual. The AToL scan of *Egernia striolata* suggest that there might be a similar convexity there; less clearly similar is the convexity in this position in certain Xantusiidae. Also, where is the superior alveolar foramen with respect to this structure? I do not see it.

See suggestion on dental terminology to clarify expression of tooth dimensions.

The pterygoid is a left one. Also, I'm not familiar with the term “pterygoid head” - Oelrich's (1956) (triangular) “anterior part” would be preferable.

Identification of vertebra. While I concur that an attribution to the scincid species is likely, it should be stated that none of the features listed are apomorphic at this level. The authors proceed by eliminating other extant Australian taxa to which the vertebra could belong. That's fine, but it should be stated explicitly.

Section 6c describes the “more anterior extent of the splenial notch” as being a plesiomorphic trait (shared with the type species) and an “egerniine” (presumably apomorphic) feature. Please clarify.

To play Devil's advocate: the striae claimed to be present labially in Martin et al.'s *Proegernia palankarinnensis* aren't clearly visible in their micrographs....

The labeling of figure sub-parts is inconsistent. Figure 4 uses -1, -2, etc., whereas Figure 10 uses A, B, etc. and Figure 11 uses nothing.

In Figure 12, it would be helpful to state where (approximately!) in the tooth row these teeth are from (e.g., middle of the tooth row).

Section 7 - abbreviation PP is for Posterior Probability

In addition to the trivial comments on language below, I have made some remarks in the annotated PDF.

Check abbreviations of names (e.g., T H Rich vs. T. H. Rich) for consistency throughout. Also, if the author's wish to use the abbreviation LF for Local Fauna, please introduce it early, explain it, and use it consistently.

The filler statement at the beginning of part 4c "Need an introductory sentence or two" should be deleted / replaced

Why are most mineral names and sometimes element names (Calcium) capitalized?

Generally: delete hyphen between adverb (-ly) and the adjective it modifies, e.g., "caudally-directed".

The word "ovular" (egg-shaped) is used where I think the authors mean "oval".

Check "ed.^eds." in references - EndNote doesn't seem to have done these properly. They might have to be manually corrected.

Reviewer: 2

Comments to the Author(s)

I have reviewed the manuscript entitled "A new species of Proegernia from the Namba Formation in South Australia and the early evolution of Australian egeriine skinks" by Thorn and co-authors. I want to begin expressing that the paper is very well written and illustrated, and is a good example of a multidisciplinary approach that can result in a highly interesting results, mainly taking into account the fragmentary nature of the studied specimens. The provided geological context is important for potential future paleontological expeditions to the area, and sets a framework for interpreting the paleoecological context. Descriptions and comparisons are extensive, but not excessive. The phylogenetic analyses (using parsimony and Bayesian methods) are sound and provide interesting results. I agree with the identification of the fossils described, and with the erection of the new species. The discussion and conclusions are justified by the results, and represent an interesting starting point for (re-)interpreting the timing of events that led to the current composition of scincid lizard faunas in Australia. As I said above, images are good, although I provide in the annotated pdf a few comments regarding the resolution of a few of them (mainly figure 3, but also fig. 4.5 and fig. 12. I recommend publication of this manuscript after the authors have considered the minor revisions suggested in the annotated pdf. I just want to congratulate the authors for a well done work, and I look forward to see this published.

Reviewer: 3

Comments to the Author(s)

First off I want to apologise to the editor and authors for my lengthy delay in reviewing this MS - last year was a special kind of hell!

Overall I think this is a really important contribution to the Australian herpetological literature. I have not made many comments on the morphology or paleontology as these are not areas that I know well. I have made a few comments and suggestions in on the pdf mainly around the biogeographic interpretations and also the framing.

In the abstract (and perhaps intro as well) I would suggest making the nature of Australia's Oligocene squamate fauna the clear central opening theme, and then finishing on this theme as well. This allows you to emphasise the broadly congruent emergent picture across molecules and paleo that most components of Australia's extent squamate biota were likely not present much before the Miocene, but with added growing evidence that the Egernia group may be an older component. This is also perhaps the inference that is most broadly interesting.

With respect to dates, I think the comparison of ages across papers should be a bit more guarded - there are lots of methodological reasons why these dates could vary - so certainly comparing the dates estimated here for Egernia group with dates for Sphenomorphines is an ok thing to do, but it needs to be caveated. In that context suggest emphasising the congruence - Skinner et al suggested that unlike other skinks and especially Sphenomorph group - the Egernia group has been in the Australia region for 50 million years, your analyses are complementing that by showing that they are also in the Australia for ~30 million years.

In comparisons with Adams 2011 paper (para II) - he dated the ages of crown radiations, this is different to arrival dates (although they may of course be linked)

I have had a little bit of a speil about Sahul in place - especially going back beyond the Miocene I think this term which describes a contemporary conglomerate, runs the risk of confounding rather than elucidating patterns - one alternative might be Australian Craton vs Melanesia island arcs and Terranes??

Re island arcs etc - latest analyses we did (Tallowin et al 2018), suggest agamids actually colonised arcs/proto-Papua from Australia, just to complicate matters further. But of course with all these ancestral state analyses, it only takes one or two extinction events to reorient the story.

Re ages of skinks see if you can track down a copy of Charles Linkem's PhD thesis (has been available online). This provides clear evidence that the Australian Sphenomorphs are recently derived from SEA groups, while the Melanesian Sphenomorphs (multiple lineages) can have much longer and deeper histories, comparable timing to Tribolonotus.

Have made a bunch of further comments in the pdf of manuscript and picked up a few small typos etc. Overall good work and great to see it, will be good to see published.

===PREPARING YOUR MANUSCRIPT===

===PREPARING YOUR REVISION IN SCHOLARONE===

- If you are requesting a discretionary waiver for the article processing charge, the waiver form must be included at this step.
- If you are providing image files for potential cover images, please upload these at this step, and inform the editorial office you have done so. You must hold the copyright to any image provided.
- A copy of your point-by-point response to referees and Editors. This will expedite the preparation of your proof.

- Ensure that your data access statement meets the requirements at <https://royalsociety.org/journals/authors/author-guidelines/#data>. You should ensure that you cite the dataset in your reference list. If you have deposited data etc in the Dryad repository, please only include the 'For publication' link at this stage. You should remove the 'For review' link.
- If you are requesting an article processing charge waiver, you must select the relevant waiver option (if requesting a discretionary waiver, the form should have been uploaded at Step 3 'File upload' above).
- If you have uploaded ESM files, please ensure you follow the guidance at <https://royalsociety.org/journals/authors/author-guidelines/#supplementary-material> to include a suitable title and informative caption. An example of appropriate titling and captioning may be found at https://figshare.com/articles/Table_S2_from_Is_there_a_trade-off_between_peak_performance_and_performance_breadth_across_temperatures_for_aerobic_scope_in_teleost_fishes_/3843624.

Author's Response to Decision Letter for (RSOS-201686.R0)

See Appendix D.

Decision letter (RSOS-201686.R1)

Dear Dr Thorn,

It is a pleasure to accept your manuscript entitled "A new species of *Proegernia* from the Namba Formation in South Australia and the early evolution and environment of Australian egeriine skinks" in its current form for publication in Royal Society Open Science.

You can expect to receive a proof of your article in the near future. Please contact the editorial office (openscience@royalsociety.org) and the production office (openscience_proofs@royalsociety.org) to let us know if you are likely to be away from e-mail

contact – if you are going to be away, please nominate a co-author (if available) to manage the proofing process, and ensure they are copied into your email to the journal.

on behalf of Prof Kevin Padian (Subject Editor)
openscience@royalsociety.org

Appendix A**ROYAL SOCIETY
OPEN SCIENCE****A new species of *Proegernia* from the Namba Formation in
South Australia and the early evolution of Australian
egerniine skinks**

Journal:	Royal Society Open Science
Manuscript ID	RSOS-201686
Article Type:	Research
Date Submitted by the Author:	20-Sep-2020
Complete List of Authors:	Thorn, Kailah; Flinders University College of Science and Engineering, ; South Australian Museum Hutchinson, Mark; South Australian Museum Lee, Michael; Flinders University, College of Science and Engineering; South Australian Museum Brown, Nathan; Flinders University, College of Science and Engineering Camens, Aaron; Flinders University, College of Science and Engineering Worthy, Trevor Henry; Flinders University, College of Science and Engineering
Subject:	Palaeontology < EARTH SCIENCES, evolution < BIOLOGY, taxonomy and systematics < BIOLOGY
Keywords:	Scincidae, Egerniinae, Oligocene, Miocene, Palaeontology, Namba
Subject Category:	Organismal and Evolutionary Biology

Author-supplied statements

Relevant information will appear here if provided.

Ethics

Does your article include research that required ethical approval or permits?:

This article does not present research with ethical considerations

Statement (if applicable):

CUST_IF_YES_ETHICS :No data available.

Data

It is a condition of publication that data, code and materials supporting your paper are made publicly available. Does your paper present new data?:

Yes

Statement (if applicable):

All specimens are registered in the Palaeontology Collection of the South Australian Museum, MicroCT files are housed in the SAM Herpetology digital collection.

All executable files for phylogenetic analyses are available as electronic supplementary material.

Conflict of interest

I/We declare we have no competing interests

Statement (if applicable):

CUST_STATE_CONFLICT :No data available.

Authors' contributions

This paper has multiple authors and our individual contributions were as below

Statement (if applicable):

KMT conceived the project, conducted fieldwork, collected the data, performed analyses, produced all figures and wrote the manuscript. THW and ABC funded the project, conducted field work, contributed to discussions, and commented on multiple drafts of the manuscript; MNH and MSYL commented on multiple drafts of the manuscript and contributed to discussions. NB collected the stratigraphic samples and information as part of a separate honours project.

Origins of Australian egeriinesROYAL SOCIETY
OPEN SCIENCE*R. Soc. open sci.*
doi:10.1098/not yet assigned**A new species of *Proegernia* from the Namba Formation in South Australia and the early evolution of Australian egeriine skinks****K. M. Thorn^{*1,2}, M. H. Hutchinson^{1,2}, M. S. Y. Lee^{1,2}, N. Brown¹, A. B. Camens¹ and T. H. Worthy¹***College of Science and Engineering, Flinders University, BEDFORD PARK, 5042, South Australia
South Australian Museum, North Terrace, ADELAIDE, 5000, South Australia***Keywords:** Scincidae, Egeriinae, Oligocene, Miocene, Palaeontology

ZOOBANK ID urn:lsid:zoobank.org:pub:9C892991-E95A-4ABA-93B9-9480308A7E73

1. Summary

New Oligo-Miocene fossil vertebrates from the Namba Formation (south of Lake Frome, South Australia) were uncovered from multiple expeditions from 2007–2018. Abundant disarticulated material of small vertebrates was concentrated in shallow lenses along the palaeo-lake edges, and is now exposed on the western shore. This fossiliferous deposit, also known from Billeroo Creek 2 km northeast of Lake Pinpa, includes abundant aquatic (such as fish, platypus *Obdurodon*, and waterfowl) and diverse terrestrial (such as possums, dasyuromorphs, and scincids) vertebrates and is hereafter recognised as the Fish Lens. The stratigraphic provenance of these deposits in relation to prior finds in the area is also established. A new egeriine scincid taxon *Proegernia mikebulli* sp. nov. described herein, is based on a near-complete reconstructed mandible, maxilla, premaxilla, and pterygoid. Postcranial scincid elements were also recovered with this material, but could not yet be confidently associated with *P. mikebulli*. This new taxon is recovered as the sister species to *P. palankarinnensis*, in a tip-dated total-evidence phylogenetic analysis, where both are recovered as stem Australian egeriines. These taxa also help pinpoint the timing of the arrival of scincids to Australia, with egeriines the first radiation to reach the continent.

2. Introduction

Major biogeographical events have shaped the Australian flora and fauna, from the separation from Antarctica ~45 million years ago, through to Pleistocene glacial cycles [1, 2]. Australia's herpetofauna can be broadly traced to two origins: relicts from the breakup of Gondwana (the southern supercontinent), or recent arrivals from Asia to Sahul (the continental mass including Australia and New Guinea) [3, 4]. Gondwanan origins are inferred for diplodactyloid geckos, chelid turtles and some frogs [5, 6]. The Asian route appears to have been taken by agamid lizards, and elapids and typhlopoid snakes sometime during the Oligo-Miocene [3, 7, 8]. The temporal origins of Australian scincid lizard radiations are still relatively poorly time-constrained, but molecular data indicate that their closest extant relatives are in south-east Asia [9].

The Australian scincids comprise three subfamilies: the Egeriinae Welch, 1982, Sphenomorphinae Welch, 1982 and Eugongylineae Welch, 1982 [10]. While these three clades have been included in large all-squamate phylogenies, detailed molecular phylogenetic analyses have also been conducted for the sphenomorphine [11, 12] and egeriine radiations [13], but no recent molecular phylogeny of the Australian eugongyline skinks has been published [see 14, which includes some Australian taxa]. A molecular phylogenetic analysis of Australian skinks [3] produced a node-calibrated molecular clock estimate of the divergence of all three subfamilies. Those results suggested that the Australian Egeriinae arrived 18.2 Ma, Eugongylineae 22.9 Ma, and the Sphenomorphinae were the first group to reach Australia 25.3 Ma. However, analyses using mid-Miocene fossil calibrations established a much earlier origin for the Australian

*Author for correspondence (Kailah.thorn@uwa.edu.au).

†Present address: Edward de Courcy Clarke Earth Science Museum, School of Earth Sciences M004, University of Western Australia, 35 Stirling Highway CRAWLEY 6009 Australia

egerniines, minimally 34.61 Ma [15]. This new date prompted the current investigation to seek fossil evidence that might further demarcate the timing of the arrival of (potentially) Australia's first scincids.

The oldest Australian fossil with distinctive scincid characters, the egeriine *Proegernia palankarinnensis* Martin et al., 2004, is from the Etadunna Formation, Lake Eyre Basin, at the Oligo-Miocene boundary 25–26 Ma [16]. An Eocene femur loosely assigned to Scincomorpha by Hocknull is not convincingly scincid [17], so is not robust evidence for inferring the temporal origin of the Australian Scincidae. Other Oligo-Miocene Australian material from the Etadunna Formation was referred to Egeriinae but remains undescribed including: a dentary (UCR 20814), broken parietal (UCR 20815), partial maxilla (UCR 20816), an assortment of vertebrae and a broken scapulocoracoid [18]. Both the *Proegernia palankarinnensis* holotype and the material referred by Estes [18] cannot currently be located. New collections of Oligo-Miocene material are required to resolve the composition and relationships of Australia's oldest scincid faunas. Recent expeditions into central South Australia have unearthed new Oligo-Miocene fossil squamates from the Namba Formation at Lake Pinpa and Billeroo Creek, southeast of Lake Frome. The Namba Formation is of a similar age to the Etadunna [19], and the vertebrate fossil material collected from the lowest layers of this Formation are termed the Pinpa Fauna [20]. This investigation revises the stratigraphic and palaeoecological setting of this deposit, and places the new Oligo-Miocene egeriine fossils into phylogenetic context, alongside *P. palankarinnensis* from the Etadunna, and the Miocene species of *Egernia* and *Tiliqua* analysed in Thorn et al. [15], to infer a more robust date for the arrival of scincids to the continent of Sahul.

3. Materials and Methods

a. Stratigraphy, excavation and fossil collection

Sediment samples from excavated trenches were taken to examine the stratigraphy of both Billeroo Creek (BC2) and Site 12, Lake Pinpa (LP12). Mineral composition was determined by XRD (X-ray diffraction) and XRF (X-ray fluorescence) analyses conducted by Flinders Analytical. Nested sieves with mesh apertures of 6, 3 and 1 mm were used to create sediment fractions from which the fossil material was sorted from the dolomitic clays. The micro-vertebrate material was sorted into fish, mammal, bird, turtle, crocodile, frog, and squamate. Of the 48 squamate cranial specimens recovered, 43 were scincids, and 5 identified as geckos (see Supplementary Information for the squamate specimen list). Most fossils discussed in this investigation were recovered by sieving excavated sediment and sorting the concentrates, however some were collected from the ground surface having been exposed by erosion.

All descriptive terminology of the cranial elements follows Evans [21] and Richter [22] and Kosma [23] for tooth crown features. Appendicular skeleton terminology follows Russell and Bauer [24], and Hoffstetter and Gasc [25] for vertebrae.

b. Scanning electron microscopy

Cranial material identified to Scincidae was further cleaned in water, dried, and then imaged using Scanning Electron Microscopy (SEM). The minute size of some specimens is beyond the capabilities of Micro CT for resolution of morphological features required for identification and descriptions. No specimens required sputter coating. All SEM work was conducted at Flinders University Microscopy facilities using an Inspect FEI F50 SEM. Maximum voltage used for image taking was limited to 2kV and a spot size of 3–4. Measurements of tooth crowns and features of the dentary mentioned in the text were taken using the SEM software.

c. Phylogenetic analyses

In order to better understand the timing of the Australian colonisation by the Egeriinae, both molecular and morphological data (including fossils) are required to generate tip-dated phylogenies. Undated parsimony and tip-dated Bayesian analyses infer, respectively, the phylogeny with the least homoplasy, and the most probable dated phylogeny.

i. Morphological characters

Morphological characters used in the following analyses consisted of 102 discrete and 48 continuous traits, forming an expanded matrix from Thorn et al. [15]. Continuous characters, derived from the measurements of the individual bones or teeth from the dentaries and maxillae, were taken from either Micro-CT scan data in Avizo Lite (v. 9.0) or SEM at Flinders Microscopy, to the nearest micrometre, or with digital callipers to the nearest ten micrometres. All measurements were converted to ratios of either dentary or maxilla length to standardise for size. Continuous characters were converted to values spanning 0–2 to replicate the average number of discrete character states, for analyses in both TNT [26] and BEAST 1.8.3 [27], so that they do not have a disproportionate weight.

ii. Molecular partitions

Molecular data sourced from Tonini et al. [28] and Gardner et al. [13] were analysed using Partition Finder 2 [29] to find optimal partitions and substitution models. The same six molecular (gene) partitions, 12s (412 base pairs [bps]), 16s (681 bps), ND4 (693 bps), BDNF (699 bps), CMOS (835 bps) and B-fibrinogen (1051 alignable bps) and substitution models [15] are used again here.

iii. Maximum Parsimony

The parsimony analyses for the combined discrete morphological, continuous morphological, and molecular data were performed using TNT v.1.5 [26]. *Eutropis multifasciata* was set as the most distant outgroup following the

phylogenetic interpretations of Gardner et al. [13] and Thorn et al. [15]. The most parsimonious tree (MPT) for the combined data was found using 1000 replicates of tree-bisection-reconnection (TBR) with up to 1000000 trees held. To assess clade support, 200 partitioned bootstrap replicates (with discrete characters, continuous characters, and each gene locus treated as a separate resampling partition), were performed using TNT, using new search methods (XMULT) with 1000 replicates and 1000000 trees held. The MPT and bootstrap trees from TNT were exported in nexus format, and continuous and discrete characters were traced [in Mesquite; 30]. The executable files for finding the Most Parsimonious tree, and for performing 200 reps of Partitioned Bootstrap resamples can be found in the SI data files Namba_Egeriines_Topology.tnt (MPT file) and Namba_Egeriines_PartitionedBootstrap.tnt.

iv. Bayesian analysis

The discrete and continuous morphological data, and molecular data were simultaneously analysed in BEAST v1.8.4 using tip-dated Bayesian approaches [27]. *Eutropis multifasciata* was again set as the furthest outgroup. Polymorphic discrete morphological data were treated exactly as coded rather than as unknown, i.e. if coded as states (0,1) it was treated as 0 or 1, but not 2. The discrete character set was analysed using the MkV-model with correction for non-sampling of constant characters [31, 32]. Despite recent disputes over the effectiveness of this model [33], it is well-tested [34, 35] and is still widely accepted and applied to morphological data [36]. Continuous characters, transformed to span values between 0 and 2, were analysed with the Brownian motion model. Bayes factors were used to test the need to accommodate among-character rate variability for both discrete and continuous morphological characters (i.e. gamma parameter).

The stratigraphic data used for tip-dating analyses were derived from fossil taxa and their associated stratigraphy noted in Table 1. No node age constraints were imposed in this analysis, all dates are retrieved from the morphological and stratigraphic age ranges from the noted fossil taxa (tips). The most appropriate available model in BEAST v.1.8.4, birth-death serial sampling [37], was applied. An uncorrelated relaxed clock [38] was separately applied to the molecular and morphological data.

Each Bayesian analysis was run for 100,000,000 generations with a burn-in of 20%. The analysis was conducted four times to confirm stationarity. The post-burnin samples of all four runs were examined in Tracer 1.7.1 [39] to ensure convergence was achieved. All four runs were combined in LogCombiner, and the consensus tree produced by TreeAnnotator [27]. The executable .xml file for BEAST, all output log files, and the final consensus tree file (.tree) are available as supplementary information.

4. Geological setting

a. The Namba Formation

The Namba Formation from north-eastern South Australia (see Figure 2) shares an unconformable lower boundary with the Eyre Formation (Paleocene-Eocene) within our study area in the Frome Basin; and is unconformably overlain by the Pleistocene Eurinilla Formation (Figure 1). The Namba sequence is a lateral equivalent to the Etadunna Formation from the north western Lake Eyre Basin [19, 40]. The Namba Formation is divided into two members [20]; Green claystones and dolomitic claystones at the top of the lower member host a locally abundant vertebrate fauna, termed the Pinpa Local Fauna [20]. This is biostratigraphically correlated with Zone A of the Etadunna Formation to be 25.5–25.7 Ma [19]. Sites visited during the late 1970s–early 1980s led by R.H. Tedford, T. H. Rich, and others, discovered and named multiple fossil localities exposed in Lakes Pinpa, Namba, Tarkarooloo, Yanda and Tinko and Billeroo Creek in the Lake Frome Basin [41]; added to these are new numbered sites from expeditions carried out in 2007 and between 2015–2018 led by T. H. Worthy and A. B. Camens.

All of the sites from Lake Pinpa and Billeroo Creek yielding scincid material for this investigation expose and sample fluvio-lacustrine sites from the Namba Formation. Five expeditions collected new material from the Namba Formation from 2007–2018. Over this period, numerous squamate fragments were collected from multiple sites at Lake Pinpa and Billeroo Creek. Two sites have yielded the majority of the material described herein (see section 4b below); Site 12 at Lake Pinpa, and the ‘Fish Lens’ at Billeroo Creek (within Wells’ Bog Site of Tedford’s, and later T.H. Rich’s, expeditions).

Deposits and the fossils therein contributing to the Pinpa Local Fauna may be classed into two clear taphonomic groups: those containing isolated bones or partial skeletons of terrestrial vertebrates (marsupials, predatory birds, wading birds, and meiolaniid turtles), and those containing localized concentrated bone accumulations which were previously thought to be derived from crocodile coprolites, encompassing mostly aquatic vertebrates (mostly fish, but including also turtle, crocodile, dolphin and rare terrestrial vertebrates). Collection of both categories of this material was predominantly from surface exposures along the western edge of Lake Pinpa and north eastern Billeroo Creek [41], with the occasional excavation of articulated or associated marsupial skeletons. In slightly younger overlying/incised fluvial units exposed as channel fill deposits, the Ericmas Local Fauna has been derived from large scale excavations at Ericmas and South Prospect quarries (Lake Namba, 5 km south of Lake Pinpa) in the 1970s and 1980s [41]. Tom O’s Quarries excavated in fluvial units at Lake Tarkarooloo unearthed the Tarkarooloo Local Fauna which is biochronologically similar to the Ericmas LF, by bulk processing and screening sediments on several expeditions led by T H Rich of Museums Victoria, once with the help of the Australian Army [42]. The quarried Ericmas LF sites contained predominantly terrestrial vertebrate remains, but until the 2007 Worthy and Camens expedition, no squamate material was recorded from the Namba Formation.

Squamate remains have predominantly been recovered alongside the smallest mammal taxa and were found in lenses of densely concentrated bones of aquatic vertebrates dominated by fish (*Actinopterygii* and *Neoceratodus* spp.). These fish lenses lie a few centimetres above dolomite of unknown depth (details below). The dolomite beds have revealed numerous fossils, including associated or partly articulated skeletons of birds and mammals, but fish bones are rare. This dolomitic bed was better exposed towards the middle of the lake bed in the 1970s when it was extensively sampled, but has since at least 2007 been buried by in-washed Quaternary sands. Both layers contain the same mammal and bird species and so the faunas from each are collectively referred to the Pinpa Local Fauna [20, 43]. Both the fish layer and underlying dolomites are exposed at Wells' Bog Site in Billeroo Creek shown in Figure 1.

Figure 1: Stratigraphic sections through the Namba Formation at Lake Pinpa at Site 12 (see Figure 2) and Billeroo Creek, Site 2, constructed in SedLog 3.1 from authors field observations in comparison with RH Tedford's sections described in notes and figures in [20, 44]. depth in meters. Grain sizes were determined in the field. Colours are converted from Munsell figures to corresponding RGB values using [45].

b. Fossil sites

Fossil sites in the Lake Pinpa–Billeroo Creek area were numbered chronologically on the 2007 trip in order of discovery, not based on geographical location (Figure 2).

i. Lake Pinpa

Site 6 (LP6) SAMA P43058 a posterior fragment of a left scincid maxilla, was recovered from Site 6 on the 2007 expedition. The fish lens was observed eroding out in patches over a broad area (200 m by 50 m) here around the margins of the overlying massive grey clay (Layer 2 on Figure 1).

Site 9 (LP9) SAMA P43057 a posterior fragment of a left scincid dentary, was recovered from Site 9 on the 2007 expedition. Here the fish lens was exposed in a narrow zone at the base and edge of eroding massive clay Layer 2.

Site 12 (LP12) Specimens SAM P57544 (partial humerus) and SAM P57545 (maxilla fragment) were found at Site 12 on the edge of Lake Pinpa. Bone was found eroding out on the surface, along the lake edge, from the lowest silty clay layer. Associated and articulated skeletons occur in both this layer and the dolomitic clay (Layer 5) beneath. An undulating clean erosional boundary was observed between these two units.

ii. Billeroo Creek Site 2 Fish Lens

Billeroo Creek Site 2 (BC2) is located on the northern side of the creek (Figure 2) and is a part of the more expansive Wells' Bog Site. Fossils derive from a concentrated lens of predominantly fish bone with limonite inclusions, within the top of Layer 3 of the Namba Formation (Figure 1). The base of this fossiliferous 'Fish Lens' is not flat, but undulated relating to depth of semi-discrete lenses exposed over an area of roughly 10 m by 3 m, excavated over three trips, with a maximum thickness of ~150 mm. The lens sits stratigraphically ~200 mm above the basal dolomite (Layer 5). This layer is laterally more extensive than the Fish lens at Billeroo Creek, and is the layer from which the majority of skeletal fossils at the site have been collected, see SI for a complete list.

Figure 2: Location maps of the Lake Pinpa and Billeroo Creek sites, Frome Downs Station, South Australia. Stratigraphic columns in Figure 1 were compiled from locations Site 12 (=LP12) and Site 2 Fish lens (=BC2) marked above.

c. Stratigraphy

Need an introductory sentence or two

Layer 1— The top layer of the stratigraphic section at LP12 (Figure 1), designated Layer 1, is not part of the Namba Formation. The red sands are reworked sediment from the nearby Quaternary dunes that unconformably overlie the Namba exposures around many of the lakes in the area. At Billeroo Creek (BC2), the fluvial Eurinilla Formation lies unconformably on the Namba Formation and eroded sediments from both it and overlying dunes mantled the Namba Formation where the section was excavated.

Layer 2— Erosion of an unknown amount of the upper part of Layer 2 means its original depth at both BC2 and LP12 cannot be assessed, but near LP07, exposures as documented in the section by Tedford et al. [20], show it was minimally ~6 m thick. It sits unconformably on Layer 3. At LP12, Layer 2 is composed of interbedded light green-grey (7/1/10Y) and yellow (5/1/10Y) medium silts that display cross bedding with very fine laminations. From a distance they have an overall uniform, pale-grey appearance. The layer at BC2 is similar but with white (8/1/10YR) medium silt rather than yellow. No inclusions or fossils have been found in this unit, but locally vertically aligned gypsum crystal plates occur. Concentrations of most soluble minerals are lowest in this stratigraphic horizon.

Layer 3— The upper boundary of Layer 3 at LP12 is erosional, the troughs filled with sediments from Layer 2. Easily distinguishable from Layer 2, it is composed of an olive-grey (4/2/5Y) clay with strong brown (4/6/7.5YR) limonite inclusions throughout. Maximum thickness at Lake Pinpa was 150 mm. The same layer at BC2 was a dark greyish-brown (4/2/2.5Y) clay that had a maximum thickness of ~100 mm and limonite presence was less consistent. The results of the XRD supported field observations of the clay content of the sediment, with clay minerals Smectite and Palygorskite combining to make up 79% of Layer 3 at Lake Pinpa. Limonite inclusions were noted in the stratigraphic section in Layer 3. The XRD analysis of Layer 3 sediment from Lake Pinpa identified the iron ore mineral Goethite (9% of total mineral composition) a common component of Limonite; this was supported by the XRF analyses which found 19.82% Iron (II) and soluble Fe content >16% at both sites. This spike in iron is not reached by any of the preceding layers. The Fish Lens at BC2 had variable thickness within the top 50 mm of this layer, but in some cases Layer 2 sits directly on Layer 3 sediments, with no Fish Lens separating the sediments of each layer. At Lake Pinpa, the Fish Lens is usually much thinner (1–10 mm thick) than at Billeroo Creek and occurs near the top of Layer 3. Exceptions occur at LP6, where the Fish Lens is locally much thicker, sometimes up to 50–120 mm thick in areas of a couple of square metres scattered over several hundred square meters on the lake bed. Layer 3 sits conformably on Layer 4 in stratigraphic sections at BC2, but the boundary is unconformable at LP12.

Layer 4— 150 mm of an olive (5Y 5/3) and orange (10YR 4/6) mottled clay that sits unconformably on Layer 5 at LP12 and BC2. Fossils were found in the lower portion of this layer, on the boundary with Layer 5 at both sites. At Billeroo Creek, Layer 4 is marked by a sharp increase in Calcium, not present until Layer 5 at Lake Pinpa.

Layer 5— This layer is composed of a dolomitic white (10YR 8/1) mudstone with extensive black manganese staining giving a mottled appearance overall. 66% of sediment in Layer 5 at Pinpa and 54% of Layer 5 at Billeroo Creek (Table 3) is derived of clay minerals Smectite and Palygorskite. Layer 5 had less Fe, but much more Mg, Ca and Mn than L3 and L4 (Figure 3 and Table 2), reflecting that it included the minerals Dolomite/Ankerite (Table 3), not present in the overlying layers at either site. The mottled appearance of the dolomitic Layer 5 in both sections is explained by the transition from Dolomite ($\text{CaMg}(\text{CO}_3)_2$) to Ankerite ($\text{Ca}(\text{Fe},\text{Mg},\text{Mn})(\text{CO}_3)_2$) with the partial replacement of magnesium with iron (II) and manganese. Carbonate presence was confirmed with a dilute hydrochloric acid in the field, and reaffirmed by the soluble mineral result from both sites (Figure 3). The primary source of fossils contributing to the Pinpa Local Fauna at Lake Pinpa and Billeroo Creek (excluding Fish Lens) is the assemblage occurring at the bottom of Layer 4 and in the top 100 mm of Layer 5.

Figure 3: Results of the soluble mineral analyses of Layers 2–4 and the white and black sediments from Layer 5 at Lake Pinpa Site 12 (left); and Billeroo Creek Site 2 Layers 2–5 (right).

5. Results

Our excavations have expanded the taxonomic diversity of the Pinpa Local Fauna (first recorded by Tedford et al. [20]). Notably, in 2015–18 we recovered both associated skeletons and isolated bones from the base of Layer 4 and the top of the dolomitic clay layer (Layer 5) at BC2 and LP12. Excavations at LP12 revealed predominantly terrestrial taxa with numerous remains of marsupials (vombatiforms, phalangeriforms, macropodiforms) and birds. In comparison, the fauna from the Fish Lens over various sites is more aquatic, a possible current-concentrated, lake-edge accumulation with bony fish abundant and lungfish, dolphins, flamingos, rails, turtles, crocodiles and the platypus *Obdurodon* represented, in addition to terrestrial marsupials (mainly dasyuromorphs, phalangeriforms) and small scincids.

6. Systematic Palaeontology

Order SQUAMATA Opper, 1811 [46]
Family SCINCIDAE Gray, 1825 [47]
Subfamily EGERNIINAE Welch, 1982 [48]
Genus PROEGERNIA Martin et al., 2004 [16]
PROEGERNIA MIKEBULLI sp. nov.

Zoobank ID: urn:lsid:zoobank.org:act:068F625A-B537-43C7-A1B1-4F20DD5F6BF9

Holotype—SAM P57502, a near complete right dentary; 27 tooth loci, 14 of which bear teeth.

Diagnosis—The species is referred to the subfamily Egeriinae because the dentary has a closed Meckel's groove and a large inferior alveolar foramen. It is referred to the genus *Proegernia* because the tooth crowns widen anteroposteriorly from the shaft, the crista lingualis and crista mesialis are near horizontal and converge on an apex slightly posterior to the centre of the tooth, and there are medially-prominent cristae on the anterior and posterior culmen laterales. *Proegernia* is further distinguished from other members of the Egeriinae (species in *Egernia*, *Bellatorias*, *Liopholis*, *Cyclodomorphus*, *Tiliqua*, *Tribolonotus*, and *Corucia*) by the combination of the following traits: a more anteriorly positioned apex of the splenial notch at >50% the anteroposterior length of the dentary; more than 22 tooth locion the dentary and 20 on the maxilla; minimal anteroposterior flaring of the tooth crown with lateral compression of the medial face; a small sliver of open Meckel's groove immediately posterior to the symphysis and a concave ventral face on the pterygoid body. *Proegernia mikebulli* differs from *P. palankarinnensis* in lacking lateral tooth striae, in having up to five more tooth loci for a total of 27 on the dentary, and a much more medially inflected anterior tip to the dentary, making it more curved in dorsal view.

Type locality—Fish Lens, subsite Billeroo Creek 2, Wells' Bog Site, northern side of Billeroo Creek, GPS coordinates S 31°6'11.76" E 140°13'53.70", See Figure 2.

Stratigraphy/Age—Namba Formation, in Layers 3–5 of the (Figure 1) as exposed in Billeroo Creek and Lake Pinpa; Pinpa Local Fauna. This LF has been biostratigraphically correlated with the "Wynyardiid" or Minkina Fauna, (Zone A) of the late Oligocene Etadunna Formation, 25.5–25.7 Ma [19, 49].

Paratype—SAM P57499 a partial right dentary from BC2, with a complete coronoid process, 23 tooth loci and 10 teeth.

Referred specimens—The following specimens are referred to this taxon based on the combination of their appropriate size, stratigraphic co-occurrence, and having a general similarity to the equivalent elements in other egeriines. Details of tooth crown shape also enable robust referral of tooth-bearing bones to the same taxon. Two right post-dentary compound bones; SAM P57543 from Lake Pinpa (LP6) preserving the dorsal surface of the surangular and glenoid, broken part-way through the retroarticular process; and SAM P57542 from Billeroo Creek (BC2) which preserves the ventral and medial face of the articular, the glenoid entirely and most of the retroarticular process. Two right partial maxillae both from BC2, SAM P57541 representing an anterior fragment with premaxillary process intact and the first 9 tooth loci holding 3 teeth; and SAM P57503 which preserves the posterior majority of the maxilla with 16 loci and 9 teeth, the facial process is broken above the row of maxillary foramina. An intact left pre-maxilla, SAM P57542 from BC2, with osteoderm fragments on the internasal process, preserving two teeth from four loci. A right pterygoid SAM P57512 from BC2, the quadrate process broken beneath the epipterygoid notch.

Etymology—The species is named after Professor Michael Bull (1947–2016) of Flinders University, South Australia, who devoted decades to documenting the ecology of Australia's egeriine skinks. Mike supervised a generation of Australian ecologists and his lectures inspired countless students to become biologists. His studies of Australian egeriine skinks and their parasites are model long-term ecological studies, and led to major discoveries such as the existence of monogamous pairs in *Tiliqua rugosa*, and parental care and family living in *Egernia* spp.; and the establishment of a successful breeding and reintroduction program for the endangered Pygmy Bluetongue, *Tiliqua adelaidensis*.

a. Description

Dentary—The most diagnostic and commonly-recovered element is the dentary bone of the lower jaw. Seventeen incomplete dentaries were recovered from the 2007–2018 trips to Lake Pinpa and Billeroo Creek. A reconstruction of a near-complete lower jaw of *Proegernia mikebulli* was made, with the dentary portion based on the holotype SAM P57502 and paratype SAM P57499 (See Figure 4). From the anterior tip of the symphysis to the posterior tip of the coronoid process, missing only the angular process, the reconstructed dentary is 12.7 mm long. In medial aspect, the dentary is convex ventrally. It is 1.6 mm wide and 2.2 mm tall at mid-length of the tooth row, and the dental sulcus is shallowly concave. From occlusal view, the symphysis is directed medially to articulate with the opposing dentary at an angle of 32°. The dental sulcus is clearly differentiated, SAM P57502 preserving 27 loci and 14 pleurodont teeth. The tooth row is 10.8 mm long.

Figure 4: 1a-c Holotype, a right dentary, SAM P57502, 2a-c paratype; a left dentary SAM P57499; two right post-dentary compound bones 3a-c SAM P57543 and 4a-c SAM P57542 of *Proegernia mikebulli* sp. nov. from the Fish Lens at BC2; and 5 complete reconstruction in medial view of the right mandible of *Proegernia mikebulli*, the coronoid and splenial (hatched area) are reconstructed based on modern egeriine specimens, no fossil representatives of these elements are known. Abbreviations: adf, adductor fossa; anf, angular facet; art., articular; d., dentary; gl, glenoid fossa; iaf, inferior alveolar foramen; mg, Meckel's groove; psf, posterior surangular foramen; rap, retroarticular process; san., surangular; and sy, symphysis.

The anterior symphysis, preserved in its entirety on SAM P57502, has a flattened, reverse '7' shape in medial view. The symphysis has two caudally-directed branches; a narrow, ventrally-directed sliver along the anterior edge of the dentary; and, dorsally, a wider process that terminates in a sharp point. Maximally, the symphysis is 1.8 mm long and 1.3 mm deep. In the notch between the caudally-directed branches of the symphysis a symphyseal foramen extends ventrally into a 1 mm long and <0.2 mm wide anterior opening of Meckel's groove. The groove is open and aligned parallel with the dorsal side of the ventral section of the symphysis; further posteriorly it is enclosed.

The inferior alveolar foramen and the dorsal and ventral margin of the splenial notch are preserved to the posterior end of the tooth row in SAM P57502 (Figure 4). The inferior alveolar foramen lies on the ventral margin of the splenial notch, below the mid-point of the tooth row. The shape of the splenial notch is narrow and roughly parallel-sided for the anterior 40% of the preserved length, expanding dorsoventrally with a convex upper edge and straight ventral margin for the remaining posterior section of length. Where the notch expands, the dorsal edge preserves a concave face, allowing the splenial to medially overlap the dentary. Posterior to this face, beneath the 3rd-last tooth, the dental sulcus is broken.

The coronoid process of the dentary is preserved on the paratype SAM P57499 and ascends posterodorsally from the position of the last tooth to a tip projecting dorsally above the posterior tooth. The ventral margin of this process preserves the anteromedial articular facet for the coronoid, which therefore can be seen to overlap the splenial and dentary anterior of the 3rd last tooth position. The angular process, although not preserved entirely on either dentary specimen, can be reconstructed with some confidence using the angle of the intact edge immediately beneath the coronoid process. The total length of this process is limited by the absence of a facet for its articulation on either of the recovered post-dentary compound bones.

The lateral face of the dentary is slightly convex and preserves a single row of eight mental foramina extending posteriorly from the anterior top of the dentary to below the 17th tooth position. The largest foramen in the row is at the posterior end.

Post-dentary (compound) bones— The post-dentary complex or compound bone is the fused surangular and articular bones making up the posterior half of the scincid mandible. This fusion develops ontogenetically; complete fusion of the two elements with no traces of a suture internally is evidence of adulthood in extant Australian scincids [50]. Two right post-dentary complexes were recovered from the Namba Formation, SAM P57543 from a right mandible at Lake Pinpa (LP6; Figure 4 3a–3c) and SAM P57542 from Billeroo Creek (BC2; Figure 4 4a–4c); these are used in the reconstruction of the complete lower jaw (Fig. 4, 5). These elements are referred to this taxon as they are of appropriate size, confirmed as scincid material by the presence of a facet for the angular bone, which is not present in gekkotans; and the alignment of the retroarticular process. The shape of the retroarticular process is similar to that of extant egeriines. When aligned horizontally in the in-vivo position, the angular in egeriines is wide and extends further laterally than the narrow ventrally positioned angular of eugongyline (i.e. *Emoia longicauda* SAMA R2352, *Eugongylus rufescens* R36735). Sphenomorphines generally have a simple, straight post-dentary complex without medial torsion of the articular, and the retroarticular is not inflected medially posterior of the glenoid. All scincid bones from Billeroo Creek and Lake Pinpa are consistent with presence of a single egeriine taxon; there is no variation among elements that might indicate more than one species being represented, so both compound bones are referred to the taxon named from the dentaries.

SAM P57543 is a near complete right adult post-dentary preserving completely fused surangular and articular bones. Anteriorly the areas for articulation with the dentary and coronoid are not preserved, neither are the angular or the anterior edge of the adductor fossa. A facet for articulation of the angular is present on the ventral half of the lateral face of the articular, not extending to reach the posterior surangular process. The posterior majority of the retroarticular process is broken off SAM P57543, so a description of its complete shape is based on SAM P57542. The dorsal surface of the surangular is relatively straight and rises slightly to meet the dorsal tip of the mandibular condyle or glenoid fossa where the quadrate articulates with the mandible. The maximum preserved length of this dorsal surface is 2.9 mm. The glenoid fossa faces posterodorsally, the surface is convex mediolaterally and concave anteroposteriorly, reaching a maximum length of 1.8 mm. The lower third of the glenoid fossa becomes slightly concave ventrally, curving up to meet the medial articular process of the surangular. The surface of the glenoid fossa has a pitted texture indicating attachment of cartilage in the mandibular cotyle. Posterovertrally the glenoid fossa ends in a clearly defined ridge between it and the retroarticular process. The preserved section of the retroarticular process is concave (SAM P57542). Medially, the foramen for the chorda tympani is preserved in SAM P57542.

Two foramina are preserved dorsally in a flattened surface, and a posterior surangular foramen is on the lateral face immediately anterior of the glenoid fossa. No sign of an anterior surangular foramen is present on either SAM P57543 or SAM P57542. From the flattened dorsal surface of the surangular, the bone curves sharply ventrally, leaving a flat medial surface above the dorsal margin of the adductor fossa. The adductor fossa is in the lower 50% of the medial face of the complex. The anterior margin of the adductor fossa is not preserved on either post-dentary compound bone. Viewed medially, the posterior edge of the narrow, ovular, fossa terminates anterior to the glenoid. The lateral face of the post-dentary complex of the mandible is convex. A discernible ridge runs from just anterior of the posterior surangular foramen, anteriorly to the broken anterior edge of SAM P57543, marking the posteroventral edge of the *M. adductor mandibulae externis*.

Maxilla— Two incomplete right maxillae (SAM P57503, SAM P57541) recovered from BC2 are referred to *Proegeria mikebulli*. SAM P57503 (Figure 5, right) preserves a near complete tooth row, a bifid posterior termination of the suborbital process (an apomorphy of the Scincidae), and complete tooth crowns from both the anterior tooth form and posterior tooth form on the same row. This specimen was recovered from sieved material in two halves, that fitted together when articulated, and so was joined with Paraloid B72 before SEM images were taken. The smoothed hemispherical shape on the dorsal edge of the palatine process of the maxilla (Figure 5) is Paraloid and not a feature of the bone.

Anteriorly, the maxilla SAM P57503 lacks the tip of the premaxillary process. The anteriormost tooth positions are missing on SAM P57541; tooth and loci counts are based on the reconstruction also using SAM P57541. The facial

process of the maxilla is incomplete preserving only 2.3 mm above the tooth row. The unbroken edge of the orbit can be traced from above the level of the 14th tooth position, to the broken dorsal tip of the suborbital process. The ventral tip of the bifid suborbital process is intact as are the last tooth positions. The tooth row preserves 16 of 20 loci, and 9 pleurodont teeth are still in position.

The lateral face of the maxilla is slightly convex in the anterior to posterior plane when viewed dorsally. The maxilla preserves nine primary foramina on the lateral face, the largest is the most anterior. Above the primary row, 11 more foramina, the largest one-third the size of the primaries, are visible in the SEM in two rows. Medially, the medial edge of the palatine shelf is broken. The palatine shelf thins posteriorly and ends in a fine point at the ventral tip of the bifid suborbital process. The tooth row occlusal surface is slightly convex posteriorly, with larger teeth beginning at the ninth position. The last two teeth decrease in size posteriorly, creating the trailing edge of the convex occlusal line. The reconstructed maxilla is 11.1 mm long from the premaxillary process to the suborbital process, with 20 tooth positions (see Figure 5). The original height of the facial process was not preserved on any specimen.

Figure 5: Right maxilla of *Proegernia mikebulli* sp. nov. reconstructed from SAM P57541 (anterior fragment; top left) and SAM P57503 (near complete; top right). 'A' marks the same tooth locus on each specimen. Enclosed by the box is the tooth 'B'. Abbreviations: cl, *cuspis labialis*; cla, *culmen lateralis anterior*; cli, *cuspis lingualis*; clp, *culmen lateralis posterior*; fp, facial process; psm, palatine shelf of the maxilla; pxp, premaxillary process; sop, suborbital process; and str, striae.

Premaxilla— A single complete left premaxilla, SAM P57542, was recovered from Fish Lens at BC2 (Figure 6). An osteoderm is fused to the anterior face of the internasal process, and would have overlapped the right premaxilla when the pair were articulated. Paired, unfused premaxillae in adulthood eliminate the possibility that this element belongs to a gekkotan. The ascending internasal process has a flattened medial side for articulation with its paired element. Viewed laterally, a foramen for the longitudinal canal is situated immediately above the maxillary process in the lateral face of the internasal process. Sharply pointed dorsally, the internasal process widens ventrally and terminates at the notch for the exit of the ethmoidal foramen laterally, and tooth row medially. The internasal process of the premaxilla rises towards the nasals at a 40° angle. The total length of the internasal process is 3.1 mm. The maxillary process is curved posterolaterally towards the maxilla and is approximately 1/3 of the height of the internasal process, reaching 1.6 mm in mediolateral width. The lateral edge of the maxilla process curves ventrally to finish beside the fourth tooth position. Four tooth loci are present and the first two teeth are preserved. These teeth are 0.53 mm long from the lateral face of the maxillary process and 0.28 mm wide. What remains of the worn crowns (see inset A, Figure 6) match those of the anterior-most teeth of the maxilla specimen SAM P57541. There are weak striae on the medial face of the crown, a mediolateral compression of the crown below the *crista lingualis*, and a sharply curved, prominent, *culmen lateralis*.

Figure 6: Left premaxilla (SAM P57542) of *Proegernia mikebulli* sp. nov. from posterior (left), anterior (centre), and lateral (right) view. A, tooth one, enlarged in box. Abbreviations: flc, foramen of longitudinal canal; inp, internasal process; mp, maxillary process; nef, notch of the ethmoidal foramen; and o, osteoderm.

Dentition— The dentition of *Proegernia mikebulli* is described using the two dentaries SAM P57502 and SAM P57499, the maxillae SAM P57541 and SAM P57503, and premaxilla SAM P57542, together comprising a near-complete upper and lower tooth row. The upper tooth row contains a total of 24–25 teeth, 4 or 5 on the left and right premaxilla, and 20 on the maxilla. The dentary tooth row has 27 tooth positions, beginning above the symphysis anteriorly and terminating just anterior of the coronoid process. The occlusal profile of the maxilla and dentary are both convex, with slightly larger teeth in the posterior half of the tooth row.

Figure 7: A and B: 24th tooth on the dentary SAM P57502, medial view, and C the 16th tooth in occlusal view with lateral and medial facies marked. Abbreviations: ai, antrum intercristatum; cl, crista lingualis; cm, crista mesialis; cul, cuspis labialis; cula, culmen lateralis anterior; and culp, culmen lateralis posterior.

On the dentary teeth, the tooth crown is similar in width to the shaft, expanding anteroposteriorly with slight mediolateral compression for the last 30% of the tooth height. Prominent anterior and posterior *culmen laterales* extend from the tips of the *crista mesialis*, turning medially and ventrally on the dentary, with a sharp angle producing ‘shoulders’ notable in medial view of the tooth (Figure 7). From the central cusp, the anterior cristae dip ventrally 20°, and posterior cristae 45°. The cusp (*cuspis labialis*; Figure 7) is slightly posterior of the centre of the tooth, and in occlusal view is positioned just posteromedial to the centre. This medial shift of the cusp creates a convex dorsal surface to the tooth, directing both cristae medially. Striae are only located on the medial face of the tooth crown, angled

dorsoventrally from the off-centre cusp. The two most prominent striae run **parallel** to the *culmen laterales* (both anterior and posterior), all others are weaker in profile with staggered lengths. Tooth crown morphology is similar on the maxilla and premaxilla.

Tooth wear occurs first on the cusp, forming a shallow rounded depression, gradually deepening medially. The *antrum intercristatum* expands in width from the central cusp. Wear is thus more noticeable in the centre of the tooth crown between the cusps, than anteriorly or posteriorly along the cristae, creating a thickening 'v' shape.

Pterygoid— A single **right** pterygoid attributed to *P. mikebulli* was recovered from Fish Lens at BC2 (SAM P57512; Figure 8). It is **near** complete, missing only the anterior-most tip of the palatine process and distal tip of the quadrate process. The pterygoid articulates with the palatine anteriorly with a pointed process extending anteriorly from a fanned, v-shaped **pterygoid head to overlap with a similar feature from the anterior element**. Laterally to the palatine process, the ectopterygoid process extends towards the ectopterygoid with a concave, curved facet for articulation with this element on the dorsal surface of the pterygoid head. Between these two processes, the fan-shaped pterygoid head has a concavity on the ventral surface that is 0.7 mm wide and 1.7 mm long. Within this concavity are three foramina, one in the deepest area of the concavity, and two are paired anteriorly, near the palatine process. Posteriorly, the pterygoid head narrows to become a parallel-sided rod, **before** bending medially at an angle of 35° towards the epipterygoid notch. The epipterygoid notch is an **ovular** concavity, with raised margins, on the dorsal surface of the pterygoid at the anterior end of the quadrate process. A sharp ridge extends from the anterior edge of the epipterygoid notch to the corner of the bend towards the pterygoid head. Posterior to the epipterygoid notch, the quadrate process becomes L-shaped in cross section and extends 1.5 mm posteriorly **before** the broken edge. The pterygoid head is 2.7 mm wide between the palatine and ectopterygoid processes. Total length of this element is **not preserved**.

Figure 8: **Right** pterygoid of *Proegernia mikebulli* sp. nov., SAM P57512 in dorsal (left) and ventral (right) views. Abbreviations: en, epipterygoid notch; pp, palatine process; pvc pterygoid ventral concavity; qp, quadrate process; and tp, ectopterygoid process.

b. Postcranial material potentially attributable to *Proegernia mikebulli*

Several procoelous vertebrae and assorted vertebral fragments were recovered from the Fish Lens at BC2. The best preserved of these is SAM P57514, a **pre-sacral** vertebra (Figure 9). This specimen is **near** complete, missing only the posterodorsal tip of the neural spine and the lateral tip of the right postzygapophysis. A thin crack runs posterolaterally on the ventral face of the centrum from just posterior of the ventral foramina, to the broken zygapophysis. The vertebra has an elongate centrum, the ventral surface is concave in lateral view. The neural canal is **a similar** diameter to the cotyle/condyle with a flat ventral surface broadening laterally before coming to a tear drop point when viewed anteriorly or posteriorly. Synapophyses are present immediately posterior to the prezygapophyses on the lateral facies. The neural spine rises posteriorly, steadily at an angle of 20°, from the dorsal margin of the neural canal until reaching the broken extremity. The prezygapophyses are angled at 32° and the postzygapophyses at 28°. Overall shape of the vertebra is relatively elongate, approximately one third longer than it is wide. Overall centrum length is 4.2 mm, the cotyle maximum width is 1.6 mm; the vertebra height anteriorly is 2.4 mm, posteriorly including the broken neural spine the height it is a maximum of 3.7 mm. The widest point of the vertebra is between the synapophyses, totalling 3.3 mm. This element is **identified as a scincoid lizard** based on procoely, absence of a zygosphenes/zygantrum, minimal dorsoventral compression of the rounded cotyle and condyle, elongated centrum, and the minimal size of the ventral foramina. Snake vertebrae have shorter centra, a round cotyle/condyle with no compression, a set of zygosphenes and zygantra, and a ventral hypapophysis. A procoelous centrum devoid of a hollow space for the notochord eliminates the possibility of the

vertebra belonging to a non-pygopodid gekkotan. Gekkotans inclusive of pygopodids also have **enlarged ventral foramina**. Procoelous Gekkota vertebrae (mostly restricted to pygopodids) tend to have smaller condyles and cotyles, the cotylar hollow often not visible in ventral view because the dorsoventral inclination is minimal [25].

Figure 9: A single vertebra (SAM P57514) potentially referable to *Proegernia mikebulli* sp. nov., recovered from Fish Lens, BC2. A, anterior view; B, posterior view; C, lateral view, anterior to the left; D, dorsal view; and E, ventral view. A proximal fragment of a right femur assigned to *Proegernia mikebulli* sp. nov. SAM P57540. F, dorsal; G, anterior; and H, ventral views.

Abbreviations: con, condyle; cot, cotyle; fec, femoral condyle; itf, intertrochanteric fossa; itr, internal trochanter, nc, neural canal; nsp, neural spine, poz, postzygapophysis; prz, prezygapophysis; and syp, synapophysis.

A proximal right femur, SAM P57540 (F–H, Figure 9), was recovered from the Fish Lens at BC2. Broken immediately beneath the intertrochanteric fossa, **no remnants of the shaft width or length are preserved**. All other proximal features are intact. The specimen is most likely from an adult **scincid** as the epiphyses are fully ossified and fused to the diaphysis. The articular surface of the femoral condyle is semi-circular viewed anteriorly, and ovular in proximal profile. The femoral head extends medially away from the shaft axis. The internal trochanter is much shorter, only rising slightly above the edge of the intertrochanteric fossa. The intertrochanteric fossa is a concave surface between the condyle and trochanter on the ventral face of the proximal femur. The dorsal side also preserves a concave surface, but this is steeper-sided with a prominent ‘v’ marking where the femoral shaft begins. The **width of the** femoral condyle is widest in anterior view, measuring 2.2 mm at the suture with the epiphysis. The internal trochanter is 1.3 mm tall from the base of the intertrochanteric fossa. Total length preserved is 3.8 mm.

The scincid apomorphy of the trochanter major gradually tapering in height distally (see Lee et al. [51]) is not visible on this specimen, as the shaft is broken immediately proximal to it.

c. Comparisons

Comparison of new taxon with *Proegernia palankarinnensis*— The type species of *Proegernia* (by monotypy) is *Proegernia palankarinnensis*. While the type specimen is currently missing, the existing description [16] suggested *Proegernia* as being representative of a transitional form between a plesiomorphic lygosomine dentary morphology and the derived egeriine condition. The plesiomorphic traits shared by both *Proegernia palankarinnensis* and *P. mikebulli* are the small (about 1 mm long) opening of Meckel’s groove immediately posterior of the symphyseal foramen and the more anterior extent of the splenial notch into the anterior half of the tooth row **length**. Both species of *Proegernia* also have egeriine features emerging in the prominent ‘shoulders’ of the *culmen lateralis* on their unicuspid tooth crowns, and a relatively deep splenial notch, extending anteriorly further than other Australasian scincids. *Proegernia palankarinnensis* was described as having 22 tooth loci, with a possible maximum of 23 teeth, uncertain due to a broken symphyseal region. The specimens of *Proegernia mikebulli* preserve up to 27 dentary tooth positions (see SAM P57502, Figure 4). Both taxa share an anterior section of crowded, smaller teeth in their lower dentition. The spacing of teeth in *P. palankarinnensis* is **marginally** wider than *P. mikebulli* at the posterior end of the tooth row. Tooth shape varies between the two taxa: *P. palankarinnensis* has tooth shafts narrowing at the base, whereas tooth shafts in *P. mikebulli* narrow beneath the crown and expand again at the base. Martin et al. [16] noted that the tooth crowns of *Proegernia* SAMA P39204 preserve distinct striations on both the lateral and medial faces. Tooth crowns for *Proegernia mikebulli* have **weakly lineated** striae on the medial face of the crown beneath the *crista lingualis*, but the lateral tooth face is smooth.

Although occurring almost concurrently, comparisons of the shape of the **holotype** dentaries of *P. palankarinnensis* and *P. mikebulli* (see Figure 10) show that the two species differ notably in the following characteristics: 1) the medial inflection of the symphysis is considerably less in *P. palankarinnensis* than in *P. mikebulli*, even after taking into account the effects of erosion. 2) The inferior alveolar foramen is positioned more anteriorly in *P. palankarinnensis*, below the 12th tooth position; that in *P. mikebulli* is level with the posterior edge of the 14th tooth. 3) No short extension of Meckel's groove can be observed anterior of the inferior alveolar foramen in *P. mikebulli*, a plesiomorphic character representing a partial remnant of Meckel's groove that was noted [16] on the holotype of *P. palankarinnensis* (Figure 10 A). **4) The internal septum of both specimens extends anteriorly to a similar degree.** 5) The overall shape of the dentary of *P. palankarinnensis* is less robust, it being narrower between the medial edge of the dental sulcus and lateral face at the height of the mental foramina; 7) The facet for articulation of the coronoid terminates beneath the **second-last** tooth in *P. palankarinnensis* and the **third last tooth locus** in *P. mikebulli*. 8) The dorsal edge of the splenial notch is widened more abruptly in *P. mikebulli* with a curve; in *P. palankarinnensis* the same space is sharply v-shaped. 9) The coronoid process preserved on *P. palankarinnensis* does not extend above the height of the final tooth crown, unlike the process on SAM P57499 of *P. mikebulli*.

Figure 10: Comparative SEM of *Proegernia mikebulli* sp. nov. (B,C) and **microphotographs of *P. palankarinnensis* (A,D) from Martin et al. [16], in medial (above) and occlusal (below) views. Note the anterior extension of the splenial notch (3) in *P. palankarinnensis* and the increased the number of teeth/loci in *P. mikebulli*. Features described in text are labelled: 1) Medial inflection of the symphysis; 2) Dentary depth; 3) Position of the inferior alveolar foramen; 4) Meckel's Groove extension anterior of the inferior alveolar foramen, present in *P. palankarinnensis* and absent in *P. mikebulli*; 5) anterior extent of the internal septum similar in both taxa, 6) Overall robustness as dentary width, 7) facet for articulation with anterior ramus of coronoid, 8) splenial notch height, and 9) height of the coronoid process relative to the last tooth.**

d. Other egeriines and other scincids

Comparisons with extant egeriines including *Lissolepis coventryi* (SAMA R57317) and *Liopholis multiscutata* (FUR168), as well as a Southeast Asian lygosomine outgroup representative *Eutropis multifasciata* (SAMA R35693), and two plesiomorphic scincids *Eumeces schneideri* (SAMA R6695), *Brachymeles schadenbergi* (SAMA R8853) and *Plestiodon fasciatus* (SAMA R66784) were conducted to determine how similar or derived the fossil taxon is from the generalised scincine (=inferred plesiomorphic) condition. Four characters notably vary among the chosen representative egeriines, the outgroup lygosomine and the extinct *P. mikebulli*: the anterior extent of the splenial notch and the inferior alveolar foramen; the shape and robustness of the symphysis, dentary depth and overall width; and, the number of teeth in the tooth row. The anterior extent of the splenial notch varies within the egeriine radiation; both species of *Lissolepis* share a further anterior-reaching inferior alveolar foramen than any other egeriine genus. The outgroup scincid condition, represented by the examined taxa *Eumeces schneideri* (SAMA R6695), *Brachymeles schadenbergi* (SAMA R8853), *Plestiodon fasciatus* (SAMA R66784), and *Eutropis multifasciata* (SAMA R35693) have a splenial notch that extends further than 60% anteriorly along the length of the tooth row, or one that stretches anteriorly into an elongate open Meckel's groove. The anterior extension of the splenial notch is a plesiomorphic state for this character, also shared by both *Proegernia* taxa.

The mandibular symphyseal joint in scincids varies between subfamilies in its overall size and robustness. Within egeriines, the variation is in the anteroposterior length of its posteroventral section, and the depth of the upper branch. In egeriine species that have a reinforced symphyseal joint and manipulate harder food, the lower branch of the symphysis is extended posteriorly and sometimes ventromedially (especially in *Tiliqua*, *Cyclodomorphus* and larger species of *Egernia*), to form a 'chin'. The dorsal branch of the symphysis deepens in *Liopholis* and in larger, omnivorous and herbivorous species of *Egernia*. This may be related to functional morphology to handle stresses on the chin during feeding, rather than a phylogenetic signal as these taxa do not form a clade. *Proegernia mikebulli* has a short and narrow posteroventral extension of the symphysis, possibly due to the presence of a foramen in the caudal notch between the limbs of the symphysis. This foramen exposes a short section of Meckel's cartilage in sphenomorphines and scincines. In egeriines this foramen is usually absent. Both *P. palankarinnensis* and *P. mikebulli* have dentaries where a slightly elongate foramen would expose the Meckel's cartilage, but in neither does this opening extend beyond the posterior edge of the symphysis.

Increased dentary depth and width, in relation to overall length, increases the robustness of the mandible. Variation in these measurements occur between genera of egeriines; species within *Liopholis* often have shorter, deeper skulls and corresponding dentaries, than similar-sized lizards within the genus *Egernia* (see the Supplementary information from [15]). *Lissolepis* has a more gracile dentary (see Figure 11). Outside of the egeriines, *Eutropis* and other SE Asian scincids with an unmodified insectivorous diet are even more gracile. *Eutropis multifasciata* has a longer dentary tooth row relative to snout-vent length than all egeriines. Derived morphologies related to dietary adaptations as seen in the herbivorous and durophagous members of the Egeriinae (i.e. *Corucia zebrata*, *Tiliqua* and *Cyclodomorphus*) result in more robust dentary dimensions and/or modified dental morphology. *Proegernia palankarinnensis* and *P. mikebulli* present dentary depths most similar to those of *Lissolepis* spp., between the ancestral shallow insectivorous *Eutropis*, and the deeper dentary typical of *Liopholis* spp. Deepening of the dentary bone and shortening of the snout noted in numerous species within the genus *Liopholis* is possibly related to the group's affinity for burrowing [52].

Eutropis multifasciata*Proegernia mikebulli* sp. nov.*Lissolepis coventryi**Liopholis multiscutata*
Figure 11: Comparative medial views of the dentaries of *Eutropis multifasciata* (SAMA R35693), *Proegernia mikebulli* sp. nov., *Lissolepis coventryi* (SAMA R57317) and *Liopholis multiscutata* (FUR168). 1 Anterior extent of splenial; 2 Symphysis shape and robustness; and 3 Dentary depth at mid-tooth row length.

e. Dentition

The teeth of *Eutropis multifasciata* (SAMA R35693) and *Proegernia mikebulli* show increasing derivation from the plesiomorphic insectivorous skink tooth type as described by Richter [22] and Kosma [23] and represented by *Plestiodon fasciatus* (SAMA R66784; Figure 12). Although many species within the extant radiations of eugongyline, sphenomorphine and egeriine scincids have modified tooth shapes for dietary specialisations, taxa exhibiting the most plesiomorphic tooth crown shapes were chosen for comparisons with fossil taxa.

Proegernia mikebulli has departed from the crown shape of basal skinks [see 22, 23] by having the cristae lingualis et mesialis directing medially, creating a sharp angle with the *culmen laterales*, and less-prominent striae mark the medial face of the tooth. *Eutropis* retain the prominent striae running dorsoventrally on the medial face of the tooth, these striae are almost all equally prominent, while *P. mikebulli* appears to have slightly stronger striae immediately adjacent and parallel to the *culmen laterales*. The tooth crowns of *Eugongylus rufescens* also demonstrate prominent *culmen laterales* and weakened striae, although the tooth shape differs; the shaft widening beneath the crown slightly and wear patterns make obvious the less medially directed cristae. The basal sphenomorphine condition is that of a narrow tooth shaft and crown, with sharply angled cristae creating a more pointed tooth profile. The medial face of the sphenomorphine tooth is anteroposteriorly convex and marked with prominent striae that do not approach the height of the lingual cristae but sit a third of the depth of the crown lower. The striae on *P. mikebulli* extend from almost directly in contact with the lingual cristae to the ventral tips of the *culmen laterales*.

The tooth crown in *P. mikebulli* is similar to that of the extant *Lissolepis coventryi* sharing medially directed cristae, sharp ‘shoulders’ to the *culmen laterales*, weak medial striae and absent lateral striae, and an expanded crown width on a narrower tooth shaft.

Figure 12: Medial view of left dentary tooth crowns of *Plestiodon fasciatus* (SAMA R66784), *Eutropis multifasciata* (SAMA R35693), *Proegernia mikebulli* sp. nov. (SAM P57502), *Lissolepis coventryi* (SAMA R57317), *Eugongylus rufescens* (SAMA R36735) and *Sphenomorphus jobiensis* (SAMA R6736). *Plestiodon* represents the basal skink tooth condition as described by Richter (1994). *Culmen lateralis* posterior is labelled culp.

7. Phylogenetic relationships

Parsimony and Bayesian analyses retrieved broadly similar trees, placing both species of *Proegernia* outside of, but close to, living (crown) Australian egeriines. Parsimony analysis of all morphological and molecular data recovered one most parsimonious tree (Figure 13) with a best score of 4726.25, found 60 times out of 1000 replicate searches. Confidence on most of the nodes indicated by the bootstrap analyses is low; only support for the clades *Bellatorias*, *Liopholis* and *Tiliqua+Cyclodomorphus* are above 50. This is almost certainly due to an unstable position of the fossils, which are missing all DNA and most morphological data (111 missing characters for *P. palankarinnensis* and 73 for *P. mikebulli*).

Figure 13: Egerniine phylogeny based on the single most parsimonious tree produced by TNT [26]. Bootstrap values >50 shown.

Proegernia mikebulli is recovered basal to the extant Australian Egerniinae (spanning *Lissolepis* to *Tiliqua*), and *Proegernia palankarinnensis* is retrieved as sister to the Solomon Islands' *Corucia zebra*, but both with bootstrap support <50%. These relationships are conservatively interpreted as an effective polytomy between crown egerniines, *Corucia*, *Proegernia*, and *Lissolepis*. Thus, rather than erect a new genus, we provisionally assign the new species to *Proegernia* (due to geographic and stratigraphic links with the type of that genus, and the Bayesian results below).

The position of both taxa outside the crown Australian egerniines is due to plesiomorphic scincid character states such as a partially open Meckel's groove (char. 11) in both fossil taxa. This feature also separates them from the outgroup taxon *Eutropis multifasciata* which has an entirely open Meckel's groove. The anterior extent of the splenial (char. 21), and the extension of Meckel's Groove anterior of the splenial notch are features supporting the separation between *Lissolepis* and *Proegernia*; *Lissolepis* with a splenial reaching approximately 50% of the tooth row length anteriorly, and *Proegernia* reaching beyond to two-thirds the tooth row length.

The crown Australian Egerniinae spanning *Lissolepis* to *Tiliqua* is diagnosed by a completely closed Meckel's Groove (char. 11), an absence of pterygoid teeth (char. 131), and a pterygoid quadrature ramus that is arcuate in cross section (char. 134). *Proegernia* (and *Corucia*) have plesiomorphic or alternative states for these characters, resulting in their position outside this clade. Potential morphological synapomorphies of the basal living Australian genus *Lissolepis* that are also present in *Proegernia* are the termination of the dentary coronoid process (char. 17), the orientation of the retroarticular process from dorsal view (char. 46), the plesiomorphic bicuspid tooth crown shape (char. 55), minimal dental cementum (char. 60), and the divot in the ventral surface of the pterygoid body that is also absent of pterygoid teeth (chars 135 and 131). These shared characters conflict with the position of *Proegernia* outside the Australian crown group, resulted in the low support for the relationships between *Corucia*, *Proegernia* and *Lissolepis*.

The Bayesian analysis of all morphological and molecular data produced the topology shown in Figure 14: the consensus of four separate runs of 100,000,000 generations, when combined in LogCombiner, then summarised with TreeAnnotator with a burnin of 20%.

Figure 14: Egerniine phylogeny based on consensus total evidence Bayesian inference tree from BEAST. †denotes extinct taxon. Small numbers at nodes denote posterior probability. Reconstructed node ages for key clades are in brackets, the green shading denotes the Eocene, after which Australia became isolated from Gondwana [4].

The extant egerniine radiation forms a well-supported clade. Within this, *Tribolonotus* is strongly supported as the sister to remaining taxa. Of those, *Corucia* is strongly supported as the sister of the remaining (Australian) egerniines. Both late-Oligocene fossil species of *Proegernia* are weakly supported as the sister to *Corucia* with a Prior Probability (PP) of 0.25. All Australian extant egerniines again form a clade that excludes both *Proegernia*. The extant Australian clade is a weakly supported (PP= 0.51) clade and retrieved as originating in the lower Oligocene (Figure 14).

Proegernia palankarinnensis and *P. mikebulli* are retrieved as sister taxa, but with little support (PP= 0.44), not unexpected given extensive missing data. Character changes affirming the sister relationship include the position of the mental foramina 50% along the length of the tooth row (char. 13), and the presence of less than 20 tooth crown striae (char. 61). Both of these features are also shared with *Lissolepis*, but not with their putative sister clade *Corucia zebrata*. The similar positioning of both taxa (i.e. basal to crown Australian egerniines) is supported by the same characters as those noted after the parsimony analysis.

The node age for all Egerniinae is retrieved as 50.12 Ma, with *Tribolonotus* as the extant stem taxon representing the arrival of the subfamily to northern Sahul (Australia, New Guinea and surrounding islands). The stem clade last shared a common ancestor with the crown Australian egerniines 40.47 Ma.

8. Discussion

Recent exploration of the Namba Formation cropping out at Lake Pinpa and Billeroo Creek has unearthed a new egerniine scincid lizard, *Proegernia mikebulli* sp. nov. Previous excavations and collection of surface material at these localities did not recover any squamate elements. This identification has increased the taxonomic diversity of the Pinpa Local Fauna in the Namba Formation to include at least one skink, with further material yet to be described attributed to the Gekkota. These discoveries are attributed to the use of fine mesh-aperture sieves and bulk screening of concentrated lenses of fossiliferous sediments.

a. Namba environment

The excavation and subsequent analyses of the sediment profile at both Billeroo Creek (BC2) and Lake Pinpa (LP12) has allowed an interpretation of palaeoenvironmental conditions in the Frome Basin during the Oligo-Miocene. Dolomite in freshwater conditions requires a flooded alkaline environment to precipitate, as does calcite [53], so the presence of these minerals in Layers 4–5 (absent in layers 2 and 3) are most likely the result of the seasonal fluctuations of the water level in the palaeo-lakes in which the sediments comprising the Namba Formation were deposited during the late Oligocene–early Miocene [see 40]. Fluctuations in the rainfall and evaporation contributed to the formation of palygorskite in the lower Layers 3–5. Smectite is the result of the weathering of minerals sourced from older rock outcrop

(most likely from the nearby Flinders, Barrier, and Olary Ranges) deposited into the Namba area as swampy soils on the lake edge. Layers where dolomite is absent, i.e. Layer 3 according to the XRD result, reflect a drier phase in the palaeo-environment without an alkaline waterbody at the surface.

Callen [40] noted that the Namba basin was once flooded, forming a larger palaeolake than the present Lake Frome. The palaeolake Namba supported a number of aquatic vertebrates including lungfish, *Obdurodon* platypus, crocodiles, cetaceans and various waterbirds [43, 54–56]; their combined presence is indicative of a deeper aquatic environment. The lake was subject to varying rainfall and evaporation rates, resulting in receding lake edges, creating cycles of dolomite precipitation (recorded in Layer 5). Callen [40] concluded that sufficient evaporation of surface water (conditions required for dolomite precipitation) could not occur in a consistently high-rainfall climate. Instead, a modern analogue might be where cool moist winters and hot dry summers result in a moderate seasonal rainfall and high evaporation rates that result in dolomite and calcite deposits such as in the Coorong lacustrine-estuarine system in southern South Australia [57]. There, lake levels rise after dry summers due to groundwater recharge, resulting in the dolomite precipitation.

The sedimentary evidence from palaeolake Namba reveal periods of evaporation and alkaline water bodies. This suggests the occurrence of periodic droughts during which fish die-offs might be expected in the evaporating water bodies and terrestrial fauna might die and accumulate along the lake-edge. The articulated skeletons found at LP12 and BC2 alongside concentrations of fish bone may be the result of these lake ‘deaths’. The remains of *Proegernia mikebulli* and other small terrestrial vertebrates are preserved in lenses of concentrated bones of aquatic vertebrates (fish, turtle, and crocodiles) within deposits reflecting these ephemeral lacustrine conditions. These lenses, as exemplified by the Fish Lens at BC2, and others exposed discontinuously at Lake Pinpa, were probably formed by littoral longshore currents disarticulating, mixing and concentrating bones. The material would not have to have been transported very far in order to mix animals from the two starkly different environments as the deposition site was in the littoral zone. Accumulation of small vertebrate material alongside associated larger vertebrate skeletons (at LP12) indicate that the transportation of material was not from a flash flood typical of tropical monsoon seasonality but rather accumulated along a shallow or ephemeral lake edge shoreline. Some small terrestrial vertebrate material is within a size range attributable to raptor predation [58, 59] and some specimens show corrosion thinning typical of that formed after raptor ingestion. Such material may be derived from raptor pellet accumulations washing into the lake where they disaggregated and mixed with the bones of the more common freshwater taxa. Undescribed accipitrids are known from the Pinpa Local Fauna and a skeleton of one was excavated from LP12 by Worthy/Camens-led expeditions, and is the subject of current research.

b. *Proegernia mikebulli* and crown group egeriines

Proegernia mikebulli was small and gracile compared to most living egeriines. Based on dentary length, an adult individual would be of a similar size to *Lissolepis coventryi*, around 100 mm SVL [60]. Examinations of tooth crown features across the type specimens reveals they share features with both *Lissolepis* and species of *Liopholis* (cf. *L. multiscutata* and *L. whitii* both of which have a cusp slightly posterior to the centre of the tooth crown). Tooth crown features of scincids potentially preserve phylogenetic and dietary signals. Detailed studies within and between the extant subfamilies may help elucidate any dietary preferences suggested by fossil dentition. Tooth striae are present on teeth in most egeriines with a specialised diet and no pattern in the number of striae between insectivorous, durophagous, or herbivorous species was obvious [see 23]. Within genera, the number and patterning of striae was fairly similar. *Cyclodomorphus* and *Tiliqua* increase the number of striae from other egeriines, partly due to the change in tooth shape leading to striae radiating on all sides from a central cusp. *Lissolepis* and *Liopholis* have fewer striae and all are restricted to the lingual face of the tooth crown, below the lingual cristae. The two fossil species of *Proegernia* have few striae, similar to *Lissolepis* and *Liopholis*. The buccal striae on the crowns of *Proegernia palankarinnensis* are unusual and were not observed on extant *Eutropis multiscutata*, *Sphenomorphus jobiensis* or *Eugongylus rufescens*. Examination of fine-scale tooth crown features may prove useful characters for morphological phylogenetic analyses of fragmentary or incomplete tooth-bearing scincid material in future.

From what can be deduced of the palaeoenvironment surrounding Lake Pinpa and Billeroo Creek in the late Oligocene–early Miocene, *Proegernia mikebulli* lived in a cool, moist climate region with seasonal hot, dry summers with high evaporation rates. Other squamates known from this region and time period include a small gekkotan, and possibly a constricting snake [43]. Although no evidence of *Proegernia palankarinnensis* was found in the Namba Formation sites, the presence of at least two Australian scincids, from neighbouring basins, covering similar time periods and environments may be an indication of the early diversity of Australian squamates. Extant Australian scincids are difficult to separate based on isolated cranial material. The same cryptic diversity may have been present in the Oligo-Miocene, so more than two species are likely present in both the Etadunna and Namba Formations.

Although neither analysis produced firm support for the phylogenetic position of *Proegernia mikebulli*, both parsimony and Bayesian analyses placed the new taxon outside of living (crown) Australian Egeriinae. *Proegernia* species have a larger tooth count than crown Australian egeriines, a splenial notch placed more anteriorly, and a ~1 mm long opening of Meckel’s groove anterior to the splenial notch. Each of these character states are representative of the predicted transitional form between the plesiomorphic outgroup morphology and the majority of crown Australian egeriines. These characters have pulled *Proegernia* outside of the Australian crown group, but not beyond the Sahul stem genera *Corucia* and *Tribolonotus*. However, their precise position outside this crown group is uncertain. If relationships of one or both *Proegernia* with *Corucia* are correct (Figure 13–Figure 14), then there were two migrations of Egeriinae into southern Sahul 40.47–32.7 Ma, or a single invasion followed by emigration of *Corucia* to the Solomon Islands. However, a large amount of missing data for these fossil taxa and low support for precise relationships means

that it is possible that these trees are wrong, and that homoplasy with *Corucia* is what pulls these taxa away from the crown Australian group. It is thus possible that *Proegermia* lies on the immediate stem to the Australian crown group, in which case only a single dispersal event to Australia need be assumed.

The tip-dated phylogeny inclusive of the two Oligo-Miocene *Proegermia* taxa retrieves the age of the Egeriinae as 50.12 Ma. The last common ancestor of Egeriinae and Lygosominae was mostly likely living in the far south-eastern edge of Asia during the early Eocene. The first egeriine evolved elsewhere when the continent of Sahul was far south of its current latitude [61]; there is no fossil evidence of their presence in Australia at this time. Crown Egeriinae most likely originated either in south-eastern Asia or on an island arc, before connecting with the Australian landmass in the late Eocene (32.7Ma). Similar origins have been hypothesised for agamids [22 Ma; 62], pythons [c. 35 Ma; 8], sphenomorphine skinks [12], and elapids [8].

9. Conclusion

Dated phylogenies with the new fossils suggest that egeriines are the oldest radiation of skinks in Australia. The diversification of the crown Australian egeriines (~33 Ma) occurred nearly 10 million years before the estimated diversification of other Australian skinks, i.e. the Sphenomorphinae (c. 25 Ma) and Eugongylinae [c. 20 Ma; 3]. However, their diversity lags behind these other, potentially more recent groups. The first egeriines would have shared the continent with pygopodid, diplodactylid and carphodactylid geckos, and madtsoiid snakes [5]. The search for more Paleogene lizard material to document the early evolution of Australia's diverse herpetofauna continues, with investigations of Eocene collections from Murgon in northern Queensland [63], and early Oligocene vertebrates from Pwerte Marnte Marnte [64].

Acknowledgments

Sue Double and Jenny Worthy for screening and sorting the fossil material. Rod Wells, Colin Doudy, Bob and Sue Tulloch, Amy Tschim, Warren Handley, Jacob Blokland, Carey Burke and Ellen Mather were all part of the collaborative field team. Dr Jason Gascooke for training on, and use of, the SEM for imaging and EDAX sedimentary analysis at Flinders Microscopy, an Australian Microscopy and Microanalysis Research Facility. We thank Sarah Harmer-Bassel and Jason Young of Flinders Analytical, for their ICPMS and X-ray diffraction (XRD) analyses of the sediment from the studied fossil sites. We thank Mary-Anne Binnie and Carolyn Kovach of the South Australian Museum, and Tim Ziegler from Museums Victoria, for access to collections, and loans of fossil and extant comparative material. The Willi Hennig Society for making TNT freely available. We especially thank Andrew Black (Blackie), manager, and Alec Wilson (prior owner) for access to Frome Downs Station that allowed this work.

Funding Statement

KMT was supported by an Australian Postgraduate Research Training Stipend. The Mark Mitchell Foundation funded part of the fieldwork component of this project, and SEM time.

Data Accessibility

All specimens are registered in the Palaeontology Collection of the South Australian Museum, MicroCT files are housed in the SAM Herpetology digital collection.

All executable files for phylogenetic analyses are available as electronic supplementary material.

Competing Interests

We have no competing interests.

Authors' Contributions

KMT conceived the project, conducted fieldwork, collected the data, performed analyses, produced all figures and wrote the manuscript. THW and ABC funded the project, conducted field work, contributed to discussions, and commented on multiple drafts of the manuscript; MNH and MSYL commented on multiple drafts of the manuscript and contributed to discussions. NB collected the stratigraphic samples and information as part of a separate honours project.

References

1. Hall, R. 2002 Cenozoic geological and plate tectonic evolution of SE Asia and the SW Pacific: computer-based reconstructions, model and animations. *Journal of Asian Earth Sciences*. **20**, 353–431.
2. White, M. E. 2006 Environments of the geological past. In *Evolution and biogeography of Australasian vertebrates*. (ed. ^eds. J. R. Merrick, Archer, A., Hickey, G. M., Lee, M. S. Y.), pp. 17–48. Oatlands, NSW: Auscipub.
3. Skinner, A., Hugall, A. F., Hutchinson, M. N. 2011 Lygosomine phylogeny and the origins of Australian scincid lizards. *Journal of Biogeography*. **38**, 1044–1058. (10.1111/j.1365-2699.2010.02471.x)

4. Oliver, P., Hugall, A. 2017 Phylogenetic evidence for mid-Cenozoic turnover of a diverse continental biota. *Nat Ecol Evol.* **1**, 1896–1902. (10.1038/s41559-017-0355-8)
5. Oliver, P. M., Sanders, K. L. 2009 Molecular evidence for Gondwanan origins of multiple lineages within a diverse Australasian gecko radiation. *Journal of Biogeography.* **36**, 2044–2055.
6. Vidal, N., Hedges, S. B. 2009 The molecular evolutionary tree of lizards, snakes, and amphisbaenians. *Comptes rendus biologiques.* **332**, 129–139.
7. Hugall, A. F., Lee, M. S. Y. 2004 Molecular claims of Gondwanan age for Australian agamid lizards are untenable. *Molecular Biology and Evolution.* **21**, 2102–2110. (10.1093/molbev/msh219)
8. Sanders, K. L., Lee, M. S. Y. 2008 Molecular evidence for a rapid late-Miocene radiation of Australasian venomous snakes (Elapidae, Colubroidea). *Molecular Phylogenetics and Evolution.* **46**, 1165–1173. (<http://dx.doi.org/10.1016/j.ympev.2007.11.013>)
9. Lambert, S. M., Reeder, T. W., Wiens, J. J. 2015 When do species-tree and concatenated estimates disagree? An empirical analysis with higher-level scincid lizard phylogeny. *Molecular phylogenetics and evolution.* **82**, 146–155.
10. Uetz, P., Freed, P., Hošek, J. e. The Reptile Database. 2019.
11. Skinner, A., Hutchinson, M. N., Lee, M. S. Y. 2013 Phylogeny and divergence times of Australian *Sphenomorphus* group skinks (Scincidae, Squamata). *Molecular Phylogenetics and Evolution.* **69**, 906–918. (<http://dx.doi.org/10.1016/j.ympev.2013.06.014>)
12. Rabosky, D. L., Donnellan, S. C., Talaba, A. L., Lovette, I. J. 2007 Exceptional among-lineage variation in diversification rates during the radiation of Australia's most diverse vertebrate clade. *Proceedings of the Royal Society of London B: Biological Sciences.* **274**, 2915–2923. (10.1098/rspb.2007.0924)
13. Gardner, M. G., Hugall, A. F., Donnellan, S. C., Hutchinson, M. N., Foster, R. 2008 Molecular systematics of social skinks: phylogeny and taxonomy of the *Egernia* group (Reptilia: Scincidae). *Zoological Journal of the Linnean Society.* **154**, 781–794. (10.1111/j.1096-3642.2008.00422.x)
14. Chapple, D. G., Ritchie, P. A., Daugherty, C. H. 2009 Origin, diversification, and systematics of the New Zealand skink fauna (Reptilia: Scincidae). *Molecular Phylogenetics and Evolution.* **52**, 470–487. (<https://doi.org/10.1016/j.ympev.2009.03.021>)
15. Thorn, K. M., Hutchinson, M. N., Archer, M., Lee, M. S. Y. 2019 A new scincid lizard from the Miocene of Northern Australia, and the evolutionary history of social skinks (Scincidae: Egerniinae). *Journal of Vertebrate Paleontology.* **39**, e1577873. (10.1080/02724634.2019.1577873)
16. Martin, J. E., Hutchinson, M. N., Meredith, R., Case, J. A., Pledge, N. S. 2004 The oldest genus of scincid lizard (Squamata) from the Tertiary Etadunna Formation of South Australia. *Journal of Herpetology.* **38**, 180–187. (10.1670/25-03A)
17. Hocknull, S. A. 2000 Remains of an Eocene skink from Queensland. *Alcheringa: An Australasian Journal of Palaeontology.* **24**, 63–64. (10.1080/03115510008619524)
18. Estes, R. 1984 Fish, amphibians and reptiles from the Etadunna Formation, Miocene of South Australia. *Australian Zoologist.* **21**, 335–343.
19. Woodburne, M. O., Macfadden, B. J., Case, J. A., Springer, M. S., Pledge, N. S., Power, J. D., Woodburne, J. M., Springer, K. B. 1994 Land mammal biostratigraphy and magnetostratigraphy of the Etadunna Formation (late Oligocene) of South Australia. *Journal of Vertebrate Paleontology.* **13**, 483–515.
20. Tedford, R. H., Archer, M., Bartholomai, A., Plane, M., Pledge, N. S., Rich, T., Rich, P., Wells, R. T. 1977 The discovery of Miocene vertebrates, Lake Frome area, South Australia. *BMR Journal of Australian Geology & Geophysics.* **2**, 53–57.
21. Evans, S. E. 2008 The skull of lizards and Tuatara. In *Biology of the Reptilia 20, Morphology H The skull of Lepidosauria.* (ed. ^eds. C. Gans, A. S. Gaunt, K. Adler), pp. 1–343. Ithaca, NY: Society for the Study of Amphibians and Reptiles.
22. Richter, A. 1994 Lacertilia aus der Unteren Kreide von Una und Galve (Spanien) und Anoual (Marokko). *Berliner geowissenschaftliche Abhandlungen (E: Paläobiologie).* **14**, 1–147.
23. Kosma, R. 2003 The dentitions of recent and fossil scinciform lizards (Lacertilia, Squamata) - Systematics, Functional Morphology, Paleocology. Hanover, Germany: University Hannover.
24. Russell, A. P., Bauer, A. M. 2008 The appendicular locomotor apparatus of *Sphenodon* and normal-limbed squamates. In *The skull and appendicular locomotor apparatus of Lepidosauria.* (ed. ^eds. C. Gans, A. S. Gaunt, K. Adler), pp. Ithaca, New York, USA: Society for the Study of Amphibians and Reptiles.
25. Hoffstetter, R., Gasc, J.-P. 1969 Vertebrae and ribs of modern reptiles. In *Biology of the Reptilia.* (ed. ^eds. C. Gans), pp. 201–310. New York: Academic Press.
26. Goloboff, P. A., Catalano, S. A. 2016 TNT version 1.5, including a full implementation of phylogenetic morphometrics. *Cladistics.* **32**, 221–238.
27. Drummond, A. J., Suchard, M. A., Xie, D., Rambaut, A. 2012 Bayesian phylogenetics with BEAUti and the BEAST 1.7. *Molecular Biology and Evolution.* **29**, 1969–1973.
28. Tonini, J. F. R., Beard, K. H., Ferreira, R. B., Jetz, W., Pyron, R. A. 2016 Fully-sampled phylogenies of squamates reveal evolutionary patterns in threat status. *Biological Conservation.* **204, Part A**, 23–31. (<https://doi.org/10.1016/j.biocon.2016.03.039>)
29. Lanfear, R., Frandsen, P. B., Wright, A. M., Senfeld, T., Calcott, B. 2016 PartitionFinder 2: new methods for selecting partitioned models of evolution for molecular and morphological phylogenetic analyses. *Molecular Biology and Evolution.* **34**, 772–773.
30. Maddison, W., Maddison, D. Mesquite: a modular system for evolutionary analysis. Version 3.2. 2017.
31. Lewis, P. O. 2001 A likelihood approach to estimating phylogeny from discrete morphological character data. *Systematic Biology.* **50**, 913–925.
32. Alekseyenko, A. V., Lee, C. J., Suchard, M. A. 2008 Wagner and Dollo: a stochastic duet by composing two parsimonious solos. *Systematic Biology.* **57**, 772–784.
33. Goloboff, P. A., Pittman, M., Pol, D., Xu, X. 2018 Morphological data sets fit a common mechanism much more poorly than DNA sequences and call into question the Mk model. *Systematic Biology.* **68**, 494–504. (10.1093/sysbio/syy077)
34. Wright, A. M., Hillis, D. M. 2014 Bayesian analysis using a simple likelihood model outperforms parsimony for estimation of phylogeny from discrete morphological data. *PLoS One.* **9**, e109210.
35. O'Reilly, J. E., Puttick, M. N., Parry, L., Tanner, A. R., Tarver, J. E., Fleming, J., Pisani, D., Donoghue, P. C. 2016 Bayesian methods outperform parsimony but at the expense of precision in the estimation of phylogeny from discrete morphological data. *Biology Letters.* **12**, 20160081.
36. Harmon, L. J. 2019 *Phylogenetic comparative methods.* 1.4 ed. Published Online: Luke Harmon.
37. Stadler, T. 2010 Sampling-through-time in birth–death trees. *Journal of Theoretical Biology.* **267**, 396–404. (<https://doi.org/10.1016/j.jtbi.2010.09.010>)
38. Drummond, A. J., Ho, S. Y. W., Phillips, M. J., Rambaut, A. 2006 Relaxed phylogenetics and dating with confidence. *PLOS Biology.* **4**, e88. (10.1371/journal.pbio.0040088)
39. Rambaut, A., Drummond, A. J., Xie, D., Baele, G., Suchard, M. A. 2018 Posterior summarisation in Bayesian phylogenetics using Tracer 1.7. *Systematic Biology.* **67**, 901–904. (doi:10.1093/sysbio/syy032)
40. Callen, R. A. 1977 Late Cainozoic environments of part of northeastern South Australia. *Journal of the Geological Society of Australia: An International Geoscience Journal of the Geological Society of Australia.* **24**, 151–169.
41. Rich, T. H. 1991 Monotremes, placentals, and marsupials: their record in Australia and its biases. In *Vertebrate Palaeontology of Australasia.* (ed. ^eds. P. Vickers-Rich, J. M. Monaghan, R. F. Baird, T. H. Rich), pp. 894–1057. Melbourne, Australia: Pioneer Design Studio Pty Ltd.
42. Rich, T. 1984 News from Foreign members: Australia, Museum of Victoria, Melbourne. *Society of Vertebrate Paleontology News Bulletin.* **130**, 28.
43. Vickers-Rich, P., Rich, P. V. 1991 *Vertebrate palaeontology of Australasia.* Melbourne, Australia: Pioneer Design Studio.
44. Callen, R. A., Tedford, R. H. 1976 New Late Cainozoic rock units and depositional environments, Lake Frome area, South Australia. *Transactions of the Royal Society of South Australia.* **100**, 125–167.
45. Centore, P. Conversions between the Munsell and sRGB colour systems.

- <http://www.munsellcoloursceinceforpainters.com/> 2013:64.
46. Opperl, M. 1811 *Die Ordnungen, Familien und Gattungen der Reptilien, als Prodrom einer Naturgeschichte derselben*. München: Joseph Lindauer.
47. Gray, J. E. 1825 A synopsis of the genera of reptiles and Amphibia, with a description of some new species. *Annals of Philosophy*. **10**, 193–217.
48. Welch, K. 1982 Herpetology of the Old World II. Preliminary comments on the classification of skinks (Family Scincidae) with specific reference to those genera found in Africa, Europe, and southwest Asia. *Herpetile*. **7**, 25–27.
49. Megirian, D., Prideaux, G., Murray, P., Smit, N. 2010 An Australian land mammal age biochronological scheme. *Paleobiology*. **36**, 658–671. (10.1666/09047.1)
50. Gelnaw, W. B. 2011 On the cranial osteology of *Eremiascincus* and its use for identification: East Tennessee State University.
51. Lee, M. S. Y., Hutchinson, M. N., Worthy, T. H., Archer, M., Tennyson, A. J. D., Worthy, J. P., Scofield, R. P. 2009 Miocene skinks and geckos reveal long-term conservatism of New Zealand's lizard fauna. *Biology Letters*. **5**, 833–837. (doi:10.1098/rsbl.2009.0440)
52. Greer, A. E. 1989 *The biology and evolution of Australian lizards*. Chipping Norton, NSW: Surrey Beatty & Sons.
53. Fukushi, K., Matsumiya, H. 2018 Control of Water Chemistry in Alkaline Lakes: Solubility of Monohydrocalcite and Amorphous Magnesium Carbonate in CaCl₂–MgCl₂–Na₂CO₃ Solutions. *ACS Earth and Space Chemistry*. **2**, 735–744. (10.1021/acsearthspacechem.8b00046)
54. Worthy, T. H. 2009 Descriptions and phylogenetic relationships of two new genera and four new species of Oligo-Miocene waterfowl (Aves: Anatidae) from Australia. *Zoological Journal of the Linnean Society*. **156**, 411–454. (10.1111/j.1096-3642.2008.00483.x)
55. Worthy, T. H. 2011 Descriptions and phylogenetic relationships of a new genus and two new species of Oligo-Miocene cormorants (Aves: Phalacrocoracidae) from Australia. *Zoological Journal of the Linnean Society*. **163**, 277–314. (10.1111/j.1096-3642.2011.00693.x)
56. Fitzgerald, E. M. G. 2004 A review of the Tertiary fossil Cetacea (Mammalia) localities in Australia. *Memoirs of Museum Victoria*. **61**, 183–208.
57. Borch, C. C. V. D., Lock, D. 1979 Geological significance of Coorong dolomites. *Sedimentology*. **26**, 813–824. (10.1111/j.1365-3091.1979.tb00974.x)
58. Andrews, P. 1990 *Owls, caves and fossils*. London: The Natural History Museum.
59. McDowell, M. C., Medlin, G. C. 2009 The effects of drought on prey selection of the barn owl (*Tyto alba*) in the Strzelecki Regional Reserve, north-eastern South Australia. *Australian Mammalogy*. **31**, 47–55. (<http://dx.doi.org/10.1071/AM08115>)
60. Wilson, S., Swan, G. 2017 *A complete guide to the Reptiles of Australia*. 5th ed. Sydney, Australia: Reed New Holland Publishers.
61. Müller, R. D., Seton, M., Zahirovic, S., Williams, S. E., Matthews, K. J., Wright, N. M., Shephard, G. E., Maloney, K. T., Barnett-Moore, N., Hosseinpour, M., et al. 2016 Ocean basin evolution and global-scale plate reorganization events since Pangea breakup. *Annual Review of Earth and Planetary Sciences*. **44**, 107–138. (10.1146/annurev-earth-060115-012211)
62. Hugall, A. F., Foster, R., Hutchinson, M., Lee, M. S. Y. 2008 Phylogeny of Australasian agamid lizards based on nuclear and mitochondrial genes: implications for morphological evolution and biogeography. *Biological Journal of the Linnean Society*. **93**, 343–358. (10.1111/j.1095-8312.2007.00911.x)
63. Godthelp, H., Archer, M., Cifelli, R., Hand, S. J., Gilkeson, C. F. 1992 Earliest known Australian Tertiary mammal fauna. *Nature*. **356**, 514–516. (10.1038/356514a0)
64. Murray, P. F., Megirian, D. 2006 The Pwerte Marnte Marnte Local Fauna: a new vertebrate assemblage of presumed Oligocene age from the Northern Territory of Australia. *Alcheringa: An Australasian Journal of Palaeontology*. **30**, 211–228. (10.1080/03115510609506864)
65. Woodhead, J., Hand, S. J., Archer, M., Graham, I., Sniderman, K., Arena, D. A., Black, K. H., Godthelp, H., Creaser, P., Price, E. 2016 Developing a radiometrically-dated chronologic sequence for Neogene biotic change in Australia, from the Riversleigh World Heritage Area of Queensland. *Gondwana Research*. **29**, 153–167. (<http://dx.doi.org/10.1016/j.gr.2014.10.004>)

Tables

Table 1: Fossil calibrations used, their minimum and maximum ages (Ma) and references for the age dates or species descriptions.

Fossil calibration	Min age	Max age	References for age
Zone A, Etadunna Formation	25.5	25.7	Woodburne MO, Macfadden BJ, Case JA, Springer MS, Pledge NS, Power JDet al. [19]
Pinpa Local Fauna, Namba Formation	25.5	25.7	Woodburne MO, Macfadden BJ, Case JA, Springer MS, Pledge NS, Power JDet al. [19]
AL 90 Locality, Carl Creek Limestone	14.17	15.11	Woodhead J, Hand SJ, Archer M, Graham I, Sniderman K, Arena DA, Black KH, Godthelp H, Creaser PPrice E [65]
Gag Locality, Carl Creek Limestone	14.47	16.86	Woodhead J, Hand SJ, Archer M, Graham I, Sniderman K, Arena DAet al. [65]

Table 2: Major and trace elements detected by X-ray fluorescence analysis of sediments sampled Layer 3 and 5 from stratigraphic sections of Lake Pinpa Site 12 and Billeroo Creek Site 2. Values are %, minor trace elements making the total of each sample 100 are not shown.

	MgO	Al ₂ O ₃	SiO ₂	CaO	MnO	Fe ₂ O ₃
Pinpa 12 L3	4.043	8.815	46.53	0.238	0.2303	19.82
Pinpa 12 L5	10.45	7.444	43.64	7.901	4.227	8.105
Billeroo Creek L5	13.65	5.875	36.08	13.02	1.634	6.982

Table 3: Mineral components of sediment samples taken from Layers 3 and 5 at Lake Pinpa and Layer 5 at Billeroo Creek Site 2.

	Clay Minerals	Dolomite/ Ankerite	Goethite	Quartz	Other	Total
Pinpa 12 L3	79	-	9	11	1	100
Pinpa 12 L5	66	23	3	6	2	100
Billeroo Creek L5	54	38	2	5	1	100

Figure and table captions

Figure 1: Stratigraphic sections through the Namba Formation at Lake Pinpa at Site 12 (see Figure 2) and Billeroo Creek, Site 2, constructed in SedLog 3.1 from authors field observations in comparison with RH Tedford's sections described in notes and figures in [20, 44], depth in meters. Grain sizes were determined in the field. Colours are converted from Munsell figures to corresponding RGB values using [45].

Figure 2: Location maps of the Lake Pinpa and Billeroo Creek sites, Frome Downs Station, South Australia. Stratigraphic columns in Figure 1 were compiled from locations Site 12 (=LP12) and Site 2 Fish lens (=BC2) marked above.

Figure 3: Results of the soluble mineral analyses of Layers 2–4 and the white and black sediments from Layer 5 at Lake Pinpa Site 12 (left); and Billeroo Creek Site 2 Layers 2–5 (right).

Figure 4: 1a-c Holotype, a right dentary, SAM P57502, 2a-c paratype; a left dentary SAM P57499; two right post-dentary compound bones 3a-c SAM P57543 and 4a-c SAM P57542 of *Proegernia mikebulli* sp. nov. from the Fish Lens at BC2; and 5 complete reconstruction in medial view of the right mandible of *Proegernia mikebulli*, the coronoid and splenial (hatched area) are reconstructed based on modern egermiine specimens, no fossil representatives of these elements are known. Abbreviations: adf, adductor fossa; anf, angular facet; art., articular; d., dentary; gl, glenoid fossa; iaf, inferior alveolar foramen; mg, Meckel's groove; psf, posterior surangular foramen; rap, retroarticular process; san., surangular; and sy, symphysis.

Figure 5: Right maxilla of *Proegernia mikebulli* sp. nov. reconstructed from SAM P57541 (anterior fragment; top left) and SAM P57503 (near complete; top right). 'A' marks the same tooth locus on each specimen. Enclosed by the box is the tooth 'B'. Abbreviations: cl, *cuspis labialis*; cla, *culmen lateralis* anterior; cli, *cuspis lingualis*; clp, *culmen lateralis* posterior; fp, facial process; psm, palatine shelf of the maxilla; pxp, premaxillary process; sop, suborbital process; and str, striae.

Figure 6: Left premaxilla (SAM P57542) of *Proegernia mikebulli* sp. nov. from posterior (left), anterior (centre), and lateral (right) view. A, tooth one, enlarged in box. Abbreviations: flc, foramen of longitudinal canal; inp, internasal process; mp, maxillary process; nef, notch of the ethmoidal foramen; and o, osteoderm.

Figure 7: A and B: 24th tooth on the dentary SAM P57502, medial view, and C the 16th tooth in occlusal view with lateral and medial facies marked. Abbreviations: ai, *antrum intercristatum*; cl, *crista lingualis*; cm, *crista mesialis*; cul, *cuspis labialis*; cula, *culmen lateralis* anterior; and culp, *culmen lateralis* posterior.

Figure 8: Right pterygoid of *Proegernia mikebulli* sp. nov., SAM P57512 in dorsal (left) and ventral (right) views. Abbreviations: en, epipterygoid notch; pp, palatine process; pvc pterygoid ventral concavity; qp, quadrate process; and tp, ectopterygoid process.

Figure 9: A single vertebra (SAM P57514) potentially referable to *Proegernia mikebulli* sp. nov., recovered from Fish Lens, BC2. A, anterior view; B, posterior view; C, lateral view, anterior to the left; D, dorsal view; and E, ventral view. A proximal fragment of a right femur assigned to *Proegernia mikebulli* sp. nov. SAM P57540. F, dorsal; G, anterior; and H, ventral views.

Figure 10: Comparative SEM of *Proegernia mikebulli* sp. nov. (B,C) and microphotographs of *P. palankarinnensis* (A,D) from Martin et al. [16], in medial (above) and occlusal (below) views. Note the anterior extension of the splenial notch (3) in *P. palankarinnensis* and the increased the number of teeth/loci in *P. mikebulli*. Features described in text are labelled: 1) Medial inflection of the symphysis; 2) Dentary depth; 3) Position of the inferior alveolar foramen; 4) Meckel's Groove extension anterior of the inferior alveolar foramen, present in *P. palankarinnensis* and absent in *P. mikebulli*; 5) anterior extent of the internal septum similar in both taxa, 6) Overall robustness as dentary width, 7) facet for articulation with anterior ramus of coronoid, 8) splenial notch height, and 9) height of the coronoid process relative to the last tooth.

Figure 11: Comparative medial views of the dentaries of *Eutropis multifasciata* (SAMA R35693), *Proegernia mikebulli* sp. nov., *Lissolepis coventryi* (SAMA R57317) and *Liopholis multiscutata* (FUR168). 1 Anterior extent of splenial; 2 Symphysis shape and robustness; and 3 Dentary depth at mid-tooth row length.

Figure 12: Medial view of left dentary tooth crowns of *Plestiodon fasciatus* (SAMA R66784), *Eutropis multifasciata* (SAMA R35693), *Proegernia mikebulli* sp. nov. (SAM P57502), *Lissolepis coventryi* (SAMA R57317), *Eugongylus rufescens* (SAMA R36735) and *Sphenomorphus jobiensis* (SAMA R6736). *Plestiodon* represents the basal skink tooth condition as described by Richter (1994). *Culmen lateralis* posterior is labelled culp.

Figure 13: Egermiine phylogeny based on the single most parsimonious tree produced by TNT [26]. Bootstrap values >50 shown.

Figure 14: Egermiine phylogeny based on consensus total evidence Bayesian inference tree from BEAST. †denotes extinct taxon. Small numbers at nodes denote posterior probability. Reconstructed node ages for key clades are in brackets, the green shading denotes the Eocene, after which Australia became isolated from Gondwana [4].

1 Table 1: Fossil calibrations used, their minimum and maximum ages (Ma) and references for the age dates or
2 species descriptions.

3 Table 2: Major and trace elements detected by X-ray fluorescence analysis of sediments sampled Layer 3 and
4 5 from stratigraphic sections of Lake Pinpa Site 12 and Billeroo Creek Site 2. Values are %, minor trace
5 elements making the total of each sample 100 are not shown.

6 Table 3: Mineral components of sediment samples taken from Layers 3 and 5 at Lake Pinpa and Layer 5 at
7 Billeroo Creek Site 2.

8
9
10
11
12
13
14
15
16
17
18
19
20
21
22
23
24
25
26
27
28
29
30
31
32
33
34
35
36
37
38
39
40
41
42
43
44
45
46
47
48
49
50
51
52
53
54
55
56
57
58
59
60

Appendix B**ROYAL SOCIETY
OPEN SCIENCE****A new species of *Proegernia* from the Namba Formation in
South Australia and the early evolution of Australian
egerniine skinks**

Journal:	Royal Society Open Science
Manuscript ID	RSOS-201686
Article Type:	Research
Date Submitted by the Author:	20-Sep-2020
Complete List of Authors:	Thorn, Kailah; Flinders University College of Science and Engineering, ; South Australian Museum Hutchinson, Mark; South Australian Museum Lee, Michael; Flinders University, College of Science and Engineering; South Australian Museum Brown, Nathan; Flinders University, College of Science and Engineering Camens, Aaron; Flinders University, College of Science and Engineering Worthy, Trevor Henry; Flinders University, College of Science and Engineering
Subject:	Palaeontology < EARTH SCIENCES, evolution < BIOLOGY, taxonomy and systematics < BIOLOGY
Keywords:	Scincidae, Egerniinae, Oligocene, Miocene, Palaeontology, Namba
Subject Category:	Organismal and Evolutionary Biology

Author-supplied statements

Relevant information will appear here if provided.

Ethics

Does your article include research that required ethical approval or permits?:

This article does not present research with ethical considerations

Statement (if applicable):

CUST_IF_YES_ETHICS :No data available.

Data

It is a condition of publication that data, code and materials supporting your paper are made publicly available. Does your paper present new data?:

Yes

Statement (if applicable):

All specimens are registered in the Palaeontology Collection of the South Australian Museum, MicroCT files are housed in the SAM Herpetology digital collection.

All executable files for phylogenetic analyses are available as electronic supplementary material.

Conflict of interest

I/We declare we have no competing interests

Statement (if applicable):

CUST_STATE_CONFLICT :No data available.

Authors' contributions

This paper has multiple authors and our individual contributions were as below

Statement (if applicable):

KMT conceived the project, conducted fieldwork, collected the data, performed analyses, produced all figures and wrote the manuscript. THW and ABC funded the project, conducted field work, contributed to discussions, and commented on multiple drafts of the manuscript; MNH and MSYL commented on multiple drafts of the manuscript and contributed to discussions. NB collected the stratigraphic samples and information as part of a separate honours project.

Origins of Australian egeriinesROYAL SOCIETY
OPEN SCIENCE*R. Soc. open sci.*
doi:10.1098/not yet assigned**A new species of *Proegeria* from the Namba Formation in South Australia and the early evolution of Australian egeriine skinks****K. M. Thorn^{*1,2}, M. H. Hutchinson^{1,2}, M. S. Y. Lee^{1,2}, N. Brown¹, A. B. Camens¹ and T. H. Worthy¹***College of Science and Engineering, Flinders University, BEDFORD PARK, 5042, South Australia
South Australian Museum, North Terrace, ADELAIDE, 5000, South Australia***Keywords:** Scincidae, Egeriinae, Oligocene, Miocene, Palaeontology

ZOOBANK ID urn:lsid:zoobank.org:pub:9C892991-E95A-4ABA-93B9-9480308A7E73

1. Summary

New Oligo-Miocene fossil vertebrates from the Namba Formation (south of Lake Frome, South Australia) were uncovered from multiple expeditions from 2007–2018. Abundant disarticulated material of small vertebrates was concentrated in shallow lenses along the palaeo-lake edges, and is now exposed on the western shore. This fossiliferous deposit, also known from Billeroo Creek 2 km northeast of Lake Pinpa, includes abundant aquatic (such as fish, platypus *Obdurodon*, and waterfowl) and diverse terrestrial (such as possums, dasyuromorphs, and scincids) vertebrates and is hereafter recognised as the Fish Lens. The stratigraphic provenance of these deposits in relation to prior finds in the area is also established. A new egeriine scincid taxon *Proegeria mikebulli* sp. nov. described herein, is based on a near-complete reconstructed mandible, maxilla, premaxilla, and pterygoid. Postcranial scincid elements were also recovered with this material, but could not yet be confidently associated with *P. mikebulli*. This new taxon is recovered as the sister species to *P. palankarinnensis*, in a tip-dated total-evidence phylogenetic analysis, where both are recovered as stem Australian egeriines. These taxa also help pinpoint the timing of the arrival of scincids to Australia, with egeriines the first radiation to reach the continent.

2. Introduction

Major biogeographical events have shaped the Australian flora and fauna, from the separation from Antarctica ~45 million years ago, through to Pleistocene glacial cycles [1, 2]. Australia's herpetofauna can be broadly traced to two origins: relicts from the breakup of Gondwana (the southern supercontinent), or recent arrivals from Asia to Sahul (the continental mass including Australia and New Guinea) [3, 4]. Gondwanan origins are inferred for diplodactyloid geckos, chelid turtles and some frogs [5, 6]. The Asian route appears to have been taken by agamid lizards, and elapids and typhlopids sometime during the Oligo-Miocene [3, 7, 8]. The temporal origins of Australian scincid lizard radiations are still relatively poorly time-constrained, but molecular data indicate that their closest extant relatives are in south-east Asia [9].

The Australian scincids comprise three subfamilies: the Egeriinae Welch, 1982, Sphenomorphinae Welch, 1982 and Eugongylineae Welch, 1982 [10]. While these three clades have been included in large all-squamate phylogenies, detailed molecular phylogenetic analyses have also been conducted for the sphenomorphine [11, 12] and egeriine radiations [13], but no recent molecular phylogeny of the Australian eugongyline skinks has been published [see 14, which includes some Australian taxa]. A molecular phylogenetic analysis of Australian skinks [3] produced a node-calibrated molecular clock estimate of the divergence of all three subfamilies. Those results suggested that the Australian Egeriinae arrived 18.2 Ma, Eugongylineae 22.9 Ma, and the Sphenomorphinae were the first group to reach Australia 25.3 Ma. However, analyses using mid-Miocene fossil calibrations established a much earlier origin for the Australian

*Author for correspondence (Kailah.thorn@uwa.edu.au).

†Present address: Edward de Courcy Clarke Earth Science Museum, School of Earth Sciences M004, University of Western Australia, 35 Stirling Highway CRAWLEY 6009 Australia

egerniines, minimally 34.61 Ma [15]. This new date prompted the current investigation to seek fossil evidence that might further demarcate the timing of the arrival of (potentially) Australia's first scincids.

The oldest Australian fossil with distinctive scincid characters, the egeriine *Proegernia palankarinnensis* Martin et al., 2004, is from the Etadunna Formation, Lake Eyre Basin, at the Oligo-Miocene boundary 25–26 Ma [16]. An Eocene femur loosely assigned to Scincomorpha by Hocknull is not convincingly scincid [17], so is not robust evidence for inferring the temporal origin of the Australian Scincidae. Other Oligo-Miocene Australian material from the Etadunna Formation was referred to Egeriinae but remains undescribed including: a dentary (UCR 20814), broken parietal (UCR 20815), partial maxilla (UCR 20816), an assortment of vertebrae and a broken scapulocoracoid [18]. Both the *Proegernia palankarinnensis* holotype and the material referred by Estes [18] cannot currently be located. New collections of Oligo-Miocene material are required to resolve the composition and relationships of Australia's oldest scincid faunas. Recent expeditions into central South Australia have unearthed new Oligo-Miocene fossil squamates from the Namba Formation at Lake Pinpa and Billeroo Creek, southeast of Lake Frome. The Namba Formation is of a similar age to the Etadunna [19], and the vertebrate fossil material collected from the lowest layers of this Formation are termed the Pinpa Fauna [20]. This investigation revises the stratigraphic and palaeoecological setting of this deposit, and places the new Oligo-Miocene egeriine fossils into phylogenetic context, alongside *P. palankarinnensis* from the Etadunna, and the Miocene species of *Egernia* and *Tiliqua* analysed in Thorn et al. [15], to infer a more robust date for the arrival of scincids to the continent of Sahul.

3. Materials and Methods

a. Stratigraphy, excavation and fossil collection

Sediment samples from excavated trenches were taken to examine the stratigraphy of both Billeroo Creek (BC2) and Site 12, Lake Pinpa (LP12). Mineral composition was determined by XRD (X-ray diffraction) and XRF (X-ray fluorescence) analyses conducted by Flinders Analytical. Nested sieves with mesh apertures of 6, 3 and 1 mm were used to create sediment fractions from which the fossil material was sorted from the dolomitic clays. The micro-vertebrate material was sorted into fish, mammal, bird, turtle, crocodile, frog, and squamate. Of the 48 squamate cranial specimens recovered, 43 were scincids, and 5 identified as geckos (see Supplementary Information for the squamate specimen list). Most fossils discussed in this investigation were recovered by sieving excavated sediment and sorting the concentrates, however some were collected from the ground surface having been exposed by erosion.

All descriptive terminology of the cranial elements follows Evans [21] and Richter [22] and Kosma [23] for tooth crown features. Appendicular skeleton terminology follows Russell and Bauer [24], and Hoffstetter and Gasc [25] for vertebrae.

b. Scanning electron microscopy

Cranial material identified to Scincidae was further cleaned in water, dried, and then imaged using Scanning Electron Microscopy (SEM). The minute size of some specimens is beyond the capabilities of Micro CT for resolution of morphological features required for identification and descriptions. No specimens required sputter coating. All SEM work was conducted at Flinders University Microscopy facilities using an Inspect FEI F50 SEM. Maximum voltage used for image taking was limited to 2kV and a spot size of 3–4. Measurements of tooth crowns and features of the dentary mentioned in the text were taken using the SEM software.

c. Phylogenetic analyses

In order to better understand the timing of the Australian colonisation by the Egeriinae, both molecular and morphological data (including fossils) are required to generate tip-dated phylogenies. Undated parsimony and tip-dated Bayesian analyses infer, respectively, the phylogeny with the least homoplasy, and the most probable dated phylogeny.

i. Morphological characters

Morphological characters used in the following analyses consisted of 102 discrete and 48 continuous traits, forming an expanded matrix from Thorn et al. [15]. Continuous characters, derived from the measurements of the individual bones or teeth from the dentaries and maxillae, were taken from either Micro-CT scan data in Avizo Lite (v. 9.0) or SEM at Flinders Microscopy, to the nearest micrometre, or with digital callipers to the nearest ten micrometres. All measurements were converted to ratios of either dentary or maxilla length to standardise for size. Continuous characters were converted to values spanning 0–2 to replicate the average number of discrete character states, for analyses in both TNT [26] and BEAST 1.8.3 [27], so that they do not have a disproportionate weight.

ii. Molecular partitions

Molecular data sourced from Tonini et al. [28] and Gardner et al. [13] were analysed using Partition Finder 2 [29] to find optimal partitions and substitution models. The same six molecular (gene) partitions, 12s (412 base pairs [bps]), 16s (681 bps), ND4 (693 bps), BDNF (699 bps), CMOS (835 bps) and B-fibrinogen (1051 alignable bps) and substitution models [15] are used again here.

iii. Maximum Parsimony

The parsimony analyses for the combined discrete morphological, continuous morphological, and molecular data were performed using TNT v.1.5 [26]. *Eutropis multifasciata* was set as the most distant outgroup following the

1 phylogenetic interpretations of Gardner et al. [13] and Thorn et al. [15]. The most parsimonious tree (MPT) for the
2 combined data was found using 1000 replicates of tree-bisection-reconnection (TBR) with up to 1000000 trees held.
3 To assess clade support, 200 partitioned bootstrap replicates (with discrete characters, continuous characters, and each
4 gene locus treated as a separate resampling partition), were performed using TNT, using new search methods (XMULT)
5 with 1000 replicates and 1000000 trees held. The MPT and bootstrap trees from TNT were exported in nexus format, and
6 continuous and discrete characters were traced [in Mesquite; 30]. The executable files for finding the Most Parsimonious
7 tree, and for performing 200 reps of Partitioned Bootstrap resamples can be found in the SI data files
8 Namba_Egeriines_Topology.tnt (MPT file) and Namba_Egeriines_PartitionedBootstrap.tnt.
9

10 iv. Bayesian analysis

11 The discrete and continuous morphological data, and molecular data were simultaneously analysed in BEAST
12 v1.8.4 using tip-dated Bayesian approaches [27]. *Eutropis multifasciata* was again set as the furthest outgroup.
13 Polymorphic discrete morphological data were treated exactly as coded rather than as unknown, i.e. if coded as states
14 (0,1) it was treated as 0 or 1, but not 2. The discrete character set was analysed using the Mkv-model with correction for
15 non-sampling of constant characters [31, 32]. Despite recent disputes over the effectiveness of this model [33], it is well-
16 tested [34, 35] and is still widely accepted and applied to morphological data [36]. Continuous characters, transformed to
17 span values between 0 and 2, were analysed with the Brownian motion model. Bayes factors were used to test the need to
18 accommodate among-character rate variability for both discrete and continuous morphological characters (i.e. gamma
19 parameter).

20 The stratigraphic data used for tip-dating analyses were derived from fossil taxa and their associated stratigraphy
21 noted in Table 1. No node age constraints were imposed in this analysis, all dates are retrieved from the morphological
22 and stratigraphic age ranges from the noted fossil taxa (tips). The most appropriate available model in BEAST v.1.8.4,
23 birth-death serial sampling [37], was applied. An uncorrelated relaxed clock [38] was separately applied to the molecular
24 and morphological data.

25 Each Bayesian analysis was run for 100,000,000 generations with a burn-in of 20%. The analysis was conducted
26 four times to confirm stationarity. The post-burnin samples of all four runs were examined in Tracer 1.7.1 [39] to ensure
27 convergence was achieved. All four runs were combined in LogCombiner, and the consensus tree produced by
28 TreeAnnotator [27]. The executable .xml file for BEAST, all output log files, and the final consensus tree file (.tree) are
29 available as supplementary information.
30

31 4. Geological setting

32 a. The Namba Formation

33 The Namba Formation from north-eastern South Australia (see Figure 2) shares an unconformable lower
34 boundary with the Eyre Formation (Paleocene-Eocene) within our study area in the Frome Basin; and is unconformably
35 overlain by the Pleistocene Eurinilla Formation (Figure 1). The Namba sequence is a lateral equivalent to the Etadunna
36 Formation from the north western Lake Eyre Basin [19, 40]. The Namba Formation is divided into two members [20];
37 Green claystones and dolomitic claystones at the top of the lower member host a locally abundant vertebrate fauna,
38 termed the Pinpa Local Fauna [20]. This is biostratigraphically correlated with Zone A of the Etadunna Formation to be
39 25.5–25.7 Ma [19]. Sites visited during the late 1970s–early 1980s led by R.H. Tedford, T. H. Rich, and others,
40 discovered and named multiple fossil localities exposed in Lakes Pinpa, Namba, Tarkarooloo, Yanda and Tinko and
41 Billeroo Creek in the Lake Frome Basin [41]; added to these are new numbered sites from expeditions carried out in 2007
42 and between 2015–2018 led by T. H. Worthy and A. B. Camens.
43

44 All of the sites from Lake Pinpa and Billeroo Creek yielding scincid material for this investigation expose and
45 sample fluvio-lacustrine sites from the Namba Formation. Five expeditions collected new material from the Namba
46 Formation from 2007–2018. Over this period, numerous squamate fragments were collected from multiple sites at Lake
47 Pinpa and Billeroo Creek. Two sites have yielded the majority of the material described herein (see section 4b below);
48 Site 12 at Lake Pinpa, and the ‘Fish Lens’ at Billeroo Creek (within Wells’ Bog Site of Tedford’s, and later T.H. Rich’s,
49 expeditions).

50 Deposits and the fossils therein contributing to the Pinpa Local Fauna may be classed into two clear taphonomic
51 groups: those containing isolated bones or partial skeletons of terrestrial vertebrates (marsupials, predatory birds, wading
52 birds, and meiolaniid turtles), and those containing localized concentrated bone accumulations which were previously
53 thought to be derived from crocodile coprolites, encompassing mostly aquatic vertebrates (mostly fish, but including also
54 turtle, crocodile, dolphin and rare terrestrial vertebrates). Collection of both categories of this material was predominantly
55 from surface exposures along the western edge of Lake Pinpa and north eastern Billeroo Creek [41], with the occasional
56 excavation of articulated or associated marsupial skeletons. In slightly younger overlying/incised fluvial units exposed
57 as channel fill deposits, the Ericmas Local Fauna has been derived from large scale excavations at Ericmas and South
58 Prospect quarries (Lake Namba, 5 km south of Lake Pinpa) in the 1970s and 1980s [41]. Tom O’s Quarries excavated in
59 fluvial units at Lake Tarkarooloo unearthed the Tarkarooloo Local Fauna which is biochronologically similar to the
60 Ericmas LF, by bulk processing and screening sediments on several expeditions led by T H Rich of Museums Victoria,
once with the help of the Australian Army [42]. The quarried Ericmas LF sites contained predominantly terrestrial
vertebrate remains, but until the 2007 Worthy and Camens expedition, no squamate material was recorded from the
Namba Formation.

Squamate remains have predominantly been recovered alongside the smallest mammal taxa and were found in lenses of densely concentrated bones of aquatic vertebrates dominated by fish (*Actinopterygii* and *Neoceratodus* spp.). These fish lenses lie a few centimetres above dolomite of unknown depth (details below). The dolomite beds have revealed numerous fossils, including associated or partly articulated skeletons of birds and mammals, but fish bones are rare. This dolomitic bed was better exposed towards the middle of the lake bed in the 1970s when it was extensively sampled, but has since at least 2007 been buried by in-washed Quaternary sands. Both layers contain the same mammal and bird species and so the faunas from each are collectively referred to the Pinpa Local Fauna [20, 43]. Both the fish layer and underlying dolomites are exposed at Wells' Bog Site in Billeroo Creek shown in Figure 1.

Figure 1: Stratigraphic sections through the Namba Formation at Lake Pinpa at Site 12 (see Figure 2) and Billeroo Creek, Site 2, constructed in SedLog 3.1 from authors field observations in comparison with RH Tedford's sections described in notes and figures in [20, 44]. depth in meters. Grain sizes were determined in the field. Colours are converted from Munsell figures to corresponding RGB values using [45].

b. Fossil sites

Fossil sites in the Lake Pinpa–Billeroo Creek area were numbered chronologically on the 2007 trip in order of discovery, not based on geographical location (Figure 2).

i. Lake Pinpa

Site 6 (LP6) SAMA P43058 a posterior fragment of a left scincid maxilla, was recovered from Site 6 on the 2007 expedition. The fish lens was observed eroding out in patches over a broad area (200 m by 50 m) here around the margins of the overlying massive grey clay (Layer 2 on Figure 1).

Site 9 (LP9) SAMA P43057 a posterior fragment of a left scincid dentary, was recovered from Site 9 on the 2007 expedition. Here the fish lens was exposed in a narrow zone at the base and edge of eroding massive clay Layer 2.

Site 12 (LP12) Specimens SAM P57544 (partial humerus) and SAM P57545 (maxilla fragment) were found at Site 12 on the edge of Lake Pinpa. Bone was found eroding out on the surface, along the lake edge, from the lowest silty clay layer. Associated and articulated skeletons occur in both this layer and the dolomitic clay (Layer 5) beneath. An undulating clean erosional boundary was observed between these two units.

ii. Billeroo Creek Site 2 Fish Lens

Billeroo Creek Site 2 (BC2) is located on the northern side of the creek (Figure 2) and is a part of the more expansive Wells' Bog Site. Fossils derive from a concentrated lens of predominantly fish bone with limonite inclusions, within the top of Layer 3 of the Namba Formation (Figure 1). The base of this fossiliferous 'Fish Lens' is not flat, but undulated relating to depth of semi-discrete lenses exposed over an area of roughly 10 m by 3 m, excavated over three trips, with a maximum thickness of ~150 mm. The lens sits stratigraphically ~200 mm above the basal dolomite (Layer 5). This layer is laterally more extensive than the Fish lens at Billeroo Creek, and is the layer from which the majority of skeletal fossils at the site have been collected, see SI for a complete list.

Figure 2: Location maps of the Lake Pinpa and Billeroo Creek sites, Frome Downs Station, South Australia. Stratigraphic columns in Figure 1 were compiled from locations Site 12 (=LP12) and Site 2 Fish lens (=BC2) marked above.

c. Stratigraphy

Need an introductory sentence or two

Layer 1— The top layer of the stratigraphic section at LP12 (Figure 1), designated Layer 1, is not part of the Namba Formation. The red sands are reworked sediment from the nearby Quaternary dunes that unconformably overlie the Namba exposures around many of the lakes in the area. At Billeroo Creek (BC2), the fluvial Eurinilla Formation lies unconformably on the Namba Formation and eroded sediments from both it and overlying dunes mantled the Namba Formation where the section was excavated.

Layer 2— Erosion of an unknown amount of the upper part of Layer 2 means its original depth at both BC2 and LP12 cannot be assessed, but near LP07, exposures as documented in the section by Tedford et al. [20], show it was minimally ~6 m thick. It sits unconformably on Layer 3. At LP12, Layer 2 is composed of interbedded light green-grey (7/1/10Y) and yellow (5/1/10Y) medium silts that display cross bedding with very fine laminations. From a distance they have an overall uniform, pale-grey appearance. The layer at BC2 is similar but with white (8/1/10YR) medium silt rather than yellow. No inclusions or fossils have been found in this unit, but locally vertically aligned gypsum crystal plates occur. Concentrations of most soluble minerals are lowest in this stratigraphic horizon.

Layer 3— The upper boundary of Layer 3 at LP12 is erosional, the troughs filled with sediments from Layer 2. Easily distinguishable from Layer 2, it is composed of an olive-grey (4/2/5Y) clay with strong brown (4/6/7.5YR) limonite inclusions throughout. Maximum thickness at Lake Pinpa was 150 mm. The same layer at BC2 was a dark greyish-brown (4/2/2.5Y) clay that had a maximum thickness of ~100 mm and limonite presence was less consistent. The results of the XRD supported field observations of the clay content of the sediment, with clay minerals Smectite and Palygorskite combining to make up 79% of Layer 3 at Lake Pinpa. Limonite inclusions were noted in the stratigraphic section in Layer 3. The XRD analysis of Layer 3 sediment from Lake Pinpa identified the iron ore mineral Goethite (9% of total mineral composition) a common component of Limonite; this was supported by the XRF analyses which found 19.82% Iron (II) and soluble Fe content >16% at both sites. This spike in iron is not reached by any of the preceding layers. The Fish Lens at BC2 had variable thickness within the top 50 mm of this layer, but in some cases Layer 2 sits directly on Layer 3 sediments, with no Fish Lens separating the sediments of each layer. At Lake Pinpa, the Fish Lens is usually much thinner (1–10 mm thick) than at Billeroo Creek and occurs near the top of Layer 3. Exceptions occur at LP6, where the Fish Lens is locally much thicker, sometimes up to 50–120 mm thick in areas of a couple of square metres scattered over several hundred square meters on the lake bed. Layer 3 sits conformably on Layer 4 in stratigraphic sections at BC2, but the boundary is unconformable at LP12.

Layer 4— 150 mm of an olive (5Y 5/3) and orange (10YR 4/6) mottled clay that sits unconformably on Layer 5 at LP12 and BC2. Fossils were found in the lower portion of this layer, on the boundary with Layer 5 at both sites. At Billeroo Creek, Layer 4 is marked by a sharp increase in Calcium, not present until Layer 5 at Lake Pinpa.

Layer 5— This layer is composed of a dolomitic white (10YR 8/1) mudstone with extensive black manganese staining giving a mottled appearance overall. 66% of sediment in Layer 5 at Pinpa and 54% of Layer 5 at Billeroo Creek (Table 3) is derived of clay minerals Smectite and Palygorskite. Layer 5 had less Fe, but much more Mg, Ca and Mn than L3 and L4 (Figure 3 and Table 2), reflecting that it included the minerals Dolomite/Ankerite (Table 3), not present in the overlying layers at either site. The mottled appearance of the dolomitic Layer 5 in both sections is explained by the transition from Dolomite ($\text{CaMg}(\text{CO}_3)_2$) to Ankerite ($\text{Ca}(\text{Fe},\text{Mg},\text{Mn})(\text{CO}_3)_2$) with the partial replacement of magnesium with iron (II) and manganese. Carbonate presence was confirmed with a dilute hydrochloric acid in the field, and reaffirmed by the soluble mineral result from both sites (Figure 3). The primary source of fossils contributing to the Pinpa Local Fauna at Lake Pinpa and Billeroo Creek (excluding Fish Lens) is the assemblage occurring at the bottom of Layer 4 and in the top 100 mm of Layer 5.

Figure 3: Results of the soluble mineral analyses of Layers 2–4 and the white and black sediments from Layer 5 at Lake Pinpa Site 12 (left); and Billeroo Creek Site 2 Layers 2–5 (right).

5. Results

Our excavations have expanded the taxonomic diversity of the Pinpa Local Fauna (first recorded by Tedford et al. [20]). Notably, in 2015–18 we recovered both associated skeletons and isolated bones from the base of Layer 4 and the top of the dolomitic clay layer (Layer 5) at BC2 and LP12. Excavations at LP12 revealed predominantly terrestrial taxa with numerous remains of marsupials (vombatiforms, phalangeriforms, macropodiforms) and birds. In comparison, the fauna from the Fish Lens over various sites is more aquatic, a possible current-concentrated, lake-edge accumulation with bony fish abundant and lungfish, dolphins, flamingos, rails, turtles, crocodiles and the platypus *Obdurodon* represented, in addition to terrestrial marsupials (mainly dasyuromorphs, phalangeriforms) and small scincids.

6. Systematic Palaeontology

Order SQUAMATA Opper, 1811 [46]
Family SCINCIDAE Gray, 1825 [47]
Subfamily EGERNIINAE Welch, 1982 [48]
Genus PROEGERNIA Martin et al., 2004 [16]
PROEGERNIA MIKEBULLI sp. nov.

Zoobank ID: urn:lsid:zoobank.org:act:068F625A-B537-43C7-A1B1-4F20DD5F6BF9

Holotype—SAM P57502, a near complete right dentary; 27 tooth loci, 14 of which bear teeth.

Diagnosis—The species is referred to the subfamily Egeriinae because the dentary has a closed Meckel's groove and a large inferior alveolar foramen. It is referred to the genus *Proegernia* because the tooth crowns widen anteroposteriorly from the shaft, the crista lingualis and crista mesialis are near horizontal and converge on an apex slightly posterior to the centre of the tooth, and there are medially-prominent cristae on the anterior and posterior culmen laterales. *Proegernia* is further distinguished from other members of the Egeriinae (species in *Egernia*, *Bellatorias*, *Liopholis*, *Cyclodomorphus*, *Tiliqua*, *Tribolonotus*, and *Corucia*) by the combination of the following traits: a more anteriorly positioned apex of the splenial notch at >50% the anteroposterior length of the dentary; more than 22 tooth locion the dentary and 20 on the maxilla; minimal anteroposterior flaring of the tooth crown with lateral compression of the medial face; a small sliver of open Meckel's groove immediately posterior to the symphysis and a concave ventral face on the pterygoid body. *Proegernia mikebulli* differs from *P. palankarinnensis* in lacking lateral tooth striae, in having up to five more tooth loci for a total of 27 on the dentary, and a much more medially inflected anterior tip to the dentary, making it more curved in dorsal view.

Type locality—Fish Lens, subsite Billeroo Creek 2, Wells' Bog Site, northern side of Billeroo Creek, GPS coordinates S 31°6'11.76" E 140°13'53.70", See Figure 2.

Stratigraphy/Age—Namba Formation, in Layers 3–5 of the (Figure 1) as exposed in Billeroo Creek and Lake Pinpa; Pinpa Local Fauna. This LF has been biostratigraphically correlated with the "Wynyardiid" or Minkina Fauna, (Zone A) of the late Oligocene Etadunna Formation, 25.5–25.7 Ma [19, 49].

Paratype—SAM P57499 a partial right dentary from BC2, with a complete coronoid process, 23 tooth loci and 10 teeth.

Referred specimens—The following specimens are referred to this taxon based on the combination of their appropriate size, stratigraphic co-occurrence, and having a general similarity to the equivalent elements in other egeriines. Details of tooth crown shape also enable robust referral of tooth-bearing bones to the same taxon. Two right post-dentary compound bones; SAM P57543 from Lake Pinpa (LP6) preserving the dorsal surface of the surangular and glenoid, broken part-way through the retroarticular process; and SAM P57542 from Billeroo Creek (BC2) which preserves the ventral and medial face of the articular, the glenoid entirely and most of the retroarticular process. Two right partial maxillae both from BC2, SAM P57541 representing an anterior fragment with premaxillary process intact and the first 9 tooth loci holding 3 teeth; and SAM P57503 which preserves the posterior majority of the maxilla with 16 loci and 9 teeth, the facial process is broken above the row of maxillary foramina. An intact left pre-maxilla, SAM P57542 from BC2, with osteoderm fragments on the internasal process, preserving two teeth from four loci. A right pterygoid SAM P57512 from BC2, the quadrate process broken beneath the epipterygoid notch.

Etymology—The species is named after Professor Michael Bull (1947–2016) of Flinders University, South Australia, who devoted decades to documenting the ecology of Australia's egeriine skinks. Mike supervised a generation of Australian ecologists and his lectures inspired countless students to become biologists. His studies of Australian egeriine skinks and their parasites are model long-term ecological studies, and led to major discoveries such as the existence of monogamous pairs in *Tiliqua rugosa*, and parental care and family living in *Egernia* spp.; and the establishment of a successful breeding and reintroduction program for the endangered Pygmy Bluetongue, *Tiliqua adelaidensis*.

a. Description

Dentary—The most diagnostic and commonly-recovered element is the dentary bone of the lower jaw. Seventeen incomplete dentaries were recovered from the 2007–2018 trips to Lake Pinpa and Billeroo Creek. A reconstruction of a near-complete lower jaw of *Proegernia mikebulli* was made, with the dentary portion based on the holotype SAM P57502 and paratype SAM P57499 (See Figure 4). From the anterior tip of the symphysis to the posterior tip of the coronoid process, missing only the angular process, the reconstructed dentary is 12.7 mm long. In medial aspect, the dentary is convex ventrally. It is 1.6 mm wide and 2.2 mm tall at mid-length of the tooth row, and the dental sulcus is shallowly concave. From occlusal view, the symphysis is directed medially to articulate with the opposing dentary at an angle of 32°. The dental sulcus is clearly differentiated, SAM P57502 preserving 27 loci and 14 pleurodont teeth. The tooth row is 10.8 mm long.

Figure 4: 1a-c Holotype, a right dentary, SAM P57502, 2a-c paratype; a left dentary SAM P57499; two right post-dentary compound bones 3a-c SAM P57543 and 4a-c SAM P57542 of *Proegernia mikebulli* sp. nov. from the Fish Lens at BC2; and 5 complete reconstruction in medial view of the right mandible of *Proegernia mikebulli*, the coronoid and splenial (hatched area) are reconstructed based on modern egeriine specimens, no fossil representatives of these elements are known. Abbreviations: adf, adductor fossa; anf, angular facet; art., articular; d., dentary; gl, glenoid fossa; iaf, inferior alveolar foramen; mg, Meckel's groove; psf, posterior surangular foramen; rap, retroarticular process; san., surangular; and sy, symphysis.

The anterior symphysis, preserved in its entirety on SAM P57502, has a flattened, reverse '7' shape in medial view. The symphysis has two caudally-directed branches; a narrow, ventrally-directed sliver along the anterior edge of the dentary; and, dorsally, a wider process that terminates in a sharp point. Maximally, the symphysis is 1.8 mm long and 1.3 mm deep. In the notch between the caudally-directed branches of the symphysis a symphyseal foramen extends ventrally into a 1 mm long and <0.2 mm wide anterior opening of Meckel's groove. The groove is open and aligned parallel with the dorsal side of the ventral section of the symphysis; further posteriorly it is enclosed.

The inferior alveolar foramen and the dorsal and ventral margin of the splenial notch are preserved to the posterior end of the tooth row in SAM P57502 (Figure 4). The inferior alveolar foramen lies on the ventral margin of the splenial notch, below the mid-point of the tooth row. The shape of the splenial notch is narrow and roughly parallel-sided for the anterior 40% of the preserved length, expanding dorsoventrally with a convex upper edge and straight ventral margin for the remaining posterior section of length. Where the notch expands, the dorsal edge preserves a concave face, allowing the splenial to medially overlap the dentary. Posterior to this face, beneath the 3rd-last tooth, the dental sulcus is broken.

The coronoid process of the dentary is preserved on the paratype SAM P57499 and ascends posterodorsally from the position of the last tooth to a tip projecting dorsally above the posterior tooth. The ventral margin of this process preserves the anteromedial articular facet for the coronoid, which therefore can be seen to overlap the splenial and dentary anterior of the 3rd last tooth position. The angular process, although not preserved entirely on either dentary specimen, can be reconstructed with some confidence using the angle of the intact edge immediately beneath the coronoid process. The total length of this process is limited by the absence of a facet for its articulation on either of the recovered post-dentary compound bones.

The lateral face of the dentary is slightly convex and preserves a single row of eight mental foramina extending posteriorly from the anterior top of the dentary to below the 17th tooth position. The largest foramen in the row is at the posterior end.

Post-dentary (compound) bones— The post-dentary complex or compound bone is the fused surangular and articular bones making up the posterior half of the scincid mandible. This fusion develops ontogenetically; complete fusion of the two elements with no traces of a suture internally is evidence of adulthood in extant Australian scincids [50]. Two right post-dentary complexes were recovered from the Namba Formation, SAM P57543 from a right mandible at Lake Pinpa (LP6; Figure 4 3a–3c) and SAM P57542 from Billeroo Creek (BC2; Figure 4 4a–4c); these are used in the reconstruction of the complete lower jaw (Fig. 4, 5). These elements are referred to this taxon as they are of appropriate size, confirmed as scincid material by the presence of a facet for the angular bone, which is not present in gekkotans; and the alignment of the retroarticular process. The shape of the retroarticular process is similar to that of extant egeriines. When aligned horizontally in the in-vivo position, the angular in egeriines is wide and extends further laterally than the narrow ventrally positioned angular of eugongyline (i.e. *Emoia longicauda* SAMA R2352, *Eugongylus rufescens* R36735). Sphenomorphines generally have a simple, straight post-dentary complex without medial torsion of the articular, and the retroarticular is not inflected medially posterior of the glenoid. All scincid bones from Billeroo Creek and Lake Pinpa are consistent with presence of a single egeriine taxon; there is no variation among elements that might indicate more than one species being represented, so both compound bones are referred to the taxon named from the dentaries.

SAM P57543 is a near complete right adult post-dentary preserving completely fused surangular and articular bones. Anteriorly the areas for articulation with the dentary and coronoid are not preserved, neither are the angular or the anterior edge of the adductor fossa. A facet for articulation of the angular is present on the ventral half of the lateral face of the articular, not extending to reach the posterior surangular process. The posterior majority of the retroarticular process is broken off SAM P57543, so a description of its complete shape is based on SAM P57542. The dorsal surface of the surangular is relatively straight and rises slightly to meet the dorsal tip of the mandibular condyle or glenoid fossa where the quadrate articulates with the mandible. The maximum preserved length of this dorsal surface is 2.9 mm. The glenoid fossa faces posterodorsally, the surface is convex mediolaterally and concave anteroposteriorly, reaching a maximum length of 1.8 mm. The lower third of the glenoid fossa becomes slightly concave ventrally, curving up to meet the medial articular process of the surangular. The surface of the glenoid fossa has a pitted texture indicating attachment of cartilage in the mandibular cotyle. Posterovertrally the glenoid fossa ends in a clearly defined ridge between it and the retroarticular process. The preserved section of the retroarticular process is concave (SAM P57542). Medially, the foramen for the chorda tympani is preserved in SAM P57542.

Two foramina are preserved dorsally in a flattened surface, and a posterior surangular foramen is on the lateral face immediately anterior of the glenoid fossa. No sign of an anterior surangular foramen is present on either SAM P57543 or SAM P57542. From the flattened dorsal surface of the surangular, the bone curves sharply ventrally, leaving a flat medial surface above the dorsal margin of the adductor fossa. The adductor fossa is in the lower 50% of the medial face of the complex. The anterior margin of the adductor fossa is not preserved on either post-dentary compound bone. Viewed medially, the posterior edge of the narrow, ovular, fossa terminates anterior to the glenoid. The lateral face of the post-dentary complex of the mandible is convex. A discernible ridge runs from just anterior of the posterior surangular foramen, anteriorly to the broken anterior edge of SAM P57543, marking the posteroventral edge of the *M. adductor mandibulae externis*.

Maxilla— Two incomplete right maxillae (SAM P57503, SAM P57541) recovered from BC2 are referred to *Proegeria mikebulli*. SAM P57503 (Figure 5, right) preserves a near complete tooth row, a bifid posterior termination of the suborbital process (an apomorphy of the Scincidae), and complete tooth crowns from both the anterior tooth form and posterior tooth form on the same row. This specimen was recovered from sieved material in two halves, that fitted together when articulated, and so was joined with Paraloid B72 before SEM images were taken. The smoothed hemispherical shape on the dorsal edge of the palatine process of the maxilla (Figure 5) is Paraloid and not a feature of the bone.

Anteriorly, the maxilla SAM P57503 lacks the tip of the premaxillary process. The anteriormost tooth positions are missing on SAM P57541; tooth and loci counts are based on the reconstruction also using SAM P57541. The facial

process of the maxilla is incomplete preserving only 2.3 mm above the tooth row. The unbroken edge of the orbit can be traced from above the level of the 14th tooth position, to the broken dorsal tip of the suborbital process. The ventral tip of the bifid suborbital process is intact as are the last tooth positions. The tooth row preserves 16 of 20 loci, and 9 pleurodont teeth are still in position.

The lateral face of the maxilla is slightly convex in the anterior to posterior plane when viewed dorsally. The maxilla preserves nine primary foramina on the lateral face, the largest is the most anterior. Above the primary row, 11 more foramina, the largest one-third the size of the primaries, are visible in the SEM in two rows. Medially, the medial edge of the palatine shelf is broken. The palatine shelf thins posteriorly and ends in a fine point at the ventral tip of the bifid suborbital process. The tooth row occlusal surface is slightly convex posteriorly, with larger teeth beginning at the ninth position. The last two teeth decrease in size posteriorly, creating the trailing edge of the convex occlusal line. The reconstructed maxilla is 11.1 mm long from the premaxillary process to the suborbital process, with 20 tooth positions (see Figure 5). The original height of the facial process was not preserved on any specimen.

Figure 5: Right maxilla of *Proegernia mikebulli* sp. nov. reconstructed from SAM P57541 (anterior fragment; top left) and SAM P57503 (near complete; top right). 'A' marks the same tooth locus on each specimen. Enclosed by the box is the tooth 'B'. Abbreviations: cl, *cuspis labialis*; cla, *culmen lateralis anterior*; cli, *cuspis lingualis*; clp, *culmen lateralis posterior*; fp, facial process; psm, palatine shelf of the maxilla; pxp, premaxillary process; sop, suborbital process; and str, striae.

Premaxilla— A single complete left premaxilla, SAM P57542, was recovered from Fish Lens at BC2 (Figure 6). An osteoderm is fused to the anterior face of the internasal process, and would have overlapped the right premaxilla when the pair were articulated. Paired, unfused premaxillae in adulthood eliminate the possibility that this element belongs to a gekkotan. The ascending internasal process has a flattened medial side for articulation with its paired element. Viewed laterally, a foramen for the longitudinal canal is situated immediately above the maxillary process in the lateral face of the internasal process. Sharply pointed dorsally, the internasal process widens ventrally and terminates at the notch for the exit of the ethmoidal foramen laterally, and tooth row medially. The internasal process of the premaxilla rises towards the nasals at a 40° angle. The total length of the internasal process is 3.1 mm. The maxillary process is curved posterolaterally towards the maxilla and is approximately 1/3 of the height of the internasal process, reaching 1.6 mm in mediolateral width. The lateral edge of the maxilla process curves ventrally to finish beside the fourth tooth position. Four tooth loci are present and the first two teeth are preserved. These teeth are 0.53 mm long from the lateral face of the maxillary process and 0.28 mm wide. What remains of the worn crowns (see inset A, Figure 6) match those of the anterior-most teeth of the maxilla specimen SAM P57541. There are weak striae on the medial face of the crown, a mediolateral compression of the crown below the *crista lingualis*, and a sharply curved, prominent, *culmen lateralis*.

Figure 6: Left premaxilla (SAM P57542) of *Proegernia mikebulli* sp. nov. from posterior (left), anterior (centre), and lateral (right) view. A, tooth one, enlarged in box. Abbreviations: flc, foramen of longitudinal canal; inp, internasal process; mp, maxillary process; nef, notch of the ethmoidal foramen; and o, osteoderm.

Dentition— The dentition of *Proegernia mikebulli* is described using the two dentaries SAM P57502 and SAM P57499, the maxillae SAM P57541 and SAM P57503, and premaxilla SAM P57542, together comprising a near-complete upper and lower tooth row. The upper tooth row contains a total of 24–25 teeth, 4 or 5 on the left and right premaxilla, and 20 on the maxilla. The dentary tooth row has 27 tooth positions, beginning above the symphysis anteriorly and terminating just anterior of the coronoid process. The occlusal profile of the maxilla and dentary are both convex, with slightly larger teeth in the posterior half of the tooth row.

Figure 7: A and B: 24th tooth on the dentary SAM P57502, medial view, and C the 16th tooth in occlusal view with lateral and medial facies marked. Abbreviations: ai, antrum intercristatum; cl, crista lingualis; cm, crista mesialis; cul, cuspis labialis; cula, culmen lateralis anterior; and culp, culmen lateralis posterior.

On the dentary teeth, the tooth crown is similar in width to the shaft, expanding anteroposteriorly with slight mediolateral compression for the last 30% of the tooth height. Prominent anterior and posterior *culmen laterales* extend from the tips of the *crista mesialis*, turning medially and ventrally on the dentary, with a sharp angle producing ‘shoulders’ notable in medial view of the tooth (Figure 7). From the central cusp, the anterior cristae dip ventrally 20°, and posterior cristae 45°. The cusp (*cuspis labialis*; Figure 7) is slightly posterior of the centre of the tooth, and in occlusal view is positioned just posteromedial to the centre. This medial shift of the cusp creates a convex dorsal surface to the tooth, directing both cristae medially. Striae are only located on the medial face of the tooth crown, angled

dorsoventrally from the off-centre cusp. The two most prominent striae run **parallel** to the *culmen laterales* (both anterior and posterior), all others are weaker in profile with staggered lengths. Tooth crown morphology is similar on the maxilla and premaxilla.

Tooth wear occurs first on the cusp, forming a shallow rounded depression, gradually deepening medially. The *antrum intercristatum* expands in width from the central cusp. Wear is thus more noticeable in the centre of the tooth crown between the cusps, than anteriorly or posteriorly along the cristae, creating a thickening 'v' shape.

Pterygoid— A single **right** pterygoid attributed to *P. mikebulli* was recovered from Fish Lens at BC2 (SAM P57512; Figure 8). It is **near** complete, missing only the anterior-most tip of the palatine process and distal tip of the quadrate process. The pterygoid articulates with the palatine anteriorly with a pointed process extending anteriorly from a fanned, v-shaped **pterygoid head to overlap with a similar feature from the anterior element**. Laterally to the palatine process, the ectopterygoid process extends towards the ectopterygoid with a concave, curved facet for articulation with this element on the dorsal surface of the pterygoid head. Between these two processes, the fan-shaped pterygoid head has a concavity on the ventral surface that is 0.7 mm wide and 1.7 mm long. Within this concavity are three foramina, one in the deepest area of the concavity, and two are paired anteriorly, near the palatine process. Posteriorly, the pterygoid head narrows to become a parallel-sided rod, **before** bending medially at an angle of 35° towards the epipterygoid notch. The epipterygoid notch is an **ovular** concavity, with raised margins, on the dorsal surface of the pterygoid at the anterior end of the quadrate process. A sharp ridge extends from the anterior edge of the epipterygoid notch to the corner of the bend towards the pterygoid head. Posterior to the epipterygoid notch, the quadrate process becomes L-shaped in cross section and extends 1.5 mm posteriorly **before** the broken edge. The pterygoid head is 2.7 mm wide between the palatine and ectopterygoid processes. Total length of this element is **not preserved**.

Figure 8: **Right** pterygoid of *Proegernia mikebulli* sp. nov., SAM P57512 in dorsal (left) and ventral (right) views. Abbreviations: en, epipterygoid notch; pp, palatine process; pvc pterygoid ventral concavity; qp, quadrate process; and tp, ectopterygoid process.

b. Postcranial material potentially attributable to *Proegernia mikebulli*

Several procoelous vertebrae and assorted vertebral fragments were recovered from the Fish Lens at BC2. The best preserved of these is SAM P57514, a **pre-sacral** vertebra (Figure 9). This specimen is **near** complete, missing only the posterodorsal tip of the neural spine and the lateral tip of the right postzygapophysis. A thin crack runs posterolaterally on the ventral face of the centrum from just posterior of the ventral foramina, to the broken zygapophysis. The vertebra has an elongate centrum, the ventral surface is concave in lateral view. The neural canal is **a similar** diameter to the cotyle/condyle with a flat ventral surface broadening laterally before coming to a tear drop point when viewed anteriorly or posteriorly. Synapophyses are present immediately posterior to the prezygapophyses on the lateral facies. The neural spine rises posteriorly, steadily at an angle of 20°, from the dorsal margin of the neural canal until reaching the broken extremity. The prezygapophyses are angled at 32° and the postzygapophyses at 28°. Overall shape of the vertebra is relatively elongate, approximately one third longer than it is wide. Overall centrum length is 4.2 mm, the cotyle maximum width is 1.6 mm; the vertebra height anteriorly is 2.4 mm, posteriorly including the broken neural spine the height it is a maximum of 3.7 mm. The widest point of the vertebra is between the synapophyses, totalling 3.3 mm. This element is **identified as a scincoid lizard** based on procoely, absence of a zygosphenes/zygantrum, minimal dorsoventral compression of the rounded cotyle and condyle, elongated centrum, and the minimal size of the ventral foramina. Snake vertebrae have shorter centra, a round cotyle/condyle with no compression, a set of zygosphenes and zygantra, and a ventral hypapophysis. A procoelous centrum devoid of a hollow space for the notochord eliminates the possibility of the

vertebra belonging to a non-pygopodid gekkotan. Gekkotans inclusive of pygopodids also have **enlarged ventral foramina**. Procoelous Gekkota vertebrae (mostly restricted to pygopodids) tend to have smaller condyles and cotyles, the cotylar hollow often not visible in ventral view because the dorsoventral inclination is minimal [25].

Figure 9: A single vertebra (SAM P57514) potentially referable to *Proegernia mikebulli* sp. nov., recovered from Fish Lens, BC2. A, anterior view; B, posterior view; C, lateral view, anterior to the left; D, dorsal view; and E, ventral view. A proximal fragment of a right femur assigned to *Proegernia mikebulli* sp. nov. SAM P57540. F, dorsal; G, anterior; and H, ventral views.

Abbreviations: con, condyle; cot, cotyle; fec, femoral condyle; itf, intertrochanteric fossa; itr, internal trochanter, nc, neural canal; nsp, neural spine, poz, postzygapophysis; prz, prezygapophysis; and syp, synapophysis.

A proximal right femur, SAM P57540 (F–H, Figure 9), was recovered from the Fish Lens at BC2. Broken immediately beneath the intertrochanteric fossa, **no remnants of the shaft width or length are preserved**. All other proximal features are intact. The specimen is most likely from an adult **scincid** as the epiphyses are fully ossified and fused to the diaphysis. The articular surface of the femoral condyle is semi-circular viewed anteriorly, and ovular in proximal profile. The femoral head extends medially away from the shaft axis. The internal trochanter is much shorter, only rising slightly above the edge of the intertrochanteric fossa. The intertrochanteric fossa is a concave surface between the condyle and trochanter on the ventral face of the proximal femur. The dorsal side also preserves a concave surface, but this is steeper-sided with a prominent ‘v’ marking where the femoral shaft begins. The **width of the** femoral condyle is widest in anterior view, measuring 2.2 mm at the suture with the epiphysis. The internal trochanter is 1.3 mm tall from the base of the intertrochanteric fossa. Total length preserved is 3.8 mm.

The scincid apomorphy of the trochanter major gradually tapering in height distally (see Lee et al. [51]) is not visible on this specimen, as the shaft is broken immediately proximal to it.

c. Comparisons

Comparison of new taxon with *Proegernia palankarinnensis*— The type species of *Proegernia* (by monotypy) is *Proegernia palankarinnensis*. While the type specimen is currently missing, the existing description [16] suggested *Proegernia* as being representative of a transitional form between a plesiomorphic lygosomine dentary morphology and the derived egeriine condition. The plesiomorphic traits shared by both *Proegernia palankarinnensis* and *P. mikebulli* are the small (about 1 mm long) opening of Meckel’s groove immediately posterior of the symphyseal foramen and the more anterior extent of the splenial notch into the anterior half of the tooth row **length**. Both species of *Proegernia* also have egeriine features emerging in the prominent ‘shoulders’ of the *culmen lateralis* on their unicuspid tooth crowns, and a relatively deep splenial notch, extending anteriorly further than other Australasian scincids. *Proegernia palankarinnensis* was described as having 22 tooth loci, with a possible maximum of 23 teeth, uncertain due to a broken symphyseal region. The specimens of *Proegernia mikebulli* preserve up to 27 dentary tooth positions (see SAM P57502, Figure 4). Both taxa share an anterior section of crowded, smaller teeth in their lower dentition. The spacing of teeth in *P. palankarinnensis* is **marginally** wider than *P. mikebulli* at the posterior end of the tooth row. Tooth shape varies between the two taxa: *P. palankarinnensis* has tooth shafts narrowing at the base, whereas tooth shafts in *P. mikebulli* narrow beneath the crown and expand again at the base. Martin et al. [16] noted that the tooth crowns of *Proegernia* SAMA P39204 preserve distinct striations on both the lateral and medial faces. Tooth crowns for *Proegernia mikebulli* have **weakly lineated** striae on the medial face of the crown beneath the *crista lingualis*, but the lateral tooth face is smooth.

Although occurring almost concurrently, comparisons of the shape of the **holotype** dentaries of *P. palankarinnensis* and *P. mikebulli* (see Figure 10) show that the two species differ notably in the following characteristics: 1) the medial inflection of the symphysis is considerably less in *P. palankarinnensis* than in *P. mikebulli*, even after taking into account the effects of erosion. 2) The inferior alveolar foramen is positioned more anteriorly in *P. palankarinnensis*, below the 12th tooth position; that in *P. mikebulli* is level with the posterior edge of the 14th tooth. 3) No short extension of Meckel's groove can be observed anterior of the inferior alveolar foramen in *P. mikebulli*, a plesiomorphic character representing a partial remnant of Meckel's groove that was noted [16] on the holotype of *P. palankarinnensis* (Figure 10 A). **4) The internal septum of both specimens extends anteriorly to a similar degree.** 5) The overall shape of the dentary of *P. palankarinnensis* is less robust, it being narrower between the medial edge of the dental sulcus and lateral face at the height of the mental foramina; 7) The facet for articulation of the coronoid terminates beneath the **second-last** tooth in *P. palankarinnensis* and the **third last tooth locus** in *P. mikebulli*. 8) The dorsal edge of the splenial notch is widened more abruptly in *P. mikebulli* with a curve; in *P. palankarinnensis* the same space is sharply v-shaped. 9) The coronoid process preserved on *P. palankarinnensis* does not extend above the height of the final tooth crown, unlike the process on SAM P57499 of *P. mikebulli*.

Figure 10: Comparative SEM of *Proegernia mikebulli* sp. nov. (B,C) and **microphotographs of *P. palankarinnensis* (A,D) from Martin et al. [16], in medial (above) and occlusal (below) views. Note the anterior extension of the splenial notch (3) in *P. palankarinnensis* and the increased number of teeth/loci in *P. mikebulli*. Features described in text are labelled: 1) Medial inflection of the symphysis; 2) Dentary depth; 3) Position of the inferior alveolar foramen; 4) Meckel's Groove extension anterior of the inferior alveolar foramen, present in *P. palankarinnensis* and absent in *P. mikebulli*; 5) anterior extent of the internal septum similar in both taxa, 6) Overall robustness as dentary width, 7) facet for articulation with anterior ramus of coronoid, 8) splenial notch height, and 9) height of the coronoid process relative to the last tooth.**

d. Other egeriines and other scincids

Comparisons with extant egeriines including *Lissolepis coventryi* (SAMA R57317) and *Liopholis multiscutata* (FUR168), as well as a Southeast Asian lygosomine outgroup representative *Eutropis multifasciata* (SAMA R35693), and two plesiomorphic scincids *Eumeces schneideri* (SAMA R6695), *Brachymeles schadenbergi* (SAMA R8853) and *Plestiodon fasciatus* (SAMA R66784) were conducted to determine how similar or derived the fossil taxon is from the generalised scincine (=inferred plesiomorphic) condition. Four characters notably vary among the chosen representative egeriines, the outgroup lygosomine and the extinct *P. mikebulli*: the anterior extent of the splenial notch and the inferior alveolar foramen; the shape and robustness of the symphysis, dentary depth and overall width; and, the number of teeth in the tooth row. The anterior extent of the splenial notch varies within the egeriine radiation; both species of *Lissolepis* share a further anterior-reaching inferior alveolar foramen than any other egeriine genus. The outgroup scincid condition, represented by the examined taxa *Eumeces schneideri* (SAMA R6695), *Brachymeles schadenbergi* (SAMA R8853), *Plestiodon fasciatus* (SAMA R66784), and *Eutropis multifasciata* (SAMA R35693) have a splenial notch that extends further than 60% anteriorly along the length of the tooth row, or one that stretches anteriorly into an elongate open Meckel's groove. The anterior extension of the splenial notch is a plesiomorphic state for this character, also shared by both *Proegernia* taxa.

The mandibular symphyseal joint in scincids varies between subfamilies in its overall size and robustness. Within egeriines, the variation is in the anteroposterior length of its posteroventral section, and the depth of the upper branch. In egeriine species that have a reinforced symphyseal joint and manipulate harder food, the lower branch of the symphysis is extended posteriorly and sometimes ventromedially (especially in *Tiliqua*, *Cyclodomorphus* and larger species of *Egernia*), to form a 'chin'. The dorsal branch of the symphysis deepens in *Liopholis* and in larger, omnivorous and herbivorous species of *Egernia*. This may be related to functional morphology to handle stresses on the chin during feeding, rather than a phylogenetic signal as these taxa do not form a clade. *Proegernia mikebulli* has a short and narrow posteroventral extension of the symphysis, possibly due to the presence of a foramen in the caudal notch between the limbs of the symphysis. This foramen exposes a short section of Meckel's cartilage in sphenomorphines and scincines. In egeriines this foramen is usually absent. Both *P. palankarinnensis* and *P. mikebulli* have dentaries where a slightly elongate foramen would expose the Meckel's cartilage, but in neither does this opening extend beyond the posterior edge of the symphysis.

Increased dentary depth and width, in relation to overall length, increases the robustness of the mandible. Variation in these measurements occur between genera of egeriines; species within *Liopholis* often have shorter, deeper skulls and corresponding dentaries, than similar-sized lizards within the genus *Egernia* (see the Supplementary information from [15]). *Lissolepis* has a more gracile dentary (see Figure 11). Outside of the egeriines, *Eutropis* and other SE Asian scincids with an unmodified insectivorous diet are even more gracile. *Eutropis multifasciata* has a longer dentary tooth row relative to snout-vent length than all egeriines. Derived morphologies related to dietary adaptations as seen in the herbivorous and durophagous members of the Egeriinae (i.e. *Corucia zebrata*, *Tiliqua* and *Cyclodomorphus*) result in more robust dentary dimensions and/or modified dental morphology. *Proegernia palankarinnensis* and *P. mikebulli* present dentary depths most similar to those of *Lissolepis* spp., between the ancestral shallow insectivorous *Eutropis*, and the deeper dentary typical of *Liopholis* spp. Deepening of the dentary bone and shortening of the snout noted in numerous species within the genus *Liopholis* is possibly related to the group's affinity for burrowing [52].

Eutropis multifasciata*Proegernia mikebulli* sp. nov.*Lissolepis coventryi**Liopholis multiscutata*
Figure 11: Comparative medial views of the dentaries of *Eutropis multifasciata* (SAMA R35693), *Proegernia mikebulli* sp. nov., *Lissolepis coventryi* (SAMA R57317) and *Liopholis multiscutata* (FUR168). 1 Anterior extent of splenial; 2 Symphysis shape and robustness; and 3 Dentary depth at mid-tooth row length.

e. Dentition

The teeth of *Eutropis multifasciata* (SAMA R35693) and *Proegernia mikebulli* show increasing derivation from the plesiomorphic insectivorous skink tooth type as described by Richter [22] and Kosma [23] and represented by *Plestiodon fasciatus* (SAMA R66784; Figure 12). Although many species within the extant radiations of eugongyline, sphenomorphine and egeriine scincids have modified tooth shapes for dietary specialisations, taxa exhibiting the most plesiomorphic tooth crown shapes were chosen for comparisons with fossil taxa.

Proegernia mikebulli has departed from the crown shape of basal skinks [see 22, 23] by having the cristae lingualis et mesialis directing medially, creating a sharp angle with the *culmen laterales*, and less-prominent striae mark the medial face of the tooth. *Eutropis* retain the prominent striae running dorsoventrally on the medial face of the tooth, these striae are almost all equally prominent, while *P. mikebulli* appears to have slightly stronger striae immediately adjacent and parallel to the *culmen laterales*. The tooth crowns of *Eugongylus rufescens* also demonstrate prominent *culmen laterales* and weakened striae, although the tooth shape differs; the shaft widening beneath the crown slightly and wear patterns make obvious the less medially directed cristae. The basal sphenomorphine condition is that of a narrow tooth shaft and crown, with sharply angled cristae creating a more pointed tooth profile. The medial face of the sphenomorphine tooth is anteroposteriorly convex and marked with prominent striae that do not approach the height of the lingual cristae but sit a third of the depth of the crown lower. The striae on *P. mikebulli* extend from almost directly in contact with the lingual cristae to the ventral tips of the *culmen laterales*.

The tooth crown in *P. mikebulli* is similar to that of the extant *Lissolepis coventryi* sharing medially directed cristae, sharp ‘shoulders’ to the *culmen laterales*, weak medial striae and absent lateral striae, and an expanded crown width on a narrower tooth shaft.

Figure 12: Medial view of left dentary tooth crowns of *Plestiodon fasciatus* (SAMA R66784), *Eutropis multifasciata* (SAMA R35693), *Proegernia mikebulli* sp. nov. (SAM P57502), *Lissolepis coventryi* (SAMA R57317), *Eugongylus rufescens* (SAMA R36735) and *Sphenomorphus jobiensis* (SAMA R6736). *Plestiodon* represents the basal skink tooth condition as described by Richter (1994). *Culmen lateralis* posterior is labelled culp.

7. Phylogenetic relationships

Parsimony and Bayesian analyses retrieved broadly similar trees, placing both species of *Proegernia* outside of, but close to, living (crown) Australian egeriines. Parsimony analysis of all morphological and molecular data recovered one most parsimonious tree (Figure 13) with a best score of 4726.25, found 60 times out of 1000 replicate searches. Confidence on most of the nodes indicated by the bootstrap analyses is low; only support for the clades *Bellatorias*, *Liopholis* and *Tiliqua+Cyclodomorphus* are above 50. This is almost certainly due to an unstable position of the fossils, which are missing all DNA and most morphological data (111 missing characters for *P. palankarinnensis* and 73 for *P. mikebulli*).

Figure 13: Egerniine phylogeny based on the single most parsimonious tree produced by TNT [26]. Bootstrap values >50 shown.

Proegernia mikebulli is recovered basal to the extant Australian Egerniinae (spanning *Lissolepis* to *Tiliqua*), and *Proegernia palankarinnensis* is retrieved as sister to the Solomon Islands' *Corucia zebata*, but both with bootstrap support <50%. These relationships are conservatively interpreted as an effective polytomy between crown egerniines, *Corucia*, *Proegernia*, and *Lissolepis*. Thus, rather than erect a new genus, we provisionally assign the new species to *Proegernia* (due to geographic and stratigraphic links with the type of that genus, and the Bayesian results below).

The position of both taxa outside the crown Australian egerniines is due to plesiomorphic scincid character states such as a partially open Meckel's groove (char. 11) in both fossil taxa. This feature also separates them from the outgroup taxon *Eutropis multifasciata* which has an entirely open Meckel's groove. The anterior extent of the splenial (char. 21), and the extension of Meckel's Groove anterior of the splenial notch are features supporting the separation between *Lissolepis* and *Proegernia*; *Lissolepis* with a splenial reaching approximately 50% of the tooth row length anteriorly, and *Proegernia* reaching beyond to two-thirds the tooth row length.

The crown Australian Egerniinae spanning *Lissolepis* to *Tiliqua* is diagnosed by a completely closed Meckel's Groove (char. 11), an absence of pterygoid teeth (char. 131), and a pterygoid quadrature ramus that is arcuate in cross section (char. 134). *Proegernia* (and *Corucia*) have plesiomorphic or alternative states for these characters, resulting in their position outside this clade. Potential morphological synapomorphies of the basal living Australian genus *Lissolepis* that are also present in *Proegernia* are the termination of the dentary coronoid process (char. 17), the orientation of the retroarticular process from dorsal view (char. 46), the plesiomorphic bicuspid tooth crown shape (char. 55), minimal dental cementum (char. 60), and the divot in the ventral surface of the pterygoid body that is also absent of pterygoid teeth (chars 135 and 131). These shared characters conflict with the position of *Proegernia* outside the Australian crown group, resulted in the low support for the relationships between *Corucia*, *Proegernia* and *Lissolepis*.

The Bayesian analysis of all morphological and molecular data produced the topology shown in Figure 14: the consensus of four separate runs of 100,000,000 generations, when combined in LogCombiner, then summarised with TreeAnnotator with a burnin of 20%.

Figure 14: Egeriine phylogeny based on consensus total evidence Bayesian inference tree from BEAST. †denotes extinct taxon. Small numbers at nodes denote posterior probability. Reconstructed node ages for key clades are in brackets, the green shading denotes the Eocene, after which Australia became isolated from Gondwana [4].

The extant egeriine radiation forms a well-supported clade. Within this, *Tribolonotus* is strongly supported as the sister to remaining taxa. Of those, *Corucia* is strongly supported as the sister of the remaining (Australian) egeriines. Both late-Oligocene fossil species of *Proegeria* are weakly supported as the sister to *Corucia* with a Prior Probability (PP) of 0.25. All Australian extant egeriines again form a clade that excludes both *Proegeria*. The extant Australian clade is a weakly supported (PP= 0.51) clade and retrieved as originating in the lower Oligocene (Figure 14).

Proegeria palankarinnensis and *P. mikebulli* are retrieved as sister taxa, but with little support (PP= 0.44), not unexpected given extensive missing data. Character changes affirming the sister relationship include the position of the mental foramina 50% along the length of the tooth row (char. 13), and the presence of less than 20 tooth crown striae (char. 61). Both of these features are also shared with *Lissolepis*, but not with their putative sister clade *Corucia zebata*. The similar positioning of both taxa (i.e. basal to crown Australian egeriines) is supported by the same characters as those noted after the parsimony analysis.

The node age for all Egeriinae is retrieved as 50.12 Ma, with *Tribolonotus* as the extant stem taxon representing the arrival of the subfamily to northern Sahul (Australia, New Guinea and surrounding islands). The stem clade last shared a common ancestor with the crown Australian egeriines 40.47 Ma.

8. Discussion

Recent exploration of the Namba Formation cropping out at Lake Pinpa and Billeroo Creek has unearthed a new egeriine scincid lizard, *Proegeria mikebulli* sp. nov. Previous excavations and collection of surface material at these localities did not recover any squamate elements. This identification has increased the taxonomic diversity of the Pinpa Local Fauna in the Namba Formation to include at least one skink, with further material yet to be described attributed to the Gekkota. These discoveries are attributed to the use of fine mesh-aperture sieves and bulk screening of concentrated lenses of fossiliferous sediments.

a. Namba environment

The excavation and subsequent analyses of the sediment profile at both Billeroo Creek (BC2) and Lake Pinpa (LP12) has allowed an interpretation of palaeoenvironmental conditions in the Frome Basin during the Oligo-Miocene. Dolomite in freshwater conditions requires a flooded alkaline environment to precipitate, as does calcite [53], so the presence of these minerals in Layers 4–5 (absent in layers 2 and 3) are most likely the result of the seasonal fluctuations of the water level in the palaeo-lakes in which the sediments comprising the Namba Formation were deposited during the late Oligocene–early Miocene [see 40]. Fluctuations in the rainfall and evaporation contributed to the formation of palygorskite in the lower Layers 3–5. Smectite is the result of the weathering of minerals sourced from older rock outcrop

(most likely from the nearby Flinders, Barrier, and Olary Ranges) deposited into the Namba area as swampy soils on the lake edge. Layers where dolomite is absent, i.e. Layer 3 according to the XRD result, reflect a drier phase in the palaeo-environment without an alkaline waterbody at the surface.

Callen [40] noted that the Namba basin was once flooded, forming a larger palaeolake than the present Lake Frome. The palaeolake Namba supported a number of aquatic vertebrates including lungfish, *Obdurodon* platypus, crocodiles, cetaceans and various waterbirds [43, 54–56]; their combined presence is indicative of a deeper aquatic environment. The lake was subject to varying rainfall and evaporation rates, resulting in receding lake edges, creating cycles of dolomite precipitation (recorded in Layer 5). Callen [40] concluded that sufficient evaporation of surface water (conditions required for dolomite precipitation) could not occur in a consistently high-rainfall climate. Instead, a modern analogue might be where cool moist winters and hot dry summers result in a moderate seasonal rainfall and high evaporation rates that result in dolomite and calcite deposits such as in the Coorong lacustrine-estuarine system in southern South Australia [57]. There, lake levels rise after dry summers due to groundwater recharge, resulting in the dolomite precipitation.

The sedimentary evidence from palaeolake Namba reveal periods of evaporation and alkaline water bodies. This suggests the occurrence of periodic droughts during which fish die-offs might be expected in the evaporating water bodies and terrestrial fauna might die and accumulate along the lake-edge. The articulated skeletons found at LP12 and BC2 alongside concentrations of fish bone may be the result of these lake ‘deaths’. The remains of *Proegernia mikebulli* and other small terrestrial vertebrates are preserved in lenses of concentrated bones of aquatic vertebrates (fish, turtle, and crocodiles) within deposits reflecting these ephemeral lacustrine conditions. These lenses, as exemplified by the Fish Lens at BC2, and others exposed discontinuously at Lake Pinpa, were probably formed by littoral longshore currents disarticulating, mixing and concentrating bones. The material would not have to have been transported very far in order to mix animals from the two starkly different environments as the deposition site was in the littoral zone. Accumulation of small vertebrate material alongside associated larger vertebrate skeletons (at LP12) indicate that the transportation of material was not from a flash flood typical of tropical monsoon seasonality but rather accumulated along a shallow or ephemeral lake edge shoreline. Some small terrestrial vertebrate material is within a size range attributable to raptor predation [58, 59] and some specimens show corrosion thinning typical of that formed after raptor ingestion. Such material may be derived from raptor pellet accumulations washing into the lake where they disaggregated and mixed with the bones of the more common freshwater taxa. Undescribed accipitrids are known from the Pinpa Local Fauna and a skeleton of one was excavated from LP12 by Worthy/Camens-led expeditions, and is the subject of current research.

b. *Proegernia mikebulli* and crown group egeriines

Proegernia mikebulli was small and gracile compared to most living egeriines. Based on dentary length, an adult individual would be of a similar size to *Lissolepis coventryi*, around 100 mm SVL [60]. Examinations of tooth crown features across the type specimens reveals they share features with both *Lissolepis* and species of *Liopholis* (cf. *L. multiscutata* and *L. whitii* both of which have a cusp slightly posterior to the centre of the tooth crown). Tooth crown features of scincids potentially preserve phylogenetic and dietary signals. Detailed studies within and between the extant subfamilies may help elucidate any dietary preferences suggested by fossil dentition. Tooth striae are present on teeth in most egeriines with a specialised diet and no pattern in the number of striae between insectivorous, durophagous, or herbivorous species was obvious [see 23]. Within genera, the number and patterning of striae was fairly similar. *Cyclodomorphus* and *Tiliqua* increase the number of striae from other egeriines, partly due to the change in tooth shape leading to striae radiating on all sides from a central cusp. *Lissolepis* and *Liopholis* have fewer striae and all are restricted to the lingual face of the tooth crown, below the lingual cristae. The two fossil species of *Proegernia* have few striae, similar to *Lissolepis* and *Liopholis*. The buccal striae on the crowns of *Proegernia palankarinnensis* are unusual and were not observed on extant *Eutropis multiscutata*, *Sphenomorphus jobiensis* or *Eugongylus rufescens*. Examination of fine-scale tooth crown features may prove useful characters for morphological phylogenetic analyses of fragmentary or incomplete tooth-bearing scincid material in future.

From what can be deduced of the palaeoenvironment surrounding Lake Pinpa and Billeroo Creek in the late Oligocene–early Miocene, *Proegernia mikebulli* lived in a cool, moist climate region with seasonal hot, dry summers with high evaporation rates. Other squamates known from this region and time period include a small gekkotan, and possibly a constricting snake [43]. Although no evidence of *Proegernia palankarinnensis* was found in the Namba Formation sites, the presence of at least two Australian scincids, from neighbouring basins, covering similar time periods and environments may be an indication of the early diversity of Australian squamates. Extant Australian scincids are difficult to separate based on isolated cranial material. The same cryptic diversity may have been present in the Oligo-Miocene, so more than two species are likely present in both the Etadunna and Namba Formations.

Although neither analysis produced firm support for the phylogenetic position of *Proegernia mikebulli*, both parsimony and Bayesian analyses placed the new taxon outside of living (crown) Australian Egeriinae. *Proegernia* species have a larger tooth count than crown Australian egeriines, a splenial notch placed more anteriorly, and a ~1 mm long opening of Meckel’s groove anterior to the splenial notch. Each of these character states are representative of the predicted transitional form between the plesiomorphic outgroup morphology and the majority of crown Australian egeriines. These characters have pulled *Proegernia* outside of the Australian crown group, but not beyond the Sahul stem genera *Corucia* and *Tribolonotus*. However, their precise position outside this crown group is uncertain. If relationships of one or both *Proegernia* with *Corucia* are correct (Figure 13–Figure 14), then there were two migrations of Egeriinae into southern Sahul 40.47–32.7 Ma, or a single invasion followed by emigration of *Corucia* to the Solomon Islands. However, a large amount of missing data for these fossil taxa and low support for precise relationships means

that it is possible that these trees are wrong, and that homoplasy with *Corucia* is what pulls these taxa away from the crown Australian group. It is thus possible that *Proegeria* lies on the immediate stem to the Australian crown group, in which case only a single dispersal event to Australia need be assumed.

The tip-dated phylogeny inclusive of the two Oligo-Miocene *Proegeria* taxa retrieves the age of the Egeriinae as 50.12 Ma. The last common ancestor of Egeriinae and Lygosominae was mostly likely living in the far south-eastern edge of Asia during the early Eocene. The first egeriine evolved elsewhere when the continent of Sahul was far south of its current latitude [61]; there is no fossil evidence of their presence in Australia at this time. Crown Egeriinae most likely originated either in south-eastern Asia or on an island arc, before connecting with the Australian landmass in the late Eocene (32.7Ma). Similar origins have been hypothesised for agamids [22 Ma; 62], pythons [c. 35 Ma; 8], sphenomorphine skinks [12], and elapids [8].

9. Conclusion

Dated phylogenies with the new fossils suggest that egeriines are the oldest radiation of skinks in Australia. The diversification of the crown Australian egeriines (~33 Ma) occurred nearly 10 million years before the estimated diversification of other Australian skinks, i.e. the Sphenomorphinae (c. 25 Ma) and Eugongylinae [c. 20 Ma; 3]. However, their diversity lags behind these other, potentially more recent groups. The first egeriines would have shared the continent with pygopodid, diplodactylid and carphodactylid geckos, and madtsoiid snakes [5]. The search for more Paleogene lizard material to document the early evolution of Australia's diverse herpetofauna continues, with investigations of Eocene collections from Murgon in northern Queensland [63], and early Oligocene vertebrates from Pwerte Marnte Marnte [64].

Acknowledgments

Sue Double and Jenny Worthy for screening and sorting the fossil material. Rod Wells, Colin Doudy, Bob and Sue Tulloch, Amy Tschim, Warren Handley, Jacob Blokland, Carey Burke and Ellen Mather were all part of the collaborative field team. Dr Jason Gascooke for training on, and use of, the SEM for imaging and EDAX sedimentary analysis at Flinders Microscopy, an Australian Microscopy and Microanalysis Research Facility. We thank Sarah Harmer-Bassel and Jason Young of Flinders Analytical, for their ICPMS and X-ray diffraction (XRD) analyses of the sediment from the studied fossil sites. We thank Mary-Anne Binnie and Carolyn Kovach of the South Australian Museum, and Tim Ziegler from Museums Victoria, for access to collections, and loans of fossil and extant comparative material. The Willi Hennig Society for making TNT freely available. We especially thank Andrew Black (Blackie), manager, and Alec Wilson (prior owner) for access to Frome Downs Station that allowed this work.

Funding Statement

KMT was supported by an Australian Postgraduate Research Training Stipend. The Mark Mitchell Foundation funded part of the fieldwork component of this project, and SEM time.

Data Accessibility

All specimens are registered in the Palaeontology Collection of the South Australian Museum, MicroCT files are housed in the SAM Herpetology digital collection.

All executable files for phylogenetic analyses are available as electronic supplementary material.

Competing Interests

We have no competing interests.

Authors' Contributions

KMT conceived the project, conducted fieldwork, collected the data, performed analyses, produced all figures and wrote the manuscript. THW and ABC funded the project, conducted field work, contributed to discussions, and commented on multiple drafts of the manuscript; MNH and MSYL commented on multiple drafts of the manuscript and contributed to discussions. NB collected the stratigraphic samples and information as part of a separate honours project.

References

1. Hall, R. 2002 Cenozoic geological and plate tectonic evolution of SE Asia and the SW Pacific: computer-based reconstructions, model and animations. *Journal of Asian Earth Sciences*. **20**, 353–431.

2. White, M. E. 2006 Environments of the geological past. In *Evolution and biogeography of Australasian vertebrates*. (ed. ^eds. J. R. Merrick, Archer, A., Hickey, G. M., Lee, M. S. Y.), pp. 17–48. Oatlands, NSW: Auscipub.

3. Skinner, A., Hugall, A. F., Hutchinson, M. N. 2011 Lygosomine phylogeny and the origins of Australian scincid lizards. *Journal of Biogeography*. **38**, 1044–1058. (10.1111/j.1365-2699.2010.02471.x)

4. Oliver, P., Hugall, A. 2017 Phylogenetic evidence for mid-Cenozoic turnover of a diverse continental biota. *Nat Ecol Evol.* **1**, 1896–1902. (10.1038/s41559-017-0355-8)
5. Oliver, P. M., Sanders, K. L. 2009 Molecular evidence for Gondwanan origins of multiple lineages within a diverse Australasian gecko radiation. *Journal of Biogeography.* **36**, 2044–2055.
6. Vidal, N., Hedges, S. B. 2009 The molecular evolutionary tree of lizards, snakes, and amphisbaenians. *Comptes rendus biologiques.* **332**, 129–139.
7. Hugall, A. F., Lee, M. S. Y. 2004 Molecular claims of Gondwanan age for Australian agamid lizards are untenable. *Molecular Biology and Evolution.* **21**, 2102–2110. (10.1093/molbev/msh219)
8. Sanders, K. L., Lee, M. S. Y. 2008 Molecular evidence for a rapid late-Miocene radiation of Australasian venomous snakes (Elapidae, Colubroidea). *Molecular Phylogenetics and Evolution.* **46**, 1165–1173. (<http://dx.doi.org/10.1016/j.ympev.2007.11.013>)
9. Lambert, S. M., Reeder, T. W., Wiens, J. J. 2015 When do species-tree and concatenated estimates disagree? An empirical analysis with higher-level scincid lizard phylogeny. *Molecular phylogenetics and evolution.* **82**, 146–155.
10. Uetz, P., Freed, P., Hošek, J. e. The Reptile Database. 2019.
11. Skinner, A., Hutchinson, M. N., Lee, M. S. Y. 2013 Phylogeny and divergence times of Australian *Sphenomorphus* group skinks (Scincidae, Squamata). *Molecular Phylogenetics and Evolution.* **69**, 906–918. (<http://dx.doi.org/10.1016/j.ympev.2013.06.014>)
12. Rabosky, D. L., Donnellan, S. C., Talaba, A. L., Lovette, I. J. 2007 Exceptional among-lineage variation in diversification rates during the radiation of Australia's most diverse vertebrate clade. *Proceedings of the Royal Society of London B: Biological Sciences.* **274**, 2915–2923. (10.1098/rspb.2007.0924)
13. Gardner, M. G., Hugall, A. F., Donnellan, S. C., Hutchinson, M. N., Foster, R. 2008 Molecular systematics of social skinks: phylogeny and taxonomy of the *Egernia* group (Reptilia: Scincidae). *Zoological Journal of the Linnean Society.* **154**, 781–794. (10.1111/j.1096-3642.2008.00422.x)
14. Chapple, D. G., Ritchie, P. A., Daugherty, C. H. 2009 Origin, diversification, and systematics of the New Zealand skink fauna (Reptilia: Scincidae). *Molecular Phylogenetics and Evolution.* **52**, 470–487. (<https://doi.org/10.1016/j.ympev.2009.03.021>)
15. Thorn, K. M., Hutchinson, M. N., Archer, M., Lee, M. S. Y. 2019 A new scincid lizard from the Miocene of Northern Australia, and the evolutionary history of social skinks (Scincidae: Egerniinae). *Journal of Vertebrate Paleontology.* **39**, e1577873. (10.1080/02724634.2019.1577873)
16. Martin, J. E., Hutchinson, M. N., Meredith, R., Case, J. A., Pledge, N. S. 2004 The oldest genus of scincid lizard (Squamata) from the Tertiary Etadunna Formation of South Australia. *Journal of Herpetology.* **38**, 180–187. (10.1670/25-03A)
17. Hocknull, S. A. 2000 Remains of an Eocene skink from Queensland. *Alcheringa: An Australasian Journal of Palaeontology.* **24**, 63–64. (10.1080/03115510008619524)
18. Estes, R. 1984 Fish, amphibians and reptiles from the Etadunna Formation, Miocene of South Australia. *Australian Zoologist.* **21**, 335–343.
19. Woodburne, M. O., Macfadden, B. J., Case, J. A., Springer, M. S., Pledge, N. S., Power, J. D., Woodburne, J. M., Springer, K. B. 1994 Land mammal biostratigraphy and magnetostratigraphy of the Etadunna Formation (late Oligocene) of South Australia. *Journal of Vertebrate Paleontology.* **13**, 483–515.
20. Tedford, R. H., Archer, M., Bartholomai, A., Plane, M., Pledge, N. S., Rich, T., Rich, P., Wells, R. T. 1977 The discovery of Miocene vertebrates, Lake Frome area, South Australia. *BMR Journal of Australian Geology & Geophysics.* **2**, 53–57.
21. Evans, S. E. 2008 The skull of lizards and Tuatara. In *Biology of the Reptilia 20, Morphology H The skull of Lepidosauria.* (ed. ^eds. C. Gans, A. S. Gaunt, K. Adler), pp. 1–343. Ithaca, NY: Society for the Study of Amphibians and Reptiles.
22. Richter, A. 1994 Lacertilia aus der Unteren Kreide von Una und Galve (Spanien) und Anoual (Marokko). *Berliner geowissenschaftliche Abhandlungen (E: Paläobiologie).* **14**, 1–147.
23. Kosma, R. 2003 The dentitions of recent and fossil scinciform lizards (Lacertilia, Squamata) - Systematics, Functional Morphology, Paleology. Hanover, Germany: University Hannover.
24. Russell, A. P., Bauer, A. M. 2008 The appendicular locomotor apparatus of *Sphenodon* and normal-limbed squamates. In *The skull and appendicular locomotor apparatus of Lepidosauria.* (ed. ^eds. C. Gans, A. S. Gaunt, K. Adler), pp. Ithaca, New York, USA: Society for the Study of Amphibians and Reptiles.
25. Hoffstetter, R., Gasc, J.-P. 1969 Vertebrae and ribs of modern reptiles. In *Biology of the Reptilia.* (ed. ^eds. C. Gans), pp. 201–310. New York: Academic Press.
26. Goloboff, P. A., Catalano, S. A. 2016 TNT version 1.5, including a full implementation of phylogenetic morphometrics. *Cladistics.* **32**, 221–238.
27. Drummond, A. J., Suchard, M. A., Xie, D., Rambaut, A. 2012 Bayesian phylogenetics with BEAUti and the BEAST 1.7. *Molecular Biology and Evolution.* **29**, 1969–1973.
28. Tonini, J. F. R., Beard, K. H., Ferreira, R. B., Jetz, W., Pyron, R. A. 2016 Fully-sampled phylogenies of squamates reveal evolutionary patterns in threat status. *Biological Conservation.* **204, Part A**, 23–31. (<https://doi.org/10.1016/j.biocon.2016.03.039>)
29. Lanfear, R., Frandsen, P. B., Wright, A. M., Senfeld, T., Calcott, B. 2016 PartitionFinder 2: new methods for selecting partitioned models of evolution for molecular and morphological phylogenetic analyses. *Molecular Biology and Evolution.* **34**, 772–773.
30. Maddison, W., Maddison, D. Mesquite: a modular system for evolutionary analysis. Version 3.2. 2017.
31. Lewis, P. O. 2001 A likelihood approach to estimating phylogeny from discrete morphological character data. *Systematic Biology.* **50**, 913–925.
32. Alekseyenko, A. V., Lee, C. J., Suchard, M. A. 2008 Wagner and Dollo: a stochastic duet by composing two parsimonious solos. *Systematic Biology.* **57**, 772–784.
33. Goloboff, P. A., Pittman, M., Pol, D., Xu, X. 2018 Morphological data sets fit a common mechanism much more poorly than DNA sequences and call into question the Mk model. *Systematic Biology.* **68**, 494–504. (10.1093/sysbio/syy077)
34. Wright, A. M., Hillis, D. M. 2014 Bayesian analysis using a simple likelihood model outperforms parsimony for estimation of phylogeny from discrete morphological data. *PLoS One.* **9**, e109210.
35. O'Reilly, J. E., Puttick, M. N., Parry, L., Tanner, A. R., Tarver, J. E., Fleming, J., Pisani, D., Donoghue, P. C. 2016 Bayesian methods outperform parsimony but at the expense of precision in the estimation of phylogeny from discrete morphological data. *Biology Letters.* **12**, 20160081.
36. Harmon, L. J. 2019 *Phylogenetic comparative methods.* 1.4 ed. Published Online: Luke Harmon.
37. Stadler, T. 2010 Sampling-through-time in birth–death trees. *Journal of Theoretical Biology.* **267**, 396–404. (<https://doi.org/10.1016/j.jtbi.2010.09.010>)
38. Drummond, A. J., Ho, S. Y. W., Phillips, M. J., Rambaut, A. 2006 Relaxed phylogenetics and dating with confidence. *PLOS Biology.* **4**, e88. (10.1371/journal.pbio.0040088)
39. Rambaut, A., Drummond, A. J., Xie, D., Baele, G., Suchard, M. A. 2018 Posterior summarisation in Bayesian phylogenetics using Tracer 1.7. *Systematic Biology.* **67**, 901–904. (doi:10.1093/sysbio/syy032)
40. Callen, R. A. 1977 Late Cainozoic environments of part of northeastern South Australia. *Journal of the Geological Society of Australia: An International Geoscience Journal of the Geological Society of Australia.* **24**, 151–169.
41. Rich, T. H. 1991 Monotremes, placentals, and marsupials: their record in Australia and its biases. In *Vertebrate Palaeontology of Australasia.* (ed. ^eds. P. Vickers-Rich, J. M. Monaghan, R. F. Baird, T. H. Rich), pp. 894–1057. Melbourne, Australia: Pioneer Design Studio Pty Ltd.
42. Rich, T. 1984 News from Foreign members: Australia, Museum of Victoria, Melbourne. *Society of Vertebrate Paleontology News Bulletin.* **130**, 28.
43. Vickers-Rich, P., Rich, P. V. 1991 *Vertebrate palaeontology of Australasia.* Melbourne, Australia: Pioneer Design Studio.
44. Callen, R. A., Tedford, R. H. 1976 New Late Cainozoic rock units and depositional environments, Lake Frome area, South Australia. *Transactions of the Royal Society of South Australia.* **100**, 125–167.
45. Centore, P. Conversions between the Munsell and sRGB colour systems.

- <http://www.munsellcoloursceinceforpainters.com/> 2013:64.
46. Opperl, M. 1811 *Die Ordnungen, Familien und Gattungen der Reptilien, als Prodrom einer Naturgeschichte derselben*. München: Joseph Lindauer.
47. Gray, J. E. 1825 A synopsis of the genera of reptiles and Amphibia, with a description of some new species. *Annals of Philosophy*. **10**, 193–217.
48. Welch, K. 1982 Herpetology of the Old World II. Preliminary comments on the classification of skinks (Family Scincidae) with specific reference to those genera found in Africa, Europe, and southwest Asia. *Herpetile*. **7**, 25–27.
49. Megirian, D., Prideaux, G., Murray, P., Smit, N. 2010 An Australian land mammal age biochronological scheme. *Paleobiology*. **36**, 658–671. (10.1666/09047.1)
50. Gelnaw, W. B. 2011 On the cranial osteology of *Eremiascincus* and its use for identification: East Tennessee State University.
51. Lee, M. S. Y., Hutchinson, M. N., Worthy, T. H., Archer, M., Tennyson, A. J. D., Worthy, J. P., Scofield, R. P. 2009 Miocene skinks and geckos reveal long-term conservatism of New Zealand's lizard fauna. *Biology Letters*. **5**, 833–837. (doi:10.1098/rsbl.2009.0440)
52. Greer, A. E. 1989 *The biology and evolution of Australian lizards*. Chipping Norton, NSW: Surrey Beatty & Sons.
53. Fukushi, K., Matsumiya, H. 2018 Control of Water Chemistry in Alkaline Lakes: Solubility of Monohydrocalcite and Amorphous Magnesium Carbonate in CaCl₂–MgCl₂–Na₂CO₃ Solutions. *ACS Earth and Space Chemistry*. **2**, 735–744. (10.1021/acsearthspacechem.8b00046)
54. Worthy, T. H. 2009 Descriptions and phylogenetic relationships of two new genera and four new species of Oligo-Miocene waterfowl (Aves: Anatidae) from Australia. *Zoological Journal of the Linnean Society*. **156**, 411–454. (10.1111/j.1096-3642.2008.00483.x)
55. Worthy, T. H. 2011 Descriptions and phylogenetic relationships of a new genus and two new species of Oligo-Miocene cormorants (Aves: Phalacrocoracidae) from Australia. *Zoological Journal of the Linnean Society*. **163**, 277–314. (10.1111/j.1096-3642.2011.00693.x)
56. Fitzgerald, E. M. G. 2004 A review of the Tertiary fossil Cetacea (Mammalia) localities in Australia. *Memoirs of Museum Victoria*. **61**, 183–208.
57. Borch, C. C. V. D., Lock, D. 1979 Geological significance of Coorong dolomites. *Sedimentology*. **26**, 813–824. (10.1111/j.1365-3091.1979.tb00974.x)
58. Andrews, P. 1990 *Owls, caves and fossils*. London: The Natural History Museum.
59. McDowell, M. C., Medlin, G. C. 2009 The effects of drought on prey selection of the barn owl (*Tyto alba*) in the Strzelecki Regional Reserve, north-eastern South Australia. *Australian Mammalogy*. **31**, 47–55. (<http://dx.doi.org/10.1071/AM08115>)
60. Wilson, S., Swan, G. 2017 *A complete guide to the Reptiles of Australia*. 5th ed. Sydney, Australia: Reed New Holland Publishers.
61. Müller, R. D., Seton, M., Zahirovic, S., Williams, S. E., Matthews, K. J., Wright, N. M., Shephard, G. E., Maloney, K. T., Barnett-Moore, N., Hosseinpour, M., et al. 2016 Ocean basin evolution and global-scale plate reorganization events since Pangea breakup. *Annual Review of Earth and Planetary Sciences*. **44**, 107–138. (10.1146/annurev-earth-060115-012211)
62. Hugall, A. F., Foster, R., Hutchinson, M., Lee, M. S. Y. 2008 Phylogeny of Australasian agamid lizards based on nuclear and mitochondrial genes: implications for morphological evolution and biogeography. *Biological Journal of the Linnean Society*. **93**, 343–358. (10.1111/j.1095-8312.2007.00911.x)
63. Godthelp, H., Archer, M., Cifelli, R., Hand, S. J., Gilkeson, C. F. 1992 Earliest known Australian Tertiary mammal fauna. *Nature*. **356**, 514–516. (10.1038/356514a0)
64. Murray, P. F., Megirian, D. 2006 The Pwerte Marnte Marnte Local Fauna: a new vertebrate assemblage of presumed Oligocene age from the Northern Territory of Australia. *Alcheringa: An Australasian Journal of Palaeontology*. **30**, 211–228. (10.1080/03115510609506864)
65. Woodhead, J., Hand, S. J., Archer, M., Graham, I., Sniderman, K., Arena, D. A., Black, K. H., Godthelp, H., Creaser, P., Price, E. 2016 Developing a radiometrically-dated chronologic sequence for Neogene biotic change in Australia, from the Riversleigh World Heritage Area of Queensland. *Gondwana Research*. **29**, 153–167. (<http://dx.doi.org/10.1016/j.gr.2014.10.004>)

Tables

Table 1: Fossil calibrations used, their minimum and maximum ages (Ma) and references for the age dates or species descriptions.

Fossil calibration	Min age	Max age	References for age
Zone A, Etadunna Formation	25.5	25.7	Woodburne MO, Macfadden BJ, Case JA, Springer MS, Pledge NS, Power JDet al. [19]
Pinpa Local Fauna, Namba Formation	25.5	25.7	Woodburne MO, Macfadden BJ, Case JA, Springer MS, Pledge NS, Power JDet al. [19]
AL 90 Locality, Carl Creek Limestone	14.17	15.11	Woodhead J, Hand SJ, Archer M, Graham I, Sniderman K, Arena DA, Black KH, Godthelp H, Creaser PPrice E [65]
Gag Locality, Carl Creek Limestone	14.47	16.86	Woodhead J, Hand SJ, Archer M, Graham I, Sniderman K, Arena DAet al. [65]

Table 2: Major and trace elements detected by X-ray fluorescence analysis of sediments sampled Layer 3 and 5 from stratigraphic sections of Lake Pinpa Site 12 and Billeroo Creek Site 2. Values are %, minor trace elements making the total of each sample 100 are not shown.

	MgO	Al ₂ O ₃	SiO ₂	CaO	MnO	Fe ₂ O ₃
Pinpa 12 L3	4.043	8.815	46.53	0.238	0.2303	19.82
Pinpa 12 L5	10.45	7.444	43.64	7.901	4.227	8.105
Billeroo Creek L5	13.65	5.875	36.08	13.02	1.634	6.982

Table 3: Mineral components of sediment samples taken from Layers 3 and 5 at Lake Pinpa and Layer 5 at Billeroo Creek Site 2.

	Clay Minerals	Dolomite/ Ankerite	Goethite	Quartz	Other	Total
Pinpa 12 L3	79	-	9	11	1	100
Pinpa 12 L5	66	23	3	6	2	100
Billeroo Creek L5	54	38	2	5	1	100

Figure and table captions

Figure 1: Stratigraphic sections through the Namba Formation at Lake Pinpa at Site 12 (see Figure 2) and Billeroo Creek, Site 2, constructed in SedLog 3.1 from authors field observations in comparison with RH Tedford's sections described in notes and figures in [20, 44], depth in meters. Grain sizes were determined in the field. Colours are converted from Munsell figures to corresponding RGB values using [45].

Figure 2: Location maps of the Lake Pinpa and Billeroo Creek sites, Frome Downs Station, South Australia. Stratigraphic columns in Figure 1 were compiled from locations Site 12 (=LP12) and Site 2 Fish lens (=BC2) marked above.

Figure 3: Results of the soluble mineral analyses of Layers 2–4 and the white and black sediments from Layer 5 at Lake Pinpa Site 12 (left); and Billeroo Creek Site 2 Layers 2–5 (right).

Figure 4: 1a-c Holotype, a right dentary, SAM P57502, 2a-c paratype; a left dentary SAM P57499; two right post-dentary compound bones 3a-c SAM P57543 and 4a-c SAM P57542 of *Proegernia mikebulli* sp. nov. from the Fish Lens at BC2; and 5 complete reconstruction in medial view of the right mandible of *Proegernia mikebulli*, the coronoid and splenial (hatched area) are reconstructed based on modern egermiine specimens, no fossil representatives of these elements are known. Abbreviations: adf, adductor fossa; anf, angular facet; art., articular; d., dentary; gl, glenoid fossa; iaf, inferior alveolar foramen; mg, Meckel's groove; psf, posterior surangular foramen; rap, retroarticular process; san., surangular; and sy, symphysis.

Figure 5: Right maxilla of *Proegernia mikebulli* sp. nov. reconstructed from SAM P57541 (anterior fragment; top left) and SAM P57503 (near complete; top right). 'A' marks the same tooth locus on each specimen. Enclosed by the box is the tooth 'B'. Abbreviations: cl, *cuspis labialis*; cla, *culmen lateralis* anterior; cli, *cuspis lingualis*; clp, *culmen lateralis* posterior; fp, facial process; psm, palatine shelf of the maxilla; pxp, premaxillary process; sop, suborbital process; and str, striae.

Figure 6: Left premaxilla (SAM P57542) of *Proegernia mikebulli* sp. nov. from posterior (left), anterior (centre), and lateral (right) view. A, tooth one, enlarged in box. Abbreviations: flc, foramen of longitudinal canal; inp, internasal process; mp, maxillary process; nef, notch of the ethmoidal foramen; and o, osteoderm.

Figure 7: A and B: 24th tooth on the dentary SAM P57502, medial view, and C the 16th tooth in occlusal view with lateral and medial facies marked. Abbreviations: ai, *antrum intercristatum*; cl, *crista lingualis*; cm, *crista mesialis*; cul, *cuspis labialis*; cula, *culmen lateralis* anterior; and culp, *culmen lateralis* posterior.

Figure 8: Right pterygoid of *Proegernia mikebulli* sp. nov., SAM P57512 in dorsal (left) and ventral (right) views. Abbreviations: en, epipterygoid notch; pp, palatine process; pvc pterygoid ventral concavity; qp, quadrate process; and tp, ectopterygoid process.

Figure 9: A single vertebra (SAM P57514) potentially referable to *Proegernia mikebulli* sp. nov., recovered from Fish Lens, BC2. A, anterior view; B, posterior view; C, lateral view, anterior to the left; D, dorsal view; and E, ventral view. A proximal fragment of a right femur assigned to *Proegernia mikebulli* sp. nov. SAM P57540. F, dorsal; G, anterior; and H, ventral views.

Figure 10: Comparative SEM of *Proegernia mikebulli* sp. nov. (B,C) and microphotographs of *P. palankarinnensis* (A,D) from Martin et al. [16], in medial (above) and occlusal (below) views. Note the anterior extension of the splenial notch (3) in *P. palankarinnensis* and the increased the number of teeth/loci in *P. mikebulli*. Features described in text are labelled: 1) Medial inflection of the symphysis; 2) Dentary depth; 3) Position of the inferior alveolar foramen; 4) Meckel's Groove extension anterior of the inferior alveolar foramen, present in *P. palankarinnensis* and absent in *P. mikebulli*; 5) anterior extent of the internal septum similar in both taxa, 6) Overall robustness as dentary width, 7) facet for articulation with anterior ramus of coronoid, 8) splenial notch height, and 9) height of the coronoid process relative to the last tooth.

Figure 11: Comparative medial views of the dentaries of *Eutropis multifasciata* (SAMA R35693), *Proegernia mikebulli* sp. nov., *Lissolepis coventryi* (SAMA R57317) and *Liopholis multiscutata* (FUR168). 1 Anterior extent of splenial; 2 Symphysis shape and robustness; and 3 Dentary depth at mid-tooth row length.

Figure 12: Medial view of left dentary tooth crowns of *Plestiodon fasciatus* (SAMA R66784), *Eutropis multifasciata* (SAMA R35693), *Proegernia mikebulli* sp. nov. (SAM P57502), *Lissolepis coventryi* (SAMA R57317), *Eugongylus rufescens* (SAMA R36735) and *Sphenomorphus jobiensis* (SAMA R6736). *Plestiodon* represents the basal skink tooth condition as described by Richter (1994). *Culmen lateralis* posterior is labelled culp.

Figure 13: Egermiine phylogeny based on the single most parsimonious tree produced by TNT [26]. Bootstrap values >50 shown.

Figure 14: Egermiine phylogeny based on consensus total evidence Bayesian inference tree from BEAST. †denotes extinct taxon. Small numbers at nodes denote posterior probability. Reconstructed node ages for key clades are in brackets, the green shading denotes the Eocene, after which Australia became isolated from Gondwana [4].

1 Table 1: Fossil calibrations used, their minimum and maximum ages (Ma) and references for the age dates or
2 species descriptions.

3 Table 2: Major and trace elements detected by X-ray fluorescence analysis of sediments sampled Layer 3 and
4 5 from stratigraphic sections of Lake Pinpa Site 12 and Billeroo Creek Site 2. Values are %, minor trace
5 elements making the total of each sample 100 are not shown.

6 Table 3: Mineral components of sediment samples taken from Layers 3 and 5 at Lake Pinpa and Layer 5 at
7 Billeroo Creek Site 2.

8
9
10
11
12
13
14
15
16
17
18
19
20
21
22
23
24
25
26
27
28
29
30
31
32
33
34
35
36
37
38
39
40
41
42
43
44
45
46
47
48
49
50
51
52
53
54
55
56
57
58
59
60

Appendix C**ROYAL SOCIETY
OPEN SCIENCE****A new species of *Proegernia* from the Namba Formation in
South Australia and the early evolution of Australian
egerniine skinks**

Journal:	Royal Society Open Science
Manuscript ID	RSOS-201686
Article Type:	Research
Date Submitted by the Author:	20-Sep-2020
Complete List of Authors:	Thorn, Kailah; Flinders University College of Science and Engineering, ; South Australian Museum Hutchinson, Mark; South Australian Museum Lee, Michael; Flinders University, College of Science and Engineering; South Australian Museum Brown, Nathan; Flinders University, College of Science and Engineering Camens, Aaron; Flinders University, College of Science and Engineering Worthy, Trevor Henry; Flinders University, College of Science and Engineering
Subject:	Palaeontology < EARTH SCIENCES, evolution < BIOLOGY, taxonomy and systematics < BIOLOGY
Keywords:	Scincidae, Egerniinae, Oligocene, Miocene, Palaeontology, Namba
Subject Category:	Organismal and Evolutionary Biology

Author-supplied statements

Relevant information will appear here if provided.

Ethics

Does your article include research that required ethical approval or permits?:

This article does not present research with ethical considerations

Statement (if applicable):

CUST_IF_YES_ETHICS :No data available.

Data

It is a condition of publication that data, code and materials supporting your paper are made publicly available. Does your paper present new data?:

Yes

Statement (if applicable):

All specimens are registered in the Palaeontology Collection of the South Australian Museum, MicroCT files are housed in the SAM Herpetology digital collection.

All executable files for phylogenetic analyses are available as electronic supplementary material.

Conflict of interest

I/We declare we have no competing interests

Statement (if applicable):

CUST_STATE_CONFLICT :No data available.

Authors' contributions

This paper has multiple authors and our individual contributions were as below

Statement (if applicable):

KMT conceived the project, conducted fieldwork, collected the data, performed analyses, produced all figures and wrote the manuscript. THW and ABC funded the project, conducted field work, contributed to discussions, and commented on multiple drafts of the manuscript; MNH and MSYL commented on multiple drafts of the manuscript and contributed to discussions. NB collected the stratigraphic samples and information as part of a separate honours project.

Origins of Australian egeriinesROYAL SOCIETY
OPEN SCIENCE*R. Soc. open sci.*
doi:10.1098/not yet assigned**A new species of *Proegernia* from the Namba Formation in South Australia and the early evolution of Australian egeriine skinks****K. M. Thorn^{*1,2}, M. H. Hutchinson^{1,2}, M. S. Y. Lee^{1,2}, N. Brown¹, A. B. Camens¹ and T. H. Worthy¹***College of Science and Engineering, Flinders University, BEDFORD PARK, 5042, South Australia
South Australian Museum, North Terrace, ADELAIDE, 5000, South Australia***Keywords:** Scincidae, Egeriinae, Oligocene, Miocene, Palaeontology

ZOOBANK ID urn:lsid:zoobank.org:pub:9C892991-E95A-4ABA-93B9-9480308A7E73

1. Summary

New Oligo-Miocene fossil vertebrates from the Namba Formation (south of Lake Frome, South Australia) were uncovered from multiple expeditions from 2007–2018. Abundant disarticulated material of small vertebrates was concentrated in shallow lenses along the palaeo-lake edges, and is now exposed on the western shore. This fossiliferous deposit, also known from Billeroo Creek 2 km northeast of Lake Pinpa, includes abundant aquatic (such as fish, platypus *Obdurodon*, and waterfowl) and diverse terrestrial (such as possums, dasyuromorphs, and scincids) vertebrates and is hereafter recognised as the Fish Lens. The stratigraphic provenance of these deposits in relation to prior finds in the area is also established. A new egeriine scincid taxon *Proegernia mikebulli* sp. nov. described herein, is based on a near-complete reconstructed mandible, maxilla, premaxilla, and pterygoid. Postcranial scincid elements were also recovered with this material, but could not yet be confidently associated with *P. mikebulli*. This new taxon is recovered as the sister species to *P. palankarinnensis*, in a tip-dated total-evidence phylogenetic analysis, where both are recovered as stem Australian egeriines. These taxa also help pinpoint the timing of the arrival of scincids to Australia, with egeriines the first radiation to reach the continent.

2. Introduction

Major biogeographical events have shaped the Australian flora and fauna, from the separation from Antarctica ~45 million years ago, through to Pleistocene glacial cycles [1, 2]. Australia's herpetofauna can be broadly traced to two origins: relicts from the breakup of Gondwana (the southern supercontinent), or recent arrivals from Asia to Sahul (the continental mass including Australia and New Guinea) [3, 4]. Gondwanan origins are inferred for diplodactyloid geckos, chelid turtles and some frogs [5, 6]. The Asian route appears to have been taken by agamid lizards, and elapids and typhlopoid snakes sometime during the Oligo-Miocene [3, 7, 8]. The temporal origins of Australian scincid lizard radiations are still relatively poorly time-constrained, but molecular data indicate that their closest extant relatives are in south-east Asia [9].

The Australian scincids comprise three subfamilies: the Egeriinae Welch, 1982, Sphenomorphinae Welch, 1982 and Eugongylineae Welch, 1982 [10]. While these three clades have been included in large all-squamate phylogenies, detailed molecular phylogenetic analyses have also been conducted for the sphenomorphine [11, 12] and egeriine radiations [13], but no recent molecular phylogeny of the Australian eugongyline skinks has been published [see 14, which includes some Australian taxa]. A molecular phylogenetic analysis of Australian skinks [3] produced a node-calibrated molecular clock estimate of the divergence of all three subfamilies. Those results suggested that the Australian Egeriinae arrived 18.2 Ma, Eugongylineae 22.9 Ma, and the Sphenomorphinae were the first group to reach Australia 25.3 Ma. However, analyses using mid-Miocene fossil calibrations established a much earlier origin for the Australian

*Author for correspondence (Kailah.thorn@uwa.edu.au).

†Present address: Edward de Courcy Clarke Earth Science Museum, School of Earth Sciences M004, University of Western Australia, 35 Stirling Highway CRAWLEY 6009 Australia

egerniines, minimally 34.61 Ma [15]. This new date prompted the current investigation to seek fossil evidence that might further demarcate the timing of the arrival of (potentially) Australia's first scincids.

The oldest Australian fossil with distinctive scincid characters, the egeriine *Proegernia palankarinnensis* Martin et al., 2004, is from the Etadunna Formation, Lake Eyre Basin, at the Oligo-Miocene boundary 25–26 Ma [16]. An Eocene femur loosely assigned to Scincomorpha by Hocknull is not convincingly scincid [17], so is not robust evidence for inferring the temporal origin of the Australian Scincidae. Other Oligo-Miocene Australian material from the Etadunna Formation was referred to Egeriinae but remains undescribed including: a dentary (UCR 20814), broken parietal (UCR 20815), partial maxilla (UCR 20816), an assortment of vertebrae and a broken scapulocoracoid [18]. Both the *Proegernia palankarinnensis* holotype and the material referred by Estes [18] cannot currently be located. New collections of Oligo-Miocene material are required to resolve the composition and relationships of Australia's oldest scincid faunas. Recent expeditions into central South Australia have unearthed new Oligo-Miocene fossil squamates from the Namba Formation at Lake Pinpa and Billeroo Creek, southeast of Lake Frome. The Namba Formation is of a similar age to the Etadunna [19], and the vertebrate fossil material collected from the lowest layers of this Formation are termed the Pinpa Fauna [20]. This investigation revises the stratigraphic and palaeoecological setting of this deposit, and places the new Oligo-Miocene egeriine fossils into phylogenetic context, alongside *P. palankarinnensis* from the Etadunna, and the Miocene species of *Egernia* and *Tiliqua* analysed in Thorn et al. [15], to infer a more robust date for the arrival of scincids to the continent of Sahul.

3. Materials and Methods

a. Stratigraphy, excavation and fossil collection

Sediment samples from excavated trenches were taken to examine the stratigraphy of both Billeroo Creek (BC2) and Site 12, Lake Pinpa (LP12). Mineral composition was determined by XRD (X-ray diffraction) and XRF (X-ray fluorescence) analyses conducted by Flinders Analytical. Nested sieves with mesh apertures of 6, 3 and 1 mm were used to create sediment fractions from which the fossil material was sorted from the dolomitic clays. The micro-vertebrate material was sorted into fish, mammal, bird, turtle, crocodile, frog, and squamate. Of the 48 squamate cranial specimens recovered, 43 were scincids, and 5 identified as geckos (see Supplementary Information for the squamate specimen list). Most fossils discussed in this investigation were recovered by sieving excavated sediment and sorting the concentrates, however some were collected from the ground surface having been exposed by erosion.

All descriptive terminology of the cranial elements follows Evans [21] and Richter [22] and Kosma [23] for tooth crown features. Appendicular skeleton terminology follows Russell and Bauer [24], and Hoffstetter and Gasc [25] for vertebrae.

b. Scanning electron microscopy

Cranial material identified to Scincidae was further cleaned in water, dried, and then imaged using Scanning Electron Microscopy (SEM). The minute size of some specimens is beyond the capabilities of Micro CT for resolution of morphological features required for identification and descriptions. No specimens required sputter coating. All SEM work was conducted at Flinders University Microscopy facilities using an Inspect FEI F50 SEM. Maximum voltage used for image taking was limited to 2kV and a spot size of 3–4. Measurements of tooth crowns and features of the dentary mentioned in the text were taken using the SEM software.

c. Phylogenetic analyses

In order to better understand the timing of the Australian colonisation by the Egeriinae, both molecular and morphological data (including fossils) are required to generate tip-dated phylogenies. Undated parsimony and tip-dated Bayesian analyses infer, respectively, the phylogeny with the least homoplasy, and the most probable dated phylogeny.

i. Morphological characters

Morphological characters used in the following analyses consisted of 102 discrete and 48 continuous traits, forming an expanded matrix from Thorn et al. [15]. Continuous characters, derived from the measurements of the individual bones or teeth from the dentaries and maxillae, were taken from either Micro-CT scan data in Avizo Lite (v. 9.0) or SEM at Flinders Microscopy, to the nearest micrometre, or with digital callipers to the nearest ten micrometres. All measurements were converted to ratios of either dentary or maxilla length to standardise for size. Continuous characters were converted to values spanning 0–2 to replicate the average number of discrete character states, for analyses in both TNT [26] and BEAST 1.8.3 [27], so that they do not have a disproportionate weight.

ii. Molecular partitions

Molecular data sourced from Tonini et al. [28] and Gardner et al. [13] were analysed using Partition Finder 2 [29] to find optimal partitions and substitution models. The same six molecular (gene) partitions, 12s (412 base pairs [bps]), 16s (681 bps), ND4 (693 bps), BDNF (699 bps), CMOS (835 bps) and B-fibrinogen (1051 alignable bps) and substitution models [15] are used again here.

iii. Maximum Parsimony

The parsimony analyses for the combined discrete morphological, continuous morphological, and molecular data were performed using TNT v.1.5 [26]. *Eutropis multifasciata* was set as the most distant outgroup following the

phylogenetic interpretations of Gardner et al. [13] and Thorn et al. [15]. The most parsimonious tree (MPT) for the combined data was found using 1000 replicates of tree-bisection-reconnection (TBR) with up to 1000000 trees held. To assess clade support, 200 partitioned bootstrap replicates (with discrete characters, continuous characters, and each gene locus treated as a separate resampling partition), were performed using TNT, using new search methods (XMULT) with 1000 replicates and 1000000 trees held. The MPT and bootstrap trees from TNT were exported in nexus format, and continuous and discrete characters were traced [in Mesquite; 30]. The executable files for finding the Most Parsimonious tree, and for performing 200 reps of Partitioned Bootstrap resamples can be found in the SI data files Namba_Egeriines_Topology.tnt (MPT file) and Namba_Egeriines_PartitionedBootstrap.tnt.

iv. Bayesian analysis

The discrete and continuous morphological data, and molecular data were simultaneously analysed in BEAST v1.8.4 using tip-dated Bayesian approaches [27]. *Eutropis multifasciata* was again set as the furthest outgroup. Polymorphic discrete morphological data were treated exactly as coded rather than as unknown, i.e. if coded as states (0,1) it was treated as 0 or 1, but not 2. The discrete character set was analysed using the Mkv-model with correction for non-sampling of constant characters [31, 32]. Despite recent disputes over the effectiveness of this model [33], it is well-tested [34, 35] and is still widely accepted and applied to morphological data [36]. Continuous characters, transformed to span values between 0 and 2, were analysed with the Brownian motion model. Bayes factors were used to test the need to accommodate among-character rate variability for both discrete and continuous morphological characters (i.e. gamma parameter).

The stratigraphic data used for tip-dating analyses were derived from fossil taxa and their associated stratigraphy noted in Table 1. No node age constraints were imposed in this analysis, all dates are retrieved from the morphological and stratigraphic age ranges from the noted fossil taxa (tips). The most appropriate available model in BEAST v.1.8.4, birth-death serial sampling [37], was applied. An uncorrelated relaxed clock [38] was separately applied to the molecular and morphological data.

Each Bayesian analysis was run for 100,000,000 generations with a burn-in of 20%. The analysis was conducted four times to confirm stationarity. The post-burnin samples of all four runs were examined in Tracer 1.7.1 [39] to ensure convergence was achieved. All four runs were combined in LogCombiner, and the consensus tree produced by TreeAnnotator [27]. The executable .xml file for BEAST, all output log files, and the final consensus tree file (.tree) are available as supplementary information.

4. Geological setting

a. The Namba Formation

The Namba Formation from north-eastern South Australia (see Figure 2) shares an unconformable lower boundary with the Eyre Formation (Paleocene-Eocene) within our study area in the Frome Basin; and is unconformably overlain by the Pleistocene Eurinilla Formation (Figure 1). The Namba sequence is a lateral equivalent to the Etadunna Formation from the north western Lake Eyre Basin [19, 40]. The Namba Formation is divided into two members [20]; Green claystones and dolomitic claystones at the top of the lower member host a locally abundant vertebrate fauna, termed the Pinpa Local Fauna [20]. This is biostratigraphically correlated with Zone A of the Etadunna Formation to be 25.5–25.7 Ma [19]. Sites visited during the late 1970s–early 1980s led by R.H. Tedford, T. H. Rich, and others, discovered and named multiple fossil localities exposed in Lakes Pinpa, Namba, Tarkarooloo, Yanda and Tinko and Billeroo Creek in the Lake Frome Basin [41]; added to these are new numbered sites from expeditions carried out in 2007 and between 2015–2018 led by T. H. Worthy and A. B. Camens.

All of the sites from Lake Pinpa and Billeroo Creek yielding scincid material for this investigation expose and sample fluvio-lacustrine sites from the Namba Formation. Five expeditions collected new material from the Namba Formation from 2007–2018. Over this period, numerous squamate fragments were collected from multiple sites at Lake Pinpa and Billeroo Creek. Two sites have yielded the majority of the material described herein (see section 4b below); Site 12 at Lake Pinpa, and the ‘Fish Lens’ at Billeroo Creek (within Wells’ Bog Site of Tedford’s, and later T.H. Rich’s, expeditions).

Deposits and the fossils therein contributing to the Pinpa Local Fauna may be classed into two clear taphonomic groups: those containing isolated bones or partial skeletons of terrestrial vertebrates (marsupials, predatory birds, wading birds, and meiolaniid turtles), and those containing localized concentrated bone accumulations which were previously thought to be derived from crocodile coprolites, encompassing mostly aquatic vertebrates (mostly fish, but including also turtle, crocodile, dolphin and rare terrestrial vertebrates). Collection of both categories of this material was predominantly from surface exposures along the western edge of Lake Pinpa and north eastern Billeroo Creek [41], with the occasional excavation of articulated or associated marsupial skeletons. In slightly younger overlying/incised fluvial units exposed as channel fill deposits, the Ericmas Local Fauna has been derived from large scale excavations at Ericmas and South Prospect quarries (Lake Namba, 5 km south of Lake Pinpa) in the 1970s and 1980s [41]. Tom O’s Quarries excavated in fluvial units at Lake Tarkarooloo unearthed the Tarkarooloo Local Fauna which is biochronologically similar to the Ericmas LF, by bulk processing and screening sediments on several expeditions led by T H Rich of Museums Victoria, once with the help of the Australian Army [42]. The quarried Ericmas LF sites contained predominantly terrestrial vertebrate remains, but until the 2007 Worthy and Camens expedition, no squamate material was recorded from the Namba Formation.

Squamate remains have predominantly been recovered alongside the smallest mammal taxa and were found in lenses of densely concentrated bones of aquatic vertebrates dominated by fish (*Actinopterygii* and *Neoceratodus* spp.). These fish lenses lie a few centimetres above dolomite of unknown depth (details below). The dolomite beds have revealed numerous fossils, including associated or partly articulated skeletons of birds and mammals, but fish bones are rare. This dolomitic bed was better exposed towards the middle of the lake bed in the 1970s when it was extensively sampled, but has since at least 2007 been buried by in-washed Quaternary sands. Both layers contain the same mammal and bird species and so the faunas from each are collectively referred to the Pinpa Local Fauna [20, 43]. Both the fish layer and underlying dolomites are exposed at Wells' Bog Site in Billeroo Creek shown in Figure 1.

Figure 1: Stratigraphic sections through the Namba Formation at Lake Pinpa at Site 12 (see Figure 2) and Billeroo Creek, Site 2, constructed in SedLog 3.1 from authors field observations in comparison with RH Tedford's sections described in notes and figures in [20, 44], depth in meters. Grain sizes were determined in the field. Colours are converted from Munsell figures to corresponding RGB values using [45].

b. Fossil sites

Fossil sites in the Lake Pinpa–Billeroo Creek area were numbered chronologically on the 2007 trip in order of discovery, not based on geographical location (Figure 2).

i. Lake Pinpa

Site 6 (LP6) SAMA P43058 a posterior fragment of a left scincid maxilla, was recovered from Site 6 on the 2007 expedition. The fish lens was observed eroding out in patches over a broad area (200 m by 50 m) here around the margins of the overlying massive grey clay (Layer 2 on Figure 1).

Site 9 (LP9) SAMA P43057 a posterior fragment of a left scincid dentary, was recovered from Site 9 on the 2007 expedition. Here the fish lens was exposed in a narrow zone at the base and edge of eroding massive clay Layer 2.

Site 12 (LP12) Specimens SAM P57544 (partial humerus) and SAM P57545 (maxilla fragment) were found at Site 12 on the edge of Lake Pinpa. Bone was found eroding out on the surface, along the lake edge, from the lowest silty clay layer. Associated and articulated skeletons occur in both this layer and the dolomitic clay (Layer 5) beneath. An undulating clean erosional boundary was observed between these two units.

ii. Billeroo Creek Site 2 Fish Lens

Billeroo Creek Site 2 (BC2) is located on the northern side of the creek (Figure 2) and is a part of the more expansive Wells' Bog Site. Fossils derive from a concentrated lens of predominantly fish bone with limonite inclusions, within the top of Layer 3 of the Namba Formation (Figure 1). The base of this fossiliferous 'Fish Lens' is not flat, but undulated relating to depth of semi-discrete lenses exposed over an area of roughly 10 m by 3 m, excavated over three trips, with a maximum thickness of ~150 mm. The lens sits stratigraphically ~200 mm above the basal dolomite (Layer 5). This layer is laterally more extensive than the Fish lens at Billeroo Creek, and is the layer from which the majority of skeletal fossils at the site have been collected, see SI for a complete list.

Figure 2: Location maps of the Lake Pinpa and Billeroo Creek sites, Frome Downs Station, South Australia. Stratigraphic columns in Figure 1 were compiled from locations Site 12 (=LP12) and Site 2 Fish lens (=BC2) marked above.

c. Stratigraphy

Need an introductory sentence or two

Layer 1— The top layer of the stratigraphic section at LP12 (Figure 1), designated Layer 1, is not part of the Namba Formation. The red sands are reworked sediment from the nearby Quaternary dunes that unconformably overlie the Namba exposures around many of the lakes in the area. At Billeroo Creek (BC2), the fluvial Eurinilla Formation lies unconformably on the Namba Formation and eroded sediments from both it and overlying dunes mantled the Namba Formation where the section was excavated.

Layer 2— Erosion of an unknown amount of the upper part of Layer 2 means its original depth at both BC2 and LP12 cannot be assessed, but near LP07, exposures as documented in the section by Tedford et al. [20], show it was minimally ~6 m thick. It sits unconformably on Layer 3. At LP12, Layer 2 is composed of interbedded light green-grey (7/1/10Y) and yellow (5/1/10Y) medium silts that display cross bedding with very fine laminations. From a distance they have an overall uniform, pale-grey appearance. The layer at BC2 is similar but with white (8/1/10YR) medium silt rather than yellow. No inclusions or fossils have been found in this unit, but locally vertically aligned gypsum crystal plates occur. Concentrations of most soluble minerals are lowest in this stratigraphic horizon.

Layer 3— The upper boundary of Layer 3 at LP12 is erosional, the troughs filled with sediments from Layer 2. Easily distinguishable from Layer 2, it is composed of an olive-grey (4/2/5Y) clay with strong brown (4/6/7.5YR) limonite inclusions throughout. Maximum thickness at Lake Pinpa was 150 mm. The same layer at BC2 was a dark greyish-brown (4/2/2.5Y) clay that had a maximum thickness of ~100 mm and limonite presence was less consistent. The results of the XRD supported field observations of the clay content of the sediment, with clay minerals Smectite and Palygorskite combining to make up 79% of Layer 3 at Lake Pinpa. Limonite inclusions were noted in the stratigraphic section in Layer 3. The XRD analysis of Layer 3 sediment from Lake Pinpa identified the iron ore mineral Goethite (9% of total mineral composition) a common component of Limonite; this was supported by the XRF analyses which found 19.82% Iron (II) and soluble Fe content >16% at both sites. This spike in iron is not reached by any of the preceding layers. The Fish Lens at BC2 had variable thickness within the top 50 mm of this layer, but in some cases Layer 2 sits directly on Layer 3 sediments, with no Fish Lens separating the sediments of each layer. At Lake Pinpa, the Fish Lens is usually much thinner (1–10 mm thick) than at Billeroo Creek and occurs near the top of Layer 3. Exceptions occur at LP6, where the Fish Lens is locally much thicker, sometimes up to 50-120 mm thick in areas of a couple of square metres scattered over several hundred square meters on the lake bed. Layer 3 sits conformably on Layer 4 in stratigraphic sections at BC2, but the boundary is unconformable at LP12.

Layer 4— 150 mm of an olive (5Y 5/3) and orange (10YR 4/6) mottled clay that sits unconformably on Layer 5 at LP12 and BC2. Fossils were found in the lower portion of this layer, on the boundary with Layer 5 at both sites. At Billeroo Creek, Layer 4 is marked by a sharp increase in Calcium, not present until Layer 5 at Lake Pinpa.

Layer 5— This layer is composed of a dolomitic white (10YR 8/1) mudstone with extensive black manganese staining giving a mottled appearance overall. 66% of sediment in Layer 5 at Pinpa and 54% of Layer 5 at Billeroo Creek (Table 3) is derived of clay minerals Smectite and Palygorskite. Layer 5 had less Fe, but much more Mg, Ca and Mn than L3 and L4 (Figure 3 and Table 2), reflecting that it included the minerals Dolomite/Ankerite (Table 3), not present in the overlying layers at either site. The mottled appearance of the dolomitic Layer 5 in both sections is explained by the transition from Dolomite ($\text{CaMg}(\text{CO}_3)_2$) to Ankerite ($\text{Ca}(\text{Fe},\text{Mg},\text{Mn})(\text{CO}_3)_2$) with the partial replacement of magnesium with iron (II) and manganese. Carbonate presence was confirmed with a dilute hydrochloric acid in the field, and reaffirmed by the soluble mineral result from both sites (Figure 3). The primary source of fossils contributing to the Pinpa Local Fauna at Lake Pinpa and Billeroo Creek (excluding Fish Lens) is the assemblage occurring at the bottom of Layer 4 and in the top 100 mm of Layer 5.

Figure 3: Results of the soluble mineral analyses of Layers 2–4 and the white and black sediments from Layer 5 at Lake Pinpa Site 12 (left); and Billeroo Creek Site 2 Layers 2–5 (right).

5. Results

Our excavations have expanded the taxonomic diversity of the Pinpa Local Fauna (first recorded by Tedford et al. [20]). Notably, in 2015-18 we recovered both associated skeletons and isolated bones from the base of Layer 4 and the top of the dolomitic clay layer (Layer 5) at BC2 and LP12. Excavations at LP12 revealed predominantly terrestrial taxa with numerous remains of marsupials (vombatiforms, phalangeriforms, macropodiforms) and birds. In comparison, the fauna from the Fish Lens over various sites is more aquatic, a possible current-concentrated, lake-edge accumulation with bony fish abundant and lungfish, dolphins, flamingos, rails, turtles, crocodiles and the platypus *Obdurodon* represented, in addition to terrestrial marsupials (mainly dasyuromorphs, phalangeriforms) and small scincids.

6. Systematic Palaeontology

Order SQUAMATA Opper, 1811 [46]
Family SCINCIDAE Gray, 1825 [47]
Subfamily EGERNIINAE Welch, 1982 [48]
Genus PROEGERNIA Martin et al., 2004 [16]
PROEGERNIA MIKEBULLI sp. nov.

Zoobank ID: urn:lsid:zoobank.org:act:068F625A-B537-43C7-A1B1-4F20DD5F6BF9

Holotype—SAM P57502, a near complete right dentary; 27 tooth loci, 14 of which bear teeth.

Diagnosis—The species is referred to the subfamily Egeriinae because the dentary has a closed Meckel's groove and a large inferior alveolar foramen. It is referred to the genus *Proegernia* because the tooth crowns widen anteroposteriorly from the shaft, the crista lingualis and crista mesialis are near horizontal and converge on an apex slightly posterior to the centre of the tooth, and there are medially-prominent cristae on the anterior and posterior culmen laterales. *Proegernia* is further distinguished from other members of the Egeriinae (species in *Egernia*, *Bellatorias*, *Liopholis*, *Cyclodomorphus*, *Tiliqua*, *Tribolonotus*, and *Corucia*) by the combination of the following traits: a more anteriorly positioned apex of the splenial notch at >50% the anteroposterior length of the dentary; more than 22 tooth loci on the dentary and 20 on the maxilla; minimal anteroposterior flaring of the tooth crown with lateral compression of the medial face; a small sliver of open Meckel's groove immediately posterior to the symphysis and a concave ventral face on the pterygoid body. *Proegernia mikebulli* differs from *P. palankarinnensis* in lacking lateral tooth striae, in having up to five more tooth loci for a total of 27 on the dentary, and a much more medially inflected anterior tip to the dentary, making it more curved in dorsal view.

Type locality—Fish Lens, subsite Billeroo Creek 2, Wells' Bog Site, northern side of Billeroo Creek, GPS coordinates S 31°6'11.76" E 140°13'53.70", See Figure 2.

Stratigraphy/Age—Namba Formation, in Layers 3–5 of the (Figure 1) as exposed in Billeroo Creek and Lake Pinpa; Pinpa Local Fauna. This LF has been biostratigraphically correlated with the "Wynyardiid" or Minkina Fauna, (Zone A) of the late Oligocene Etadunna Formation, 25.5–25.7 Ma [19, 49].

Paratype—SAM P57499 a partial right dentary from BC2, with a complete coronoid process, 23 tooth loci and 10 teeth.

Referred specimens—The following specimens are referred to this taxon based on the combination of their appropriate size, stratigraphic co-occurrence, and having a general similarity to the equivalent elements in other egeriines. Details of tooth crown shape also enable robust referral of tooth-bearing bones to the same taxon. Two right post-dentary compound bones; SAM P57543 from Lake Pinpa (LP6) preserving the dorsal surface of the surangular and glenoid, broken part-way through the retroarticular process; and SAM P57542 from Billeroo Creek (BC2) which preserves the ventral and medial face of the articular, the glenoid entirely and most of the retroarticular process. Two right partial maxillae both from BC2, SAM P57541 representing an anterior fragment with premaxillary process intact and the first 9 tooth loci holding 3 teeth; and SAM P57503 which preserves the posterior majority of the maxilla with 16 loci and 9 teeth, the facial process is broken above the row of maxillary foramina. An intact left pre-maxilla, SAM P57542 from BC2, with osteoderm fragments on the internasal process, preserving two teeth from four loci. A right pterygoid SAM P57512 from BC2, the quadrate process broken beneath the epipterygoid notch.

Etymology—The species is named after Professor Michael Bull (1947–2016) of Flinders University, South Australia, who devoted decades to documenting the ecology of Australia's egeriine skinks. Mike supervised a generation of Australian ecologists and his lectures inspired countless students to become biologists. His studies of Australian egeriine skinks and their parasites are model long-term ecological studies, and led to major discoveries such as the existence of monogamous pairs in *Tiliqua rugosa*, and parental care and family living in *Egernia* spp.; and the establishment of a successful breeding and reintroduction program for the endangered Pygmy Bluetongue, *Tiliqua adelaidensis*.

a. Description

Dentary—The most diagnostic and commonly-recovered element is the dentary bone of the lower jaw. Seventeen incomplete dentaries were recovered from the 2007–2018 trips to Lake Pinpa and Billeroo Creek. A reconstruction of a near-complete lower jaw of *Proegernia mikebulli* was made, with the dentary portion based on the holotype SAM P57502 and paratype SAM P57499 (See Figure 4). From the anterior tip of the symphysis to the posterior tip of the coronoid process, missing only the angular process, the reconstructed dentary is 12.7 mm long. In medial aspect, the dentary is convex ventrally. It is 1.6 mm wide and 2.2 mm tall at mid-length of the tooth row, and the dental sulcus is shallowly concave. From occlusal view, the symphysis is directed medially to articulate with the opposing dentary at an angle of 32°. The dental sulcus is clearly differentiated, SAM P57502 preserving 27 loci and 14 pleurodont teeth. The tooth row is 10.8 mm long.

Figure 4: 1a-c Holotype, a right dentary, SAM P57502, 2a-c paratype; a left dentary SAM P57499; two right post-dentary compound bones 3a-c SAM P57543 and 4a-c SAM P57542 of *Proegernia mikebulli* sp. nov. from the Fish Lens at BC2; and 5 complete reconstruction in medial view of the right mandible of *Proegernia mikebulli*, the coronoid and splenial (hatched area) are reconstructed based on modern egeriine specimens, no fossil representatives of these elements are known. Abbreviations: adf, adductor fossa; anf, angular facet; art., articular; d., dentary; gl, glenoid fossa; iaf, inferior alveolar foramen; mg, Meckel's groove; psf, posterior surangular foramen; rap, retroarticular process; san., surangular; and sy, symphysis.

The anterior symphysis, preserved in its entirety on SAM P57502, has a flattened, reverse '7' shape in medial view. The symphysis has two caudally-directed branches; a narrow, ventrally-directed sliver along the anterior edge of the dentary; and, dorsally, a wider process that terminates in a sharp point. Maximally, the symphysis is 1.8 mm long and 1.3 mm deep. In the notch between the caudally-directed branches of the symphysis a symphyseal foramen extends ventrally into a 1 mm long and <0.2 mm wide anterior opening of Meckel's groove. The groove is open and aligned parallel with the dorsal side of the ventral section of the symphysis; further posteriorly it is enclosed.

The inferior alveolar foramen and the dorsal and ventral margin of the splenial notch are preserved to the posterior end of the tooth row in SAM P57502 (Figure 4). The inferior alveolar foramen lies on the ventral margin of the splenial notch, below the mid-point of the tooth row. The shape of the splenial notch is narrow and roughly parallel-sided for the anterior 40% of the preserved length, expanding dorsoventrally with a convex upper edge and straight ventral margin for the remaining posterior section of length. Where the notch expands, the dorsal edge preserves a concave face, allowing the splenial to medially overlap the dentary. Posterior to this face, beneath the 3rd-last tooth, the dental sulcus is broken.

The coronoid process of the dentary is preserved on the paratype SAM P57499 and ascends posterodorsally from the position of the last tooth to a tip projecting dorsally above the posterior tooth. The ventral margin of this process preserves the anteromedial articular facet for the coronoid, which therefore can be seen to overlap the splenial and dentary anterior of the 3rd last tooth position. The angular process, although not preserved entirely on either dentary specimen, can be reconstructed with some confidence using the angle of the intact edge immediately beneath the coronoid process. The total length of this process is limited by the absence of a facet for its articulation on either of the recovered post-dentary compound bones.

The lateral face of the dentary is slightly convex and preserves a single row of eight mental foramina extending posteriorly from the anterior top of the dentary to below the 17th tooth position. The largest foramen in the row is at the posterior end.

Post-dentary (compound) bones— The post-dentary complex or compound bone is the fused surangular and articular bones making up the posterior half of the scincid mandible. This fusion develops ontogenetically; complete fusion of the two elements with no traces of a suture internally is evidence of adulthood in extant Australian scincids [50]. Two right post-dentary complexes were recovered from the Namba Formation, SAM P57543 from a right mandible at Lake Pinpa (LP6; Figure 4 3a–3c) and SAM P57542 from Billeroo Creek (BC2; Figure 4 4a–4c); these are used in the reconstruction of the complete lower jaw (Fig. 4, 5). These elements are referred to this taxon as they are of appropriate size, confirmed as scincid material by the presence of a facet for the angular bone, which is not present in gekkotans; and the alignment of the retroarticular process. The shape of the retroarticular process is similar to that of extant egeriines. When aligned horizontally in the in-vivo position, the angular in egeriines is wide and extends further laterally than the narrow ventrally positioned angular of eugongyline (i.e. *Emoia longicauda* SAMA R2352, *Eugongylus rufescens* R36735). Sphenomorphines generally have a simple, straight post-dentary complex without medial torsion of the articular, and the retroarticular is not inflected medially posterior of the glenoid. All scincid bones from Billeroo Creek and Lake Pinpa are consistent with presence of a single egeriine taxon; there is no variation among elements that might indicate more than one species being represented, so both compound bones are referred to the taxon named from the dentaries.

SAM P57543 is a near complete right adult post-dentary preserving completely fused surangular and articular bones. Anteriorly the areas for articulation with the dentary and coronoid are not preserved, neither are the angular or the anterior edge of the adductor fossa. A facet for articulation of the angular is present on the ventral half of the lateral face of the articular, not extending to reach the posterior surangular process. The posterior majority of the retroarticular process is broken off SAM P57543, so a description of its complete shape is based on SAM P57542. The dorsal surface of the surangular is relatively straight and rises slightly to meet the dorsal tip of the mandibular condyle or glenoid fossa where the quadrate articulates with the mandible. The maximum preserved length of this dorsal surface is 2.9 mm. The glenoid fossa faces posterodorsally, the surface is convex mediolaterally and concave anteroposteriorly, reaching a maximum length of 1.8 mm. The lower third of the glenoid fossa becomes slightly concave ventrally, curving up to meet the medial articular process of the surangular. The surface of the glenoid fossa has a pitted texture indicating attachment of cartilage in the mandibular cotyle. Posterovertrally the glenoid fossa ends in a clearly defined ridge between it and the retroarticular process. The preserved section of the retroarticular process is concave (SAM P57542). Medially, the foramen for the chorda tympani is preserved in SAM P57542.

Two foramina are preserved dorsally in a flattened surface, and a posterior surangular foramen is on the lateral face immediately anterior of the glenoid fossa. No sign of an anterior surangular foramen is present on either SAM P57543 or SAM P57542. From the flattened dorsal surface of the surangular, the bone curves sharply ventrally, leaving a flat medial surface above the dorsal margin of the adductor fossa. The adductor fossa is in the lower 50% of the medial face of the complex. The anterior margin of the adductor fossa is not preserved on either post-dentary compound bone. Viewed medially, the posterior edge of the narrow, ovular, fossa terminates anterior to the glenoid. The lateral face of the post-dentary complex of the mandible is convex. A discernible ridge runs from just anterior of the posterior surangular foramen, anteriorly to the broken anterior edge of SAM P57543, marking the posteroventral edge of the *M. adductor mandibulae externis*.

Maxilla— Two incomplete right maxillae (SAM P57503, SAM P57541) recovered from BC2 are referred to *Proegeria mikebulli*. SAM P57503 (Figure 5, right) preserves a near complete tooth row, a bifid posterior termination of the suborbital process (an apomorphy of the Scincidae), and complete tooth crowns from both the anterior tooth form and posterior tooth form on the same row. This specimen was recovered from sieved material in two halves, that fitted together when articulated, and so was joined with Paraloid B72 before SEM images were taken. The smoothed hemispherical shape on the dorsal edge of the palatine process of the maxilla (Figure 5) is Paraloid and not a feature of the bone.

Anteriorly, the maxilla SAM P57503 lacks the tip of the premaxillary process. The anteriormost tooth positions are missing on SAM P57541; tooth and loci counts are based on the reconstruction also using SAM P57541. The facial

process of the maxilla is incomplete preserving only 2.3 mm above the tooth row. The unbroken edge of the orbit can be traced from above the level of the 14th tooth position, to the broken dorsal tip of the suborbital process. The ventral tip of the bifid suborbital process is intact as are the last tooth positions. The tooth row preserves 16 of 20 loci, and 9 pleurodont teeth are still in position.

The lateral face of the maxilla is slightly convex in the anterior to posterior plane when viewed dorsally. The maxilla preserves nine primary foramina on the lateral face, the largest is the most anterior. Above the primary row, 11 more foramina, the largest one-third the size of the primaries, are visible in the SEM in two rows. Medially, the medial edge of the palatine shelf is broken. The palatine shelf thins posteriorly and ends in a fine point at the ventral tip of the bifid suborbital process. The tooth row occlusal surface is slightly convex posteriorly, with larger teeth beginning at the ninth position. The last two teeth decrease in size posteriorly, creating the trailing edge of the convex occlusal line. The reconstructed maxilla is 11.1 mm long from the premaxillary process to the suborbital process, with 20 tooth positions (see Figure 5). The original height of the facial process was not preserved on any specimen.

Figure 5: Right maxilla of *Proegernia mikebulli* sp. nov. reconstructed from SAM P57541 (anterior fragment; top left) and SAM P57503 (near complete; top right). 'A' marks the same tooth locus on each specimen. Enclosed by the box is the tooth 'B'. Abbreviations: cl, *cuspis labialis*; cla, *culmen lateralis anterior*; cli, *cuspis lingualis*; clp, *culmen lateralis posterior*; fp, *facial process*; psm, *palatine shelf of the maxilla*; pxp, *premaxillary process*; sop, *suborbital process*; and str, *striae*.

Premaxilla— A single complete left premaxilla, SAM P57542, was recovered from Fish Lens at BC2 (Figure 6). An osteoderm is fused to the anterior face of the internasal process, and would have overlapped the right premaxilla when the pair were articulated. Paired, unfused premaxillae in adulthood eliminate the possibility that this element belongs to a gekkotan. The ascending internasal process has a flattened medial side for articulation with its paired element. Viewed laterally, a foramen for the longitudinal canal is situated immediately above the maxillary process in the lateral face of the internasal process. Sharply pointed dorsally, the internasal process widens ventrally and terminates at the notch for the exit of the ethmoidal foramen laterally, and tooth row medially. The internasal process of the premaxilla rises towards the nasals at a 40° angle. The total length of the internasal process is 3.1 mm. The maxillary process is curved posterolaterally towards the maxilla and is approximately 1/3 of the height of the internasal process, reaching 1.6 mm in mediolateral width. The lateral edge of the maxilla process curves ventrally to finish beside the fourth tooth position. Four tooth loci are present and the first two teeth are preserved. These teeth are 0.53 mm long from the lateral face of the maxillary process and 0.28 mm wide. What remains of the worn crowns (see inset A, Figure 6) match those of the anterior-most teeth of the maxilla specimen SAM P57541. There are weak striae on the medial face of the crown, a mediolateral compression of the crown below the *crista lingualis*, and a sharply curved, prominent, *culmen lateralis*.

Figure 6: Left premaxilla (SAM P57542) of *Proegernia mikebulli* sp. nov. from posterior (left), anterior (centre), and lateral (right) view. A, tooth one, enlarged in box. Abbreviations: flc, foramen of longitudinal canal; inp, internasal process; mp, maxillary process; nef, notch of the ethmoidal foramen; and o, osteoderm.

Dentition— The dentition of *Proegernia mikebulli* is described using the two dentaries SAM P57502 and SAM P57499, the maxillae SAM P57541 and SAM P57503, and premaxilla SAM P57542, together comprising a near-complete upper and lower tooth row. The upper tooth row contains a total of 24–25 teeth, 4 or 5 on the left and right premaxilla, and 20 on the maxilla. The dentary tooth row has 27 tooth positions, beginning above the symphysis anteriorly and terminating just anterior of the coronoid process. The occlusal profile of the maxilla and dentary are both convex, with slightly larger teeth in the posterior half of the tooth row.

Figure 7: A and B: 24th tooth on the dentary SAM P57502, medial view, and C the 16th tooth in occlusal view with lateral and medial facies marked. Abbreviations: ai, antrum intercristatum; cl, crista lingualis; cm, crista mesialis; cul, cuspis labialis; cula, culmen lateralis anterior; and culp, culmen lateralis posterior.

On the dentary teeth, the tooth crown is similar in width to the shaft, expanding anteroposteriorly with slight mediolateral compression for the last 30% of the tooth height. Prominent anterior and posterior *culmen laterales* extend from the tips of the *crista mesialis*, turning medially and ventrally on the dentary, with a sharp angle producing ‘shoulders’ notable in medial view of the tooth (Figure 7). From the central cusp, the anterior cristae dip ventrally 20°, and posterior cristae 45°. The cusp (*cuspis labialis*; Figure 7) is slightly posterior of the centre of the tooth, and in occlusal view is positioned just posteromedial to the centre. This medial shift of the cusp creates a convex dorsal surface to the tooth, directing both cristae medially. Striae are only located on the medial face of the tooth crown, angled

dorsoventrally from the off-centre cusp. The two most prominent striae run parallel to the *culmen laterales* (both anterior and posterior), all others are weaker in profile with staggered lengths. Tooth crown morphology is similar on the maxilla and premaxilla.

Tooth wear occurs first on the cusp, forming a shallow rounded depression, gradually deepening medially. The *antrum intercristatum* expands in width from the central cusp. Wear is thus more noticeable in the centre of the tooth crown between the cusps, than anteriorly or posteriorly along the cristae, creating a thickening 'v' shape.

Pterygoid— A single right pterygoid attributed to *P. mikebulli* was recovered from Fish Lens at BC2 (SAM P57512; Figure 8). It is near complete, missing only the anterior-most tip of the palatine process and distal tip of the quadrate process. The pterygoid articulates with the palatine anteriorly with a pointed process extending anteriorly from a fanned, v-shaped pterygoid head to overlap with a similar feature from the anterior element. Laterally to the palatine process, the ectopterygoid process extends towards the ectopterygoid with a concave, curved facet for articulation with this element on the dorsal surface of the pterygoid head. Between these two processes, the fan-shaped pterygoid head has a concavity on the ventral surface that is 0.7 mm wide and 1.7 mm long. Within this concavity are three foramina, one in the deepest area of the concavity, and two are paired anteriorly, near the palatine process. Posteriorly, the pterygoid head narrows to become a parallel-sided rod, before bending medially at an angle of 35° towards the epipterygoid notch. The epipterygoid notch is an ovular concavity, with raised margins, on the dorsal surface of the pterygoid at the anterior end of the quadrate process. A sharp ridge extends from the anterior edge of the epipterygoid notch to the corner of the bend towards the pterygoid head. Posterior to the epipterygoid notch, the quadrate process becomes L-shaped in cross section and extends 1.5 mm posteriorly before the broken edge. The pterygoid head is 2.7 mm wide between the palatine and ectopterygoid processes. Total length of this element is not preserved.

Figure 8: Right pterygoid of *Proegernia mikebulli* sp. nov., SAM P57512 in dorsal (left) and ventral (right) views. Abbreviations: en, epipterygoid notch; pp, palatine process; pvc pterygoid ventral concavity; qp, quadrate process; and tp, ectopterygoid process.

b. Postcranial material potentially attributable to *Proegernia mikebulli*

Several procoelous vertebrae and assorted vertebral fragments were recovered from the Fish Lens at BC2. The best preserved of these is SAM P57514, a pre-sacral vertebra (Figure 9). This specimen is near complete, missing only the posterodorsal tip of the neural spine and the lateral tip of the right postzygapophysis. A thin crack runs posterolaterally on the ventral face of the centrum from just posterior of the ventral foramina, to the broken zygapophysis. The vertebra has an elongate centrum, the ventral surface is concave in lateral view. The neural canal is a similar diameter to the cotyle/condyle with a flat ventral surface broadening laterally before coming to a tear drop point when viewed anteriorly or posteriorly. Synapophyses are present immediately posterior to the prezygapophyses on the lateral facies. The neural spine rises posteriorly, steadily at an angle of 20°, from the dorsal margin of the neural canal until reaching the broken extremity. The prezygapophyses are angled at 32° and the postzygapophyses at 28°. Overall shape of the vertebra is relatively elongate, approximately one third longer than it is wide. Overall centrum length is 4.2 mm, the cotyle maximum width is 1.6 mm; the vertebra height anteriorly is 2.4 mm, posteriorly including the broken neural spine the height it is a maximum of 3.7 mm. The widest point of the vertebra is between the synapophyses, totalling 3.3 mm. This element is identified as a scincid lizard based on procoely, absence of a zygosphenes/zygantrum, minimal dorsoventral compression of the rounded cotyle and condyle, elongated centrum, and the minimal size of the ventral foramina. Snake vertebrae have shorter centra, a round cotyle/condyle with no compression, a set of zygosphenes and zygantra, and a ventral hypapophysis. A procoelous centrum devoid of a hollow space for the notochord eliminates the possibility of the

vertebra belonging to a non-pygopodid gekkotan. Gekkotans inclusive of pygopodids also have enlarged ventral foramina. Procoelous Gekkota vertebrae (mostly restricted to pygopodids) tend to have smaller condyles and cotyles, the cotylar hollow often not visible in ventral view because the dorsoventral inclination is minimal [25].

Figure 9: A single vertebra (SAM P57514) potentially referable to *Proegernia mikebulli* sp. nov., recovered from Fish Lens, BC2. A, anterior view; B, posterior view; C, lateral view, anterior to the left; D, dorsal view; and E, ventral view. A proximal fragment of a right femur assigned to *Proegernia mikebulli* sp. nov. SAM P57540. F, dorsal; G, anterior; and H, ventral views.

Abbreviations: con, condyle; cot, cotyle; fec, femoral condyle; itf, intertrochanteric fossa; itr, internal trochanter, nc, neural canal; nsp, neural spine, poz, postzygapophysis; prz, prezygapophysis; and syp, synapophysis.

A proximal right femur, SAM P57540 (F–H, Figure 9), was recovered from the Fish Lens at BC2. Broken immediately beneath the intertrochanteric fossa, no remnants of the shaft width or length are preserved. All other proximal features are intact. The specimen is most likely from an adult scincid as the epiphyses are fully ossified and fused to the diaphysis. The articular surface of the femoral condyle is semi-circular viewed anteriorly, and ovular in proximal profile. The femoral head extends medially away from the shaft axis. The internal trochanter is much shorter, only rising slightly above the edge of the intertrochanteric fossa. The intertrochanteric fossa is a concave surface between the condyle and trochanter on the ventral face of the proximal femur. The dorsal side also preserves a concave surface, but this is steeper-sided with a prominent ‘v’ marking where the femoral shaft begins. The width of the femoral condyle is widest in anterior view, measuring 2.2 mm at the suture with the epiphysis. The internal trochanter is 1.3 mm tall from the base of the intertrochanteric fossa. Total length preserved is 3.8 mm.

The scincid apomorphy of the trochanter major gradually tapering in height distally (see Lee et al. [51]) is not visible on this specimen, as the shaft is broken immediately proximal to it.

c. Comparisons

Comparison of new taxon with *Proegernia palankarinnensis*—The type species of *Proegernia* (by monotypy) is *Proegernia palankarinnensis*. While the type specimen is currently missing, the existing description [16] suggested *Proegernia* as being representative of a transitional form between a plesiomorphic lygosomine dentary morphology and the derived egeriine condition. The plesiomorphic traits shared by both *Proegernia palankarinnensis* and *P. mikebulli* are the small (about 1 mm long) opening of Meckel’s groove immediately posterior of the symphyseal foramen and the more anterior extent of the splenial notch into the anterior half of the tooth row length. Both species of *Proegernia* also have egeriine features emerging in the prominent ‘shoulders’ of the *culmen lateralis* on their unicuspid tooth crowns, and a relatively deep splenial notch, extending anteriorly further than other Australasian scincids. *Proegernia palankarinnensis* was described as having 22 tooth loci, with a possible maximum of 23 teeth, uncertain due to a broken symphyseal region. The specimens of *Proegernia mikebulli* preserve up to 27 dentary tooth positions (see SAM P57502, Figure 4). Both taxa share an anterior section of crowded, smaller teeth in their lower dentition. The spacing of teeth in *P. palankarinnensis* is marginally wider than *P. mikebulli* at the posterior end of the tooth row. Tooth shape varies between the two taxa: *P. palankarinnensis* has tooth shafts narrowing at the base, whereas tooth shafts in *P. mikebulli* narrow beneath the crown and expand again at the base. Martin et al. [16] noted that the tooth crowns of *Proegernia SAMA P39204* preserve distinct striations on both the lateral and medial faces. Tooth crowns for *Proegernia mikebulli* have weakly lineated striae on the medial face of the crown beneath the *crista lingualis*, but the lateral tooth face is smooth.

Although occurring almost concurrently, comparisons of the shape of the holotype dentaries of *P. palankarinnensis* and *P. mikebulli* (see Figure 10) show that the two species differ notably in the following characteristics: 1) the medial inflection of the symphysis is considerably less in *P. palankarinnensis* than in *P. mikebulli*, even after taking into account the effects of erosion. 2) The inferior alveolar foramen is positioned more anteriorly in *P. palankarinnensis*, below the 12th tooth position; that in *P. mikebulli* is level with the posterior edge of the 14th tooth. 3) No short extension of Meckel's groove can be observed anterior of the inferior alveolar foramen in *P. mikebulli*, a plesiomorphic character representing a partial remnant of Meckel's groove that was noted [16] on the holotype of *P. palankarinnensis* (Figure 10 A). 4) The internal septum of both specimens extends anteriorly to a similar degree. 5) The overall shape of the dentary of *P. palankarinnensis* is less robust, it being narrower between the medial edge of the dental sulcus and lateral face at the height of the mental foramina; 7) The facet for articulation of the coronoid terminates beneath the second-last tooth in *P. palankarinnensis* and the third last tooth locus in *P. mikebulli*. 8) The dorsal edge of the splenial notch is widened more abruptly in *P. mikebulli* with a curve; in *P. palankarinnensis* the same space is sharply v-shaped. 9) The coronoid process preserved on *P. palankarinnensis* does not extend above the height of the final tooth crown, unlike the process on SAM P57499 of *P. mikebulli*.

Figure 10: Comparative SEM of *Proegernia mikebulli* sp. nov. (B,C) and microphotographs of *P. palankarinnensis* (A,D) from Martin et al. [16], in medial (above) and occlusal (below) views. Note the anterior extension of the splenial notch (3) in *P. palankarinnensis* and the increased number of teeth/loci in *P. mikebulli*. Features described in text are labelled: 1) Medial inflection of the symphysis; 2) Dentary depth; 3) Position of the inferior alveolar foramen; 4) Meckel's Groove extension anterior of the inferior alveolar foramen, present in *P. palankarinnensis* and absent in *P. mikebulli*; 5) anterior extent of the internal septum similar in both taxa, 6) Overall robustness as dentary width, 7) facet for articulation with anterior ramus of coronoid, 8) splenial notch height, and 9) height of the coronoid process relative to the last tooth.

d. Other egeriines and other scincids

Comparisons with extant egeriines including *Lissolepis coventryi* (SAMA R57317) and *Liopholis multiscutata* (FUR168), as well as a Southeast Asian lygosomine outgroup representative *Eutropis multifasciata* (SAMA R35693), and two plesiomorphic scincids *Eumeces schneideri* (SAMA R6695), *Brachymeles schadenbergi* (SAMA R8853) and *Plestiodon fasciatus* (SAMA R66784) were conducted to determine how similar or derived the fossil taxon is from the generalised scincine (=inferred plesiomorphic) condition. Four characters notably vary among the chosen representative egeriines, the outgroup lygosomine and the extinct *P. mikebulli*: the anterior extent of the splenial notch and the inferior alveolar foramen; the shape and robustness of the symphysis, dentary depth and overall width; and, the number of teeth in the tooth row. The anterior extent of the splenial notch varies within the egeriine radiation; both species of *Lissolepis* share a further anterior-reaching inferior alveolar foramen than any other egeriine genus. The outgroup scincid condition, represented by the examined taxa *Eumeces schneideri* (SAMA R6695), *Brachymeles schadenbergi* (SAMA R8853), *Plestiodon fasciatus* (SAMA R66784), and *Eutropis multifasciata* (SAMA R35693) have a splenial notch that extends further than 60% anteriorly along the length of the tooth row, or one that stretches anteriorly into an elongate open Meckel's groove. The anterior extension of the splenial notch is a plesiomorphic state for this character, also shared by both *Proegernia* taxa.

The mandibular symphyseal joint in scincids varies between subfamilies in its overall size and robustness. Within egeriines, the variation is in the anteroposterior length of its posteroventral section, and the depth of the upper branch. In egeriine species that have a reinforced symphyseal joint and manipulate harder food, the lower branch of the symphysis is extended posteriorly and sometimes ventromedially (especially in *Tiliqua*, *Cyclodomorphus* and larger species of *Egernia*), to form a 'chin'. The dorsal branch of the symphysis deepens in *Liopholis* and in larger, omnivorous and herbivorous species of *Egernia*. This may be related to functional morphology to handle stresses on the chin during feeding, rather than a phylogenetic signal as these taxa do not form a clade. *Proegernia mikebulli* has a short and narrow posteroventral extension of the symphysis, possibly due to the presence of a foramen in the caudal notch between the limbs of the symphysis. This foramen exposes a short section of Meckel's cartilage in sphenomorphines and scincines. In egeriines this foramen is usually absent. Both *P. palankarinnensis* and *P. mikebulli* have dentaries where a slightly elongate foramen would expose the Meckel's cartilage, but in neither does this opening extend beyond the posterior edge of the symphysis.

Increased dentary depth and width, in relation to overall length, increases the robustness of the mandible. Variation in these measurements occur between genera of egeriines; species within *Liopholis* often have shorter, deeper skulls and corresponding dentaries, than similar-sized lizards within the genus *Egernia* (see the Supplementary information from [15]). *Lissolepis* has a more gracile dentary (see Figure 11). Outside of the egeriines, *Eutropis* and other SE Asian scincids with an unmodified insectivorous diet are even more gracile. *Eutropis multifasciata* has a longer dentary tooth row relative to snout-vent length than all egeriines. Derived morphologies related to dietary adaptations as seen in the herbivorous and durophagous members of the Egeriinae (i.e. *Corucia zebrata*, *Tiliqua* and *Cyclodomorphus*) result in more robust dentary dimensions and/or modified dental morphology. *Proegernia palankarinnensis* and *P. mikebulli* present dentary depths most similar to those of *Lissolepis* spp., between the ancestral shallow insectivorous *Eutropis*, and the deeper dentary typical of *Liopholis* spp. Deepening of the dentary bone and shortening of the snout noted in numerous species within the genus *Liopholis* is possibly related to the group's affinity for burrowing [52].

Eutropis multifasciata*Proegernia mikebulli* sp. nov.*Lissolepis coventryi**Liopholis multiscutata*
Figure 11: Comparative medial views of the dentaries of *Eutropis multifasciata* (SAMA R35693), *Proegernia mikebulli* sp. nov., *Lissolepis coventryi* (SAMA R57317) and *Liopholis multiscutata* (FUR168). 1 Anterior extent of splenial; 2 Symphysis shape and robustness; and 3 Dentary depth at mid-tooth row length.

e. Dentition

The teeth of *Eutropis multifasciata* (SAMA R35693) and *Proegernia mikebulli* show increasing derivation from the plesiomorphic insectivorous skink tooth type as described by Richter [22] and Kosma [23] and represented by *Plestiodon fasciatus* (SAMA R66784; Figure 12). Although many species within the extant radiations of eugongyline, sphenomorphine and egeriine scincids have modified tooth shapes for dietary specialisations, taxa exhibiting the most plesiomorphic tooth crown shapes were chosen for comparisons with fossil taxa.

Proegernia mikebulli has departed from the crown shape of basal skinks [see 22, 23] by having the cristae lingualis et mesialis directing medially, creating a sharp angle with the *culmen laterales*, and less-prominent striae mark the medial face of the tooth. *Eutropis* retain the prominent striae running dorsoventrally on the medial face of the tooth, these striae are almost all equally prominent, while *P. mikebulli* appears to have slightly stronger striae immediately adjacent and parallel to the *culmen laterales*. The tooth crowns of *Eugongylus rufescens* also demonstrate prominent *culmen laterales* and weakened striae, although the tooth shape differs; the shaft widening beneath the crown slightly and wear patterns make obvious the less medially directed cristae. The basal sphenomorphine condition is that of a narrow tooth shaft and crown, with sharply angled cristae creating a more pointed tooth profile. The medial face of the sphenomorphine tooth is anteroposteriorly convex and marked with prominent striae that do not approach the height of the lingual cristae but sit a third of the depth of the crown lower. The striae on *P. mikebulli* extend from almost directly in contact with the lingual cristae to the ventral tips of the *culmen laterales*.

The tooth crown in *P. mikebulli* is similar to that of the extant *Lissolepis coventryi* sharing medially directed cristae, sharp ‘shoulders’ to the *culmen laterales*, weak medial striae and absent lateral striae, and an expanded crown width on a narrower tooth shaft.

Figure 12: Medial view of left dentary tooth crowns of *Plestiodon fasciatus* (SAMA R66784), *Eutropis multifasciata* (SAMA R35693), *Proegernia mikebulli* sp. nov. (SAM P57502), *Lissolepis coventryi* (SAMA R57317), *Eugongylus rufescens* (SAMA R36735) and *Sphenomorphus jobiensis* (SAMA R6736). *Plestiodon* represents the basal skink tooth condition as described by Richter (1994). *Culmen lateralis* posterior is labelled culp.

7. Phylogenetic relationships

Parsimony and Bayesian analyses retrieved broadly similar trees, placing both species of *Proegernia* outside of, but close to, living (crown) Australian egeriines. Parsimony analysis of all morphological and molecular data recovered one most parsimonious tree (Figure 13) with a best score of 4726.25, found 60 times out of 1000 replicate searches. Confidence on most of the nodes indicated by the bootstrap analyses is low; only support for the clades *Bellatorias*, *Liopholis* and *Tiliqua+Cyclodomorphus* are above 50. This is almost certainly due to an unstable position of the fossils, which are missing all DNA and most morphological data (111 missing characters for *P. palankarinnensis* and 73 for *P. mikebulli*).

Figure 13: Egerniine phylogeny based on the single most parsimonious tree produced by TNT [26]. Bootstrap values >50 shown.

Proegernia mikebulli is recovered basal to the extant Australian Egerniinae (spanning *Lissolepis* to *Tiliqua*), and *Proegernia palankarinnensis* is retrieved as sister to the Solomon Islands' *Corucia zebra*, but both with bootstrap support <50%. These relationships are conservatively interpreted as an effective polytomy between crown egerniines, *Corucia*, *Proegernia*, and *Lissolepis*. Thus, rather than erect a new genus, we provisionally assign the new species to *Proegernia* (due to geographic and stratigraphic links with the type of that genus, and the Bayesian results below).

The position of both taxa outside the crown Australian egerniines is due to plesiomorphic scincid character states such as a partially open Meckel's groove (char. 11) in both fossil taxa. This feature also separates them from the outgroup taxon *Eutropis multifasciata* which has an entirely open Meckel's groove. The anterior extent of the splenial (char. 21), and the extension of Meckel's Groove anterior of the splenial notch are features supporting the separation between *Lissolepis* and *Proegernia*; *Lissolepis* with a splenial reaching approximately 50% of the tooth row length anteriorly, and *Proegernia* reaching beyond to two-thirds the tooth row length.

The crown Australian Egerniinae spanning *Lissolepis* to *Tiliqua* is diagnosed by a completely closed Meckel's Groove (char. 11), an absence of pterygoid teeth (char. 131), and a pterygoid quadrate ramus that is arcuate in cross section (char. 134). *Proegernia* (and *Corucia*) have plesiomorphic or alternative states for these characters, resulting in their position outside this clade. Potential morphological synapomorphies of the basal living Australian genus *Lissolepis* that are also present in *Proegernia* are the termination of the dentary coronoid process (char. 17), the orientation of the retroarticular process from dorsal view (char. 46), the plesiomorphic bicuspid tooth crown shape (char. 55), minimal dental cementum (char. 60), and the divot in the ventral surface of the pterygoid body that is also absent of pterygoid teeth (chars 135 and 131). These shared characters conflict with the position of *Proegernia* outside the Australian crown group, resulted in the low support for the relationships between *Corucia*, *Proegernia* and *Lissolepis*.

The Bayesian analysis of all morphological and molecular data produced the topology shown in Figure 14: the consensus of four separate runs of 100,000,000 generations, when combined in LogCombiner, then summarised with TreeAnnotator with a burnin of 20%.

Figure 14: Egeriine phylogeny based on consensus total evidence Bayesian inference tree from BEAST. †denotes extinct taxon. Small numbers at nodes denote posterior probability. Reconstructed node ages for key clades are in brackets, the green shading denotes the Eocene, after which Australia became isolated from Gondwana [4].

The extant egeriine radiation forms a well-supported clade. Within this, *Tribolonotus* is strongly supported as the sister to remaining taxa. Of those, *Corucia* is strongly supported as the sister of the remaining (Australian) egeriines. Both late-Oligocene fossil species of *Proegeria* are weakly supported as the sister to *Corucia* with a Prior Probability (PP) of 0.25. All Australian extant egeriines again form a clade that excludes both *Proegeria*. The extant Australian clade is a weakly supported (PP= 0.51) clade and retrieved as originating in the lower Oligocene (Figure 14).

Proegeria palankarinnensis and *P. mikebulli* are retrieved as sister taxa, but with little support (PP= 0.44), not unexpected given extensive missing data. Character changes affirming the sister relationship include the position of the mental foramina 50% along the length of the tooth row (char. 13), and the presence of less than 20 tooth crown striae (char. 61). Both of these features are also shared with *Lissolepis*, but not with their putative sister clade *Corucia zebra*. The similar positioning of both taxa (i.e. basal to crown Australian egeriines) is supported by the same characters as those noted after the parsimony analysis.

The node age for all Egeriinae is retrieved as 50.12 Ma, with *Tribolonotus* as the extant stem taxon representing the arrival of the subfamily to northern Sahul (Australia, New Guinea and surrounding islands). The stem clade last shared a common ancestor with the crown Australian egeriines 40.47 Ma.

8. Discussion

Recent exploration of the Namba Formation cropping out at Lake Pinpa and Billeroo Creek has unearthed a new egeriine scincid lizard, *Proegeria mikebulli* sp. nov. Previous excavations and collection of surface material at these localities did not recover any squamate elements. This identification has increased the taxonomic diversity of the Pinpa Local Fauna in the Namba Formation to include at least one skink, with further material yet to be described attributed to the Gekkota. These discoveries are attributed to the use of fine mesh-aperture sieves and bulk screening of concentrated lenses of fossiliferous sediments.

a. Namba environment

The excavation and subsequent analyses of the sediment profile at both Billeroo Creek (BC2) and Lake Pinpa (LP12) has allowed an interpretation of palaeoenvironmental conditions in the Frome Basin during the Oligo-Miocene. Dolomite in freshwater conditions requires a flooded alkaline environment to precipitate, as does calcite [53], so the presence of these minerals in Layers 4–5 (absent in layers 2 and 3) are most likely the result of the seasonal fluctuations of the water level in the palaeo-lakes in which the sediments comprising the Namba Formation were deposited during the late Oligocene–early Miocene [see 40]. Fluctuations in the rainfall and evaporation contributed to the formation of palygorskite in the lower Layers 3–5. Smectite is the result of the weathering of minerals sourced from older rock outcrop

(most likely from the nearby Flinders, Barrier, and Olary Ranges) deposited into the Namba area as swampy soils on the lake edge. Layers where dolomite is absent, i.e. Layer 3 according to the XRD result, reflect a drier phase in the palaeo-environment without an alkaline waterbody at the surface.

Callen [40] noted that the Namba basin was once flooded, forming a larger palaeolake than the present Lake Frome. The palaeolake Namba supported a number of aquatic vertebrates including lungfish, *Obdurodon* platypus, crocodiles, cetaceans and various waterbirds [43, 54–56]; their combined presence is indicative of a deeper aquatic environment. The lake was subject to varying rainfall and evaporation rates, resulting in receding lake edges, creating cycles of dolomite precipitation (recorded in Layer 5). Callen [40] concluded that sufficient evaporation of surface water (conditions required for dolomite precipitation) could not occur in a consistently high-rainfall climate. Instead, a modern analogue might be where cool moist winters and hot dry summers result in a moderate seasonal rainfall and high evaporation rates that result in dolomite and calcite deposits such as in the Coorong lacustrine-estuarine system in southern South Australia [57]. There, lake levels rise after dry summers due to groundwater recharge, resulting in the dolomite precipitation.

The sedimentary evidence from palaeolake Namba reveal periods of evaporation and alkaline water bodies. This suggests the occurrence of periodic droughts during which fish die-offs might be expected in the evaporating water bodies and terrestrial fauna might die and accumulate along the lake-edge. The articulated skeletons found at LP12 and BC2 alongside concentrations of fish bone may be the result of these lake ‘deaths’. The remains of *Proegernia mikebulli* and other small terrestrial vertebrates are preserved in lenses of concentrated bones of aquatic vertebrates (fish, turtle, and crocodiles) within deposits reflecting these ephemeral lacustrine conditions. These lenses, as exemplified by the Fish Lens at BC2, and others exposed discontinuously at Lake Pinpa, were probably formed by littoral longshore currents disarticulating, mixing and concentrating bones. The material would not have to have been transported very far in order to mix animals from the two starkly different environments as the deposition site was in the littoral zone. Accumulation of small vertebrate material alongside associated larger vertebrate skeletons (at LP12) indicate that the transportation of material was not from a flash flood typical of tropical monsoon seasonality but rather accumulated along a shallow or ephemeral lake edge shoreline. Some small terrestrial vertebrate material is within a size range attributable to raptor predation [58, 59] and some specimens show corrosion thinning typical of that formed after raptor ingestion. Such material may be derived from raptor pellet accumulations washing into the lake where they disaggregated and mixed with the bones of the more common freshwater taxa. Undescribed accipitrids are known from the Pinpa Local Fauna and a skeleton of one was excavated from LP12 by Worthy/Camens-led expeditions, and is the subject of current research.

b. *Proegernia mikebulli* and crown group egeriines

Proegernia mikebulli was small and gracile compared to most living egeriines. Based on dentary length, an adult individual would be of a similar size to *Lissolepis coventryi*, around 100 mm SVL [60]. Examinations of tooth crown features across the type specimens reveals they share features with both *Lissolepis* and species of *Liopholis* (cf. *L. multiscutata* and *L. whitii* both of which have a cusp slightly posterior to the centre of the tooth crown). Tooth crown features of scincids potentially preserve phylogenetic and dietary signals. Detailed studies within and between the extant subfamilies may help elucidate any dietary preferences suggested by fossil dentition. Tooth striae are present on teeth in most egeriines with a specialised diet and no pattern in the number of striae between insectivorous, durophagous, or herbivorous species was obvious [see 23]. Within genera, the number and patterning of striae was fairly similar. *Cyclodomorphus* and *Tiliqua* increase the number of striae from other egeriines, partly due to the change in tooth shape leading to striae radiating on all sides from a central cusp. *Lissolepis* and *Liopholis* have fewer striae and all are restricted to the lingual face of the tooth crown, below the lingual cristae. The two fossil species of *Proegernia* have few striae, similar to *Lissolepis* and *Liopholis*. The buccal striae on the crowns of *Proegernia palankarinnensis* are unusual and were not observed on extant *Eutropis multiscutata*, *Sphenomorphus jobiensis* or *Eugongylus rufescens*. Examination of fine-scale tooth crown features may prove useful characters for morphological phylogenetic analyses of fragmentary or incomplete tooth-bearing scincid material in future.

From what can be deduced of the palaeoenvironment surrounding Lake Pinpa and Billeroo Creek in the late Oligocene–early Miocene, *Proegernia mikebulli* lived in a cool, moist climate region with seasonal hot, dry summers with high evaporation rates. Other squamates known from this region and time period include a small gekkotan, and possibly a constricting snake [43]. Although no evidence of *Proegernia palankarinnensis* was found in the Namba Formation sites, the presence of at least two Australian scincids, from neighbouring basins, covering similar time periods and environments may be an indication of the early diversity of Australian squamates. Extant Australian scincids are difficult to separate based on isolated cranial material. The same cryptic diversity may have been present in the Oligo-Miocene, so more than two species are likely present in both the Etadunna and Namba Formations.

Although neither analysis produced firm support for the phylogenetic position of *Proegernia mikebulli*, both parsimony and Bayesian analyses placed the new taxon outside of living (crown) Australian Egeriinae. *Proegernia* species have a larger tooth count than crown Australian egeriines, a splenial notch placed more anteriorly, and a ~1 mm long opening of Meckel’s groove anterior to the splenial notch. Each of these character states are representative of the predicted transitional form between the plesiomorphic outgroup morphology and the majority of crown Australian egeriines. These characters have pulled *Proegernia* outside of the Australian crown group, but not beyond the Sahul stem genera *Corucia* and *Tribolonotus*. However, their precise position outside this crown group is uncertain. If relationships of one or both *Proegernia* with *Corucia* are correct (Figure 13–Figure 14), then there were two migrations of Egeriinae into southern Sahul 40.47–32.7 Ma, or a single invasion followed by emigration of *Corucia* to the Solomon Islands. However, a large amount of missing data for these fossil taxa and low support for precise relationships means

that it is possible that these trees are wrong, and that homoplasy with *Corucia* is what pulls these taxa away from the crown Australian group. It is thus possible that *Proegermia* lies on the immediate stem to the Australian crown group, in which case only a single dispersal event to Australia need be assumed.

The tip-dated phylogeny inclusive of the two Oligo-Miocene *Proegermia* taxa retrieves the age of the Egermiinae as 50.12 Ma. The last common ancestor of Egermiinae and Lygosominae was mostly likely living in the far south-eastern edge of Asia during the early Eocene. The first egermiine evolved elsewhere when the continent of Sahul was far south of its current latitude [61]; there is no fossil evidence of their presence in Australia at this time. Crown Egermiinae most likely originated either in south-eastern Asia or on an island arc, before connecting with the Australian landmass in the late Eocene (32.7Ma). Similar origins have been hypothesised for agamids [22 Ma; 62], pythons [c. 35 Ma; 8], sphenomorphine skinks [12], and elapids [8].

9. Conclusion

Dated phylogenies with the new fossils suggest that egermiines are the oldest radiation of skinks in Australia. The diversification of the crown Australian egermiines (~33 Ma) occurred nearly 10 million years before the estimated diversification of other Australian skinks, i.e. the Sphenomorphinae (c. 25 Ma) and Eugongylinae [c. 20 Ma; 3]. However, their diversity lags behind these other, potentially more recent groups. The first egermiines would have shared the continent with pygopodid, diplodactylid and carphodactylid geckos, and madtsoiid snakes [5]. The search for more Paleogene lizard material to document the early evolution of Australia's diverse herpetofauna continues, with investigations of Eocene collections from Murgon in northern Queensland [63], and early Oligocene vertebrates from Pwerte Marnte Marnte [64].

Acknowledgments

Sue Double and Jenny Worthy for screening and sorting the fossil material. Rod Wells, Colin Doudy, Bob and Sue Tulloch, Amy Tschim, Warren Handley, Jacob Blokland, Carey Burke and Ellen Mather were all part of the collaborative field team. Dr Jason Gascooke for training on, and use of, the SEM for imaging and EDAX sedimentary analysis at Flinders Microscopy, an Australian Microscopy and Microanalysis Research Facility. We thank Sarah Harmer-Bassel and Jason Young of Flinders Analytical, for their ICPMS and X-ray diffraction (XRD) analyses of the sediment from the studied fossil sites. We thank Mary-Anne Binnie and Carolyn Kovach of the South Australian Museum, and Tim Ziegler from Museums Victoria, for access to collections, and loans of fossil and extant comparative material. The Willi Hennig Society for making TNT freely available. We especially thank Andrew Black (Blackie), manager, and Alec Wilson (prior owner) for access to Frome Downs Station that allowed this work.

Funding Statement

KMT was supported by an Australian Postgraduate Research Training Stipend. The Mark Mitchell Foundation funded part of the fieldwork component of this project, and SEM time.

Data Accessibility

All specimens are registered in the Palaeontology Collection of the South Australian Museum, MicroCT files are housed in the SAM Herpetology digital collection.

All executable files for phylogenetic analyses are available as electronic supplementary material.

Competing Interests

We have no competing interests.

Authors' Contributions

KMT conceived the project, conducted fieldwork, collected the data, performed analyses, produced all figures and wrote the manuscript. THW and ABC funded the project, conducted field work, contributed to discussions, and commented on multiple drafts of the manuscript; MNH and MSYL commented on multiple drafts of the manuscript and contributed to discussions. NB collected the stratigraphic samples and information as part of a separate honours project.

References

1. Hall, R. 2002 Cenozoic geological and plate tectonic evolution of SE Asia and the SW Pacific: computer-based reconstructions, model and animations. *Journal of Asian Earth Sciences*. **20**, 353–431.

2. White, M. E. 2006 Environments of the geological past. In *Evolution and biogeography of Australasian vertebrates*. (ed. eds. J. R. Merrick, Archer, A., Hickey, G. M., Lee, M. S. Y.), pp. 17–48. Oatlands, NSW: Auscipub.

3. Skinner, A., Hugall, A. F., Hutchinson, M. N. 2011 Lygosomine phylogeny and the origins of Australian scincid lizards. *Journal of Biogeography*. **38**, 1044–1058. (10.1111/j.1365-2699.2010.02471.x)

4. Oliver, P., Hugall, A. 2017 Phylogenetic evidence for mid-Cenozoic turnover of a diverse continental biota. *Nat Ecol Evol.* **1**, 1896–1902. (10.1038/s41559-017-0355-8)
5. Oliver, P. M., Sanders, K. L. 2009 Molecular evidence for Gondwanan origins of multiple lineages within a diverse Australasian gecko radiation. *Journal of Biogeography.* **36**, 2044–2055.
6. Vidal, N., Hedges, S. B. 2009 The molecular evolutionary tree of lizards, snakes, and amphisbaenians. *Comptes rendus biologiques.* **332**, 129–139.
7. Hugall, A. F., Lee, M. S. Y. 2004 Molecular claims of Gondwanan age for Australian agamid lizards are untenable. *Molecular Biology and Evolution.* **21**, 2102–2110. (10.1093/molbev/msh219)
8. Sanders, K. L., Lee, M. S. Y. 2008 Molecular evidence for a rapid late-Miocene radiation of Australasian venomous snakes (Elapidae, Colubroidea). *Molecular Phylogenetics and Evolution.* **46**, 1165–1173. (<http://dx.doi.org/10.1016/j.ympev.2007.11.013>)
9. Lambert, S. M., Reeder, T. W., Wiens, J. J. 2015 When do species-tree and concatenated estimates disagree? An empirical analysis with higher-level scincid lizard phylogeny. *Molecular phylogenetics and evolution.* **82**, 146–155.
10. Uetz, P., Freed, P., Hošek, J. e. The Reptile Database. 2019.
11. Skinner, A., Hutchinson, M. N., Lee, M. S. Y. 2013 Phylogeny and divergence times of Australian *Sphenomorphus* group skinks (Scincidae, Squamata). *Molecular Phylogenetics and Evolution.* **69**, 906–918. (<http://dx.doi.org/10.1016/j.ympev.2013.06.014>)
12. Rabosky, D. L., Donnellan, S. C., Talaba, A. L., Lovette, I. J. 2007 Exceptional among-lineage variation in diversification rates during the radiation of Australia's most diverse vertebrate clade. *Proceedings of the Royal Society of London B: Biological Sciences.* **274**, 2915–2923. (10.1098/rspb.2007.0924)
13. Gardner, M. G., Hugall, A. F., Donnellan, S. C., Hutchinson, M. N., Foster, R. 2008 Molecular systematics of social skinks: phylogeny and taxonomy of the *Egernia* group (Reptilia: Scincidae). *Zoological Journal of the Linnean Society.* **154**, 781–794. (10.1111/j.1096-3642.2008.00422.x)
14. Chapple, D. G., Ritchie, P. A., Daugherty, C. H. 2009 Origin, diversification, and systematics of the New Zealand skink fauna (Reptilia: Scincidae). *Molecular Phylogenetics and Evolution.* **52**, 470–487. (<https://doi.org/10.1016/j.ympev.2009.03.021>)
15. Thorn, K. M., Hutchinson, M. N., Archer, M., Lee, M. S. Y. 2019 A new scincid lizard from the Miocene of Northern Australia, and the evolutionary history of social skinks (Scincidae: Egerniinae). *Journal of Vertebrate Paleontology.* **39**, e1577873. (10.1080/02724634.2019.1577873)
16. Martin, J. E., Hutchinson, M. N., Meredith, R., Case, J. A., Pledge, N. S. 2004 The oldest genus of scincid lizard (Squamata) from the Tertiary Etadunna Formation of South Australia. *Journal of Herpetology.* **38**, 180–187. (10.1670/25-03A)
17. Hocknull, S. A. 2000 Remains of an Eocene skink from Queensland. *Alcheringa: An Australasian Journal of Palaeontology.* **24**, 63–64. (10.1080/03115510008619524)
18. Estes, R. 1984 Fish, amphibians and reptiles from the Etadunna Formation, Miocene of South Australia. *Australian Zoologist.* **21**, 335–343.
19. Woodburne, M. O., Macfadden, B. J., Case, J. A., Springer, M. S., Pledge, N. S., Power, J. D., Woodburne, J. M., Springer, K. B. 1994 Land mammal biostratigraphy and magnetostratigraphy of the Etadunna Formation (late Oligocene) of South Australia. *Journal of Vertebrate Paleontology.* **13**, 483–515.
20. Tedford, R. H., Archer, M., Bartholomai, A., Plane, M., Pledge, N. S., Rich, T., Rich, P., Wells, R. T. 1977 The discovery of Miocene vertebrates, Lake Frome area, South Australia. *BMR Journal of Australian Geology & Geophysics.* **2**, 53–57.
21. Evans, S. E. 2008 The skull of lizards and Tuatara. In *Biology of the Reptilia 20, Morphology H The skull of Lepidosauria.* (ed. ^eds. C. Gans, A. S. Gaunt, K. Adler), pp. 1–343. Ithaca, NY: Society for the Study of Amphibians and Reptiles.
22. Richter, A. 1994 Lacertilia aus der Unteren Kreide von Una und Galve (Spanien) und Anoual (Marokko). *Berliner geowissenschaftliche Abhandlungen (E: Paläobiologie).* **14**, 1–147.
23. Kosma, R. 2003 The dentitions of recent and fossil scinciform lizards (Lacertilia, Squamata) - Systematics, Functional Morphology, Paleocology. Hanover, Germany: University Hannover.
24. Russell, A. P., Bauer, A. M. 2008 The appendicular locomotor apparatus of *Sphenodon* and normal-limbed squamates. In *The skull and appendicular locomotor apparatus of Lepidosauria.* (ed. ^eds. C. Gans, A. S. Gaunt, K. Adler), pp. Ithaca, New York, USA: Society for the Study of Amphibians and Reptiles.
25. Hoffstetter, R., Gasc, J.-P. 1969 Vertebrae and ribs of modern reptiles. In *Biology of the Reptilia.* (ed. ^eds. C. Gans), pp. 201–310. New York: Academic Press.
26. Goloboff, P. A., Catalano, S. A. 2016 TNT version 1.5, including a full implementation of phylogenetic morphometrics. *Cladistics.* **32**, 221–238.
27. Drummond, A. J., Suchard, M. A., Xie, D., Rambaut, A. 2012 Bayesian phylogenetics with BEAUti and the BEAST 1.7. *Molecular Biology and Evolution.* **29**, 1969–1973.
28. Tonini, J. F. R., Beard, K. H., Ferreira, R. B., Jetz, W., Pyron, R. A. 2016 Fully-sampled phylogenies of squamates reveal evolutionary patterns in threat status. *Biological Conservation.* **204**, Part A, 23–31. (<https://doi.org/10.1016/j.biocon.2016.03.039>)
29. Lanfear, R., Frandsen, P. B., Wright, A. M., Senfeld, T., Calcott, B. 2016 PartitionFinder 2: new methods for selecting partitioned models of evolution for molecular and morphological phylogenetic analyses. *Molecular Biology and Evolution.* **34**, 772–773.
30. Maddison, W., Maddison, D. Mesquite: a modular system for evolutionary analysis. Version 3.2. 2017.
31. Lewis, P. O. 2001 A likelihood approach to estimating phylogeny from discrete morphological character data. *Systematic Biology.* **50**, 913–925.
32. Alekseyenko, A. V., Lee, C. J., Suchard, M. A. 2008 Wagner and Dollo: a stochastic duet by composing two parsimonious solos. *Systematic Biology.* **57**, 772–784.
33. Goloboff, P. A., Pittman, M., Pol, D., Xu, X. 2018 Morphological data sets fit a common mechanism much more poorly than DNA sequences and call into question the Mk model. *Systematic Biology.* **68**, 494–504. (10.1093/sysbio/syy077)
34. Wright, A. M., Hillis, D. M. 2014 Bayesian analysis using a simple likelihood model outperforms parsimony for estimation of phylogeny from discrete morphological data. *PLoS One.* **9**, e109210.
35. O'Reilly, J. E., Puttick, M. N., Parry, L., Tanner, A. R., Tarver, J. E., Fleming, J., Pisani, D., Donoghue, P. C. 2016 Bayesian methods outperform parsimony but at the expense of precision in the estimation of phylogeny from discrete morphological data. *Biology Letters.* **12**, 20160081.
36. Harmon, L. J. 2019 *Phylogenetic comparative methods.* 1.4 ed. Published Online: Luke Harmon.
37. Stadler, T. 2010 Sampling-through-time in birth–death trees. *Journal of Theoretical Biology.* **267**, 396–404. (<https://doi.org/10.1016/j.jtbi.2010.09.010>)
38. Drummond, A. J., Ho, S. Y. W., Phillips, M. J., Rambaut, A. 2006 Relaxed phylogenetics and dating with confidence. *PLOS Biology.* **4**, e88. (10.1371/journal.pbio.0040088)
39. Rambaut, A., Drummond, A. J., Xie, D., Baele, G., Suchard, M. A. 2018 Posterior summarisation in Bayesian phylogenetics using Tracer 1.7. *Systematic Biology.* **67**, 901–904. (doi:10.1093/sysbio/syy032)
40. Callen, R. A. 1977 Late Cainozoic environments of part of northeastern South Australia. *Journal of the Geological Society of Australia: An International Geoscience Journal of the Geological Society of Australia.* **24**, 151–169.
41. Rich, T. H. 1991 Monotremes, placentals, and marsupials: their record in Australia and its biases. In *Vertebrate Palaeontology of Australasia.* (ed. ^eds. P. Vickers-Rich, J. M. Monaghan, R. F. Baird, T. H. Rich), pp. 894–1057. Melbourne, Australia: Pioneer Design Studio Pty Ltd.
42. Rich, T. 1984 News from Foreign members: Australia, Museum of Victoria, Melbourne. *Society of Vertebrate Paleontology News Bulletin.* **130**, 28.
43. Vickers-Rich, P., Rich, P. V. 1991 *Vertebrate palaeontology of Australasia.* Melbourne, Australia: Pioneer Design Studio.
44. Callen, R. A., Tedford, R. H. 1976 New Late Cainozoic rock units and depositional environments, Lake Frome area, South Australia. *Transactions of the Royal Society of South Australia.* **100**, 125–167.
45. Centore, P. Conversions between the Munsell and sRGB colour systems.

- <http://www.munsellcolourscienceforpainters.com/> 2013:64.
46. Opperl, M. 1811 *Die Ordnungen, Familien und Gattungen der Reptilien, als Prodrom einer Naturgeschichte derselben*. München: Joseph Lindauer.
47. Gray, J. E. 1825 A synopsis of the genera of reptiles and Amphibia, with a description of some new species. *Annals of Philosophy*. **10**, 193–217.
48. Welch, K. 1982 Herpetology of the Old World II. Preliminary comments on the classification of skinks (Family Scincidae) with specific reference to those genera found in Africa, Europe, and southwest Asia. *Herpetile*. **7**, 25–27.
49. Megirian, D., Prideaux, G., Murray, P., Smit, N. 2010 An Australian land mammal age biochronological scheme. *Paleobiology*. **36**, 658–671. (10.1666/09047.1)
50. Gelnaw, W. B. 2011 On the cranial osteology of *Eremiascincus* and its use for identification: East Tennessee State University.
51. Lee, M. S. Y., Hutchinson, M. N., Worthy, T. H., Archer, M., Tennyson, A. J. D., Worthy, J. P., Scofield, R. P. 2009 Miocene skinks and geckos reveal long-term conservatism of New Zealand's lizard fauna. *Biology Letters*. **5**, 833–837. (doi:10.1098/rsbl.2009.0440)
52. Greer, A. E. 1989 *The biology and evolution of Australian lizards*. Chipping Norton, NSW: Surrey Beatty & Sons.
53. Fukushi, K., Matsumiya, H. 2018 Control of Water Chemistry in Alkaline Lakes: Solubility of Monohydrocalcite and Amorphous Magnesium Carbonate in CaCl₂–MgCl₂–Na₂CO₃ Solutions. *ACS Earth and Space Chemistry*. **2**, 735–744. (10.1021/acsearthspacechem.8b00046)
54. Worthy, T. H. 2009 Descriptions and phylogenetic relationships of two new genera and four new species of Oligo-Miocene waterfowl (Aves: Anatidae) from Australia. *Zoological Journal of the Linnean Society*. **156**, 411–454. (10.1111/j.1096-3642.2008.00483.x)
55. Worthy, T. H. 2011 Descriptions and phylogenetic relationships of a new genus and two new species of Oligo-Miocene cormorants (Aves: Phalacrocoracidae) from Australia. *Zoological Journal of the Linnean Society*. **163**, 277–314. (10.1111/j.1096-3642.2011.00693.x)
56. Fitzgerald, E. M. G. 2004 A review of the Tertiary fossil Cetacea (Mammalia) localities in Australia. *Memoirs of Museum Victoria*. **61**, 183–208.
57. Borch, C. C. V. D., Lock, D. 1979 Geological significance of Coorong dolomites. *Sedimentology*. **26**, 813–824. (10.1111/j.1365-3091.1979.tb00974.x)
58. Andrews, P. 1990 *Owls, caves and fossils*. London: The Natural History Museum.
59. McDowell, M. C., Medlin, G. C. 2009 The effects of drought on prey selection of the barn owl (*Tyto alba*) in the Strzelecki Regional Reserve, north-eastern South Australia. *Australian Mammalogy*. **31**, 47–55. (<http://dx.doi.org/10.1071/AM08115>)
60. Wilson, S., Swan, G. 2017 *A complete guide to the Reptiles of Australia*. 5th ed. Sydney, Australia: Reed New Holland Publishers.
61. Müller, R. D., Seton, M., Zahirovic, S., Williams, S. E., Matthews, K. J., Wright, N. M., Shephard, G. E., Maloney, K. T., Barnett-Moore, N., Hosseinpour, M., et al. 2016 Ocean basin evolution and global-scale plate reorganization events since Pangea breakup. *Annual Review of Earth and Planetary Sciences*. **44**, 107–138. (10.1146/annurev-earth-060115-012211)
62. Hugall, A. F., Foster, R., Hutchinson, M., Lee, M. S. Y. 2008 Phylogeny of Australasian agamid lizards based on nuclear and mitochondrial genes: implications for morphological evolution and biogeography. *Biological Journal of the Linnean Society*. **93**, 343–358. (10.1111/j.1095-8312.2007.00911.x)
63. Godthelp, H., Archer, M., Cifelli, R., Hand, S. J., Gilkeson, C. F. 1992 Earliest known Australian Tertiary mammal fauna. *Nature*. **356**, 514–516. (10.1038/356514a0)
64. Murray, P. F., Megirian, D. 2006 The Pwerte Marnte Marnte Local Fauna: a new vertebrate assemblage of presumed Oligocene age from the Northern Territory of Australia. *Alcheringa: An Australasian Journal of Palaeontology*. **30**, 211–228. (10.1080/03115510609506864)
65. Woodhead, J., Hand, S. J., Archer, M., Graham, I., Sniderman, K., Arena, D. A., Black, K. H., Godthelp, H., Creaser, P., Price, E. 2016 Developing a radiometrically-dated chronologic sequence for Neogene biotic change in Australia, from the Riversleigh World Heritage Area of Queensland. *Gondwana Research*. **29**, 153–167. (<http://dx.doi.org/10.1016/j.gr.2014.10.004>)

Tables

Table 1: Fossil calibrations used, their minimum and maximum ages (Ma) and references for the age dates or species descriptions.

Fossil calibration	Min age	Max age	References for age
Zone A, Etadunna Formation	25.5	25.7	Woodburne MO, Macfadden BJ, Case JA, Springer MS, Pledge NS, Power JDet al. [19]
Pinpa Local Fauna, Namba Formation	25.5	25.7	Woodburne MO, Macfadden BJ, Case JA, Springer MS, Pledge NS, Power JDet al. [19]
AL 90 Locality, Carl Creek Limestone	14.17	15.11	Woodhead J, Hand SJ, Archer M, Graham I, Sniderman K, Arena DA, Black KH, Godthelp H, Creaser PPrice E [65]
Gag Locality, Carl Creek Limestone	14.47	16.86	Woodhead J, Hand SJ, Archer M, Graham I, Sniderman K, Arena DAet al. [65]

Table 2: Major and trace elements detected by X-ray fluorescence analysis of sediments sampled Layer 3 and 5 from stratigraphic sections of Lake Pinpa Site 12 and Billeroo Creek Site 2. Values are %, minor trace elements making the total of each sample 100 are not shown.

	MgO	Al ₂ O ₃	SiO ₂	CaO	MnO	Fe ₂ O ₃
Pinpa 12 L3	4.043	8.815	46.53	0.238	0.2303	19.82
Pinpa 12 L5	10.45	7.444	43.64	7.901	4.227	8.105
Billeroo Creek L5	13.65	5.875	36.08	13.02	1.634	6.982

Table 3: Mineral components of sediment samples taken from Layers 3 and 5 at Lake Pinpa and Layer 5 at Billeroo Creek Site 2.

	Clay Minerals	Dolomite/ Ankerite	Goethite	Quartz	Other	Total
Pinpa 12 L3	79	-	9	11	1	100
Pinpa 12 L5	66	23	3	6	2	100
Billeroo Creek L5	54	38	2	5	1	100

Figure and table captions

Figure 1: Stratigraphic sections through the Namba Formation at Lake Pinpa at Site 12 (see Figure 2) and Billeroo Creek, Site 2, constructed in SedLog 3.1 from authors field observations in comparison with RH Tedford's sections described in notes and figures in [20, 44], depth in meters. Grain sizes were determined in the field. Colours are converted from Munsell figures to corresponding RGB values using [45].

Figure 2: Location maps of the Lake Pinpa and Billeroo Creek sites, Frome Downs Station, South Australia. Stratigraphic columns in Figure 1 were compiled from locations Site 12 (=LP12) and Site 2 Fish lens (=BC2) marked above.

Figure 3: Results of the soluble mineral analyses of Layers 2–4 and the white and black sediments from Layer 5 at Lake Pinpa Site 12 (left); and Billeroo Creek Site 2 Layers 2–5 (right).

Figure 4: 1a–c Holotype, a right dentary, SAM P57502, 2a–c paratype; a left dentary SAM P57499; two right post-dentary compound bones 3a–c SAM P57543 and 4a–c SAM P57542 of *Proegernia mikebulli* sp. nov. from the Fish Lens at BC2; and 5 complete reconstruction in medial view of the right mandible of *Proegernia mikebulli*, the coronoid and splenial (hatched area) are reconstructed based on modern egermiine specimens, no fossil representatives of these elements are known. Abbreviations: adf, adductor fossa; anf, angular facet; art., articular; d., dentary; gl, glenoid fossa; iaf, inferior alveolar foramen; mg, Meckel's groove; psf, posterior surangular foramen; rap, retroarticular process; san., surangular; and sy, symphysis.

Figure 5: Right maxilla of *Proegernia mikebulli* sp. nov. reconstructed from SAM P57541 (anterior fragment; top left) and SAM P57503 (near complete; top right). 'A' marks the same tooth locus on each specimen. Enclosed by the box is the tooth 'B'. Abbreviations: cl, *cuspis labialis*; cla, *culmen lateralis* anterior; cli, *cuspis lingualis*; clp, *culmen lateralis* posterior; fp, facial process; psm, palatine shelf of the maxilla; pxp, premaxillary process; sop, suborbital process; and str, striae.

Figure 6: Left premaxilla (SAM P57542) of *Proegernia mikebulli* sp. nov. from posterior (left), anterior (centre), and lateral (right) view. A, tooth one, enlarged in box. Abbreviations: flc, foramen of longitudinal canal; inp, internasal process; mp, maxillary process; nef, notch of the ethmoidal foramen; and o, osteoderm.

Figure 7: A and B: 24th tooth on the dentary SAM P57502, medial view, and C the 16th tooth in occlusal view with lateral and medial facies marked. Abbreviations: ai, *antrum intercristatum*; cl, *crista lingualis*; cm, *crista mesialis*; cul, *cuspis labialis*; cula, *culmen lateralis* anterior; and culp, *culmen lateralis* posterior.

Figure 8: Right pterygoid of *Proegernia mikebulli* sp. nov., SAM P57512 in dorsal (left) and ventral (right) views. Abbreviations: en, epipterygoid notch; pp, palatine process; pvc pterygoid ventral concavity; qp, quadrate process; and tp, ectopterygoid process.

Figure 9: A single vertebra (SAM P57514) potentially referable to *Proegernia mikebulli* sp. nov., recovered from Fish Lens, BC2. A, anterior view; B, posterior view; C, lateral view, anterior to the left; D, dorsal view; and E, ventral view. A proximal fragment of a right femur assigned to *Proegernia mikebulli* sp. nov. SAM P57540. F, dorsal; G, anterior; and H, ventral views.

Figure 10: Comparative SEM of *Proegernia mikebulli* sp. nov. (B,C) and microphotographs of *P. palankarinnensis* (A,D) from Martin et al. [16], in medial (above) and occlusal (below) views. Note the anterior extension of the splenial notch (3) in *P. palankarinnensis* and the increased the number of teeth/loci in *P. mikebulli*. Features described in text are labelled: 1) Medial inflection of the symphysis; 2) Dentary depth; 3) Position of the inferior alveolar foramen; 4) Meckel's Groove extension anterior of the inferior alveolar foramen, present in *P. palankarinnensis* and absent in *P. mikebulli*; 5) anterior extent of the internal septum similar in both taxa, 6) Overall robustness as dentary width, 7) facet for articulation with anterior ramus of coronoid, 8) splenial notch height, and 9) height of the coronoid process relative to the last tooth.

Figure 11: Comparative medial views of the dentaries of *Eutropis multifasciata* (SAMA R35693), *Proegernia mikebulli* sp. nov., *Lissolepis coventryi* (SAMA R57317) and *Liopholis multiscutata* (FUR168). 1 Anterior extent of splenial; 2 Symphysis shape and robustness; and 3 Dentary depth at mid-tooth row length.

Figure 12: Medial view of left dentary tooth crowns of *Plestiodon fasciatus* (SAMA R66784), *Eutropis multifasciata* (SAMA R35693), *Proegernia mikebulli* sp. nov. (SAM P57502), *Lissolepis coventryi* (SAMA R57317), *Eugongylus rufescens* (SAMA R36735) and *Sphenomorphus jobiensis* (SAMA R6736). *Plestiodon* represents the basal skink tooth condition as described by Richter (1994). *Culmen lateralis* posterior is labelled culp.

Figure 13: Egermiine phylogeny based on the single most parsimonious tree produced by TNT [26]. Bootstrap values >50 shown.

Figure 14: Egermiine phylogeny based on consensus total evidence Bayesian inference tree from BEAST. †denotes extinct taxon. Small numbers at nodes denote posterior probability. Reconstructed node ages for key clades are in brackets, the green shading denotes the Eocene, after which Australia became isolated from Gondwana [4].

1 Table 1: Fossil calibrations used, their minimum and maximum ages (Ma) and references for the age dates or
2 species descriptions.

3 Table 2: Major and trace elements detected by X-ray fluorescence analysis of sediments sampled Layer 3 and
4 5 from stratigraphic sections of Lake Pinpa Site 12 and Billeroo Creek Site 2. Values are %, minor trace
5 elements making the total of each sample 100 are not shown.

6 Table 3: Mineral components of sediment samples taken from Layers 3 and 5 at Lake Pinpa and Layer 5 at
7 Billeroo Creek Site 2.

8
9
10
11
12
13
14
15
16
17
18
19
20
21
22
23
24
25
26
27
28
29
30
31
32
33
34
35
36
37
38
39
40
41
42
43
44
45
46
47
48
49
50
51
52
53
54
55
56
57
58
59
60

Appendix D

Reviewer: 1

Comments to the Author(s)

The present paper is a careful work that combines paleontology and modern systematic approaches. It provides a great new perspective on the evolution of a charismatic clade of skinks native to Australia and the Solomon Islands. In some respects the results are surprising; for instance, with the new fossils Sahul clades have greater inferred ages, but the divergence of Sahul clades from *Tribolonotus* is younger in the present analysis (compared to Skinner, A., Hugall, A. F., & Hutchinson, M. N. (2011). Lygosomine phylogeny and the origins of Australian scincid lizards. *Journal of Biogeography*, 38(6), 1044-1058.).

My comments below are minor, or easily addressed.

One issue is crucial for reproducibility: the character list (missing, unless I'm much mistaken). The morphological data set is expanded from 95/40 to 102/48 discrete/continuous characters. The new discrete characters are not explained, and the new continuous characters can only vaguely be gleaned from the BEAST xlm input file. Please provide a character list similar to that in Thorn et al. (2019) on *Egernia* (unless I've overlooked something).

Response

The character list may have been missing from the uploaded SI documents, this has been rectified. The complete list of morphological characters, including the new characters (after Thorn et al. 2019) have been uploaded.

I think the title sells the paper short. To be sure, nothing in the title is not also found in the paper, but there's also a lot of apparently novel information about the environment that should be reflected in the title too. Maybe: "... and the early evolution and environment of Australian *egerniine* skinks" ? It might broaden the impact of the paper.

Response

Helpful feedback, the change has been made. The title now reads 'A new species of *Proegernia* from the Namba Formation in South Australia and the early evolution and environment of Australian *egerniine* skinks'.

Some clarification of the XRF analyses might be useful. I had a geological training but don't remember encountering the term 'soluble minerals' as used here. A literature search suggests they are a principal concern of soil science. As I understand, the authors used XRF to measure heavier elemental concentrations (Na, Mg, Ca, K, Cl), which are somehow equated with "soluble salts". To the uninitiated (me), it seems like these are probably ions and occur in lots of minerals, some readily soluble, some not. But at the end it becomes clear that the authors are concerned especially with dolomite, the formation of which is related to paleoenvironment. Correct? Some additional clarification of goals in the methods (3a) would be helpful.

Response

Clarifications added to the methods section to make aims more clear. Changed to:

‘Soluble mineral composition was determined by XRD (X-ray diffraction) and XRF (X-ray fluorescence) analyses conducted by Flinders Analytical, to test for fluctuations in evaporite concentrations indicative of changing environments.’

There are a number of cases where characters are stated or implied to be apomorphic at some level where this is inaccurate. For instance:

A fused Meckelian groove is more broadly distributed in Scincidae (Lygosominae sensu Pyron et al. 2013, including Egerniinae and Eugongylineae of the present authors). Thus, the first sentence of the Diagnosis (!) is difficult to follow.

The “bifid posterior termination of the suborbital process (an apomorphy of Scincidae)” - Gauthier et al. (2012, char. 123/1) found this to be a synapomorphy of Scincoidea.

See also below.

Response

The first sentence of the diagnosis has been modified to reflect the shift towards the plesiomorphic scincid state represented by this feature. The “bifid posterior termination of the suborbital process (an apomorphy of Scincidae)” has been changed to “...Scincoidea” and citation for Gauthier added.

To justify referrals of other specimens to the new species, the authors list characters that are frequently not apomorphic. For examples, see below. It seems that geography and process of elimination has played a role in these identifications. In other words, we assume it’s a member of an extant Australian clade; we rule out Gekkota (and Varanus), leaving only scincid, and the (usually plesiomorphic) characters are consistent with that. I think the authors’ conclusions are likely valid, but this line of argumentation should be made more explicit.

Response

We have now added the following paragraph:

To justify referrals of specimens to the new species, we took a pragmatic approach and prioritised when available, but also used distinct combinations of characters (some of uncertain polarity or plesiomorphic) in order to eliminate some competing possibilities. We assumed that the taxon most plausibly belongs to one of the extant clades of Australian squamates, and that the broad patterns of morphological variation seen in these living Australian squamate clades are a reliable guide to the limits of variation in the Miocene.

The term “Lygosominae” has been used in the literature for a very different (and more inclusive) set of taxa. The authors appear to follow a taxonomy promulgated in an obscure publication (in the journal “Herptile”) to which I do not have access. Perhaps a brief summary of that taxonomy and what the authors mean by various terms would be helpful for the reader.

Response

We do not use Lygosominae anymore and just simply refer to the actual taxon ie *Eutrophis* (*Mabuya*-group)

"The last common ancestor of Egerniinae and *Eutrophis* (*Mabuya*-group) ..."

I find myself confused by the term “splenial notch” (which is not defined or labelled). I think of a notch as a nick or short incision, like at the back end of an arrow. So when I read “splenial notch” I think of the V-shaped remnant of the Meckelian groove into which the splenial inserts. But really I think the authors mean the dorsal articulation facet of the splenial on the dentary (along the upper margin of the Meckelian groove). Please clarify.

Response

The term ‘splenial notch’ is widely used (e.g. to describe this feature, and it is illustrated on images in the character list in SI) if they require clarification. To clarify, we have added reference citations to previous use of the term when we first use it.

In Section 8 the word “stem” is sometimes used for extant taxa that branched basal to a clade in question (like Australian Egerniinae). I think “stem” can only refer to extinct forms. Please re-write: for instance, “earlier-branching” or something similar (I know some people have strong opinions).

Response

We have rephrased as

"These characters have pulled *Proegernia* outside of the Australian crown group, but not beyond the more basal Sahul genera *Corucia* and *Tribolonotus*."

Part 3c(i). The scaling of continuous character weight (Thorn et al. 2019) was, in my opinion, a clever way to deal with the issue (moving beyond the equal-character / equal-state weighting dichotomy). So the mean or median number of discrete character states per character is 3 (i.e., 0-1-2)? The value should be stated.

Response

Continuous character states were linearly scaled to values spanning 0–2 to replicate the mean number of discrete character states (three)...

Part 3c(ii). The molecular data come from other sources. The authors state that Partition Finder 2 was used, but that the same partitions were “used again” in this work. Did the authors re-run Partition Finder 2 on the data and come up with the same results? If so, maybe write “... were discovered again here.” Otherwise, maybe refer to the previous works for details of data partitioning.

Response

The molecular data were not rerun in partition finder since Thorn et al. 2019, but for clarification the method is repeated here. Wording changed to reflect this detail.

The same six molecular (gene) partitions, 12s (412 base pairs [bps]), 16s (681 bps), ND4 (693 bps), BDNF (699 bps), CMOS (835 bps) and B-fibrinogen (1051 alignable bps) and substitution models chosen in that study [15] are used again here.

Part 3c(iii). Please compare file names to supplementary files, which do not match.

Part 3c(iv). I don't see the log files mentioned at the end. These would help follow synapomorphies.

The BEAST log files do not optimise the synapomorphies (and are huge), and so were not included as SI, but the ancestral state values for the continuous characters are viewable in the consensus tree file.

First citations of Figure 1 and 2 reversed. However, the map first makes sense, so I'd consider switching the figures.

Response

We now cite figure 1 first. The stratigraphic column is required to provide context to the Namba Formation description, this Formation extends beyond the map range and as such the map is site-specific and secondary to the geological context. This is why the map is Figure 2 and the stratigraphy is Figure 1.

"GPS coordinates" - since coordinates are not presented on the map (figure 2), I think it ought to be stated which datum is being referred to (WGS 84 ?).

Response

We have added the WGS 84 datum information immediately after the GPS coordinates.

An angular bone (under description of Post-dentary / compound bone) is plesiomorphic, so its presence can only rule out Gekkota if we assume that there are no other groups present. I would suggest "compound bone" (also used by the author) over "post-dentary", as the latter could also be understood to include the angular.

Response

Fair suggestion, however we clarify that "the post-dentary complex or compound bone is the fused surangular, articular and prearticular" in scincids immediately afterwards. So there can be no confusion that this bone includes the angular.

The maxilla seems to show an unusual feature that is not mentioned (or shown in the reconstruction): the convexity of the palatal shelf just in front of the tooth position labeled "B" (fig. 5). It does not seem like this is artifactual. The AToL scan of *Egernia striolata* suggest that there might be a similar convexity there; less clearly similar is the convexity in this position in certain Xantusiidae. Also, where is the superior alveolar foramen with respect to this structure? I do not see it.

Response

The convexity is a blob of Paraloid B72, a glue used to repair the broken maxilla. This is stated in the text already as “The smoothed hemispherical shape on the dorsal edge of the palatine process of the maxilla (Figure 5) is Paraloid and not a feature of the bone.”

See suggestion on dental terminology to clarify expression of tooth dimensions.

The pterygoid is a left one. Also, I'm not familiar with the term “pterygoid head” - Oelrich's (1956) (triangular) “anterior part” would be preferable.

Response

Yes it is a left, thank you for picking up the error, now fixed. The term ‘pterygoid head’ has been replaced with ‘pterygoid body’ to avoid confusing anterior/posterior descriptives.

Identification of vertebra. While I concur that an attribution to the scincid species is likely, it should be stated that none of the features listed are apomorphic at this level. The authors proceed by eliminating other extant Australian taxa to which the vertebra could belong. That's fine, but it should be stated explicitly.

Response

Explicit statement made as suggested, and mention of the exclusion of *Varanus* due to centrum shape also added.

Section 6c describes the “more anterior extent of the splenial notch” as being a plesiomorphic trait (shared with the type species) and an “egerniine” (presumably apomorphic) feature. Please clarify.

Response

This is now clarified as

While the type specimen is currently missing, the existing description [16] suggested *Proegernia* as being representative of a transitional form between a plesiomorphic scincid dentary morphology (e.g. as exemplified by *Mabuaya*-group lygosomines) and the derived egerniine condition. The plesiomorphic traits shared by both *Proegernia palankarinnensis* and *P. mikebulli* are the small (about 1 mm long) opening of Meckel's groove immediately posterior of the symphyseal foramen and the more anterior extent of the splenial notch into the anterior half of the tooth row.

To play Devil's advocate: the striae claimed to be present labially in Martin et al.'s *Proegernia palankarinnensis* aren't clearly visible in their micrographs....

Response

If we could source the missing Holotype, we could check their presence. For now, we refer to their description where image resolution is not clear enough to determine striae.

The labeling of figure sub-parts is inconsistent. Figure 4 uses -1, -2, etc., whereas Figure 10

uses A, B, etc. and Figure 11 uses nothing.

In Figure 12, it would be helpful to state where (approximately!) in the tooth row these teeth are from (e.g., middle of the tooth row).

Response

‘mid-tooth row’ clarification added to caption.

Section 7 - abbreviation PP is for Posterior Probability

Response

Yes, correction made.

In addition to the trivial comments on language below, I have made some remarks in the annotated PDF.

Check abbreviations of names (e.g., T H Rich vs. T. H. Rich) for consistency throughout. Also, if the author’s wish to use the abbreviation LF for Local Fauna, please introduce it early, explain it, and use it consistently.

Response

Correction made to the sets of initials throughout the document. The first instance of Pinpa Local Fauna is now followed by (LF), which is used throughout document unless it is the first time a different named Local Fauna is mentioned (then stated in full, i.e. Ericmas Local Fauna and subsequently Ericmas LF thereafter).

The filler statement at the beginning of part 4c “Need an introductory sentence or two” should be deleted / replaced

Response

Change made. Intro sentence or two written.

Why are most mineral names and sometimes element names (Calcium) capitalized?

Response

Mineral names have all been changed to lower case.

Generally: delete hyphen between adverb (-ly) and the adjective it modifies, e.g., “caudally-directed”.

Response

Changes made, hyphens have been deleted.

The word “ovular” (egg-shaped) is used where I think the authors mean “oval”.

Response

Correct, change made.

Check “ed.^eds.” in references - EndNote doesn’t seem to have done these properly. They might have to be manually corrected.

Response

The reference list has been reformatted with EndNote and then errors removed after conversion to plain text.

Reviewer: 2

Comments to the Author(s)

I have reviewed the manuscript entitled "A new species of Proegernia from the Namba Formation in South Australia and the early evolution of Australian egerniine skinks" by Thorn and co-authors. I want to begin expressing that the paper is very well written and illustrated, and is a good example of a multidisciplinary approach that can result in a highly interesting results, mainly taking into account the fragmentary nature of the studied specimens. The provided geological context is important for potential future paleontological expeditions to the area, and sets a framework for interpreting the paleoecological context. Descriptions and comparisons are extensive, but not excessive. The phylogenetic analyses (using parsimony and Bayesian methods) are sound and provide interesting results. I agree with the identification of the fossils described, and with the erection of the new species. The discussion and conclusions are justified by the results, and represent an interesting starting point for (re-)interpreting the timing of events that led to the current composition of scincid lizard faunas in Australia. As I said above, images are good, although I provide in the annotated pdf a few comments regarding the resolution of a few of them (mainly figure 3, but also fig. 4.5 and fig. 12). I recommend publication of this manuscript after the authors have considered the minor revisions suggested in the annotated pdf. I just want to congratulate the authors for a well done work, and I look forward to see this published.

Annotated PDF (presumably from Reviewer 2)

Minor comment regarding clarification of wording in abstract brought to our attention the need to restructure some sentences regarding the ‘Lens’ and its occurrence within the Namba Formation across sites.

‘Age’ used to replace the complicated terms, as suggested by reviewer.

‘Formation’ added to referrals to the ‘Etadunna’ as suggested.

Leading space removed from paragraph and minor grammatical changes made as suggested.

Geological setting - Sentence regarding earlier site visits has been revisited, and rephrased to make sense.

Clarification of Fish Lens thickness at Billeroo Creek, and its relation to Layer 5, made.

Description – minor changes to terminology, spelling, and grammar made throughout as suggested in PDF and in written reviewer response. Detailed responses to other comment have been address above as they were also noted in the text reviewer #1 response.

Reviewer: 3

Comments to the Author(s)

First off I want to apologise to the editor and authors for my lengthy delay in reviewing this MS - last year was a special kind of hell!

Overall I think this is a really important contribution to the Australian herpetological literature. I have not made many comments on the morphology or paleontology as these are not areas that I know well. I have made a few comments and suggestions in on the pdf mainly around the biogeographic interpretations and also the framing.

In the abstract (and perhaps intro as well) I would suggest making the nature of Australia's Oligocene squamate fauna the clear central opening theme, and then finishing on this theme as well. This allows you to emphasise the broadly congruent emergent picture across molecules and paleo that most components of Australias extent squamate biota were likely not present much before the Miocene, but with added grwoing evidence that the Egernia group may be an older component. This is also perhaps the inference that is most broadly interesting.

Response

Abstract now includes:

The diverse living Australian lizard fauna contrasts greatly with their limited Oligo-Miocene fossil record.

These taxa also help pinpoint the timing of the arrival of scincids to Australia, with egerniines the first radiation to reach the continent.

With respect to dates, I think the comparison of ages across papers should be a bit more guarded - there are lots of methodological reasons why these dates could vary - so certainly comparing the dates estimated here for egernia group with dates for sphenomorphines is an ok thing to do, but its needs to be caveated. In that context suggest emphasising the congruence - skinner et al suggested that unlike other skinks and especially spheno group - the Egernia group has been in the Australia region for 50 million years, your analyses are complementing that by showing that they are also in the Australia for ~30 million years.

In comparisons with Adams 2011 paper (para II) - He dated the ages of crown radiations, this is different to arrival dates (although they may of course be linked)

Now rephrased:

the Sphenomorphinae were the first group to reach Australia, **before 25.3 Ma.**

I have had a little bit of a speil about Sahul in place - especially going back beyond the Miocene I think this term which describes a contemporary conglomerate, runs the risk of confounding rather than elucidating patterns - one alternative might be Australian Craton vs Melanesia island arcs and Terranes??

We change Sahul to Australasia or Australia where appropriate; finer distinction is not important in much of our discussion.

Re island arcs etc - latest analyses we did (Tallowin et al 2018), suggest agamids actually colonised arcs/proto-Papua from Australia, just to complicate matters further. But of course with all these ancestral state analyses, it only takes one or two extinction events to reorient the story.

Re ages of skinks see if you can track down a copy of Charles Linkem's PhD thesis (has been available online). This provides clear evidence that the Australian sphenos are recently derived from SEA groups, while the Melanesian sphenos (multiple lineages) can have much longer and deeper histories, comparable timing to Tribolonutis.

Have made a bunch of further comments in the pdf of manuscript and picked up a few small typos etc. Overall good work and great to see it, will be good to see published.

Summary - an introductory sentence on diverse contemporary skink/squamate fauna juxtaposed against a rubbish fossil record would certainly help orient the uninitiated here

Response

Added: The diverse living Australian lizard fauna contrasts greatly with their limited Oligo-Miocene fossil record.

You could argue this is not true for Egernia group, they aside from a few odd lineages in Melanesia, they don't appear to have any friends anywhere - indeed when we working on this we considered the Egernia group as somewhat lineball in the Gondwanan vs immigrant dichotomy

Likewise for Eugongylus group - Australia and Islands!

Response

Minor changes to the wording have been made in response to this reviewer and the others, see below.

'The ages and origins of Australian scincid lizard radiations are still relatively poorly constrained, and molecular data are unclear beyond a few island-dwelling extant relatives in south-east Asia [9].'

are radiation ages being confounded with arrival dates - perhaps 'arrived before..' might be better phrasing? e.g would certainly say that Skinner et al does not strongly indicate that egernia group arrived after eugongylus groups, however it does argue that the contemporary radiation is younger (if Corucia is excluded)

do they have to have arrived, could they be a Gondwanan origin with high level of turnover?
again not sure if arrival is best word - maybe minimum estimates for how long scincids have been in Australia

Response

Have now clarified, that these dates are all minimum ages and actual arrival order (ie constrained by max stem age) might be different.

rephrased "to infer a more robust estimate for the age of the Australian radiation of egeriines, which represents the latest possible date for their arrival in Australia."

Geological setting - 'previously' -implies there is now an alternative explanation, or at least the coprolite theory has been rejected?

Response

The coprolite idea was mentioned in some of the first publications to come from the Namba Formation including Vickers-Rich et al. 1991. The presence of large lenses of this material somewhat negates the isolated 'coprolite' pockets idea. The term 'previously' has been replaced with 'first'.

Could the sections for the two different sites get an A and a B - and be split up a bit more to emphasise differentiation?

Response

This figure exists to show the similarity between the two sections. Although minor changes in the thickness of units is present, they do correlate across sites with similar fauna and in the same stratigraphic order.

maybe this is just a draft issue, but resolution on some text in this figure is rough

Response

Document resolution is with compressed figures to reduce working document size. Higher resolution images will be uploaded with final manuscript.

Is this striking? If you randomly sampled skink skulls from a contemporary Australian reptile fauna, my guess is that you would not get just one species (unless there were ecological biases driving deposition)

Response

Yes, finding only one living scincid species in a site is unusual in modern Australia, but only finding one egeriine is not uncommon in Holocene deposits. Taphonomy favours preservation of larger taxa or taxa accumulation via selective predators, and furthermore many extant species are cannot be differentiated based on single skull elements. Other lizards may have been present, but possibly not preferred prey, or they were not living close enough to the palaeo-lake.

presumably material examined, but not assigned to the palankarinnesis? Make this clear perhaps?

Response

It is the Holotype specimen, will change to 'type specimen' to make clear. Only the Holotype is known for *Proegernia palankarinnensis*.